# Ocean Iron Fertilization Experiments: Past–Present–Future looking to a future Korean Iron Fertilization Experiment in the Southern Ocean (KIFES) Project

Joo-Eun Yoon[1], Kyu-Cheul Yoo[2], Alison M. Macdonald[3], Ho-Il Yoon[2], Ki-Tae Park[2], Eun Jin Yang[2], Hyun-Cheol Kim[2], Jae Il Lee[2], Min Kyung Lee[2], Jinyoung Jung[2], Jisoo Park[2], Jiyoung Lee[1], Soyeon Kim[1], Kitae Kim[2], and Il-Nam Kim[1*]

[1]Department of Marine Science, Incheon National University, Incheon 22012, Republic of Korea
[2]Korea Polar Research Institute, Incheon 21990, Republic of Korea
[3]WHOI, MS 21, 266 Woods Hold Rd., Woods Hole, MA 02543, USA

*Correspondence to*: Il-Nam Kim (ilnamkim@inu.ac.kr)

**Abstract.** Since the start of the industrial revolution, human activities have caused a rapid increase in atmospheric $CO_2$ concentrations, which have, in turn, had an impact on climate leading to global warming and ocean acidification. Various approaches have been proposed to reduce atmospheric $CO_2$. The 'Martin (or Iron) Hypothesis' suggests that ocean iron fertilization (OIF) could be an effective method for stimulating oceanic carbon sequestration through the biological pump in iron-limited, high-nutrient, low-chlorophyll (HNLC) regions. To test the Martin hypothesis, 13 artificial OIF (aOIF) experiments have been performed since 1990 in HNLC regions. These aOIF field experiments have demonstrated that primary production can be significantly enhanced by the artificial addition of iron. However, except in the Southern Ocean European Iron Fertilization Experiment, the effectiveness of aOIF (i.e., the amount of iron-induced carbon export flux below the winter mixed layer depth) has been unexpectedly low compared to that achieved by natural phytoplankton blooms. These results, including possible side effects have been debated amongst those who support and oppose aOIF experimentation, and many questions such as effectiveness of scientific aOIF, environmental side effects, and international aOIF law frameworks remain. In the context of increasing global and political concerns associated with climate change, it is valuable to examine the validity and usefulness of the aOIF experiments. To maximize the effectiveness of aOIF experiments under international aOIF regulations in the future, we suggest a design that incorporates several components. (1) Experiments conducted in the center of an eddy structure when grazing pressure is low and silicate levels are high (e.g., in the Southern Ocean south of polar front during early summer). (2) Shipboard observations extending over a minimum of ~40 days, with multiple iron injections (at least 2 (or 3) iron infusions of ~2,000 kg with an interval of ~10–15 days to fertilize a patch of 300 km$^2$ and obtain a ~2 nM concentration). (3) Tracing of the iron fertilized patch using both physical (e.g., a drifting buoy) and biogeochemical (e.g., sulfur hexafluoride and photosynthetic quantum efficiency) tracers. (4) Employment of neutrally buoyant sediment traps and application of the water-column derived [234]Thorium method at two depths (i.e., just below the *in situ* mixed layer depth and at the winter mixed layer depth), with autonomous profilers equipped with an underwater video profiler and a transmissometer. (5) Monitoring of side effects on marine/ocean ecosystems, including production of climate-relevant gases (e.g., $N_2O$, dimethyl sulfide, and halogenated volatile organic compounds), decline in oxygen inventory, and development of toxic algae blooms, with optical sensor equipped autonomous moored profilers and/or autonomous benthic vehicles. Lastly, we introduce the scientific aOIF experimental design guidelines for a future Korean Iron Fertilization Experiment in the Southern Ocean.

**Keywords:** Ocean Iron Fertilization; High-Nutrient and Low-Chlorophyll regions; Biological Pump; Phytoplankton; Iron

## 1 Introduction

Since the start of the industrial revolution, human activities have caused a rapid increase in atmospheric carbon dioxide ($CO_2$, a major greenhouse gas) from ~280 ppm (pre-industrial revolution) to ~400 ppm (present day) (http://www.esrl.noaa.gov/), which has, in turn, led to global warming and ocean acidification, indicating that there is an urgent need to reduce global greenhouse gas emissions (IPCC, 2013) (Fig. 1). As the Anthropocene climate system has rapidly become more unpredictable, the scientific consensus is that the negative outcomes are a globally urgent issue that should be resolved in a timely manner for the sake of all life on Earth (IPCC, 1990, 1992, 1995, 2001, 2007, 2013). The various ideas/approaches that have been proposed to relieve/resolve the problem of global warming (Matthews, 1996; Lenton and Vaughan, 2009; Vaughan and Lenton, 2011; IPCC, 2014; Leung et al., 2014; Ming et al., 2014) largely fall into two categories: (1) reduction of atmospheric $CO_2$ by the enhancement of biological $CO_2$ uptake (including ocean fertilization) and/or the direct capture or storage of atmospheric $CO_2$ through chemically engineered processes, and (2) control of solar radiation by artificial aerosol injection into the atmosphere to augment cloud formation and cloud brightening to elevate albedo (Fig. 2). One of the most attractive methods among the proposed approaches is ocean fertilization (https://web.whoi.edu/ocb-fert/), which targets the drawdown of atmospheric $CO_2$ by nutrient addition (e.g., iron, nitrogen, or phosphorus compounds) to stimulate phytoplankton growth and, subsequently, carbon export to the deep ocean or sediments via the ocean biological pump (ACE CRC, 2015).

The ocean biological pump is frequently depicted as a single combined process, whereby organic matter produced by phytoplankton during photosynthesis in surface waters is quickly transported to intermediate and/or deep waters (Fig. 3a) (Volk and Hoffert, 1985; De La Rocha, 2007). Although the effectiveness of the biological pump is primarily controlled by the supply of macro-nutrients (i.e., nitrate, phosphate, and silicate) from the deep ocean into the mixed layer (ML) leading to new production (Sarmiento and Gruber, 2006), iron acts as an essential micro-nutrient to stimulate the uptake of macro-nutrients for phytoplankton growth (Fig. 3b) (Martin and Fitzwater, 1988; Martin, 1990; Morel and Price, 2003). In the subarctic North Pacific (NP), equatorial Pacific (EP), and Southern Ocean (SO), which are well known as high-nutrient and low-chlorophyll (HNLC) regions (Figs. 4a and b), phytoplankton cannot completely utilize the available macro-nutrients (particularly nitrate) for photosynthesis due to a lack of iron. As a consequence, primary production (PP) in these HNLC regions is relatively low, despite the high availability of macronutrients (in particular nitrate and phosphate) (Figs. 4a and b).

Analyses of trapped air bubbles in Arctic/Antarctic ice cores have revealed that atmospheric $CO_2$ (~180 ppm) during the Last Glacial Maximum (LGM; ~20,000 years ago) was much lower than during pre-industrial times (~280 ppm) (Neftel et al., 1982; Barnola et al., 1987; Petit et al., 1999). Over the last 25 years, several hypotheses have been proposed to explain the lowered atmospheric $CO_2$ level during the LGM (Broecker, 1982; McElroy, 1983; Falkowski, 1997; Broecker and Henderson, 1998; Sigman and Boyle, 2000). Dust inputs are generally regarded as a major natural iron source for ocean fertilization, and Martin (1990) hypothesized that during the LGM increased dust inputs relieved iron-limitation and, thereby, substantially enhanced the biological pump in HNLC regions, particularly in the SO (Fig. 3b). Since Martin's hypothesis was first published, there has been an enormous interest in ocean iron fertilization (OIF) because only a small amount of iron (C:Fe ratios = 100,000:1, Anderson and Morel, 1982) is needed to stimulate a strong phytoplankton response. Therefore, much of the investigative focus has centered on the artificial addition of iron to HNLC regions as a means of enhancing carbon fixation and subsequent export via the biological pump (ACE CRC, 2008).

To test Martin's hypothesis, six natural OIF (nOIF) and 13 artificial OIF (aOIF) experiments have been performed to date in the subtropical North Atlantic (NA), EP, subarctic NP, and SO (Blain et al., 2007; Boyd et al., 2007; Pollard et al., 2009; Strong et al., 2009; Smetacek et al., 2012; Martin et al., 2013; Blain et al., 2015) (Fig. 4a and Table 1). These OIF experiments demonstrated, particularly for the SO, that PP could be significantly increased after iron addition (de Baar et al.,

2005; Boyd et al., 2007). However, for aOIF to be considered as a useful geoengineering approach (IPCC, 2007), in the long run, the most critical issue is the 'effectiveness of aOIF'. That is, whether a significant portion of the organic carbon produced by aOIF in the surface waters is exported below the winter mixed layer depth (MLD) to intermediate/deep layers for long-term (~1,000 years) storage (Fig. 3c) (Lampitt et al., 2008). A high carbon export was observed in the nOIF experiments in the SO near the Kerguelen Plateau and Crozet Islands (Blain et al., 2007; Pollard et al., 2009). However, all aOIF experiments have shown unexpectedly low carbon exports compared to natural systems (de Baar et al., 2005; Boyd et al., 2007), except for the SO European Iron Fertilization Experiment, EIFEX (Smetacek et al., 2012). The results of these experiments, as well as the potential side effects (e.g., $N_2O$ production and development of hypoxia) (Fuhrman and Capone, 1991), have been scientifically debated amongst those who support and oppose aOIF experimentation (Chisholm et al., 2001; Johnson and Karl, 2002; Lawrence, 2002; Buesseler and Boyd, 2003; Smetacek and Naqvi, 2008; Williamson et al., 2012).

In the context of increasing global (social-political-economic) concerns associated with rapid climate change, it is necessary to examine the validity and usefulness of aOIF experimentation as a climate change mitigation strategy. Therefore, the purpose of this paper is to: (1) provide a thorough overview of the aOIF experiments conducted over the last 25 years; (2) discuss aOIF-related important unanswered questions, including carbon export measurement methods, potential side effects, and international law; (3) suggest considerations for the design of future aOIF experiments to maximize the effectiveness of the technique and begin to answer open questions; and (4) introduce design guidelines for a future Korean Iron Fertilization Experiment in the Southern Ocean (KIFES) project.

## 2 Past: Overview of previous aOIF experiments

A total of 13 aOIF experiments have been conducted in the following areas: 12 experiments were conducted in the three main HNLC (i.e., nitrate >~10 μM) regions: two in the EP, three in the subarctic NP, and seven in the SO (Table 1, Figs. 4a and b). One experiment was conducted in the subtropical NA, known to be a low-nutrient and low-chlorophyll (LNLC) (i.e., nitrate <1 μM) region. These aOIF experiments have been conducted with various/multiple objectives/hypotheses to investigate the biogeochemical responses of ocean environments to artificial iron additions (Table 2). This overview of past aOIF experimentation begins in Section 2.1, with a presentation of the reasons why each experiment was performed and the main hypotheses (Table 2). The unique ocean conditions for the various experiments are described in Section 2.2. Iron addition and tracing methods are described in Section 2.3. The biogeochemical responses to the aOIF experiments are presented in Section 2.4, and finally the significant findings from these experiments are summarized in Section 2.5.

### 2.1 Objectives/hypotheses of previous aOIF experiments

#### 2.1.1 Equatorial Pacific

Initially, Martin's hypothesis was supported by the results of laboratory and shipboard iron-enrichment bottle experiments (Hudson and Morel, 1990; Brand, 1991; Sunda et al., 1991; DiTullio et al., 1993; Hutchins et al., 1993). However, the extrapolation of these results based on bottle incubations that exclude higher trophic levels has been strongly criticized due to possible underestimates in grazing rates and other bottle effects. To deal with these issues, *in situ* iron fertilization experiments at the whole-ecosystem level are required. Under the hypothesis that aOIF would increase phytoplankton productivity by relieving iron limitations on phytoplankton in HNLC regions, the first aOIF experiment, Iron Enrichment Experiment (IronEx-1), was conducted over 10 days in October 1993 in the EP where high light intensity and

temperatures would promote rapid phytoplankton growth (Table 1 and Fig. 4a) (Martin et al., 1994; Coale et al., 1998).

However, the magnitude of the biogeochemical responses in IronEx-1 was not as large as expected (Martin et al., 1994). Four hypotheses were advanced to explain the weak responses observed: (1) the possibility of unforeseen micronutrient (e.g., zinc, cadmium, and manganese) or macro-nutrient (e.g., silicate) limitations, (2) the short residence time of bioavailable iron in the surface patch due to colloidal aggregation and/or sinking of larger particles containing iron, (3) insufficient light brought about by subduction of the patch, and (4) high grazing pressure by zooplankton (Martin et al., 1994; Cullen, 1995; Coale et al., 1996; Gordon et al., 1998). To test the four hypotheses, a second aOIF experiment, IronEx-2, was conducted in May 1995 (Coale et al., 1996). The IronEx-2 research cruise investigated the same area for a longer period (17 days), providing more time to collect information about the biogeochemical, physiological, and ecological responses to the aOIF.

### 2.1.2 Southern Ocean

The SO plays an important role in intermediate and deep-water formation, and has the greatest potential of any of the major ocean basins for carbon sequestration associated with artificial iron addition (Martin, 1990; Sarmiento and Orr, 1991; Cooper et al., 1996; Marshall and Speer, 2012). It is known as the largest HNLC region in the World Ocean and models simulating aOIF have predicted that among all HNLC regions, the effect of OIF on carbon sequestration is greatest in the SO (Sarmiento and Orr, 1991; Aumont and Bopp, 2006). However, a simple extrapolation of the IronEx-2 results to the SO was not deemed appropriate because of the vastly different environmental conditions (Coale et al., 1996), therefore, based on the lessons from the EP experiments, several aOIF experiments were carried out in the SO (Frost, 1996; Boyd et al., 2000; Smetacek, 2001; Coale et al., 2004; Harvey et al., 2010; Smetacek et al., 2012; Martin et al., 2013). To test the roles of iron and light availability as key factors controlling phytoplankton dynamics, community structure, and grazing in the SO, the Southern Ocean Iron Release Experiment (SOIREE) (Table 1 and Fig. 4a), the first *in situ* aOIF experiment performed in the SO, took place in February 1999 (13 days) in the Australasian-Pacific sector (Boyd et al., 2000).

The following year, a second aOIF experiment in the SO, EisenEx ('Eisen' means iron in German), was performed in November within an Antarctic Circumpolar Current eddy in the Atlantic sector (Smetacek, 2001). This region is considered to have a relatively high iron supply, which is supported by dust inputs and possibly icebergs (de Baar et al., 1995; Quéguiner et al., 1997; Smetacek et al., 2002). EisenEx was designed to test how atmospheric dust, an important source of iron in ocean environments, might have led to a dramatic increase in ocean productivity during the LGM due to the relief of iron-limiting conditions for phytoplankton growth (Abelmann et al., 2006).

In addition to iron availability, the supply of silicate is also considered to be an important factor controlling PP in the SO. Silicate-requiring diatoms, which are large-sized phytoplankton, play an important role in the biological pump and are responsible for ~75% of the annual PP in the SO (Tréguer et al., 1995). The silicate concentrations in the SO show a decreasing northward gradient, in particular, on either side of the Antarctic Polar Front (PF), with low silicate concentrations ($<5$ μM) in the sub-Antarctic waters north of the PF ($<61°$S) and high silicate concentrations ($>60$ μM) to the south of the PF (Fig. 4c). Therefore, to address the impact of iron and silicate on phytoplankton communities and export, two aOIF experiments were conducted during January–February 2002 in two distinct regions: the Southern Ocean iron experiment-north (SOFeX-N) and -south (SOFeX-S) of the PF (Table 1) (Coale et al., 2004; Hiscock and Millero, 2005). Two years later, the Surface Ocean Lower Atmosphere Study (SOLAS) Air–Sea Gas Exchange (SAGE) experiment was conducted during March–April 2004 (15 days) in sub-Antarctic waters, which are typically HNLC with low silicate concentrations (HNLCLSi). The aim was to determine the response of phytoplankton dynamics to iron addition in an HNLCLSi region (Fig. 4c) (Law et al., 2011). SAGE was designed with the assumption that the response of phytoplankton blooms to aOIF could be detected by enhanced air-sea exchanges of climate-relevant gases (e.g., $CO_2$ and dimethyl sulfide (DMS)) (Harvey et al.,

2010; Law et al., 2011).

These early aOIF experiments resulted in clear increases in phytoplankton biomass and PP, but the impact on export production (i.e., carbon export from the surface waters to below the winter MLD) was not evident (Fig. 3c) (de Baar et al., 2005; Boyd et al., 2007). To determine if aOIF could increase export production, EIFEX was carried out in the closed core of a cyclonic eddy near the PF during the austral summer of 2004 (Fig. 5). Because it was designed to investigate the termination of a bloom and resulting export production, EIFEX was much longer (39 days) than earlier experiments (mean ± SD = 22 ± 10 days; SD represents standard deviation) (Smetacek et al., 2012).

Of similar duration, the Indo-German iron fertilization experiment (LOHAFEX; 'Loha' is iron in Hindi) was conducted during January–March 2009 (40 days), also in a PF cyclonic eddy in HNLCLSi waters (Smetacek and Naqvi, 2010; Martin et al., 2013).

### 2.1.3 Subarctic North Pacific

The subarctic NP shows a strong longitudinal gradient in aeolian dust deposition (i.e., high dust deposition in the west, but low in the east) (Duce and Tindale, 1991; Tsuda et al., 2003; Takeda and Tsuda, 2005), which is different from the other two HNLC regions (i.e., EP and SO). To investigate the relationship between the phytoplankton biomass/community and dust deposition, the Subarctic Pacific iron Experiment for Ecosystem Dynamics Study-1 (SEEDS-1) was conducted in July–August 2001 (13 days) in the western subarctic gyre (Tsuda et al., 2003, 2005). In 2004, the experiment was repeated (SEEDS-2) in almost the same location and season. In the intervening year, the Subarctic Ecosystem Response to Iron Enrichment Study (SERIES) was performed in July–August 2002 (25 days) in the Gulf of Alaska (representing the eastern subarctic gyre ecosystem) to compare the response of phytoplankton in this area with that in the western subarctic (Boyd et al., 2004, 2005). The SEEDS-1/-2 experiments focused on changes in phytoplankton composition, vertical carbon flux, and climate-relevant gas production stimulated by artificial iron addition (Tsuda et al., 2005, 2007). The main objective of SEEDS-2 and SERIES was to determine the most significant factor (i.e., nutrient supply and/or grazing) controlling the iron induced phytoplankton bloom from its beginning to its end (Tsuda et al., 2003; Boyd et al., 2004).

### 2.1.4 Subtropical North Atlantic

Unlike HNLC regions, PP in LNLC regions, which are predominantly occupied by $N_2$ fixers, is generally co-limited by phosphate and iron (Mills et al., 2004). To investigate the impact of iron and phosphate co-limitation on PP, the *in situ* phosphate and iron addition experiment (FeeP) was conducted by adding both phosphate and iron in a LNLC region of the subtropical NA during April–May 2004 (21 days) (Rees et al., 2007). The location of the subtropical NA experiment corresponded to a typical LNLC region (Figs. 4a and b, Tables 3 and 4) with low nutrients (nitrate: <0.01 μM, phosphate: ~0.01 μM, and iron: <0.4 nM) and chlorophyll-a (<0.1 mg m$^{-3}$) conditions much lower than other experimental sites. The FeeP experiment reported that pico-plankton (0.2–2.0 μm) abundances increased after iron and phosphate additions (Rees et al., 2007); however, no other details on the biogeochemical response to aOIF in FeeP have been reported. This experiment will, therefore not be discussed further.

### 2.2 Environmental conditions prior to iron addition

The initial environment (~1–7 days before iron addition) can affect the outcome of an aOIF experiment, and the experiments described above were conducted under a wide range of physical and biogeochemical conditions. Below we consider the similarities and differences in these environments according to the physical and biogeochemical properties of

the sites (Coale et al., 1998; Steinberg et al., 1998; Bakker et al., 2001; Boyd and Law, 2001; Gervais et al., 2002; Coale et al., 2004; Boyd et al., 2005; Takeda and Tsuda, 2005; Tsuda et al., 2007; Cisewski et al., 2008; Harvey et al., 2010; Cavagna et al., 2011) (Fig. 6, Tables 3 and 4).

### 2.2.1 Equatorial Pacific

The first two aOIF experiments, IronEx-1 and IronEx-2, which were both conducted in the EP, were performed in different seasons (i.e., IronEx-1: October, IronEx-2: May). However, the initial surface physical conditions were similar, with warm temperatures ($24.1 \pm 1.2°C$), high surface photosynthetic available radiation values (~$51.7 \pm 2.1$ mol m$^{-2}$ d$^{-1}$), and shallow MLDs ($27.5 \pm 2.5$ m) (Figs. 6c and d) (Coale et al., 1996; Coale et al., 1998; Steinberg et al., 1998; de Baar et al., 2005).

The initial surface biogeochemical conditions were high nutrients (i.e., nitrate = $10.6 \pm 0.2$ μM, phosphate = $0.9 \pm 0.06$ μM, and silicate = $4.5 \pm 0.6$ μM) and low chlorophyll-a concentrations ($0.2 \pm 0.05$ mg m$^{-3}$) (Tables 3 and 4). The pico-phytoplankton community, including *Synechococcus* and *Prochlorococcus*, was dominant (Martin et al., 1994; Coale et al., 1996; Cavender-Bares et al., 1999). Initial surface nutrient concentrations were relatively low compared with other ocean basin aOIF sites (Table 3 and Fig. 6e). Initial photosynthetic quantum efficiency (i.e., Fv/Fm ratio, where Fm is the maximum chlorophyll fluorescence yield and Fv is the difference between Fm and the minimum chlorophyll fluorescence yield) (Butler, 1978), which is widely used to determine the degree to which iron is the limiting nutrient for phytoplankton growth (the Fv/Fm ratio ranges from 0.2 to 0.65 where conditions are less iron limited as Fv/Fm approaches 0.65), was less than ~0.3 (Fig. 6g and Table 4), suggesting severe iron limitation (Behrenfeld et al., 1996; Barber and Hiscock, 2006; Aiken et al., 2008). In the EP, initial surface partial pressure of $CO_2$ ($pCO_2$) values were $504.5 \pm 33.5$ μatm, which were much higher than those observed in the SO ($355.6 \pm 11.7$ μatm) or the subarctic NP ($370.0 \pm 16.3$ μatm) (Table 3) (Steinberg et al., 1998).

### 2.2.2 Southern Ocean

The initial physical conditions for the aOIF experiments in the SO (SOIREE, EisenEx, SOFeX-N/-S, EIFEX, SAGE, and LOHAFEX) were very different from those found in the EP; MLDs were much deeper ($57.9 \pm 19.2$ m) (Fig. 6c) and sea surface temperature (SST) was much lower ($4.7 \pm 3.4$ °C) (Fig. 6d). During SOFeX-N/-S, which were conducted along the same line of longitude, on either side of the PF, there were distinct differences in SST: 5.0°C in SOFeX-N and −0.5°C in SOFeX-S (Coale et al., 2004). SAGE was the northernmost of the aOIF experiments in the SO (Table 1) and, therefore, had the highest SST (11.5°C) (Fig. 6d) (Harvey et al., 2010).

The locations for the aOIF experiments were selected following preliminary surveys to confirm the HNLC conditions, i.e., based on satellite imagery, nutrient concentrations, and Fv/Fm. Initial nitrate concentrations ranged from 7.9 μM (SAGE) to 26.3 μM (SOFeX-S) (Fig. 6e and Table 3). Among the various aOIF HNLC experiment sites, the SO had the highest initial nitrate concentrations ($21.4 \pm 5.8$ μM), while the EP had the lowest ($10.6 \pm 0.2$ μM). Initial nitrate and phosphate concentrations at aOIF sites in the SO followed a latitudinal gradient, with higher values to the south of 50°S (nitrate: $24.6 \pm 1.6$ μM and phosphate: $1.6 \pm 0.2$ μM) and lower values to the north (nitrate: $17.1 \pm 6.7$ μM and phosphate: $1.1 \pm 0.4$ μM) (Table 3, Figs. 4b and 6e). The full range of initial silicate concentrations has been covered by the various SO aOIF experiments, with values ranging from ~1.0 μM (SAGE) in the most northernmost site to ~60 μM (SOFeX-S) in the most southernmost (Table 3, Figs. 4c and 6f). With the specific intent of investigating the co-limitation of iron and silicate, SOFeX-N, SAGE, and LOHAFEX were all conducted in HNLCLSi regions, with initial silicate concentrations less than 2.5 μM (Figs. 4c and 6f) (Coale et al., 2004; Harvey et al., 2010; Martin et al., 2013; Ebersbach et al., 2014). Initial $pCO_2$ values were low in the SO ($355.6 \pm 11.7$ μatm) ranging from 330 μatm (SAGE) to 367 μatm (SOFeX-N) (Table 3).

As in the EP, initial Fv/Fm values were below ~0.33 (Table 4 and Fig. 6g), indicating severe iron limitation. Prior to iron addition, initial chlorophyll-a concentrations ranged from ~0.15 to 0.70 mg m$^{-3}$. The maximum initial chlorophyll concentrations occurred in EIFEX, which started with a community dominated by diatoms (Hoffmann et al., 2006; Assmy et al., 2013), while the lowest initial chlorophyll concentrations occurred in SOFeX-N, with a community dominated by a nano-plankton (2.0–20 μm), such as prymnesiophytes, pelagophytes, and dinoflagellates (Coale et al., 2004).

### 2.2.3 Subarctic North Pacific

The subarctic NP aOIF experiments (i.e., SEEDS-1/-2 and SERIES) were performed in regions with high nitrate (15.6 ± 4.0 μM) and low chlorophyll-a concentrations (0.7 ± 0.2 mg m$^{-3}$) (Tables 3 and 4, Figs. 6e and h). Compared with the other aOIF experiments, these subarctic experiments had much higher initial silicate concentrations (27.3 ± 9.6 μM) (Table 3 and Fig. 6f) and shallower MLDs (Fig. 6c). Although SEEDS-1 and SEEDS-2 were conducted in almost the same location and season in the western basin (Tsuda et al., 2007), the MLD in SEEDS-1 (8.5 m) was shallower than in SEEDS-2 (28 m).

Unlike the latitudinal gradients seen in the aOIF experiments in the SO, there were longitudinal gradients in physical and biogeochemical properties in the subarctic NP experiments (Tables 3–4, Figs. 4b–c and 6d–h). Initial SSTs in the subarctic NP were lower in the western region (7.5°C in SEEDS-1 and 8.4°C in SEEDS-2) than in the eastern region (12.5°C in SERIES) (Fig. 6d). Initial nutrient concentrations were much higher in the west (nitrate: 18.5 ± 0.1 μM and silicate: 34.0 ± 2.2 μM) compared to the east (nitrate: 10 μM and silicate: 14 μM) (Table 3, Figs. 4b–c and 6e–f). There was also a longitudinal gradient in chlorophyll-a concentrations, with relatively high values in the west (SEEDS-1: 0.8 mg m$^{-3}$ and SEEDS-2: 0.8 mg m$^{-3}$) and low value in the east (SERIES: 0.4 mg m$^{-3}$) (Fig. 6h). Before the first SEEDS-1 iron infusion, micro-phytoplanktons (20–200 μm), such as the pennate diatom "*Pseudo-nitzschia turgidula*", were dominant, whereas the areas for SERIES and SEEDS-2 were exclusively occupied by pico- and nano-phytoplankton, such as *Synechococcus* and haptophytes (Boyd et al., 2005; Tsuda et al., 2005; Sato et al., 2009). Initial Fv/Fm ratios in the subarctic NP aOIF experiments were <0.3, indicating a severe iron limitation (Fig. 6g).

### 2.3 Iron addition and tracing methods

### 2.3.1 Iron addition

Iron(II) and sulfate aerosols are ubiquitous in the atmosphere and, therefore, iron-sulfate (FeSO$_4$·H$_2$O), a common form of combined iron that enters the ocean environment via dust deposition, has been frequently regarded as a bioavailable iron source during glacial periods (Zhuang et al., 1992; Zhuang and Duce, 1993; Spolaor et al., 2013). Iron-sulfate is a common inexpensive agricultural fertilizer that is relatively soluble in acidified seawater (Coale et al., 1998). Therefore, all aOIF experiments have been conducted by releasing commercial iron-sulfate dissolved in acidified seawater into the propeller wash of a moving ship (Fig. 5), to ensure mixing with surface waters during iron additions.

In general, background dissolved iron concentrations in HNLC regions are <0.2 nM (Table 1). Iron-enrichment bottle incubation experiments performed in deck incubators using *in situ* seawater have indicated the maximum phytoplankton growth rates in response to iron additions of 1–2 nM (Fitzwater et al., 1996). In aOIF experiments performed in the ocean, targeted iron concentrations within the ML have ranged between ~1 to 4 nM, depending on the site (Table 1 and Fig. 6b) (Martin et al., 1994; Coale et al., 1996; Boyd et al., 2000; Bowie et al., 2001; Tsuda et al., 2003; Coale et al., 2004; Nishioka et al., 2005; Law et al., 2006; Smetacek et al., 2012; Martin et al., 2013). If injected iron is well dispersed throughout the ML within 24 hours by convective mixing (Martin and Chisholm, 1992), the amount of added iron required to raise the

background iron concentration to the target level can be calculated using a volume estimate (i.e., iron-fertilized water patch area × MLD) (Watson et al., 1991). To minimize uncertainty between the first iron addition and phytoplankton response, aOIF experiments have involved multiple-small iron injections to the surface waters in the study area at ~0.4 to ~1.5 km intervals over a 1–2-day period (Coale et al., 1998). The patch size fertilized by the first iron addition varied from 25 km$^2$ (e.g., FeeP; Iron(II) addition of 1840 kg) to 300 km$^2$ (e.g., LOHAFEX; Iron(II) addition of 2,000 kg), and by the end of these experiments had spread to a maximum ~2500 km$^2$ (Coale et al., 2004; Boyd et al., 2007; Martin et al., 2013) (Table 1, Fig. 6a).

During the experiments, dissolved iron concentrations increased to the target ~1.0–4.0 nM (Table 1 and Fig. 6b), but decreased to background concentrations within days. The fast decrease in dissolved iron concentrations indicates that iron was horizontally dispersed and/or rapidly incorporated into particles. These processes occur more rapidly in warmer waters (ACE CRC, 2015). For example, the first aOIF experiment, IronEx-1, showed that the dissolved iron concentration rapidly decreased from 3.6 to 0.25 nM ~4 days after iron addition in the center of the fertilized patch, suggesting a limit to the level required for phytoplankton growth (Coale et al., 1998; Gordon et al., 1998). As a result, except for the single iron addition experiments of IronEx-I, SEEDS-1, and FeeP (Martin et al., 1994; Tsuda et al., 2003; Rees et al., 2007), most aOIF experiments have involved multiple iron additions at the patch center, to continuously derive the stimulation of phytoplankton during the experiments. These experiments included: (2 additions) EIFEX, SERIES, SEEDS-2, LOHAFEX (Boyd et al., 2005; Tsuda et al., 2007; Smetacek et al., 2012; Martin et al., 2013); (3 additions) IronEx-2, EisenEx, SOFeX-N (Coale et al., 1996; Gervais et al., 2002; Coale et al., 2004; Nishioka et al., 2005); and (4 additions) SOIREE, SOFeX-S, SAGE (Boyd et al., 2000; Coale et al., 2004; Bakker et al., 2005; Harvey et al., 2010) (Table 1).

**2.3.2 Tracing iron-fertilized patch**

To trace the iron-fertilized patch, aOIF experiments have used a combination of physical and biogeochemical approaches. All the aOIF experiments except EIFEX have used sulfur hexafluoride (SF$_6$) as a chemical tracer (Table 1) (Martin et al., 1994; de Baar et al., 2005; Smetacek et al., 2012). The SF$_6$, which is not naturally found in oceanic waters, is a useful tracer for investigating physical mixing and advection-diffusion processes in the ocean environment due to its nontoxicity, biogeochemically inert characteristics, and low detection limit (Law et al., 1998). The injected SF$_6$ is continuously monitored using gas chromatography with an electron capture detector system (Law et al., 1998; Tsumune et al., 2005). Usually only one SF$_6$ injection is necessary because background levels are generally extremely low in the ocean (<1.2 fM; f: femto-, 10$^{-15}$) (Law et al., 1998; Law et al., 2003; Boyd et al., 2004); however, in the SAGE experiment, with its higher mixing and lateral dilution, there were 3 injections (Harvey et al., 2010). Although these earlier experiments demonstrated that the injection of artificial SF$_6$ is a useful technique for following iron-fertilized patches, SF$_6$ can only be used for limited period (~2 weeks) due to the loss at the surface through air-sea gas exchange (Law et al., 2006; Tsumune et al., 2009; Martin et al., 2013). Furthermore, caution is required because artificially high levels of SF$_6$ injection may negatively impact the interpretation of low-level SF$_6$ signals dissolved in seawater via air-sea exchange to estimate tracer-based water mass ages for understanding physical circulation (Fine, 2011). These techniques have been widely used to estimate anthropogenic carbon invasion as well as to understand ocean circulation in various ocean environments, with SF$_6$ being an important time-dependent tracer that has a well-recorded atmospheric history. Thus, continuous sampling systems, measuring biogeochemical parameters such as Fv/Fm, $p$CO$_2$, and chlorophyll fluorescence, have also been used as an alternative means of following iron-fertilized patches (Gervais et al., 2002; Boyd et al., 2005; Tsuda et al., 2007; Harvey et al., 2010; Smetacek et al., 2012). The Fv/Fm ratio displays a particularly rapid increase (within 24 hours) in response to a first iron addition (Kolber et al., 1994; Behrenfeld et al., 1996; Smetacek et al., 2012), suggesting that it is an easy and convenient tracer for following a fertilized patch.

In addition, surface-drifting buoys equipped with Argos or GPS positioning systems have been successfully used to track the movement of fertilized patches along with biogeochemical tracers (Coale et al., 1998; Boyd and Law, 2001; Law et al., 2006; Martin et al., 2013). However, floats tend to drift out of the fertilized patches under strong wind forcing (Watson et al., 1991; Law et al., 1998; Stanton et al., 1998). NASA airborne oceanographic Lidar and ocean-color satellites have also been employed to assess the large-scale effects of iron addition on surface chlorophyll in fertilized patches, as compared to surrounding regions (Martin et al., 1994; Westberry et al., 2013).

## 2.4 Biogeochemical responses

Biogeochemical responses to artificial iron addition, in particular, Fv/Fm ratio, chlorophyll-a, PP, nutrients, $CO_2$ variables, and carbon export fluxes, are given in Tables 3–5 and Figs. 7–8. The results are important, as they have been used as a basis to determine whether the aOIF is effective. Here we address the biogeochemical response in each of the ocean basins to the aOIF experiments to date.

## 2.4.1 Equatorial Pacific

The IronEx-1/-2 experiments, which were conducted in similar initial conditions (refer to Section 2.2.1), presented quite different biogeochemical responses (Tables 3–4 and Fig. 7). In IronEx-1, there were small increases in the Fv/Fm ratio, chlorophyll-a concentration, PP, and $pCO_2$ concentrations, but no significant changes in nutrients (Martin et al., 1994). On the other hand, IronEx-2 found dramatic changes in biogeochemical responses, providing support for Martin's hypothesis (Coale et al., 1996). The extremely different results from the two experiments are likely to be associated with additional iron injections (IronEx-1: no extra addition; IronEx-2: 2 additional injections) and different experimental durations (IronEx-1: 10 days; IronEx-2: 17 days).

The Fv/Fm ratios provided further detail. In IronEx-1 and IronEx-2, Fv/Fm rapidly increased within ~24 hours of iron addition and reached a maximum of ~0.60 on the second day (Table 4) (Barber and Hiscock, 2006; Aiken et al., 2008). While the elevated IronEx-1 Fv/Fm ratios promptly disappeared, suggesting rapid iron loss due to the subduction of the fertilized patch and/or adsorption onto colloidal particles (perhaps indicative of insufficient iron supply), increased IronEx-2 Fv/Fm ratios were maintained for eight days through multiple iron additions, suggesting that additional iron enrichments are likely to be a determining factor in successfully artificially increasing PP through OIF (Kolber et al., 1994; Behrenfeld et al., 1996).

During IronEx-1, chlorophyll-a concentrations increased significantly (3-fold) reaching a maximum value of 0.65 mg $m^{-3}$ in the first four days following iron addition (Martin et al., 1994). In IronEx-2, surface chlorophyll-a increased <27-fold with a maximum of 4 mg $m^{-3}$ after day 7 (Table 4 and Fig. 7c) (Coale et al., 1996). To quantify the changes in carbon fixation following iron addition, the depth-integrated PP (from the surface to the critical depth, euphotic depth, or MLD) was estimated in the iron-fertilized patches. The depth-integrated PP values increased significantly compared to the initial values. The IronEx-2 ΔPP (where ΔPP = PP$_{post-fertilization (postf)}$ − PP$_{pre-fertilization (pref)}$) was the highest (~1800 mg C $m^{-2}$ $d^{-1}$) of all the aOIF experiments discussed here (Table 4 and Fig. 7e).

Changes in $pCO_2$ during IronEx-1 were less than expected (Δ$pCO_2$ = [$pCO_2$]$_{postf}$ − [$pCO_2$]$_{pref}$ = -13 μatm) (Martin et al., 1994). However, substantial drawdowns of $pCO_2$ (Δ$pCO_2$ = -73 μatm) and dissolved inorganic carbon (DIC) (ΔDIC = [DIC]$_{postf}$ − [DIC]$_{pref}$ = -27 μM) during IronEx-2 were derived through the increased PP (Table 3 and Fig. 7f) (Steinberg et al., 1998). As the bloom developed, a significant nitrate uptake (e.g., Δ$NO_3^-$ = [$NO_3^-$]$_{postf}$ − [$NO_3^-$]$_{pref}$ = -4.0 μM) was observed (Table 3 and Fig. 7b) and silicate concentrations also gradually decreased from 5.1 to 1.1 μM (i.e., limiting diatom growth)

over eight days (Coale et al., 1996; Boyd, 2002). The depletion of macro-nutrients in fertilized patches provides indirect evidence that phytoplankton growth in surface waters was driven by aOIF (Boyd and Law, 2001).

Although no phytoplankton community change was observed in IronEx-1, after iron addition in IronEx-2 a shift from a pico-phytoplankton dominated community to a micro-phytoplankton dominated community was observed, resulting in a diatom-dominated bloom (Behrenfeld et al., 1996; Coale et al., 1996; Cavender-Bares et al., 1999). Diatom biomass increased <70-fold over eight days early in the experiment, compared to a less than a 2-fold increase for the pico-phytoplankton (Landry et al., 2000). The biomass of meso-zooplankton (200–2,000 μm), such as copepods, grew simultaneously, substantially increasing the community grazing effect of larger animals on phytoplankton standing stocks from 7.8% d$^{-1}$ outside patch to 11.4% d$^{-1}$ in the patch (Coale et al., 1996). However, grazing did not prevent the development of a diatom bloom over eight days early in the IronEx-2 experiment (Table 4) (Coale et al., 1996; Rollwagen Bollens and Landry, 2000). The iron-induced diatom bloom began to decline after day ~8 of the experiment. The decline was probably associated with the combined effects of both the elevated grazing pressure and the onset of nutrient depletion (i.e., limitation in silicate and/or iron) (Cavender-Bares et al., 1999; Boyd, 2002).

To determine whether the biological pump (i.e., export production) is enhanced after iron addition, the export flux of particulate organic carbon (POC) was estimated using chemical tracer, the natural radiotracer thorium-234 ($^{234}$Th; half-life = 24.1 days) (Table 5) (Bidigare et al., 1999). The $^{234}$Th radionuclide has a strong affinity for particles, and the extent of $^{234}$Th removal in the water column is indicative of the export of POC associated with surface PP out of the ML (Buesseler, 1998). IronEx-2 was the first aOIF experiment in which the POC flux from the surface to 25 m was measured (Table 5). However, no $^{234}$Th measurements were made in the unfertilized patch for comparison, and no measurements in the deep ocean were undertaken to demonstrate deep carbon export (Bidigare et al., 1999).

**2.4.2 Southern Ocean**

As in the EP IronEx-1/-2 experiments, there were initial rapid increases in the Fv/Fm ratio within 24 hours of iron addition, indicating that phytoplankton growth was mainly limited by iron availability. Maximum values of the Fv/Fm ratio ranged from 0.5 (SOFeX-N and LOHAFEX) to 0.65 (SOIREE and SOFeX-S) (Table 4 and Fig. 7a). However, the time taken to reach the maximum Fv/Fm ratio was usually longer than ~10 days, i.e., much slower than in IronEx-1/-2 (~2 days) (Boyd and Abraham, 2001; Gervais et al., 2002; Coale et al., 2004; Smetacek et al., 2005; Peloquin et al., 2011a; Martin et al., 2013). The slower response time in the SO compared to the EP might be attributed to the colder temperatures (~5°C vs. ~24°C) and/or the deeper MLDs (~60 m vs. ~30 m) (Figs. 6c and d) (Boyd and Abraham, 2001; Boyd, 2002).

The aOIF experiments in the SO recorded >2-fold increases in chlorophyll-a concentrations compared to initial levels (<0.7 mg m$^{-3}$), and maximum values between 1.25 mg m$^{-3}$ (LOHAFEX) and ~3.8 mg m$^{-3}$ (SOFeX-S) were obtained after artificial iron additions (Table 4 and Fig.7c). Satellite observations were used to investigate the changing spatial and temporal distribution of chlorophyll-a concentration in response to iron fertilization in the fertilized patches compared to the surrounding waters; for example, SOFeX-N/-S found elevated chlorophyll-a concentrations in fertilized patches after iron addition through satellite images (Fig. 7d) (Boyd et al., 2000; Coale et al., 2004; Westberry et al., 2013).

Following artificial iron enrichment in the SO, ΔPP ranged from 360 (SAGE) to ~1356 mg C m$^{-2}$ d$^{-1}$ (SOFeX-N) (Table 4 and Fig. 7e). During SOIREE, EisenEx, and SOFeX-N/-S, PP increased continuously throughout the duration of the experiments (Boyd et al., 2000; Gall et al., 2001a; Gervais et al., 2002; Coale et al., 2004; Assmy et al., 2007). However, in EIFEX, SAGE, and LOHAFEX there was a significant increase in PP for ~10 (SAGE) to 20 (EIFEX) days in response to the iron addition, and decreasing trends after day ~12 (SAGE) – 25 (EIFEX). The decrease was due to various processes such as

export (e.g., EIFEX), lateral dilution with surrounding waters (e.g., SAGE), and high grazing pressure and bacterial respiration (e.g., LOHAFEX) (Boyd, 2002; Gervais et al., 2002; Buesseler et al., 2004; Coale et al., 2004; Peloquin et al., 2011a; Smetacek et al., 2012; Thiele et al., 2012; Assmy et al., 2013; Martin et al., 2013; Latasa et al., 2014).

Using both microscopes and high-performance liquid chromatography pigment analysis, changes in phytoplankton community affected by iron addition have also been investigated. Most SO aOIF experiments have resulted in blooms of diatoms (Boyd et al., 2007). During SOIREE and EisenEx, the dominant phytoplankton community shifted from pico- and nano-phytoplankton (e.g., pico-eukaryotes and prymnesiophytes) to micro-phytoplankton (i.e., diatoms) (Gall et al., 2001a; Gervais et al., 2002; Assmy et al., 2007). In SOFeX-S and EIFEX, diatoms were already the most abundant group prior to iron addition (Coale et al., 2004; Hoffmann et al., 2006; Assmy et al., 2013). The contribution of large diatoms became especially clear in EIFEX where ~97% of the phytoplankton bloom was attributed to this group (Smetacek et al., 2012; Assmy et al., 2013). However, no taxonomic shift toward diatom-dominated communities (<5% of total phytoplankton community) was observed during SAGE and LOHAFEX, which were conducted under silicate-limited conditions (Harvey et al., 2010; Peloquin et al., 2011a; Martin et al., 2013; Ebersbach et al., 2014). Although SOFeX-N was conducted under low silicate conditions (Fig. 6f), the diatom biomass increased remarkably making up ~44% of the total phytoplankton community (Coale et al., 2004). This result was partly influenced by the temporary relief of silicate limitation through lateral mixing of the iron-fertilized waters with surrounding waters, with relatively higher silicate concentrations (Coale et al., 2004).

Iron-mediated increases in PP resulted in a significant uptake in macronutrients and $p$CO$_2$ throughout the aOIF experiments in the SO (except for SAGE) (Table 3, Figs. 7b and f). $\Delta$NO$_3^-$ ranged from -3.5 μM (e.g., SOFeX-S) to -1.4 μM (e.g., SOFeX-N) and $\Delta p$CO$_2$ ranged from -38 μatm (e.g., SOIREE) to -7 μatm (e.g., LOHAFEX). Although both initially dominated by diatoms, SOFeX-S had a somewhat greater $\Delta$NO$_3^-$ (-3.5 μM) and $\Delta p$CO$_2$ (-36 μatm) than EIFEX ($\Delta$NO$_3^-$: -1.6 μM and $\Delta p$CO$_2$: -30 μatm) (Coale et al., 2004; Hoffmann et al., 2006; Smetacek et al., 2012; Assmy et al., 2013). However, the smaller silicate uptake ($\Delta$Si = [Si]$_{postf}$ − [Si]$_{pref}$) observed during SOFeX-S (-4 μM) compared to EIFEX (-11 μM) was associated with a decrease in silicification (i.e., changes in frustule thickness of the dominant diatom species, *Fragilariopsis* sp., Twining et al., 2004). During EIFEX, the ratio of heavily silicified diatoms (e.g., *Thalassiothrix antarctica*) to total diatom biomass increased from 0.24 (day 0) to 0.46 (day 37) leading to the higher Si uptake (Hoffmann et al., 2006; Assmy et al., 2013). Interestingly, the biogeochemical responses in SAGE were totally different from those seen in other experiments as increases in $\Delta$NO$_3^-$ (+3.9 μM), $\Delta p$CO$_2$ (+8 μatm), and $\Delta$DIC (+25 μM) were observed (Table 3, Figs. 7b and f). These contrasting results were thought to be the result of entrainment through vertical and horizontal physical mixing into the iron-fertilized patch of surrounding waters with higher nutrient and $p$CO$_2$ concentrations (Currie et al., 2011; Law et al., 2011).

SOIREE was the first aOIF experiment in the SO to estimate the downward carbon flux into deep waters (Fig. 3c). A comprehensive suite of methods was used: drifting traps, $^{234}$Th and the stable carbon isotope of particulate organic matter ($\delta^{13}$C$_{org}$) estimates derived from high-volume pump sampling, and a beam transmissometer (Nodder and Waite, 2001). However, no measurable change in carbon export was observed in response to iron-stimulated PP (Table 5 and Fig. 8b) (Charette and Buesseler, 2000; Nodder and Waite, 2001; Trull and Armand, 2001; Waite and Nodder, 2001). During EisenEx, an increased downward carbon flux estimated from $^{234}$Th deficiency was observed in the iron-fertilized patch as the experiment progressed. However, there were no clear differences between in- and outside-patch carbon fluxes (Buesseler et al., 2005). During SOFeX-S, significantly enhanced POC fluxes below the MLD similar to those observed in natural blooms, were estimated from $^{234}$Th measurements after iron enrichment (Buesseler et al., 2005). During SOFeX-N autonomous

profilers equipped with transmissometers recorded a downward carbon flux between day ~27 and ~45 after the first iron addition (Bishop et al., 2004; Coale et al., 2004). However, it was unclear whether surface-fixed carbon was well and truly delivered below the winter MLD. During SAGE and LOHAFEX, which were conducted under silicate limited conditions (Table 3, Figs. 4c and 6f), no significant enhancement of carbon export was detected (Table 5) (Peloquin et al., 2011a; Martin et al., 2013). This result was likely due to the dominance of pico-plankton and grazing that led to rapid recycling of organic matter in the ML. In contrast to the other aOIF experiments, EIFEX, which was conducted within the core of an eddy, showed clear evidence of carbon export well below 500 m, stimulated by artificial iron addition (Jacquet et al., 2008; Smetacek et al., 2012). During EIFEX, the initial export flux, estimated from $^{234}$Th in the upper 100 m of the fertilized patch, was ~340 mg C m$^{-2}$ d$^{-1}$ (Table 5 and Fig. 8a) (Smetacek et al., 2012). This value remained constant for about 24 days after iron addition. Between day 28 and 32 a massive increase in carbon export flux (maximum of ~1692 mg C m$^{-2}$ d$^{-1}$) was observed in the fertilized patch, while the initial value remained constant in the unfertilized patch (Table 5 and Fig. 8a). The profiling transmissometer with high-resolution coverage confirmed this result, showing an increase in exported POC below 200 m after day 24. At least half the iron-induced biomass sank (via the formation of aggregates of diatom species, in particular '*Chaetoceros dichaeta*') to a depth of 1,000 m, with a tenfold higher sinking rate (500 m d$^{-1}$), compared to the initial conditions (Smetacek et al., 2012). Significant changes in export production were not found in any of the other aOIF experiments and, therefore, the impact of artificial iron addition on this component of the biological pump needs to be resolved in future aOIF experiments (Boyd et al., 2004; Smetacek et al., 2012; Martin et al., 2013).

### 2.4.3 Subarctic North Pacific

The observed increase in the Fv/Fm ratio in response to aOIF in the subarctic NP suggests that the relief in iron limitation may have assisted phytoplankton growth (Table 4 and Fig. 7a). SEEDS-1/-2, which were conducted in the western basin, showed continuous increases in the Fv/Fm ratio, with a maximum value of ~0.4 approximately 10 days after the first iron addition (Tsuda et al., 2003, 2007). During SERIES, which was conducted in the eastern basin, the Fv/Fm ratio rapidly increased and reached a maximum value of 0.55 within 24 hours of the first iron addition (Boyd et al., 2005). However, the Fv/Fm ratio returned toward the initial value of <0.3 as the dissolved iron concentrations decreased to background levels (<0.2 nM) after about day 10 (Tsuda et al., 2003; Boyd et al., 2005; Tsuda et al., 2007).

Increases in chlorophyll-a concentrations were detected in the subarctic NP aOIF experiments in both basins after about the fifth day (Tsuda et al., 2003; Boyd et al., 2004; Suzuki et al., 2009). These increases were especially apparent in SEEDS-1, where they reached a maximum value of 21.8 mg m$^{-3}$ (27 times the initial value of 0.8 mg m$^{-3}$) (Table 4 and Fig. 7c). This augmentation was the largest among all the aOIF experiments (Tsuda et al., 2003). The dramatic surface chlorophyll-a increase observed during SEEDS-1 was partly attributed to the particular range of seawater temperature in the region, which was conducive to diatom growth (i.e., 8–13°C) as well as to the shallower MLD (~10 m), which provided a relatively longer surface water residence time for the additional iron (Figs. 6c and d) (Noiri et al., 2005; Takeda and Tsuda, 2005; Tsumune et al., 2005). During SERIES, chlorophyll-a concentrations increased substantially from the initial value of 0.35 to ~5 mg m$^{-3}$ over 17 days, the second highest concentration recorded in all aOIF experiments (Table 4 and Fig. 7c) (Boyd et al., 2004). However, on the 18th day there was a downturn in chlorophyll-a as silicate concentrations decreased to <2 μM (Boyd et al., 2005). Although SEEDS-2 was conducted under similar initial conditions to SEEDS-1 (refer to Section 2.2.3), there was a minimal increase in chlorophyll-a (i.e., maximum value of less than 3 mg m$^{-3}$) (Fig. 7c). This smaller increase was thought to be the result of strong copepod grazing (SEEDS-2 had almost five times more copepod biomass than SEEDS-1) (Table 4) (Tsuda et al., 2007). A similar range was seen in depth-integrated PP, which increased 7-fold or more after iron addition in the subarctic NP aOIF experiments (e.g., from 300–420 to 1,000–2,000 mg C m$^{-2}$ d$^{-1}$) (Table 4 and Fig. 7e).

Changes in the composition of phytoplankton groups were investigated in the subarctic NP aOIF experiments. In SEEDS-1 there was a shift from oceanic diatoms (e.g., *Pseudo-nitzschia turgidula*), with growth rates of 0.5–0.9 $d^{-1}$, to faster-growing neritic diatoms (e.g., *Chaetoceros debilis*, 1.8 $d^{-1}$) (Tsuda et al., 2005). The shift in the dominant phytoplankton species during SEEDS-1 was an important contributor to the recorded increase in phytoplankton biomass. During SERIES, the phytoplankton community changed from *Synechococcus* and haptophytes to diatoms, and the highest SERIES chlorophyll-a concentration (day 17) was associated with a peak in diatom abundance (Boyd et al., 2005). However, during SEEDS-2, no significant iron-induced diatom bloom was observed. Instead, pico-phytoplankton (e.g., *Synechococcus*) (67% of the total community) dominated throughout the duration of the experiment due to the heavy grazing pressure on diatoms (Table 4) (Tsuda et al., 2007; Sato et al., 2009).

In the subarctic NP experiments, significant changes in macro-nutrient uptake (i.e., $\Delta NO_3^-$ and $\Delta Si$), $\Delta DIC$, and $\Delta pCO_2$ in response to aOIF were observed (Table 3 and Figs. 7b and f). SEEDS-1, which exhibited the largest increases in chlorophyll-a concentrations, also had the largest $\Delta pCO_2$ (-130 µatm) and $\Delta DIC$ (-58 µM) (Table 3 and Fig. 7f). These changes led, in turn, to the largest $\Delta NO_3^-$ (-15.8 µM) (Fig. 7b) and $\Delta Si$ (-26.8 µM) (Table 3) (Tsuda et al., 2003). The second largest increase in the chlorophyll-a concentration was observed in SERIES, where drawdowns of $pCO_2$ (-85 µatm), DIC (-37 µM), nitrate (-8.5 µM), and silicate (-13.6 µM) were recorded. During SEEDS-2, the nitrate concentration decreased remarkably from 18.4 µM to 12.7 µM after day 5; however, there was no significant change in silicate concentrations, which would have been expected as a signal of an iron-induced diatom bloom (Tsuda et al., 2007; Suzuki et al., 2009).

Despite the formation of a massive iron-induced phytoplankton bloom during SEEDS-1, there was no large POC export flux during the observation period (Table 5) (Tsuda et al., 2003; Aono et al., 2005; Aramaki et al., 2009). During SERIES and SEEDS-2, which allowed comprehensive time-series measurements of the development and decline of the iron-stimulated bloom, POC fluxes estimated by the drifting traps in the fertilized patch displayed temporal variations (Boyd et al., 2004; Aramaki et al., 2009). The results suggested that subsequently, the drifting trap captured only a small part of the decrease in ML POC and POC flux losses were mainly governed by bacterial remineralization and mesozooplankton grazing (Boyd et al., 2004; Tsuda et al., 2007).

## 2.5 Summary of the significant results from aOIF experiments

aOIF experiments have generally led to changes in the size of the phytoplankton community from pico and nano-phytoplankton to micro-phytoplankton. This effect was particularly noticeable as diatoms became the dominant species during IronEx-2, SOIREE, EisenEx, SEEDS-1, SOFeX-S, EIFEX, and SERIES. Diatom-dominated blooms induced >4.5-fold increases in chlorophyll-a concentrations and accounted for >65% of the chlorophyll-a increase (Boyd et al., 2000; Gervais et al., 2002; Coale et al., 2004; Smetacek et al., 2012). The shift to a diatom-dominated community appears to be related to initial availability of silicate (i.e., initial silicate was >~5 µM in all the experiments listed above). However, as silicate concentrations decreased to <2 µM due to removal by phytoplankton, diatom blooms rapidly declined. SAGE and LOHAFEX had low initial levels of silicate (<2 µM). As a consequence, pico and nano-phytoplankton dominated their communities and diatom growth was limited by the lack of available silicate. However during SOFeX-N, initial silicate limitation (< ~3 µM) in the iron-fertilized waters was temporarily relieved through lateral mixing with the surrounding waters that had relatively higher silicate concentrations (Coale et al., 2004), which contributed to a taxonomic shift toward diatom-dominated communities (from 16% to 44% of total phytoplankton community). These results suggest that to develop large phytoplankton bloom, changeover to a diatom-dominated community after iron addition is needed. A necessary, but not sufficient condition, for such a change to occur is the availability of silicate. Silicate alone is not expected to be sufficient

because diatom-dominated blooms were not observed in all experiments with high initial silicate concentrations. IronEx-1 and SEEDS-2 had the high initial silicate levels (>~4 µM) considered conducive to the development of a diatom-dominated bloom, but blooms were suppressed due to high grazing pressure. Taken together, the aOIF results suggest that both mesozooplankton grazing rates and initial silicate concentrations play a role in limiting the stimulation of diatom-dominated blooms after artificial iron enrichment.

In experiments with smaller increases (<~3 times) in plankton biomass (IronEx-1, SEEDS-2, SAGE, and LOHAFEX) there was little change in the carbon export flux. Among previous aOIF experiments, the subarctic NP SEEDS-1 experiment, which was conducted under temperature conditions ideal for diatom growth (~8°C) and with shallow MLDs (~10 m), produced the greatest changes in surface phytoplankton biomass. However, influence of iron addition on the phytoplankton growth covers from surface to euphotic depth as added iron is mixed within the ML by physical processes (Coale et al., 1998). Although maximum surface chlorophyll-a concentration during SEEDS-1 (~22 mg m$^{-3}$) was much higher than EIFEX (~3.2 mg m$^{-3}$), the MLD-integrated chlorophyll-a concentrations were similar to ~250 mg m$^{-2}$ between two experiments. Therefore, to quantify the exact changes in phytoplankton biomass in response to iron addition, it would be eligible to consider the MLD-integrated PP for comparison. During IronEx-2, SOIREE, EisenEx, SEEDS-1, SOFeX-N/-S, EIFEX, and SERIES, a >2-fold increase in PP within the ML, with massive diatom-dominated blooms was observed. However, changes in the carbon export varied substantially and differed from experiment to experiment. In SEEDS-1 and SOIREE there was little increase in export flux. These two experiments were conducted over only about two weeks. The short duration of each experiment could have prevented the detection of downward carbon export. In SERIES, there was a distinct increase in the POC export flux within the ML (MLD = 30 m), but there was no increase in the carbon export flux below MLD because the produced POC was rapidly remineralized due to elevated heterotrophic bacteria respiration within the ML (Boyd et al., 2004). In SOFeX-S the export flux was enhanced at 100 m, below the MLD (45 m). However, the changes in export flux, after iron addition, were not dramatic compared to natural values (Buesseler et al., 2005). It is possible that the duration of SOFeX-S was also insufficient (~4 weeks) (Table 2). EIFEX was the only aOIF experiment that produced significant carbon export to deeper layers (down to 3,000 m). This high flux was due to aggregate formation with fast sinking rates (Smetacek et al., 2012). EIFEX observed an entire cycle (i.e., development – decline – fate) of the iron-induced phytoplankton bloom during the 39 days of the experiment, which strongly suggests that a sufficient experimental duration is a prerequisite for detecting fully formed diatom aggregates (i.e., carbon export). It should also be noted that the rates of bacterial remineralization and grazing pressure on the diatoms were in the same range inside the fertilized patch as outside, which might have assisted the delivery of iron-induced POC from the ML to deep layers (Smetacek et al., 2012). These results suggest that to detect significant carbon exported below the winter MLD following an increase in PP, at least three conditions are necessary: (1) a shift to a diatom-dominated community, (2) low bacterial respiration and grazing pressure rates within the ML, and (3) a sufficient experimental duration, enabling both immediate and delayed responses to iron addition to be observed.

## 3 Present: Unanswered aOIF questions - export flux, possible side effects, and international law

OIF has been proposed as a potential technique for rapidly and efficiently reducing atmospheric $CO_2$ levels at a relatively low cost (Buesseler and Boyd, 2003), but there is still much debate. Over the past 25 years, controlled aOIF experiments have shown that substantial increases in phytoplankton biomass can be stimulated in HNLC regions through iron addition, resulting in the drawdown of DIC and macronutrients (de Baar et al., 2005; Boyd et al., 2007; Smetacek et al., 2012; Martin et al., 2013). However, the impact on the net transfer of $CO_2$ from the atmosphere to below the winter MLD through the 'biological pump' (Fig. 3c) is not yet fully understood or quantified and appears to vary with environmental

conditions, export flux measurement techniques, and other unknown factors (Smetacek et al., 2012). There have also been a wide range of the estimates of atmospheric $CO_2$ drawdown resulting from large-scale and long-term aOIF based on model simulations (Joos et al., 1991; Peng and Broecker, 1991; Sarmiento and Orr, 1991; Kurz and Maier-Reimer, 1993; Gnanadesikan et al., 2003; Aumont and Bopp, 2006; Denman, 2008; Jin et al., 2008; Zahariev et al., 2008; Strong et al., 2009; Sarmiento et al., 2010). While it is generally agreed that OIF effectiveness needs to be determined through quantification of export fluxes, there has been no discussion about which export flux measurement techniques are the most effective. Meanwhile, concern has been expressed regarding possible environmental side effects in response to iron addition (Fuhrman and Capone, 1991). These side effects include the production of greenhouse gases (e.g., $N_2O$ and $CH_4$) (Lawrence, 2002; Jin and Gruber, 2003; Liss et al., 2005; Law, 2008; Oschlies et al., 2010), the development of hypoxia/anoxia in the water column (Sarmiento and Orr, 1991; Oschlies et al., 2010; Keller et al., 2014), and toxic algal blooms (e.g., *Pseudo-nitzschia*) (Silver et al., 2010; Trick et al., 2010). These unwanted side effects could lead to negative climate and ecosystem changes (Fuhrman and Capone, 1991; Sarmiento and Orr, 1991; Jin and Gruber, 2003; Schiermeier, 2003; Oschlies et al., 2010). Model studies suggested that the unintended ecological and biogeochemical consequences in response to large-scale aOIF might cancel out the effectiveness of aOIF. For example, aOIF enhanced $N_2O$ production may have offset (up to ~40%) the benefits of $CO_2$ sequestration in the EP (Sarmiento and Orr, 1991; Jin and Gruber, 2003; Oschlies et al., 2010; Hauck et al., 2016). Core unanswered questions remain concerning the different carbon export flux results from different measurement techniques (Nodder and Waite, 2001; Aono et al., 2005), the possible side effects that could directly influence the aOIF effectiveness, and the legal framework that is in place to regulate aOIF operations while simultaneously supporting further studies to increase our understanding of the potential risks and benefits of aOIF (Williamson et al., 2012). With the design of future aOIF experiments in mind, the following section discusses the these core questions: (1) which of the methods are optimal for tracking and quantifying carbon export flux, (2) which of the possible side effects have negative impacts on aOIF effectiveness, and (3) what are the international aOIF experimentation laws and can they be ignored?

### 3.1 Export flux measurement methods

A traditional, direct method for estimating POC export fluxes in the water column is a sediment trap that collects sinking particles (Suess, 1980). Sediment traps are generally deployed at specific depths for days to years to produce estimates of total dried mass, POC, particulate inorganic carbon (PIC), particulate organic nitrogen (PON), particulate biogenic silica, $\delta^{13}C_{org}$, and $^{234}$Th. A basic assumption for the use of a sediment trap is that it exclusively collects settling particles, resulting from the gravitational sinking of organic matter produced in surface waters. However, although they are designed to ensure the well-defined collection/conservation of sinking particles, they have accuracy issues due to: 1) interference of the hydrodynamic flow across the trap (i.e., strong advective flow), 2) inclusion/invasion (accounting for 14–90% of the total POC collected) of metazoan zooplankton (e.g., copepods, amphipods, and euphausiids) capable of vertical migration (Karl and Knauer, 1989; Buesseler, 1991; Buesseler et al., 2007), and 3) loss of trapped particles by bacterial decay and/or dissolution during trap deployment and storage periods (Gardner et al., 1983; Knauer et al.,1984; Kähler and Bauerfeind, 2001). The application of sediment traps for the determination of the carbon export flux is relatively more biased in the ML where ocean currents are generally faster and zooplankton are much more active than deep water. These issues suggest that sediment traps alone may not accurately determine carbon export fluxes within the ML.

Even when used at the same depth, traditional sediment traps, such as the surface-tethered drifting trap and bottom-moored trap, can greatly over- or under-estimate particulate $^{234}$Th fluxes compared to water-column based estimates (Buesseler, 1991). The water column-based total $^{234}$Th deficiency method (the sum of dissolved and particulate activities) is

less sensitive than sediment traps to the issues mentioned above, and provides better spatial and temporal resolution in flux estimates (Buesseler, 1998). For these reasons, traditional sediment trap POC flux estimates have often been calibrated using the total $^{234}$Th deficiency measured using a rosette bottle or high-volume pump samples (Coale and Bruland, 1985; Buesseler et al., 2006) as a reference. However, the water column-based $^{234}$Th method is sensitive to the characterization of the POC to $^{234}$Th ratio on sinking particles and/or the choice of $^{234}$Th flux models (Buesseler et al., 2006). Therefore, sampling to estimate the POC to $^{234}$Th ratio should be conducted below MLD to accurately detect downward carbon export flux into intermediate/deep waters.

Several aOIF experiments have used both sediment traps and $^{234}$Th deficiency to estimate the iron-induced POC export flux (Table 5). SOIREE reported distinct differences in POC fluxes estimated from drifting traps (185 mg m$^{-2}$ d$^{-1}$) at a 110 m over day 11–13 of the experiment and $^{234}$Th (~87 mg m$^{-2}$ d$^{-1}$) at 100 m (Charette and Buesseler, 2000; Nodder and Waite, 2001). While there was no measurable change in $^{234}$Th-based POC fluxes during the 13 days of the SOIREE experiment (Fig. 8b), the traps suggested a 27% increase over the course of the experiment (from 146 to 185 mg m$^{-2}$ d$^{-1}$) (Table 5). It was later discovered that the sediment trap-based sampling biases caused this supposed increase (Nodder et al., 2001; Nodder and Waite, 2001). Likewise, in SEEDS-1 $^{234}$Th-based POC fluxes at 50 m over day 9–13 were estimated to be 423 mg m$^{-2}$ d$^{-1}$, but the drifting trap only recorded 141 mg m$^{-2}$ d$^{-1}$ at 40 m over day 12–14, 3 times lower (Table 5) (Aono et al., 2005; Aramaki et al., 2009). This large discrepancy between the two methods might be caused by the under-sampling of POC into the drifting traps (Aono et al., 2005).

To resolve the potential biases in traditional sediment traps, a neutrally buoyant (and freely drifting) sediment trap (NBST) was developed (Valdes and Price, 2000; Valdes and Buesseler, 2006). Through preliminary experiments conducted in June and October 1997 at the Bermuda Atlantic Time-series Study site, Buesseler et al. (2000) showed that an NBST system could reduce the horizontal flow and invasion/inclusion of zooplankton into the trap samplers, and that NBST-based $^{234}$Th fluxes were comparable with water-column based estimates. LOHAFEX has been the only aOIF experiment so far that has measured particle export using PELAGRA (Particle Export measurement using a LAGRAngian trap) sediment traps based on the NBST system deployed at two depths of 200 m and 450 m (below the winter MLD) (Martin et al., 2013). However, the PELAGRA sediment traps did not detect aOIF-induced carbon export even though PP did increase within the ML. Water-column based $^{234}$Th measurements estimated the POC flux at a 100 m to be ~94 mg m$^{-2}$ d$^{-1}$, whereas the PELAGRA sediment traps estimated the flux at 200 m and 450 m to be <~12 mg m$^{-2}$ d$^{-1}$ (Table 5) (Martin et al., 2013). It should be noted that both sediment traps and water-column based $^{234}$Th measurements have a limited ability to fully scan the vertical profile of POC fluxes and, therefore, these methods should ideally be complemented with additional techniques that can measure particle stocks at high depth resolution throughout the water column.

To resolve the full column more effectively, LOHAFEX employed an underwater video profiler (UVP), which provided photographic evidence of sinking particles (particle size ≥100 μm) from the surface down to ~3,000 m, with ~0.2 m vertical resolution (Smetacek and Naqvi, 2010; Martin et al., 2013). Through an analysis of particle size distributions, the UVP also allowed particles to be classified into fecal pellets, aggregates, and live zooplankton. Total vertical particle volume profiles obtained from the UVP indicated a maximum concentration at 75 m (~0.3 mm$^3$ L$^{-1}$), with a gradual decrease to 150 m (~0.15 mm$^3$ L$^{-1}$). Interestingly, large particles (i.e., zooplankton) were copious between 75 m and 100 m, suggesting that there might be high grazing pressure. Heavy grazing might explain the large discrepancy between the 100 m (water-column based $^{234}$Th method) and 200 m / 450 m (PELAGRA sediment trap) POC flux estimates (i.e., rather than a sampling bias in sediment trap data) (Martin et al., 2013). To continuously monitor vertical changes in POC stocks following iron addition, EIFEX used a transmissometer, providing high vertical resolution (~24 data points per meter) and tracking of the iron-

induced stocks down to ~3,000 m, even though, unlike UVPs, transmissometers do not allow classification of particles (Smetacek et al., 2012). Improving on this method, SOFeX-N applied autonomous carbon explorers equipped with transmissometers, designed to float along with the currents. Three autonomous carbon explorers were deployed, two explored the 'iron fertilized in-patch' and one acted as a 'control' outside the patch. Carbon explorers could continuously monitor carbon flux in the field for up to 18 months beyond the initial deployment, which allowed SOFeX-N to observe 'episodic raining' in the iron-fertilized waters (Bishop et al., 2004), indicating a high carbon export flux after artificial iron addition. Furthermore, recent studies also reported that use of optical spike signals in particulate backscattering and fluorescence, measured from autonomous platforms such as gliders and floats, can provide high-resolution observations of POC flux (Briggs et al., 2011; Dall'Olmo and Mork, 2014).

The combination of multiple approaches is essential to the successful detection of POC produced in response to iron addition and its fate. NBST systems (e.g., the PELAGRA sediment trap) should be deployed at two depths (i.e., below both the *in situ* MLD and the winter MLD) to quantify the aOIF-induced POC flux. This technique is improved, when accompanied by calibration using water-column based $^{234}$Th. Particle profiling systems (e.g., a transmissometer and an UVP) can provide continuous quantitative and qualitative information about sinking particles, with high vertical resolution and full coverage of the water column (>3,000 m). They are therefore useful for indirectly identifying deep carbon transport. Autonomous carbon explorers are an excellent alternative, allowing for continuous observation of POC fluxes during and after an aOIF experiment.

## 3.2 Considering environmental side effects

The purpose of aOIF is to reduce the atmospheric $CO_2$ level by stimulating the sequestration of oceanic carbon through artificial iron additions in the HNLC regions, mitigating the global warming threat. Beyond the benefits of aOIF experimentation, scientists have debated the unintended secondary consequences of aOIF, such as production of climate-relevant gases and ocean ecosystem changes. Therefore, it is important to consider the possible negative consequences of aOIF to evaluate whether the aOIF experiments are effective (i.e., net profit: positives > negatives).

To investigate changes in climate-relevant gas emissions produced by biological activities and/or photochemical reactions before and after iron additions, the production of $CH_4$, $N_2O$, DMS, and halogenated volatile organic compounds (HVOCs) was measured during aOIF experiments (Liss et al., 2005), because their emission may lead to unintended consequences negating the desired effects of aOIF experiments on carbon sequestration. Among the climate-relevant gases, $CH_4$ has a ~20 times greater warming potential than $CO_2$ (IPCC, 1990). However, $CH_4$ has been considered to be relatively low risk because most of the $CH_4$ formed in the ocean is used as an energy source for microorganisms and is converted to $CO_2$ before reaching the sea surface (Smetacek and Naqvi, 2008; Williamson et al., 2012). The SO nOIF experiment conducted in 2011 year (i.e., Kerguelen Ocean and Plateau compared Study-2: KEOPS-2) (Table 1) showed that $CH_4$ concentrations were 4-fold higher in the naturally iron-fertilized patch than in the control area (Farías et al., 2015). During the SOFeX-N experiment, measurements of dissolved $CH_4$ indicated concentrations were slightly elevated, i.e., by less than 1% (1.74 ppmv in fertilized patch and 1.72 ppmv outside fertilized patch) (Wingenter et al., 2004). Simulated SO large-scale aOIF has suggested that a 20% enhancement of $CH_4$ emissions would offset only <1% (~4 Tg C $yr^{-1}$) of the resulting carbon sequestration (Oschlies et al., 2010). Hence, additional $CH_4$ production from aOIF experiments is not likely to be significant.

On the other hand, $N_2O$ has a relatively long lifetime in the atmosphere (~110 years) and has a global warming potential about 300 times greater than $CO_2$ (Forster et al., 2007). The ocean is already a significant source of atmospheric

N₂O (Nevison et al., 2003; Bange, 2006). Oceanic $N_2O$ is mainly produced by bacterial remineralization. Therefore, increases in $N_2O$ production after iron additions are expected and, in the long run, contribute to an increase rather than a decrease in the greenhouse effect (Fuhrman and Capone, 1991). During the SOIREE experiment, a significant increase (~4%) in mean $N_2O$ saturation in the pycnocline (65–80 m) of the fertilized patch (104.4 ± 2.4%), as compared to outside the fertilized patch (100.3 ± 1.7%), was associated with an increased phytoplankton biomass (Law and Ling, 2001). Measurements of $N_2O$ saturation during SERIES also showed increases of 8% at 30–50 m, which were coincident with the accumulation of ammonium and nitrite attributable to increases in bacterial remineralization following increased POC levels (Boyd et al., 2004; Law, 2008). SOIREE-based model estimates suggested that potential $N_2O$ production at timescales longer than six weeks would subsequently offset carbon reduction benefits resulting from the bacterial remineralization of additional carbon fixation by 6–12% (Law and Ling, 2001). This estimate is in line with the $N_2O$ offset of 6–18% suggested by a modeling study (Jin and Gruber, 2003) and the 5–9% suggested by a more recent modeling study investigating the effectiveness of long-term and large-scale SO aOIF (Oschlies et al., 2010). However, the SO nOIF experiment (i.e., KEOPS-2) suggested that nOIF acts as both a sink and a source for $N_2O$ (Farías et al., 2015). Excess $N_2O$ was not found after iron addition in EIFEX, where significant vertical export through the formation of rapidly sinking aggregates was found (Walter et al., 2005; Law, 2008). One explanation for the absence of $N_2O$ accumulation below the EIFEX patch might be the limited bacterial remineralization due to the rapid export of organic matter well below the 500 m to the seafloor (Law, 2008). Based on the results of previous studies, no consensus has yet been reached on the exact extent of additional $N_2O$ production after iron additions. However, because there is the potential for excessive $N_2O$ production that would not only impact the effectiveness of aOIF experiments but also positively contribute to global warming, further studies are required to reach a conclusion.

Unlike $N_2O$ emissions, which have the potential to offset the effectiveness of aOIF, DMS, a potential precursor of sulfate aerosols that cause cloud formation, may contribute to the homeostasis of the earth's climate by countering the warming due to increased $CO_2$ emissions (Charlson et al., 1987). DMS is produced by the enzymatic cleavage of planktonic dimethylsulfoniopropionate (DMSP). Microzooplankton grazing on nano-phytoplankton (e.g., haptophytes) is a key factor controlling oceanic DMS production (Dacey and Wakeham, 1986; Gall et al., 2001b; Park et al., 2014). The production of DMS in response to iron addition was measured during all aOIF experiments. In the EP and SO, DMS production increased, but in the subarctic NP, it remained constant or decreased (Boyd et al., 2007; Law, 2008). There were significant short-term increases in DMS production in IronEx-2 (from 2.5 to 4.2 nM), SOIREE (from 0.5 to 3.4 nM), EisenEx (from 1.9 to 3.1 nM), and SOFeX-N (7.7 nM in the fertilized patch and 1.6 nM outside the fertilized patch) (Turner et al., 1996; Turner et al., 2004; Wingenter et al., 2004; Liss et al., 2005; Wingenter et al., 2007). The maximum DMS production observed was a 6.8-fold increase after iron addition in SOIREE (Turner et al., 2004). During an early SOIREE experiment, the dominant phytoplankton species were haptophytes, and DMS production was increased by microzooplankton grazing on DMSP-rich haptophyte species (i.e., Prymnesiophyceae) (Gall et al., 2001b). Similarly, a 4.8-fold enhancement of DMS production was observed in SOFeX-N. Estimates derived by the extrapolation of SOFeX-N DMS production results suggested that fertilizing ~2% of the SO area over the course of a week would derive a 20% increase of the total SO DMS flux, which would lead to a 2°C decrease in air temperature over the SO (Wingenter et al., 2007). On the other hand, the SO nOIF experiment (KEOPS-1) conducted in 2005 year (Table 1) showed that DMS production was not markedly higher in the naturally fertilized area compared to the surrounding waters (Belviso et al., 2008). Twenty years simulated SO aOIF did not produce accumulation of DMS in surface waters (Bopp et al., 2008). Interestingly, there were no significant changes in DMS production after iron additions in the western subarctic NP SEEDS-1/-2 experiments, despite increases in PP (Takeda and Tsuda, 2005; Nagao et al., 2009). Furthermore, in the eastern subarctic NP, SERIES DMS production increased from 8.5–10.9 nM on day 1 to a maximum of 41.2 nM on day 10, but decreased to <0.03 nM by the end of the experiment due to an

increase in bacterial abundance (Table 4) (Levasseur et al., 2006). It is therefore difficult to predict the iron-induced DMS response, because OIF itself is not the only source of DMS. Based on the results of previous aOIF experiments, DMS production was sensitive in the EP and SO, but was less sensitive in the subarctic NP (Law, 2008). These results indicate that further process and modeling studies for each region are required to determine the production and degradation of DMS, both following iron fertilization and in the natural environment.

HVOCs, such as $CH_3Cl$, $CH_3Br$, and $CH_3I$, are well known for their ability to destroy ozone in the lower stratosphere and marine boundary layer (Solomon et al., 1994), and were also measured during past aOIF experiments (Wingenter et al., 2004; Liss et al., 2005). However, no consistent results have been reported for HVOCs production (Liss et al., 2005). In SOFeX-N, the impact of iron addition on HVOCs was complicated, with $CH_3Cl$ concentrations remaining unchanged, and $CH_3Br$ concentrations increasing by 14% (6.5 pptv in the fertilized patch and 5.7 pptv outside the fertilized patch), while $CH_3I$ concentrations decreased by 23% (4.9 pptv in fertilized patch and 6.4 pptv outside the fertilized patch) (Wingenter et al., 2004). In contrast, $CH_3I$ concentrations increased ~2-fold during EisenEx (Liss et al., 2005). Such a complicated response suggests that, as for DMS, further study is needed to fully understand natural cycling of HVOCs and their responses to aOIF.

Another important consideration is the extent to which the effectiveness of aOIF is cancelled out by its tendency to lead to ocean ecosystem changes such as a decrease in dissolved oxygen and an increase in domoic acid (DA) levels. The decomposition of iron addition-enhanced biomass may cause decreased oxygen concentrations in subsurface waters, but mid-water oxygen depletion has not been reported from aOIF experiments to date (Williamson et al., 2012). Early modeling studies suggest that anoxic conditions may develop after long-term, large-scale aOIF (Fuhrman and Capone, 1991; Sarmiento and Orr, 1991), whereas a recent study based on more sophisticated models showed sustained well-oxygenated conditions ($O_2 \approx 120$ μM) even under simulated aOIF south of 30°S on a 100 year timescale from 2010 to 2110 (Oschlies et al., 2010). Keller et al. (2014) found that simulated SO large-scale aOIF south of 40°S from the year 2020 to 2100 under a high $CO_2$ emissions scenario (Meinshausen et al., 2011) may develop suboxia ($O_2 <10$ μM) in the year 2125. Clearly, the circumstances under which a substantial decline in oxygen inventory can be caused by large-scale aOIF need further study.

The changes in phytoplankton community composition after iron addition discussed in Section 2.4 may also have unintended consequences. For example, such changes could lead to potentially toxic species dominating plankton assemblages (Silver et al., 2010; Trick et al., 2010). Some aOIF experiments (e.g., IronEx-2, SOIREE, EisenEx, SOFeX-N/-S, and SERIES) generated large blooms dominated by pennate diatoms belonging to the genus '*Pseudo-nitzschia*' (de Baar et al., 2005; Silver et al., 2010; Trick et al., 2010). Some '*Pseudo-nitzschia*' species have the capacity to produce the neurotoxin DA that can detrimentally affect marine ecosystems. However, no DA was found during EisenEx and SERIES, even though '*Pseudo-nitzschia*' were dominant (Gervais et al., 2002; Assmy et al., 2007; Marchetti et al., 2008). However, phytoplankton samples used to estimate DA production have sometimes been stored for a long time before the analysis, for example, 12 years in IronEx-2 and four years in SOFeX-S (Silver et al., 2010). Trick et al. (2010) argued that storage might have affected the DA content in the samples, which led to an underestimation in DA concentrations. Nevertheless, discernable changes in DA production were found in IronEx-2 and SOFeX-S experiments (Silver et al., 2010). It is likely that detection was possible because these samples were collected with net tows (20- to 30-μm mesh phytoplankton nets), which provided concentrated samples of larger phytoplankton including *Pseudo-nitzschia* (e.g., *Pseudo-nitzschia* abundance: $1.3 \times 10^6$ cells $L^{-1}$ in IronEx-2 and $7.5 \times 10^4$ cells $L^{-1}$ in SOFeX-S). During IronEx-2 and SOFeX-S, high cell abundances of *Pseudo-nitzschia* ($10^6$ and $10^5$ cells $l^{-1}$, respectively) combined with moderate DA cell quotas (0.05 and 1 pg DA $cell^{-1}$, respectively) produced toxin levels as high as 45 ng DA $l^{-1}$ and 220 ng DA $l^{-1}$ in the water, respectively, i.e., toxin levels high enough to damage marine

communities in coastal waters (Scholin et al., 2000; Schnetzer et al., 2007). Trick et al. (2010) suggested that large-scale OIF may induce DA accumulation with developing toxic *Pseudo-nitzschia* blooms. However, large uncertainties remain as Trick et al. (2010) simply extrapolated DA concentration based on bottle incubation experiments with HNLC surface waters to the DA production expected from large-scale OIF. As a result, it is necessary to clarify/quantify DA production in response to aOIF, with concentrated larger phytoplankton samples collected using net tows (20- to 30-μm mesh phytoplankton net). Here again, existing research indicates that the processes involved need to be better understood in the natural environment before the ramifications of aOIF can be fully understood.

Whether aOIF is a viable carbon removal strategy is still under debate (Boyd et al., 2007; Smetacek and Naqvi, 2008). The production of climate-relevant gases such as $N_2O$, DMS, and HVOCs, which is influenced by the remineralization of sinking particles that follows OIF-induced blooms, the decline in oxygen inventory, and the production of DA are particularly important to understand. These processes can directly and indirectly modify the effectiveness of carbon sequestration, with either positive or negative effects. Therefore, monitoring declines in oxygen content and production of climate-relevant gases and DA to evaluate the effectiveness of aOIF as a geoengineering approach is essential. The processes discussed here represent the current state of knowledge concerning aOIF side effects. The direct and indirect environmental consequences remain largely unresolved due to the inconsistent and highly uncertain outcomes of the experiments conducted so far, as well as our poor understanding of the processes involved under both nOIF and aOIF conditions (Chisholm et al., 2001; Johnson and Karl, 2002; Williamson et al., 2012). Therefore, considering the increasing evidence for the necessity to keep warming at or below 1.5°C (Rogelj et al., 2015), there continues to be a need to quantitatively determine the effectiveness of aOIF as a long-term means for reducing atmospheric $CO_2$ through the quantification of aOIF side effects.

### 3.3 Regulation of aOIF: International law of the sea as it applies to aOIF

To prevent pollution of the sea from human activities, the international Convention on the Prevention of Marine Pollution by Dumping of Wastes and Other Matter (London Convention, 1972) was amended in 1972. In 1996, contracting parties to the London Convention adopted the Protocol to the London Convention (London Protocol, 1996). This places legal restrictions on the dumping of wastes and other matter that may cause hazard, harm, and damage in the ocean and/or interfere with the marine environment. However, the London Convention & Protocol (LC/LP) did not establish specific laws to protect the ocean environment against the side effects of fertilization activities. In 2007, several commercial companies (e.g., GreenSea Venture [http://www.greenseaventure.com] and Climos [http://www.climos.com]) promoted large-scale (10,000 km$^2$) commercial aOIF as a climate mitigation strategy and as a means to gain carbon credits (Chisholm et al., 2001; Buesseler and Boyd, 2003; Freestone and Rayfuse, 2008). Meanwhile, assessments of the effectiveness of aOIF have been limited to small fertilized patches (25–300 km$^2$) (Fig. 6a) due to the time and expense of comparing fertilized and unfertilized areas (ACE CRC, 2008). As discussed earlier, these small-scale experiments have left many unanswered scientific questions regarding both the effectiveness and the potential impacts of aOIF (Lawrence, 2002; Buesseler and Boyd, 2003). In the same year, noting the potential risks and benefits, the LC/LP scientific group released a statement on large-scale ocean fertilization and recommended that ocean fertilization activities be evaluated carefully to ensure that such operations were not contrary to the aims of the LC/LP.

At the 2008 LC/LP meeting, the contracting parties adopted Resolution LC-LP.1 (2008) on the regulation of ocean fertilization. This resolution prohibited ocean fertilization activities until such time that specific guidance could be developed to justify legitimate scientific research. There was an exception for 'small-scale scientific research studies within coastal

waters' to permit the development of proposals that would lead to an assessment framework for scientific ocean fertilization research (Resolution LC-LP.1, 2008). In the meantime, there was a call to develop an assessment framework for ocean fertilization experiments to assess, accurately, scientific research proposals (Resolution LC-LP.1, 2008). In 2010, LC/LP parties developed Resolution LC-LP.2 (2010), adopting an "Assessment Framework for Scientific Research Involving Ocean Fertilization" to be used to assess, on a case-by-case basis, whether any proposed ocean fertilization activity constitutes legitimate scientific research falling within the aims and scope of Resolution LC-LP.1 (2008) (Fig. 9) (Resolution LC-LP.2, 2010). This framework demands preliminary scientific research prior to any aOIF experimentation. There must be a transparent/reasonable scientific rationale/purpose to the experiment and a risk analysis must be undertaken using parameters such as problem formulation, site selection, exposure and effect assessment, and risk characterization and management. Monitoring is also required as an integral component of all approved (i.e., legitimate) scientific aOIF research activity to assess ecological impacts and to review actual vs. intended geo-engineering benefits (ACE CRC, 2015). In October 2013, the LC/LP parties adopted amendments that categorize aOIF as marine geo-engineering, thereby prohibiting operational aOIF activities, but enabling aOIF scientific research that meets the permit conditions through the environmental assessment framework (Resolution LP.4 (8), 2013). This means that large-scale (i.e., >300 km$^2$ based on previous aOIF experiments; exact areal sizes are not determined in the LC/LP) and/or commercial aOIF (e.g., 'the 2012 Haida Gwaii Iron Dump' off the west coast of Canada) are currently banned by international regulations. Under LC/LP, commercial aOIF efforts cannot proceed because of the large uncertainties related to large-scale aOIF.

## 4 Future: Designing future aOIF experiments

Scientific aOIF research has focused on improving our understanding of the effectiveness, capacity, and risks of OIF as an atmospheric $CO_2$ removal strategy both in the future and the past (in particular glacial periods). Although the first aOIF experiments took place more than twenty years ago, the legal and economic aspects of such a strategy in terms of the international laws of the sea and carbon offset markets are not yet clear (ACE CRC, 2015). Nonetheless, previous small-scale aOIF experiments have demonstrated a considerable potential for easily and effectively reducing atmospheric $CO_2$ levels. Accordingly, physical/biogeochemical/ecological models and nOIF experiments (long-term) have been conducted in an effort to overcome some of the limitations of short-term aOIF experiments (e.g., spatial and temporal scales) and to predict the effectiveness of long-term and large-scale fertilization (Aumont and Bopp, 2006; Blain et al., 2007; Denman, 2008; Pollard et al., 2009; Sarmiento et al., 2010). For example, earlier global biogeochemical models have indicated that massive fertilization could draw down atmospheric $CO_2$ by as much as 107 μatm in 100 years (Joos et al., 1991; Peng and Broecker, 1991; Sarmiento and Orr, 1991; Kurz and Maier-Reimer, 1993). Recent global models, with more realistic ecosystem and biogeochemical cycles predict values closer to a 33 μatm drawdown in atmospheric $CO_2$ (Aumont and Bopp, 2006). These results suggest that the amount of carbon sequestration resulting from aOIF represents only a modest offset, i.e., a contribution of ~10% over the range of IPCC future emission scenarios (IPCC, 2000; Aumont and Bopp, 2006; Denman, 2008; Zahariev et al., 2008). The nOIF experiments have also produced much higher carbon sequestration rates than the small-scale aOIF experiments (Morris and Charette, 2013). Furthermore, the results from nOIF experiments do not support the potential negative impacts proposed for OIF experiments, even at larger scales (Belviso et al., 2008). However, these nOIF results do not guarantee that aOIF as a geoengineering approach is able to achieve the high effectiveness associated with carbon sequestration and enables a simple scaling-up as a prediction tool, because the nOIF experiments differ from the aOIF experiments in the mode of iron supply. In particular, nOIF is a continuous and slow process and its iron source is based on the upwelling of iron-rich subsurface waters to the surface layer, whereas aOIF is intended to be episodic, with massive short-term iron additions (Blain et al., 2007). In addition, in nOIF it is difficult to accurately identify iron sources

due to the complexity of the system, whereas in aOIF there is quantitative and qualitative information about iron additions and sources (Blain et al., 2008). Contrary to the results of aOIF experiments in the SO (e.g., SOIREE and SOFeX-N), no increase in DMS emissions was found in SO nOIF experiment (i.e., KEOPS-1) (Belviso et al., 2008), suggesting that it might be difficult to identify the potential long-term negative effects of aOIF through the study of naturally fertilized systems, at least in the SO. Therefore, it is important to continue undertaking small-scale studies to obtain a better understanding of natural processes in the SO as well as to assess the associated risks, and so lay the groundwork for evaluating the potential effectiveness and impacts of large-scale aOIF as a geoengineering solution to anthropogenic climate change. It is therefore of paramount importance that future aOIF experiments continue to focus on the effectiveness and capacity of aOIF as a means of reducing atmospheric $CO_2$, but they should also carefully consider the location (i.e., 'where'), timing (i.e., 'when'), and duration (i.e., 'how long'), as well as modes of iron addition (i.e., 'how'), tracing methods/parameters measurements/protocols (i.e., 'what'), and side effects on marine/ocean ecosystems (i.e., 'what concerns'). They should build on the results of previous aOIF experiments to develop our understanding of the magnitude and sources of uncertainties, and provide confidence in our ability to reproduce results.

Where: The first consideration for a successful aOIF experiment is the location. The dominance of diatoms in phytoplankton communities plays a major role in increasing the biological pump because diatom species can sink rapidly as aggregates or by forming resting spores to efficiently bypass the intense grazing pressure of mesozooplankton (e.g., copepods, salps, and krill) and export carbon out of the winter ML (Tréguer et al., 1995; Salter et al. 2007; Assmy et al., 2013; Rembauville et al., 2015; Rembauville et al., 2016). Previous aOIF experiments have shown that silicate concentration and mesozooplankton stocks (i.e., copepods) are the crucial factors controlling diatom blooms (Boyd et al., 2000; Gervais et al., 2002; Coale et al., 2004; Tsuda et al., 2007; Smetacek et al., 2012). Therefore, to obtain the greatest possible carbon export flux in response to iron addition, aOIF experiments should be designed in regions with high silicate concentrations and low grazing pressure. It will be important to conduct initial surveys to measure the degree of grazing pressure in HNLC region with high silicate concentrations such as in the subarctic NP (e.g., SEEDS-1 experiment) and the south of SO PF (e.g., SOFeX-S experiment) >~15 μM (Fig. 4c). In selecting sites for aOIF, it is also important to distinguish the iron-fertilized patch from the surrounding unfertilized waters to easily and efficiently observe iron-induced changes (Coale et al., 1996). Ocean eddies provide an excellent setting for aOIF experimentation because they tend to naturally isolate interior waters from the surrounding waters. Mesoscale eddies range from 25 to 250 km in diameter and maintain their characteristics for 10–100 days after formation (Morrow and Le Traon, 2012; Faghmous et al., 2015). Eddy centers tend to be subject to relatively slow current speeds, with low shear and high vertical coherence, providing ideal conditions for tracing the same water from the surface to below the winter MLD, while simultaneously minimizing lateral stirring and advection (Smetacek et al, 2012). Finding an appropriate eddy setting in a study area should be a high priority consideration when designing an aOIF experiment (Smetacek and Naqvi, 2008).

When: The second consideration for a successful aOIF experiment is timing, which includes when an experiment starts. PP in ocean environment is generally limited by nutrient availability and/or by light availability, often referred to as a single- or co-limitation. PP in the SO, a representative HNLC region, is subject to co-limitation by micro/macro-nutrients (i.e., iron and/or silicate) and light availability (Mitchell et al., 1991). To the south of the SO PF, phytoplankton blooms usually occur during early summer (i.e., from late December to early January) due to an increase in the nutrient flux from subsurface waters induced by winter mixing, along with the favorable light conditions provided by a shoaling of ML (Moore and Abbott, 2002). Prior to December, phytoplankton growth is mainly limited due to light availability (Mitchell et al., 1991; Veth et al., 1997; Abbott et al., 2000), while after January (i.e., during late summer and early autumn from February to March) it is mainly limited due to iron and silicate availability (Abbott et al., 2000; Mengelt et al., 2001; Nelson et al., 2001). In previous

SO aOIF experiments conducted between spring and early autumn, PP was mainly limited by iron and/or silicate availability rather than light availability (except when heavy clouds led to severe light limitation, only occurred for a few days during EisenEx) (Gervais et al., 2002; Bakker et al., 2005; Smetacek and Naqvi, 2008; Peloquin et al., 2011b). The grazing pressure of mesozooplankton on large diatoms was also a major limiting factor in diatom production (Schultes et al., 2006; Smetacek and Naqvi, 2010), and was generally higher during late summer and early autumn (February to March) (Hunt and Hosie, 2006; Rembauville et al., 2015). Considering the key factors (i.e., micro/macro nutrient availability, light availability, and grazing pressure) controlling PP in the SO, the most appropriate timing for the start of a SO aOIF experiment is likely to be the early summertime (i.e., late December to early January).

How long: The third consideration for a successful aOIF experiment is the duration. Although the periods that phytoplankton blooms have been maintained by aOIF have lasted from ~10 to ~40 days (Kolber et al., 1994; Martin et al., 1994; Coale et al., 1996; Boyd et al., 2000; Boyd et al., 2004; Coale et al., 2004; Tsuda et al., 2005; Smetacek et al., 2012), it has also been suggested that most aOIF experiments did not cover the full response times from onset to termination (Boyd et al., 2005). For example, SOIREE and SEEDS-1 had relatively short observation periods (13 days) and saw increasing trends in PP throughout the experiments (Fig. 10a), suggesting that the observation period should have been extended. Furthermore, after the end of SOIREE, ocean color satellite images showed continued high chlorophyll-a concentrations (>1 mg m$^{-3}$) in the iron fertilized patch, which was visible as a long ribbon shape that extended some 150 km for >40 days (~6 weeks) after the first iron addition (Fig. 10b) (Abraham et al., 2000; Westberry et al., 2013). This indicates that short experimental durations may not be sufficient for detecting the full influence of aOIF on PP and ecosystem (Figs. 8b and 10) (Boyd et al., 2000; Tsuda et al., 2003; de Baar et al., 2005). SOFeX-S also resulted in relatively low export production despite the high PP due to the experimental duration being insufficient to cover the termination of the phytoplankton bloom. However, SERIES, SEEDS-2, EIFEX, and LOHAFEX did fully monitor all phases of the phytoplankton bloom from onset to termination. EIFEX, the third-longest aOIF experiment, at 39 days, was the only one that observed iron-induced deep export production between day 28 and 32 (Table 5 and Fig. 8a) (Smetacek et al., 2012; Assmy et al., 2013). Furthermore, long-term observations covering the later stage of bloom development during nOIF experiments resulted in much higher C:Fe export efficiencies compared to the short-term aOIF (Blain et al., 2007; Pollard et al., 2009). Based on previous aOIF experiments, it would, therefore, be important to detect the full phase of a phytoplankton bloom to determine accurately the amount of iron-induced POC exported out of the winter ML. The observation period is, therefore, an importantly considering factor with regard to budget and effectiveness estimates. It is suggested that the experimental duration should be a minimum of ~40 days based on the SOIREE experiment, which produced the longest iron-induced bloom (>40 days). In addition, autonomous observation platforms are essential to monitor post-assessment of effectiveness, capacity, and risks of aOIF for at least 12 months after experiment termination.

How: The fourth consideration for a successful aOIF experiment lies in the strategy/approach of adding and maintaining dissolved iron within the ML to produce a phytoplankton bloom. First, the chemical form for iron addition should be acidified iron-sulfate, which is less expensive and more bioavailable than other iron compounds. The amount of iron-sulfate required is calculated according to the target concentration of the dissolved iron and volume (MLD × patch size). Based on bottle incubation experiments, target iron concentrations of ~2–4 nM are recommended to stimulate maximum phytoplankton growth due to the rapid losses of added iron by horizontal advection/diffusion and oxidation to poorly bioavailable iron(III) (Coale et al., 1996; Coale et al., 1998; Bowie et al., 2001). For patch size, a biogeochemical model study showed that a fertilized patch size of 156 km$^2$ maintained an iron concentration above 0.3 nM for 56 days, while a longer period of 194 days required a fertilized patch size of 160,000 km$^2$ (Xiu and Chai, 2010). As a consequence of expansion and dilution, previous aOIF experiments also produced similar results to this model study. The lateral dilution rate

($<0.25$ $d^{-1}$) during SAGE, which had the smallest fertilized patch size (36 $km^2$) of the SO experiments, was two times higher than the rates ($<0.11$ $d^{-1}$) in the SO experiments with a larger fertilized patch size (e.g., EIFEX fertilized with a patch size of 167 $km^2$ and SOFeX-S fertilized with a patch size of 225 $km^2$) (Coale et al., 2004; Harvey et al., 2010; Law et al., 2011; Smetacek et al., 2012). Therefore, it would be more appropriate to fertilize a large area (e.g., LOHAFEX had the largest aOIF experiment at 300 $km^2$), which would reduce the dilution effect with unfertilized waters (Xiu and Chai, 2010). Based on a ~2 nM iron concentration for a patch size of 300 $km^2$ and MLD of ~60 m, it would need ~2,000 kg of iron(II) to be applied in a fertilization experiment. Iron should be released into the wake of a ship, with the release track describing an expanding spiral (or square) in the eddy center, with a regular interval of ~1 km throughout the patch, because it is easier to locate a fertilized patch than a point release (Watson et al., 1991). In addition, it should be completed within ~24 hours because of the time-dependent phytoplankton response within the iron-fertilized patch. Previous aOIF experiments have shown that multiple iron additions ($\geq 2$ infusions) are needed to maintain the dissolved iron concentration required to derive maximum phytoplankton growth within the fertilized patch. For example, in SOIREE it was found that 4 additions of iron at intervals of about three days led to persistently high levels of both dissolved and particulate iron within the ML, with a rapid reduction at the end of the experiment, combined with an increase in the concentration of iron-binding ligands (Bowie et al., 2001). In both EIFEX and SOFeX-S, it was also found that multiple iron(II) infusions (in particular, 2 infusions with intervals of 13 days in EIFEX and 4 infusions with intervals of four days in SOFeX-S) allowed iron to persist in the ML longer than its expected oxidation kinetics. The relatively low oxidation rates were related to a combination of photochemical production, slow oxidation and, possibly, organic complexation (Croot et al., 2008). Blain et al. (2007) explained that the higher carbon sequestration effectiveness of nOIF experiments compared to aOIF experiments partly resulted from the slow and continuous iron addition that occurs in the natural environment. Large amounts of iron addition at one time can lead to a substantial loss of artificially added iron. Therefore, for an experimental duration of $>$~40 days, a minimum of 2 (or 3) iron infusions at intervals of ~10–15 days would be required to prevent the iron limitation on phytoplankton growth, based on the EIFEX experiment (Smetacek et al., 2012).

What: The fifth consideration for a successful aOIF experiment is the effective tracing of the fertilized patch, including the detection of carbon sequestration (Buesseler and Boyd, 2003). All previous aOIF experiments used physical tracers, in particular GPS and Argos equipped drifting buoys, to follow the iron fertilized patch. The release of GPS and Argos equipped drifting buoys at the center of the patch after the iron infusions would provide a visual map showing the tracked positions of the fertilized patch, because a drifting buoy is a natural and passive system moving along with the currents. However, it can be escaped from the fertilized patch due to the action of strong winds (Tsumune et al., 2005). An inert chemical tracer, such as $SF_6$, would also be an excellent option for following the fertilized patch after iron addition. Previous aOIF experiments have shown that the $SF_6$ measurements based on underway sampling systems can be used to accurately determine time-dependent vertical and lateral transport of iron-fertilized patches. However, tracing via $SF_6$ allows for only a limited period (~2 weeks) due to air-sea gas exchange (Law et al., 2006; Tsumune et al., 2009; Martin et al., 2013). Thus, many subsequent aOIF experiments have also used tracing methods based on the observation of biogeochemical parameters (such as the Fv/Fm ratio, chlorophyll fluorescence, and underway $p$CO$_2$) before and after iron addition (Martin et al., 1994; Coale et al., 1996; Boyd et al., 2000; Boyd et al., 2004; Coale et al., 2004; Tsuda et al., 2005; Smetacek et al., 2012). The Fv/Fm ratio can be easily and promptly used as an indicator to track the fertilized patch due to its rapid response to iron addition. Direct measurements of carbon export fluxes to determine the effectiveness of aOIF should be conducted by deploying an NBST at two depths: (1) just below the *in situ* MLD to detect increases in iron-induced POC in the surface layer along with the calibration of the water-column based $^{234}$Th method, and (2) at the winter MLD to detect iron-induced carbon export fluxes below winter MLD (Bidigare et al., 1999; Nodder et al., 2001; Boyd et al., 2004; Buesseler et al., 2004; Coale et al., 2004; Aono et al., 2005; Buesseler et al., 2005; Tsuda et al., 2007; Smetacek et al., 2012; Martin et al., 2013).

Sinking-particle profiling systems (e.g., transmissometers mounted on autonomous floats and gliders) that measure sinking particles could provide a record of the temporal and vertical evolution of iron-induced POC stocks through successive depth layers down to ~3,000 m for ~20 months after deployment, once calibrated using POC fluxes measured from sediment traps and/or the water-column based $^{234}$Th method (Bishop et al., 2004; Smetacek et al., 2012). Repeat casts with UVPs mounted on the rosette could also serve a similar purpose providing a photographic history of the water column (Martin et al., 2013). Future aOIF experiments would benefit from these technological advances, enabling a more efficient tracing of the carbon export flux and particle size and composition at higher vertical and temporal resolution than has been possible in the past. Hence, the application of an NBST system and water-column based $^{234}$Th method to direct flux estimates, combined with autonomous sinking-particle profilers of a transmissometer and an UVP, will enable the quantitative and qualitative evaluation of the effectiveness of aOIF and direct observation of iron-induced carbon export fluxes after artificial iron additions.

What concerns: The sixth consideration for a successful aOIF experiment is the monitoring of possible side effects. The LC/LP parties recently adopted Resolution LC-LP.2 (2010), which includes the "Assessment Framework for Scientific Research Involving Ocean Fertilization". This considers possible side effects on marine/ocean ecosystems after artificial iron additions, such as the production of climate-relevant gases and negative ecosystem changes, which are vital to assess when proposing an aOIF experiment. The emissions of climate-relevant gases, such as $N_2O$, DMS, and HVOCs, may directly contribute to warming or cooling effects, and oxygen decrease and toxic DA production may have a negative impact on marine/ocean ecosystems (Law, 2008; Silver et al., 2010; Trick et al., 2010; Williamson et al., 2012), resulting in significant offsets against the benefits of aOIF experiments. However, there is little quantitative and qualitative information regarding possible side effects following the previous aOIF experiments. Therefore, the future monitoring of these potential side effects is a prerequisite to evaluate accurately the effectiveness of an aOIF experiment in the future. The possible side effects after an aOIF experiment can be continuously monitored from optical sensors equipped autonomous moored profiler and/or autonomous benthic vehicle (e.g., crawler, which is capable to perform a long-term benthic oxygen measurements for ~12 months) (Dunne et al., 2002; Purser et al., 2013; Wenzhöfer et al., 2016).

In summary, to maximize the effectiveness of aOIF experiments in the future, we suggest a design that incorporates several conditions. (1) Experiments are conducted in the center of an eddy structure when grazing pressure is low and silicate levels are high (e.g., in the case of SO, at the south of PF during the early summer). (2) Shipboard observations are made during a minimum of ~40 days, with multiple iron injections (iron infusions of ~2,000 kg at least 2 (or 3) times, with an interval of ~10–15 days, to fertilize a patch of 300 km$^2$ to obtain a ~2 nM concentration). (3) The iron-fertilized patch is traced using both physical (e.g., a drifting buoy) and biogeochemical (e.g., $SF_6$ and the Fv/Fm ratio) tracers. (4) NBST system and water-column derived $^{234}$Th method are employed at two depths (i.e., just below the *in situ* MLD and at the winter MLD), with autonomous profilers equipped with an UVP and a transmissometer to estimate accurately the carbon export flux. (5) The side effects on marine/ocean ecosystems, including decline in oxygen contents and the production of climate-relevant gases (e.g., $N_2O$, DMS, and HVOCs) and toxic DA, are monitored using optical sensors equipped autonomous moored profiler and/or autonomous benthic vehicle.

## 5. Design of the Korean Iron Fertilization Experiment in the Southern Ocean (KIFES)

### 5.1 Background - Bransfield Basin

A science-oriented aOIF project, KIFES (Fig. 11), was launched in 2016 with research funding from the Korean

Ministry of Oceans and Fisheries. This project was largely managed by the Korea Polar Research Institute (KOPRI) with domestic collaborators (i.e., Incheon National University, Inha University, Pusan National University, Hanyang University, and Yeonsei University) and strengthened by international collaborators (i.e., Alfred-Wegener-Institut (AWI), Institute of Geological and Nuclear Sciences, Woods Hole Oceanographic Institution (WHOI), University of Otago, University of California at Irvine, McMaster University, University of South Florida, Royal Netherlands Institute for Sea Research, and Dalhousie University). KIFES had four main aims. (1) To conduct the first scientific aOIF experiment complying with the "Assessment Framework for Scientific Research Involving Ocean Fertilization" after the framework was accepted from the LC/LP in 2010. (2) To evaluate the effectiveness of scientific aOIF in terms of atmospheric carbon sequestration (i.e., to identify/quantify significant increases in iron-induced carbon export fluxes below the winter MLD) in the SO. (3) To determine the environmental conditions that would maximize the effectiveness of aOIF. (4) To quantitatively and qualitatively monitor short- and long-term possible side effects derived from previous aOIF experiments.

A location near the eastern Bransfield Basin was considered for the site of KIFES based on the following three criteria: (1) the possibility of diatom blooms, (2) the proximity to meso-scale eddies, and (3) the availability of historical oceanographic data. The development of a diatom bloom is the first prerequisite to maximize the effectiveness of an aOIF experiment. The paleoclimate team at KOPRI had found geological evidence of massive amounts of organic carbon buried in the sediments, especially in the diatomaceous ooze layer, in the eastern Bransfield Basin on the Antarctic Peninsula (Yoo et al., 2016). The well-preserved diatomaceous ooze layer (Bahk et al., 2003; Kang et al., 2003; Bak et al., 2015) indicates high accumulation rates of fast sinking diatoms, suggesting the existence of a strong 'biological pump (i.e., significant export production)' in this basin. In addition, this basin has a high silicate concentration of ~30 μM (Fig. 4c), which is a fundamental condition to produce a massive diatom bloom. Despite the favorable environmental conditions, the Fv/Fm ratio in/near the eastern Bransfield Basin (<~0.43) (Park et al., 2013) was lower than the maximum value of 0.65 measured during the aOIF experiments in the SO (e.g., SOIREE and SOFeX-S), suggesting an iron limitation on diatom growth. Therefore, we hypothesized that the input of bioavailable iron enabling an increase in productivity and export would lead to a massive enhancement of the diatom flux in this basin. Accordingly, we expected that an aOIF in the diatom-dominated region with high sinking rates near the eastern Bransfield Basin would be more effective for carbon export, as compared to the previous aOIF experiments conducted in the SO. This hypothesis, based on sedimentary evidence, was not considered in the site selection for previous experiments. A second important factor was the presence of stable eddies in/near the eastern Bransfield Basin (Kahru et al., 2007; Sangrà et al., 2011), providing coherent structures that made it possible to track effectively the iron-induced carbon export fluxes (Smetacek et al., 2012). For example, Thompson et al. (2009) showed that a large standing eddy (~40 km in diameter) was centered at ~62°S and 54°W and remained for ~30 days using historical drifters released during the period 1989–2005, and 40 drifters released in February 2007 as part of the Antarctic Drifter Experiment: Links to Isobaths and Ecosystems project. Satellite sea-level height images have indicated meso-scale eddies with long lifespans in/near the eastern Bransfield Basin (https://www.aviso.altimetry.fr/). Additionally, several historical oceanographic datasets are available for this basin (Grelowski et al., 1986; Helbling et al., 1995; Figueiras et al., 1999; Kang et al., 2001; Varela et al., 2002; Khim et al., 2005). The historical datasets provide general oceanographic characteristics about sites for an aOIF experiment as well as basic information helpful for designing the experiment, including a preliminary hydrographic survey. Unfortunately, KIFES has lost its source of funding. Nevertheless, optimism prevails that alternative funding will be found at a future date and the following section is intended, therefore, to provide a basic set of design guidelines, with the expectation that an opportunity to move forward with KIFES will occur in the near future.

**5.2 A plan for the future: KIFES**

The KIFES design entails a five-year project plan modeled on the 'EIFEX' program that found deep carbon by conducting an aOIF experiment in the center of an eddy structure. The KIFES project would include a preliminary environmental survey both outside and inside the center of an eddy structure formed in/near the eastern Bransfield Basin, a scientific aOIF experiment, and an assessment of the full KIFES project. In this section, we introduce the major goals, objectives, and main tasks of KIFES.

### 5.2.1 Year one plan

Goals: (1) Data collection with regard to oceanographic conditions in/near the eastern Bransfield Basin, including both eddy development and distribution. (2) Establishment of the study aims, hypothesis, and site for the KIFES experiment.

Objective: To understand the physical and biogeochemical oceanography of relevance to the eastern Bransfield Basin as an aOIF site through an analysis of earlier datasets and a review of published papers.

Main tasks: (1) Review databases of physical and biogeochemical parameters from previous surveys conducted in/near the eastern Bransfield Basin. (2) Review the eastern Bransfield Basin oceanographic conditions using data analysis and references. (3) Establish the study aims, hypothesis, and site in/near the eastern Bransfield Basin for an aOIF experiment, based on the results obtained from tasks (1) and (2). (4) Design an oceanographic cruise map for the first preliminary survey in/near the eastern Bransfield Basin. (5) Analyze eddy development and distribution using satellite data in/near the eastern Bransfield Basin. (6) Prepare scientific instruments for ocean physical and biogeochemical monitoring. (7) Establish an international collaborative aOIF network. (8) Submit KIFES field program proposal for the 'Initial Assessment' to determine that KIFES falls within the remit of ocean fertilization and should be evaluated in the LC/LP assessment framework based on the results from tasks (1)–(4).

### 5.2.2 Year two plan

Goal: First preliminary hydrographic survey to provide a foundational understanding of oceanographic conditions in/near the eastern Bransfield Basin.

Objectives: (1) To obtain information about oceanographic conditions from *in situ* measurements in/near the eastern Bransfield Basin. (2) To provide background information before the KIFES experiment.

Main tasks: (1) Using the ice breaker RV *ARAON*, undertake a field investigation in/near the eastern Bransfield Basin to determine physical and biogeochemical parameters associated with both carbon sequestration and aOIF side effects (e.g., decline in oxygen inventory and production of $N_2O$, DMS, HVOCs, and DA), based on the first-year results. (2) Prepare an 'Environmental Assessment' for the LC/LP assessment framework based on the first-year results and a preliminary hydrographic survey.

### 5.2.3 Year three plan

Goals: (1) Preliminary hydrographic survey outside/inside the center of an eddy structure prior to the KIFES experiment. (2) Approval of KIFES from LC/LP.

Objectives: (1) To compare oceanographic conditions inside and outside the center of an eddy structure formed in/near the eastern Bransfield Basin prior to the KIFES experiment. (2) To obtain a permission on the basis that the proposed KIFES is legitimate scientific research from the LC/LP.

Main tasks: (1) Using the ice breaker RV *ARAON*, detect an eddy formed in/near the eastern Bransfield Basin using observations from acoustic Doppler current profilers (ADCPs) and satellites. (2) Conduct intensive physical and biogeochemical field investigations both inside and outside the center of an eddy structure. (3) Assess the physical and biogeochemical properties outside vs. inside the center of an eddy structure prior to KIFES. (4) Establish a final design for KIFES. (5) Submit the research results for 'Environmental Assessment' stage of the LC/LP assessment framework and obtain approval for the KIFES experiment via the 'Decision Making' process from the LC/LP.

### 5.2.4 Year four plan

Goal: Conduction of the KIFES scientific aOIF experiment in the center of an eddy structure during the early summertime (Fig. 11).

Objective: To conduct a scientific aOIF experiment in the center of an eddy structure formed near/in the eastern Bransfield Basin.

Main tasks: (1) Using the ice breaker RV *ARAON*, detect an eddy formed in/near the eastern Bransfield Basin using observations from ADCPs and satellites, and investigate the initial environmental conditions for ~4 days before KIFES. (2) Execute the KIFES field campaign during a >~40-day period with the eddy structure. (3) At least 2 (or 3) iron additions at intervals of ~15 days, with each iron injection being ~2,000 kg following a spiral ship track, with a regular interval of ~1 km to create a patch size of 300 $km^2$ (target dissolved iron concentration of ~2 nM). (4) Trace the fertilized patch with deployments of GPS and Argos equipped drifting buoys, biogeochemical tracers ($SF_6$ and Fv/Fm ratio) employing underway-sampling systems, and gliders equipped with biogeochemical sensors. (5) Measure iron-induced carbon export fluxes for the regions both inside and outside the center of an eddy structure using NBST systems at two depths (i.e., just below the *in situ* MLD and at the winter MLD) along with the calibration of water-column based $^{234}$Th measurements and autonomous profilers equipped with a transmissometer and an UVP. (6) Monitor possible side effects, such as the decline in oxygen contents and the production of climate-relevant gases and toxic DA, using optical sensors equipped autonomous moored profiler and/or autonomous benthic vehicle. (7) Monitor continued responses after KIFES termination using satellite observations and autonomous profilers. (8) Assess the effectiveness of carbon sequestration and environmental (ocean and atmosphere) side effects for KIFES and prepare the KIFES assessment for the 'Results of Monitoring' stage of the LC/LP assessment framework.

### 5.2.5 Year five plan

Goal: Integrated assessment of the KIFES project.

Objective: To evaluate whether small-scale scientific aOIF experimentation can be an effective tool for detecting the effectiveness of artificially iron-induced export production and determining any negative impacts on climate change.

Main tasks: (1) Submit the KIFES assessment report. (2) Submit scientific results to international journals. (3) Collect feedback regarding the KIFES project from international scientific/oceanographic communities. (4) Produce a final aOIF experimental summary (including main tasks (1)–(3)). (5) Evaluate effectiveness and environmental side effects of large-scale SO aOIF via more realistic simulations under various scenarios with ocean biogeochemical models using the integrated results of KIFES. (6) Submit a final report of the KIFES assessment to the LC/LP.

**5.3 Final Remark**

None of the KIFES scientists have commercial interests (i.e., carbon credits) related to aOIF experiments. The interests of KIFES participants all lie in the detailed investigation of the biogeochemical effects of scientific aOIF in the SO and in aOIF as a possible geo-engineering method to mitigate the climate change effects we will face in the future. We envisage a future where the KIFES, or similar projects, can be resumed, enabling a more robust assessment of the potential of aOIF as a geo-engineering solution to help reduce atmospheric $CO_2$ concentrations. A continuation of the KIFES project would provide fundamental information and guidelines for future scientific aOIF experiments in HNLC regions, as well as improving our understanding of SO biogeochemistry. The risks and side effects of aOIF should be thoroughly investigated to calm international concerns. Finally, we emphasize that international cooperation is essential for a project as organizationally and scientifically complex as KIFES, and that we seek to improve our knowledge and provide a positive outlook for the Earth's future.

**6 Summary**

To test Martin's hypothesis, a total of 13 scientific aOIF experiments have been conducted in HNLC regions during the last 25 years. These aOIF experiments have resulted in increases of PP and drawdowns of macro-nutrients and DIC. In most experiments, the phytoplankton group has tended to shift from small-sized to large-sized plankton cells (mostly diatom-dominated). However, their effectiveness in enhancing export production has not been confirmed, except for EIFEX. Likewise, the possible environmental negative side effects in response to iron addition, such as decline in oxygen contents and the production of climate-relevant gases and toxic DA, could not be fully evaluated due to the widely differing outcomes, with large uncertainties depending on aOIF experimental conditions and settings. In particular, the monitoring of $N_2O$, DMS, and HVOCs is essential to determine the effectiveness of aOIF as a geoengineering approach, because these potential trace gas emissions can directly and indirectly modify the carbon reduction benefits resulting from aOIF. Furthermore, oxygen decline and toxic DA production may cause serious damage to marine/ocean ecosystems. Therefore, the validation and suitability of aOIF for the mitigation of rapidly increasing atmospheric $CO_2$ levels is a subject of vigorous debate. At present, large-scale and/or commercial aOIF is prohibited by international regulation, while small-scale aOIF experimentation for scientific purposes is permitted. To maximize the effectiveness of aOIF, future aOIF experiments should be conducted by carefully considering the major factors including the methods for iron addition, tracing methods, measurement parameters, location, timing, and experimental duration, under international aOIF regulations. Finally, we envisage a future where the KIFES project, or a similar alternative, becomes a reality so that we may determine whether aOIF is a promising geo-engineering solution.

*Acknowledgements*: We thank two reviewers for their valuable comments on the manuscript. We also thank Dr. Victor Smetacek (AWI) for his valuable comments and time on the development of KIFES project. Thanks to all the people who contributed to the scientific OIF experiments. This research was a part of the project titled 'Korean Iron Fertilization Experiment in the Southern Ocean (KOPRI, PM 16060)' funded by the Ministry of Oceans and Fisheries, Korea. This work was partly supported by the National Research Foundation of Korea (NRF) grant funded by the Korea government (MSIP) (No. 2015R1C1A1A01052051), the K-AOOS (KOPRI, PM17040) project funded by the MOF, and the KOPRI projects (PE17030 and PE17140). A. M. Macdonald was supported by NOAA grant #NA11OAR4310063 and internal WHOI funding. The last two authors (Il-Nam Kim and Kitae Kim) act as co-corresponding authors for this work.

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

**Table 1.** Summary of ocean iron fertilization (OIF) experiments; time, location, research vessel, added iron(II) (values in brackets correspond to the number of days from the first iron addition, e.g., the first iron addition becomes (0)), initial iron concentrations, target iron concentrations (iron concentrations after iron addition), tracer, initial patch size, experiment duration, and regional characteristics (HNLC: high-nutrient and low-chlorophyll).

| | Experiment | Time | Location | Research vessel | Added iron(II) (kg) (day) | Initial iron (nM) | Target iron (nM) | Tracer | Patch size (km$^2$) | Duration (days) | Regional characteristics |
|---|---|---|---|---|---|---|---|---|---|---|---|
| 1 | IronEx-1 | Oct 1993 | Equatorial Pacific 5° S, 90° W | RV *Columbus Iselin* | ①450 (0) | 0.06 | 3.60 | SF$_6$ | 64 | 10 | HNLC |
| 2 | IronEx-2 | May 1995 | Equatorial Pacific 3.5° S, 104° W | RV *Melville* | ①225 (0) ②112 (3) ③112 (7) | 0.02 | 2.00 1.00 1.00 | SF$_6$ | 72 | 17 | HNLC |
| 3 | SOIREE | Feb 1999 | Southern Ocean-Australasian-Pacific sector 61° S, 140° E | RV *Astrolab* | ①768 (0) ②312 (3) ③312 (5) ④353 (7) | 0.08 | 3.80 2.60 2.60 2.50 | SF$_6$ | 50 | 13 | HNLC |
| 4 | EisenEx | Nov 2000 | Southern Ocean-Atlantic sector 48° S, 21° E | RV *Polarstern* | ①780 (0) ②780 (7) ③780 (16) | 0.06 | 2.00 | SF$_6$ | 50 | 23 | HNLC |
| 5 | SOFeX-N | Jan–Feb 2002 | Southern Ocean-Pacific sector 56.23° S, 172° W | RV *Revelle* RV *Melville* | ①631 (0) ②631 (4) ③450 (29) | | 1.20 1.20 1.50 | SF$_6$ | 225 | 40 | HNLCLSi[a] |
| 6 | SOFeX-S | Jan–Feb 2002 | Southern Ocean-Pacific sector 66.45° S, 171.8° W | RV *Revelle* RV *Melville* RV *Polar star* | ①315 (0) ②315 (5) ③315 (8) ④315 (12) | | 0.70 0.70 0.70 0.70 | SF$_6$ | 225 | 28 | HNLC |
| 7 | EIFEX | Feb–Mar 2004 | Southern Ocean-Atlantic sector 50° S, 2° E | RV *Polarstern* | ①1410 (0) ②1410 (13) | 0.08–0.20 | 1.50 0.34 | | 167 | 39 | HNLC |

| | Experiment | Time | Location | Research vessel | Added iron(II) (kg) (day) | Initial iron (nM) | Target iron (nM) | Tracer | Patch size (km$^2$) | Duration (days) | Regional characteristics |
|---|---|---|---|---|---|---|---|---|---|---|---|
| 8 | SAGE | Mar–Apr 2004 | Southern Ocean-Southeast of New Zealand 46.5° S 172.5° E | RV *Tangaroa* | ①265 (0) ②265 (6) ③265 (9) ④265 (12) | 0.09 | 3.03 1.59 0.55 1.01 | SF$_6$ | 36 | 15 | HNLCLSi[a] |
| 9 | LOHAFEX | Jan–Mar 2009 | Southern Ocean-Atlantic sector 48° S, 15° W | RV *Polarstern* | ①2000 (0) ②2000 (18) | | 2.00 | SF$_6$ | 300 | 40 | HNLCLSi[a] |
| 10 | SEEDS-1 | Jul–Aug 2001 | Subarctic North Pacific-Western basin 48.5° N, 165° E | RV *Kaiyo-Maru* | ①350 (0) | 0.05 | 2.90 | SF$_6$ | 80 | 13 | HNLC |
| 11 | SERIES | Jul-Aug 2002 | Subarctic North Pacific-Eastern basin 50.14° N, 144.75° W | RV *John P. Tully* RV *El Puma* RV *Kaiyo-Maru* | ①245 (0) ②245 (6) | <0.10 | >1.00 0.60 | SF$_6$ | 77 | 25 | HNLC |
| 12 | SEEDS-2 | Jul–Aug 2004 | Subarctic North Pacific-Western basin 48° N, 166° E | RV *Hakuho-Maru* RV *Kilo-Moana* | ①332 (0) ②159 (6) | 0.17 | 1.38 | SF$_6$ | 64 | 26 | HNLC |
| 13 | FeeP | Apr–May 2004 | Subtropical North Atlantic-North-east Atlantic 27.5° N 22.5° W | RV *Charles Darwin* RV *Poseidon* | ①1840 (0) | 0.20–0.40 | 3.00 | SF$_6$ | 25 | 21 | LNLC[b] |
| I | Polar Front[c] | Oct–Nov 1992 | Southern Ocean-Atlantic Sector 48° S, 6° W | RV *Polarstern* | | | | | | | HNLC |
| II | PlumeEx[c] | Nov 1993 | Equatorial Pacific 2° S, 89° W | RV *Columbus Iselin* | | 0.05 | 0.2 | | | | HNLC |

To be continued

| | Experiment | Time | Location | Research vessel | Added iron(II) (kg) (day) | Initial iron (nM) | Target iron (nM) | Tracer | Patch size (km$^2$) | Duration (days) | Regional characteristics |
|---|---|---|---|---|---|---|---|---|---|---|---|
| III | CROZEX[c] | Nov 2004–Jan 2005 | Southern Ocean-Crozet Plateau 44° S, 50° E | RV *Discovery* | | | 0.55 | | | | HNLC |
| IV | KEOPS-1[c] | Jan–Feb 2005 | Southern Ocean-Kerguelen Plateau 50° S, 73° E | RV *Marion Dufresne* | | 0.09 | 0.35 | | | | HNLC |
| V | DynaLiFe[c] | Jan–Feb 2009 | Southern Ocean-Pacific sector 74° S, 105° W | RV *Nathaniel B. Palmer* | | 0.2 | 0.4 | | | | HNLC |
| VI | KEOPS-2[c] | Oct–Nov 2011 | Southern Ocean-Kerguelen Plateau 50.63° S, 72.08° E | RV *Marion Dufresne* | | | | | | | HNLC |

[a]High-Nutrient Low-Chlorophyll and Low Silicate (HNLCLSi) region. [b]Low-Nutrient Low-Chlorophyll (LNLC) region. [c]Natural OIF experiments (PlumeEx: Natural iron enrichment experiment near Galapagos Islands; CROZEX: CROZet natural iron bloom and EXport experiment; KEOPS-1/2: KErguelen Ocean and Plateau compared Study-1/2; DynaLiFe: Dynamic Light on Iron Limitation program).

Sources are Martin et al. (1994); de Baar et al. (1995); Coale et al. (1996); Coale et al. (1998); Gordon et al. (1998); Boyd et al. (2000); Boyd and Law (2001); Gervais et al. (2002); Tsuda et al. (2003); Boyd et al. (2004); Coale et al. (2004); Bakker et al. (2005); Boyd et al. (2005); de Baar et al. (2005); Nishioka et al. (2005); Hoffmann et al. (2006); Law et al. (2006); Blain et al. (2007); Boyd et al. (2007); Rees et al. (2007); Tsuda et al. (2007); Pollard et al. (2009); Strong et al. (2009); Harvey et al. (2010); Gerringa et al. (2012); Smetacek et al. (2012); Martin et al. (2013); and Blain et al. (2015).

**Table 2.** Summary of artificial ocean iron fertilization (aOIF) experiments; objectives, significant results, and limitations.

| | Experiment | Objectives | Significant results | Limitations |
|---|---|---|---|---|
| 1 | IronEx-1 | • To test the hypothesis that artificial iron addition will increase phytoplankton productivity by relieving the iron limitation of phytoplankton in high-nutrient low chlorophyll regions | • Small increases in the $pCO_2$ concentrations, Fv/Fm ratio, chlorophyll-a concentration, and primary production (PP)<br>• Insignificant changes in nutrients | • Single iron addition<br>• Insufficient experimental periods to observe the full phases of biogeochemical responses from the onset to termination after iron additions<br>• Micro/macro-nutrient limitations |
| 2 | IronEx-2 | • To test four hypotheses that were advanced to explain the weak biogeochemical response observed during IronEx-1 | • Dramatic changes in biogeochemical responses; close to support for Martin's hypothesis<br>• Taxonomic shift toward diatom-dominated phytoplankton communities | • No export flux measurements in the deep ocean<br>• Insufficient experimental duration |
| 3 | SOIREE | • To test the iron hypothesis in the Southern Ocean | • Diatom-dominated bloom<br>• No measurable change in carbon export | • Insufficient experimental duration |
| 4 | EisenEx | • To test the hypothesis that atmospheric dust inputs might have led to a dramatic increase in ocean productivity during the Last Glacial Maximum due to the relief of iron-limited conditions for phytoplankton growth | • Diatom-dominated bloom<br>• No clear differences in carbon flux between in-patch and outside-patch | • Light limitation by storms<br>• Insufficient experimental duration |
| 5 | SOFeX-N | • To address the potential for iron and silicate interactions to regulate the diatom bloom | • Remarkable increase in diatom biomass<br>• Observation of large export flux event with transmissometers | • Entrainment of dissolved silicate into the fertilized patch by physical mixing<br>• No direct measurement of export fluxes with sediment traps |
| 6 | SOFeX-S | • To address the potential for iron and silicate interactions to regulate the diatom bloom | • Significantly enhanced export fluxes out of the mixed layer (ML), but similar to those for natural blooms | • Insufficient experimental duration |
| 7 | EIFEX | • To confirm that aOIF experiments can increase export production | • Observation of all the phases of the phytoplankton bloom from onset to termination<br>• Significant carbon export to deeper layers (down to 3,000 m) due to the formation of aggregates with rapid sinking rates | |

| | Experiment | Objective | Significant results | Limitations |
|---|---|---|---|---|
| 8 | SAGE | • To determine the response of phytoplankton dynamics to iron addition in high-nutrient low-chlorophyll and low silicate (HNLCLSi) regions<br>• To test the assumption that the response of phytoplankton blooms to artificial iron addition can be detected by the enhanced air-sea exchanges of climate-relevant gases | • No shift to a diatom-dominated community<br>• No detection of fertilization-induced export | • High dilution rate by small patch size |
| 9 | LOHAFEX | • To trace the fate of iron-stimulated phytoplankton blooms and deep carbon export in HNLCLSi regions | • Observation of all the phases of the phytoplankton bloom from onset to termination<br>• No shift to a diatom-dominated community<br>• No detection of fertilization-induced export<br>• High grazing pressure | |
| 10 | SEEDS-1 | • To investigate the relationship between phytoplankton biomass/community and dust deposition in the subarctic North Pacific (NP)<br>• To investigate changes in phytoplankton composition and vertical carbon flux | • A shift from oceanic diatoms to fast-growing neritic ones<br>• The largest changes in biogeochemical parameters of all aOIF experiments<br>• No detection of large carbon export flux | • Single iron addition<br>• Insufficient experimental duration |
| 11 | SERIES | • To compare the response of phytoplankton in eastern subarctic with that in the western subarctic ecosystem<br>• To investigate the most significant factor that controls the beginning to the ending of the phytoplankton bloom induced by iron addition | • Observation of all phases of the phytoplankton bloom from onset to termination<br>• No significant increases in export fluxes below the ML<br>• High bacterial remineralization and mesozooplankton grazing pressure | |
| 12 | SEEDS-2 | • To investigate the most significant factor that controls the beginning to the ending of the phytoplankton bloom induced by iron addition | • Observation of all phases of the phytoplankton bloom from onset to termination<br>• No shift to a diatom-dominated community<br>• No significant increases in export fluxes<br>• Extensive copepod grazing | |
| 13 | FeeP | • To investigate the impact of iron and phosphate co-limitation on PP | • Increases in pico-phytoplankton abundances | |

Sources are Martin et al. (1994); Coale et al. (1996); Coale et al. (1998); Bidigare et al. (1999); Boyd et al. (2000); Charette and Buesseler (2000); Gervais et al. (2002); Tsuda et al. (2003); Boyd et al. (2004); Coale et al. (2004); Bakker et al. (2005); Boyd et al. (2005); de Baar et al. (2005); Hiscock and Millero (2005); Nishioka et al. (2005); Tsuda et al. (2005); Tsumune et al. (2005); Boyd et al. (2007); Rees et al. (2007); Tsuda et al. (2007); Harvey et al. (2010); Law et al. (2011); Smetacek et al. (2012); and Martin et al. (2013).

**Table 3.** Initial conditions and changes (Δ values) in chemical parameters during the artificial ocean iron fertilization (aOIF) experiments

| | Experiment | Initial NO$_3^-$ (µM) | ΔNO$_3^-$ (µM) | Initial PO$_4^{3-}$ (µM) | ΔPO$_4^{3-}$ (µM) | Initial Si (µM) | ΔSi (µM) | Initial $p$CO$_2$ (µatm) | Δ$p$CO$_2$ (µatm) | Initial DIC (µM) | ΔDIC (µM) |
|---|---|---|---|---|---|---|---|---|---|---|---|
| 1 | IronEx-1 | 10.8 | -0.70 | 0.92 | -0.02 | 3.90 | -0.02 | 471 | -13.0 | 2044[a] | -6.00[a] |
| 2 | IronEx-2 | 10.4 | -4.00 | 0.80 | -0.25 | 5.10 | -4.00 | 538 | -73.0 | 2051[a] | -27.0[a] |
| 3 | SOIREE | 25.0 | -2.90 | 1.50 | -0.24 | 10.0 | -2.90 | 349 | -(38.0–32.0) | 2137 | -(18.0–15.0) |
| 4 | EisenEx | 23.5 | -1.60 | 1.60 | -0.16 | 14.2 | ~0 | ~360 | -(20.0–18.0) | 2131 | -(15.0–12.0) |
| 5 | SOFeX-N | 21.9 | -1.40 | 1.40 | -0.09 | 2.50 | -1.10 | 367 | -26.0 | 2109 | -14.0 |
| 6 | SOFeX-S | 26.3 | -3.50 | 1.87 | -0.21 | 62.8 | -4.00 | 365 | -36.0 | 2176 | -21.0 |
| 7 | EIFEX | 25.0 | -1.60 | 1.80 | ~-0.30[b] | 19.0 | -11.0 | 360 | -30.0 | 2135 | -13.5 |
| 8 | SAGE | 7.90–10.5 | 1.30–3.90 | 0.62–0.85 | | 0.83–0.97 | | 330 | 8.00 | 2057 | 25.0 |
| 9 | LOHAFEX | 20.0 | -2.50 | 1.20–1.30 | ~-0.15[c] | 0.60–1.60 | | ~358[d] | -(15.0–7.00) | | |
| 10 | SEEDS-1 | 18.5 | -15.8 | | | 31.8 | -26.8 | 390 | -130 | | -58.0 |
| 11 | SERIES | 10.0–12.0 | -(8.50–6.50) | 1.00 | -0.50 | 14.0–16.0 | -(13.6–11.6) | 350 | -85.0 | 2030 | -37.0 |
| 12 | SEEDS-2 | 18.4 | -5.70 | | | 36.1 | | 370 | -6.00 | | |
| 13 | FeeP | <0.01 | | ~0.01 | | | | | | | ~-1.00 |

[a]Dissolved inorganic carbon (DIC) values in IronEx-1/-2 indicate normalized DIC (normalized DIC = DIC × 35/Salinity). [b]ΔPO$_4^{3-}$ in EIFEX was digitized from Figure 3 of Smetacek et al. (2012). [c]ΔPO$_4^{3-}$ in LOHAFEX was digitized from Figure 5.1 of Smetacek and Naqvi (2010). [d]Δ$p$CO$_2$ in LOHAFEX was digitized from Figure 6.1 of Smetacek and Naqvi (2010). Sources are Martin et al. (1994); Steinberg et al. (1998); Boyd et al. (2000); Bakker et al. (2001); Frew et al. (2001); Bakker et al. (2005); Boyd et al. (2005); Bozec et al. (2005); Hiscock and Millero (2005); Smetacek et al. (2005); Takeda and Tsuda (2005); Tsuda et al. (2005); Marchetti et al. (2006); Wong et al. (2006); Boyd et al. (2007); Tsuda et al. (2007); Tsumune et al. (2009); Harvey et al. (2010); Smetacek and Naqvi (2010); Berg et al. (2011); Currie et al. (2011); Law et al. (2011); Smetacek et al. (2012); Assmy et al. (2013); Ebersbach et al. (2014); and Latasa et al. (2014).

**Table 4.** Initial values of biological parameters and the values after fertilization. Note that maximum values were attained after fertilization.

| | Experiment | Initial Fv/Fm | After Fv/Fm | Initial Chlorophyll-a (mg m$^{-3}$) | After Chlorophyll-a (mg m$^{-3}$) | Initial PP (mg C m$^{-2}$ d$^{-1}$) | After PP (mg C m$^{-2}$ d$^{-1}$) | Initial Mesozooplankton biomass (mg C m$^{-3}$) | After Mesozooplankton biomass (mg C m$^{-3}$) | Initial Heterotrophic bacteria abundance (× 10$^5$ cells ml$^{-1}$) | After Heterotrophic bacteria abundance (× 10$^5$ cells ml$^{-1}$) |
|---|---|---|---|---|---|---|---|---|---|---|---|
| 1 | IronEx-1 | ~0.30 | 0.63 | 0.24 | 0.65 | 300–450[a] | 805–1330[a] | | | | |
| 2 | IronEx-2 | 0.25 | ~0.57[b] | 0.15–0.20 | 4.00 | ~630[c] | ~2430[c] | ~6[d] (0–55 m) | ~14[d] (0–55 m) | 6.5 | 10.8 |
| 3 | SOIREE | 0.22 | 0.65 | 0.25 | 2.00 | ~120[e] | ~1300[e] | 22.8[f] (0–65 m) | 30.1[f] (0–65 m) | 1.7 | 3.9 |
| 4 | EisenEx | 0.30 | 0.56 | 0.50 | 2.50 | 130–220 | 790 | | | 2.0 | 6.2 |
| 5 | SOFeX-N | 0.20 | 0.5 | ~0.15[g] | ~2.60[g] | ~144[h] | ~1500[h] | | | | 10.9 |
| 6 | SOFeX-S | 0.25 | 0.65 | ~0.30[g] | ~3.80[g] | ~216[h] | ~972[h] | | | 3.3 | 5.3 |
| 7 | EIFEX | ~0.28[i] | ~0.6[i] | 0.70 | 3.16 | ~750 | 1500 | | | | |
| 8 | SAGE | 0.27 | 0.61 | 0.63 | 1.33 | 540 | 900 | | | | |
| 9 | LOHAFEX | ~0.33 | 0.50 | 0.50 | 1.25 | <960 | 1560 | | | | |
| 10 | SEEDS-1 | ~0.19[j] | ~0.42[j] | 0.80–0.90 | 21.8 | 420 | 1670 | 6.8[f] (0–20 m) | 7.5[f] (0–20 m) | ~3.2 | 8.1 |
| 11 | SERIES | 0.24 | 0.55 | 0.35 | ~5.00 | 300 | >2000 | 7.3[f] (0–30 m) | | 5.5 | 12 |
| 12 | SEEDS-2 | 0.29 | ~0.43[k] | 0.80 | 2.48 | 390 | >1000 | 18.9[f] (0–20 m) | 38[f] (0–20 m) | | |
| 13 | FeeP | | | 0.06 | 0.07 | | | | | | |

[a]Primary productivity (PP) in IronEx-1 was estimated by multiplying PP (mg C m$^{-3}$ d$^{-1}$) with the mixed layer depth (initial: 30 m and after: 35 m). [b]Fv/Fm in IronEx-2 was digitized from the Figure 3 of Behrenfeld et al. (1996). [c]PP in IronEx-2 was digitized from the Figure 2 of Boyd (2002). [d]Mesozooplankton biomass in IronEx-2 was digitized from the Figure 1 of Rollwagen Bollens and Landry (2000); Values in brackets correspond to the sampling layer. [e]PP in SOIREE was digitized from the Figure 3 of Gall et al. (2001a). [f]Mesozooplankton biomass indicates copepod biomass; Values in brackets correspond to the sampling layer; After mesozooplankton biomass is the mean value averaged for the experimental period after iron addition. [g]Chlorophyll-a concentrations in SOFeX-N/-S were digitized from the supplementary Figure 5 of Coale et al. (2004). [h]PP values in SOFeX-N/-S were digitized from the Figure 4 of Coale et al. (2004). [i]Fv/Fm in EIFEX was digitized from the Figure 2 of Berg et al. (2011). [j]Fv/Fm in SEEDS-1 was digitized from the Figure 2 of Tsuda et al. (2003). [k]Fv/Fm in SEEDS-2 was digitized from the Figure 6 of Tsuda et al. (2007).

Sources are Kolber et al. (1994); Behrenfeld et al. (1996); Coale et al. (1996); Steinberg et al. (1998); Boyd et al. (2000); Rollwagen Bollens and Landry (2000); Boyd and Law (2001); Cochlan (2001); Gall et al. (2001a); Hall and Safi (2001); Zeldis (2001); Boyd (2002); Gervais et al. (2002); Tsuda et al. (2003); Arrieta et al. (2004); Boyd et al. (2004); Coale et al. (2004); Oliver et al. (2004); Boyd et al. (2005); de Baar et al. (2005); Suzuki et al. (2005); Takeda and Tsuda (2005); Tsuda et al. (2005); Levasseur et al. (2006); Boyd et al. (2007); Tsuda et al. (2007); Kudo et al. (2009); Tsuda et al. (2009); Harvey et al. (2010); Berg et al. (2011); Currie et al. (2011); Peloquin et al. (2011a); Smetacek et al. (2012); Thiele et al. (2012);

5    Martin et al. (2013); and Latasa et al. (2014).

**Table 5**. Initial values of the export flux and the values after fertilization (mg C m$^{-2}$ d$^{-1}$), the corresponding depth inside and outside the fertilized patch for artificial ocean iron fertilization (aOIF) experiments, and measurement method. Values in brackets correspond to the day of measurement after fertilization.

| | Experiment | In-patch Initial (day) | In-patch After (day) | Outside-patch Initial (day) | Outside-patch After (day) | Depth (m) | Method |
|---|---|---|---|---|---|---|---|
| 1 | IronEx-1 | | | | | | |
| 2 | IronEx-2 | 84 (0) | 600 (7–14) | | | 25 | Water-column $^{234}$Th |
| 3 | SOIREE | | ~87 | | | 100 | Water-column $^{234}$Th |
| | | | 185 (11–13) | 146 (0–2) | 78 (11–13) | 110 | Drifting trap |
| | | | 74 (11–13) | 73 (0–2) | 38 (11–13) | 310 | Drifting trap |
| 4 | EisenEx | | | | | | |
| 5 | SOFeX-N | | | | | | |
| 6 | SOFeX-S | 36 (5) | 112 (27) | 48 (6) | 49 (26) | 50 | Water-column $^{234}$Th |
| | | 19 (5) | 142 (27) | 38 (6) | 56 (26) | 100 | Water-column $^{234}$Th |
| 7 | EIFEX | ~340 (0)[a] | ~1692 (32)[a] | ~396 (0)[a] | ~516 (32)[a] | 100 | Water-column $^{234}$Th |
| 8 | SAGE | | | | | | |
| 9 | LOHAFEX | ~62 (0)[b] | ~94 (25)[b] | ~77 (4)[b] | ~54 (34)[b] | 100 | Water-column $^{234}$Th |
| | | ~6 (0–2)[c] | ~5 (13–15)[c] | | ~29 (26–27)[c] | 200 | Neutrally buoyant sediment trap |
| | | | ~12 (28–37)[c] | | ~11 (24–29)[c] | 450 | Neutrally buoyant sediment trap |
| 10 | SEEDS-1 | 234 (1–3) | 141 (12–14) | 148 (1–6) | 154 (10–14) | 40 | Drifting trap |
| | | 100 (0–2) | 423 (9–13) | | | 50 | Water-column $^{234}$Th[d] |
| | | 68 (1–3) | 85 (12–14) | 61 (1–6) | 91 (10–14) | 100 | Drifting trap |
| | | 121 (0–2) | 460 (2–9) | | | 200 | Water-column $^{234}$Th |
| 11 | SERIES | ~138 (3)[e] | 480 (24) | 192 (3) | 139 (15) | 50 | Drifting trap |
| | | ~48 (3)[e] | ~192 (24)[e] | | | 100 | Drifting trap |
| 12 | SEEDS-2 | 290 (1–4) | 580 (19–22) | 300 (1–8) | 509 (18–31) | 40 | Drifting trap |
| | | 316 (1–4) | 337 (19–22) | 213 (1–8) | 204 (18–31) | 100 | Drifting trap |
| 13 | FeeP | | | | | | |

[a]Export flux in EIFEX was digitized from the supplementary Figure 5.1 of Smetacek et al. (2012). [b]Export flux in LOHAFEX was digitized from the Figure 4 of Martin et al. (2013). [c]Export flux in LOHAFEX was digitized from the Figure 6 of Martin et al. (2013). [d]Export flux in SEEDS-1 was determined from the suspended particles. [e]Export flux in SERIES was digitized from the Figure 2 of Boyd et al. (2004).

Sources are Bidigare et al. (1999); Charette and Buesseler (2000); Nodder and Waite (2001); Boyd et al. (2004); Aono et al. (2005); Buesseler et al. (2005); Aramaki et al. (2009); Smetacek et al. (2012); and Martin et al. (2013).

**Figure Captions**

**Fig. 1.** Diagram showing the monthly atmospheric $CO_2$ concentrations (ppm) (blue) measured at the Mauna Loa Observatory, Hawaii (http://www.esrl.noaa.gov/gmd/ccgg/trends/data.html), global monthly land-surface air and sea surface temperature anomalies (°C) (red) (http://data.giss.nasa.gov/gistemp/), and pH (green) measured at Station ALOHA in the central North Pacific (http://hahana.soest.hawaii.edu/hot/products/HOT_surface_CO2.txt). The data values represent moving average values for 12 months and shading indicates the standard deviation for 12 months.

**Fig. 2.** Schematic representation of several proposed climate-engineering methods (modified from Matthews (1996)).

**Fig. 3.** The iron hypothesis, as suggested by Martin (1990). (a) Effectiveness of the biological pump under normal conditions. (b) Effectiveness of the biological pump following iron enrichment (modified from Sarmiento and Gruber (2006)). (c) Schematic diagram of the decrease in the downward flux of organic carbon as a function of depth in the water column (modified from Lampitt et al. (2008)). OM is organic matter and DIC is dissolved inorganic carbon.

**Fig. 4.** Global annual distribution of surface (a) Chlorophyll concentrations (mg m$^{-3}$), (b) Nitrate concentrations (μM), and (c) Silicate concentrations (μM). The chlorophyll-a concentration distribution was obtained from the Aqua MODIS chlorophyll-a composite from July 2002 to February 2016 (https://oceancolor.gsfc.nasa.gov/cgi/l3), nitrate and silicate were obtained from the World Ocean Atlas 2013 dataset (https://odv.awi.de/en/data/ocean/world-ocean-atlas-2013) and plotted using Ocean Data View (Schlitzer, 2017). The white circles indicate the locations of 13 artificial ocean iron fertilization (aOIF) experiments and the black triangles indicate the locations of six natural OIF (nOIF) experiments. Note that the numbers indicate the order of the aOIF experiments and the Roman-numerals indicate the order of the nOIF experiments (see Table 1).

**Fig. 5.** Photographs of the iron addition procedure. Panels a-f taken during the European Iron Fertilization Experiment (EIFEX), Surface Ocean Lower Atmosphere Study (SOLAS) Air–Sea Gas Exchange (SAGE), and Indo-German iron fertilization experiment (LOHAFEX): (a) Iron(II) sulfate bags. (b) The funnel used to pour iron and hydrochloric acid. (c) Tank system used for mixing Iron(II) sulfate, hydrochloric acid, and seawater (Smetacek, 2015). (d) Preparation for release: the deck of RV *Tangaroa* with the iron tanks on the left and the $SF_6$ tracer tanks on the right (Photo: Matt Walkington) (https://www.niwa.co.nz/coasts-and-oceans/research-projects/sage). (e) Outlet pipe connected to the tank system. (f) Pumping iron into the prop wash during EIFEX (Smetacek, 2015).

**Fig. 6.** (a) Maximum (bar with dotted line) and initial (bar with solid line) patch size (km$^2$) during artificial ocean iron fertilization (aOIF) experiments. (b) First target iron concentrations (nM). (c) Maximum (bar with dotted line) and minimum (bar with solid line) mixed layer depth (MLD, m) during aOIF experiments. (d) Initial sea surface temperature (SST, °C). (e) Initial nitrate concentrations (μM). (f) Initial silicate concentrations (μM). (g) Initial Fv/Fm ratios. (h) Initial chlorophyll-a concentrations (mg m$^{-3}$). Note that the numbers on the X axis indicate the order of aOIF experiments as given in Fig. 4 and Table 1 and are grouped according to ocean basins; Equatorial Pacific (EP) (yellow bar), Southern Ocean (SO) (blue bar), subarctic North Pacific (NP) (red bar), and subtropical North Atlantic (NA) (green bar). Sources are Kolber et al. (1994); Martin et al. (1994); Behrenfeld et al. (1996); Coale et al. (1996); Coale et al. (1998); Steinberg et al. (1998); Boyd et al. (2000); Boyd and Law (2001); Gall et al. (2001b); Gervais et al. (2002); Law et al. (2003); Tsuda et al. (2003); Coale et al. (2004); Turner et al. (2004); Bakker et al. (2005); Boyd et al. (2005); Bozec et al. (2005); de Baar et al. (2005); Hiscock and Millero (2005); Takeda and Tsuda (2005); Tsuda et al. (2005); Tsumune et al. (2005); Law et al. (2006); Marchetti et al. (2006); Boyd et al. (2007); Rees et al. (2007); Tsuda et al. (2007); Suzuki et al. (2009); Tsumune et al. (2009); Harvey et al. (2010); Smetacek and Naqvi (2010); Berg et al. (2011); Hadfield (2011); Law et al. (2011); Peloquin et al. (2011a); Smetacek et al. (2012); Thiele et al. (2012); Martin et al. (2013); Ebersbach et al. (2014); and Latasa et al. (2014).

**Fig. 7.** (a) Maximum (bar with dotted line) and initial (bar with solid line) Fv/Fm ratios during artificial ocean iron fertilization (aOIF) experiments. (b) Changes in nitrate concentrations ($\Delta NO_3^- = [NO_3^-]_{post-fertilization (posf)} - [NO_3^-]_{pre-fertilization (pref)}$; μM). (c) Maximum (bar with dotted line) and initial (bar with solid line) chlorophyll-a concentrations (mg m$^{-3}$). (d) Distributions of chlorophyll-a concentrations (mg m$^{-3}$) on day 24 after iron addition in the Southern Ocean iron experiment-north (SOFeX-N) from MODIS Terra Level-2 daily image and on day 20 in the SOFeX-south (SOFeX-S) from SeaWiFS Level-2 daily image (white dotted box indicates phytoplankton bloom during aOIF experiments). (e) Changes in primary productivity (PP) ($\Delta PP = [PP]_{posf} - [PP]_{pref}$; mg C m$^{-2}$ d$^{-1}$). (f) Changes in partial pressure of $CO_2$ ($pCO_2$) ($\Delta pCO_2 = [pCO_2]_{posf} - [pCO_2]_{pref}$; μatm). The color bar indicates changes in dissolved inorganic carbon (DIC) ($\Delta DIC = [DIC]_{posf} - [DIC]_{pref}$; μM). Note that the PP (mg C m$^{-2}$ d$^{-1}$) of aOIF experiment number 1 (IronEx-1) was estimated by multiplying the PP (mg C m$^{-3}$ d$^{-1}$) with the mixed layer depth (initial: 30 m and after: 35 m). The numbers on the X axis indicate the order of

aOIF experiments as given in Fig. 4 and Table 1 and are grouped according to ocean basins; Equatorial Pacific (EP) (yellow bar), Southern Ocean (SO) (blue bar), subarctic North Pacific (NP) (red bar), and subtropical North Atlantic (NA) (green bar). Sources are Kolber et al. (1994); Martin et al. (1994); Behrenfeld et al. (1996); Coale et al. (1996); Steinberg et al. (1998); Boyd et al. (2000); Boyd and Law (2001); Frew et al. (2001); Gall et al. (2001a); Boyd (2002); Gervais et al. (2002); Tsuda et al. (2003); Coale et al. (2004); Boyd et al. (2004); Bakker et al. (2005); Boyd et al. (2005); Bozec et al. (2005); de Baar et al. (2005); Hiscock and Millero (2005); Smetacek et al. (2005); Takeda and Tsuda (2005); Tsuda et al. (2005); Wong et al. (2006); Boyd et al. (2007); Tsuda et al. (2007); Kudo et al. (2009); Tsumune et al. (2009); Harvey et al. (2010); Smetacek and Naqvi (2010); Berg et al. (2011); Currie et al. (2011); Law et al. (2011); Peloquin et al. (2011a); Smetacek et al. (2012); Thiele et al. (2012); Assmy et al. (2013); Martin et al. (2013); Ebersbach et al. (2014); and Latasa et al. (2014).

**Fig. 8.** (a) Time-series of particulate organic carbon (POC) fluxes estimated from the water-column based $^{234}$Th method (mg C m$^{-2}$ d$^{-1}$) of the upper 100 m layer inside (red bar) and outside the fertilized patch (blue bar) during the European Iron Fertilization Experiment (EIFEX) (modified from Smetacek et al. (2012)). (b) Time-series of vertically integrated $^{234}$Th (dpm l$^{-1}$) inside (red circles) and outside the fertilized patch (blue diamonds) relative to the parent $^{238}$U (dpm l$^{-1}$; dotted black line) during the Southern Ocean Iron Release Experiment (SOIREE) (modified from Nodder et al. (2001)).

**Fig. 9.** Assessment framework for scientific research involving ocean fertilization (OF) (modified from Resolution LC-LP.2 (2010)).

**Fig. 10.** (a) Time-series of mixed layer depth-integrated chlorophyll-a concentrations (mg m$^{-2}$) during the Southern Ocean Iron Release Experiment (SOIREE) (brown line), Subarctic Pacific iron Experiment for Ecosystem Dynamics Study-1 (SEEDS-1) (coral line), Subarctic Ecosystem Response to Iron Enrichment Study (SERIES) (cyan line), SEEDS-2 (blue line), and European Iron Fertilization Experiment (EIFEX) (teal line). (b) The distributions of chlorophyll-a concentrations (mg m$^{-3}$) on day 5 and day 42 during SOIREE from SeaWiFS Level-2 daily images. Sources are Gall et al. (2001b); Tsuda et al. (2007); and Assmy et al. (2013).

**Fig. 11.** Schematic diagram of the Korean Iron Fertilization Experiment in the Southern Ocean (KIFES) representing the experiment target site (eddy structure) and survey methods (underway sampling systems, multiple sediment traps, sub-bottom profilers, sediment coring systems, and satellite observations).

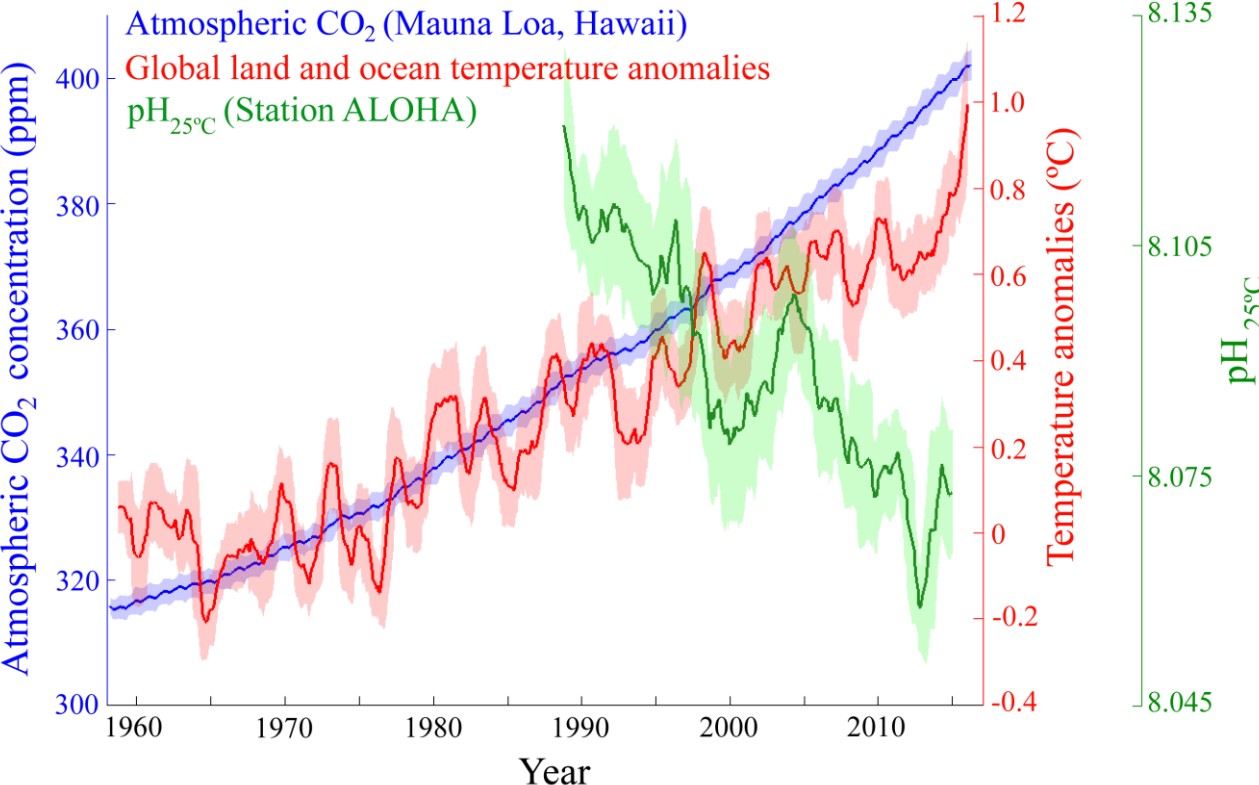

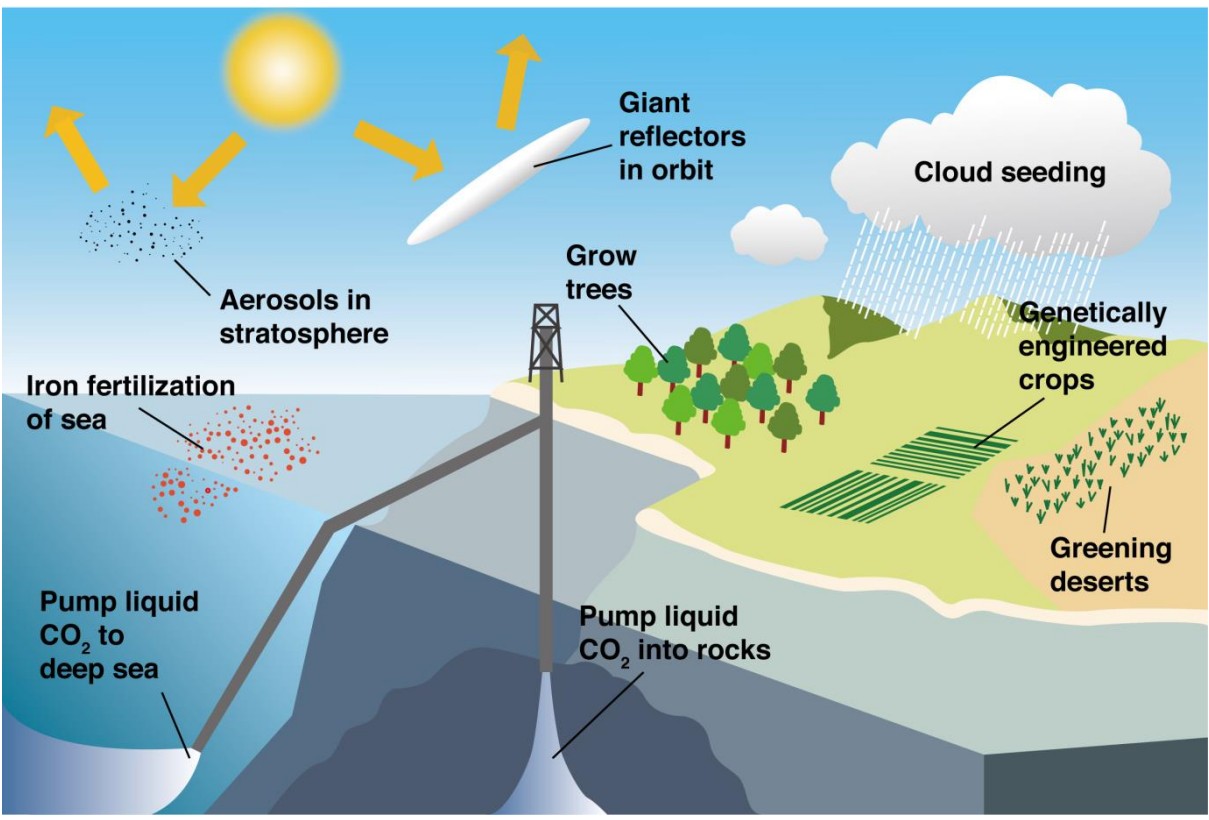

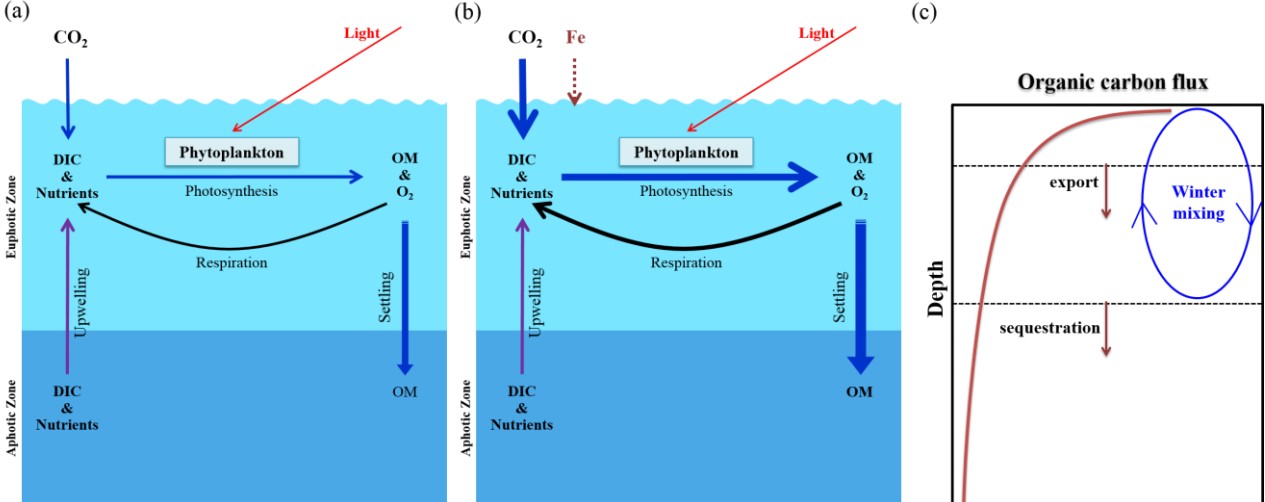

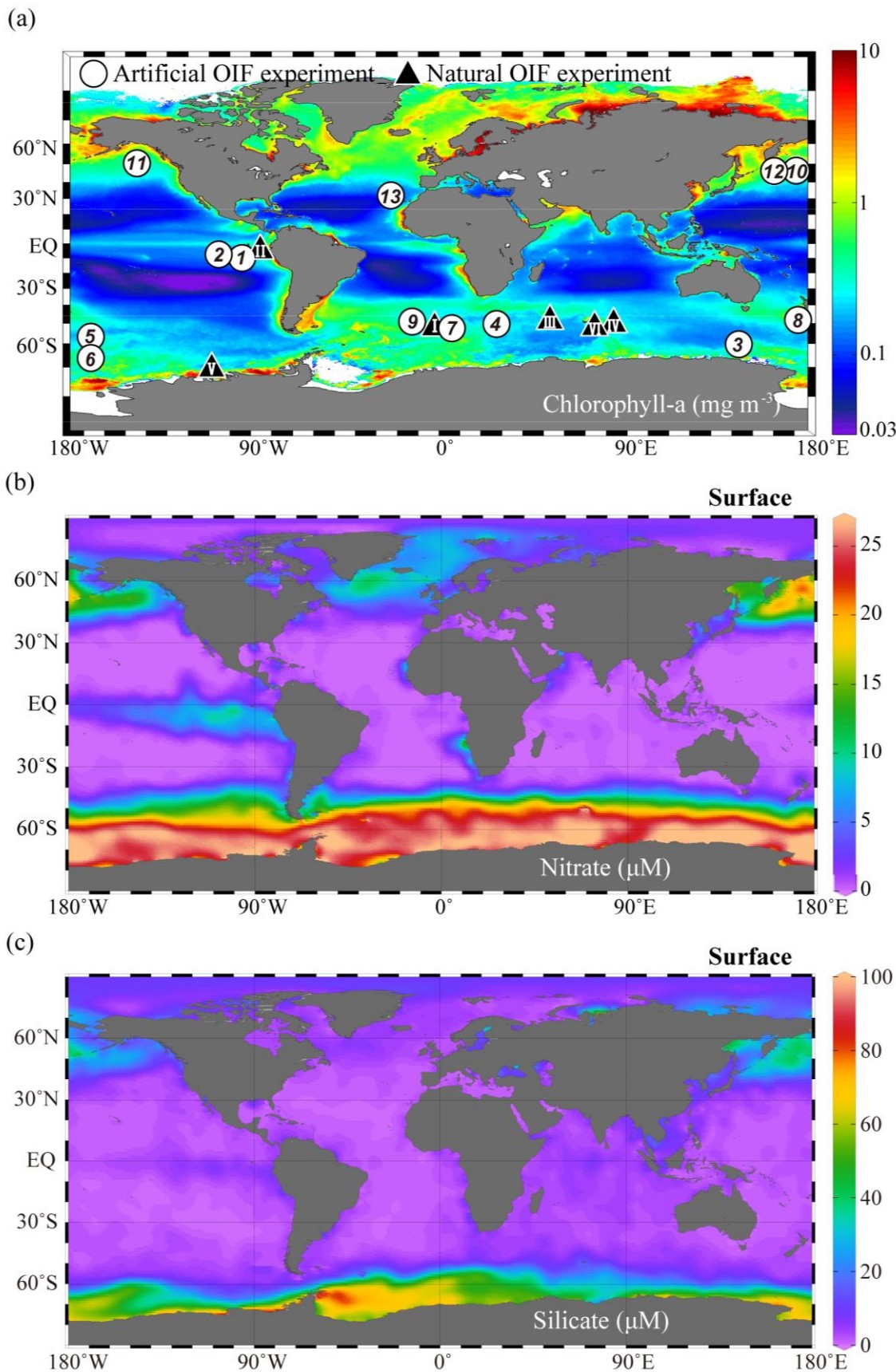

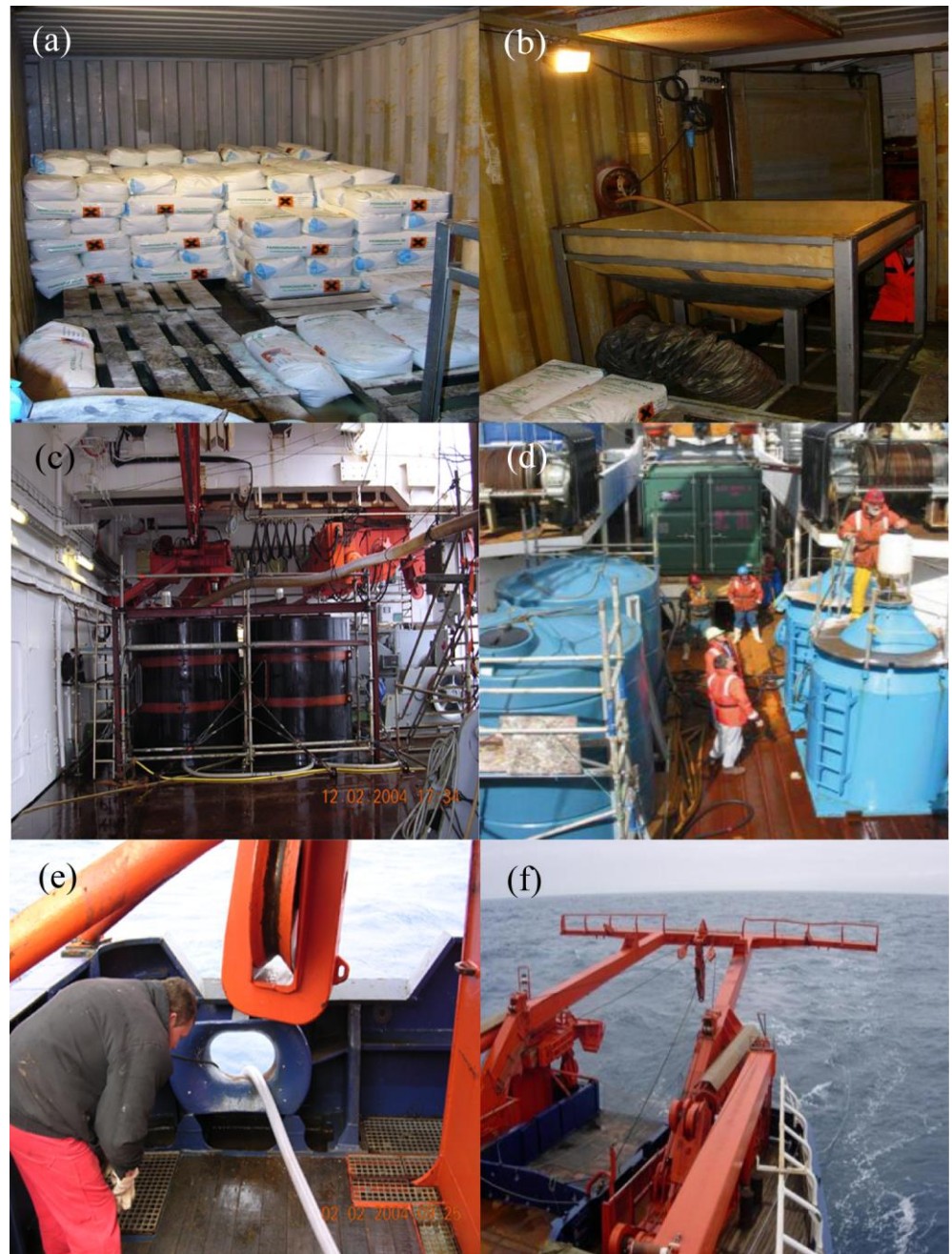

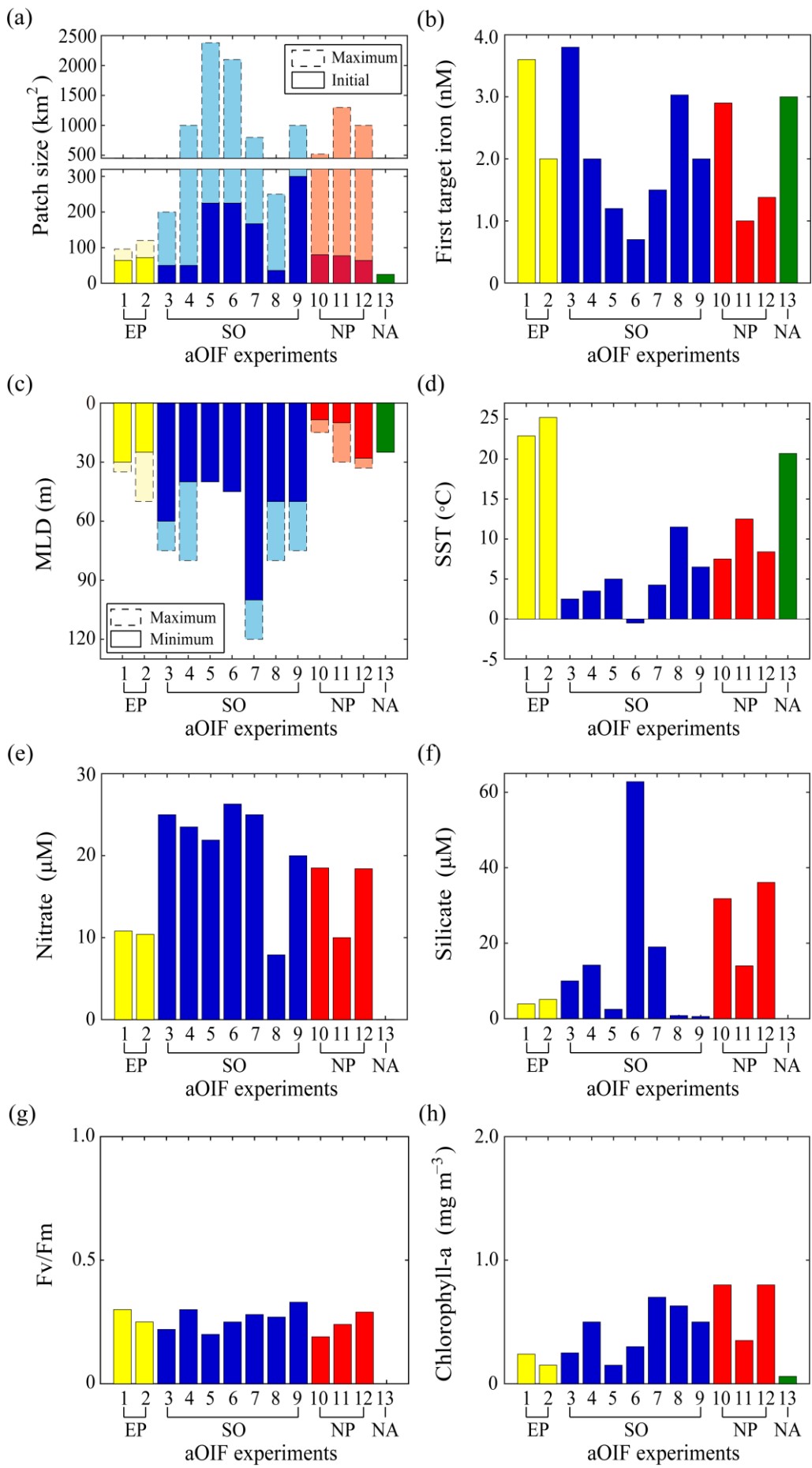

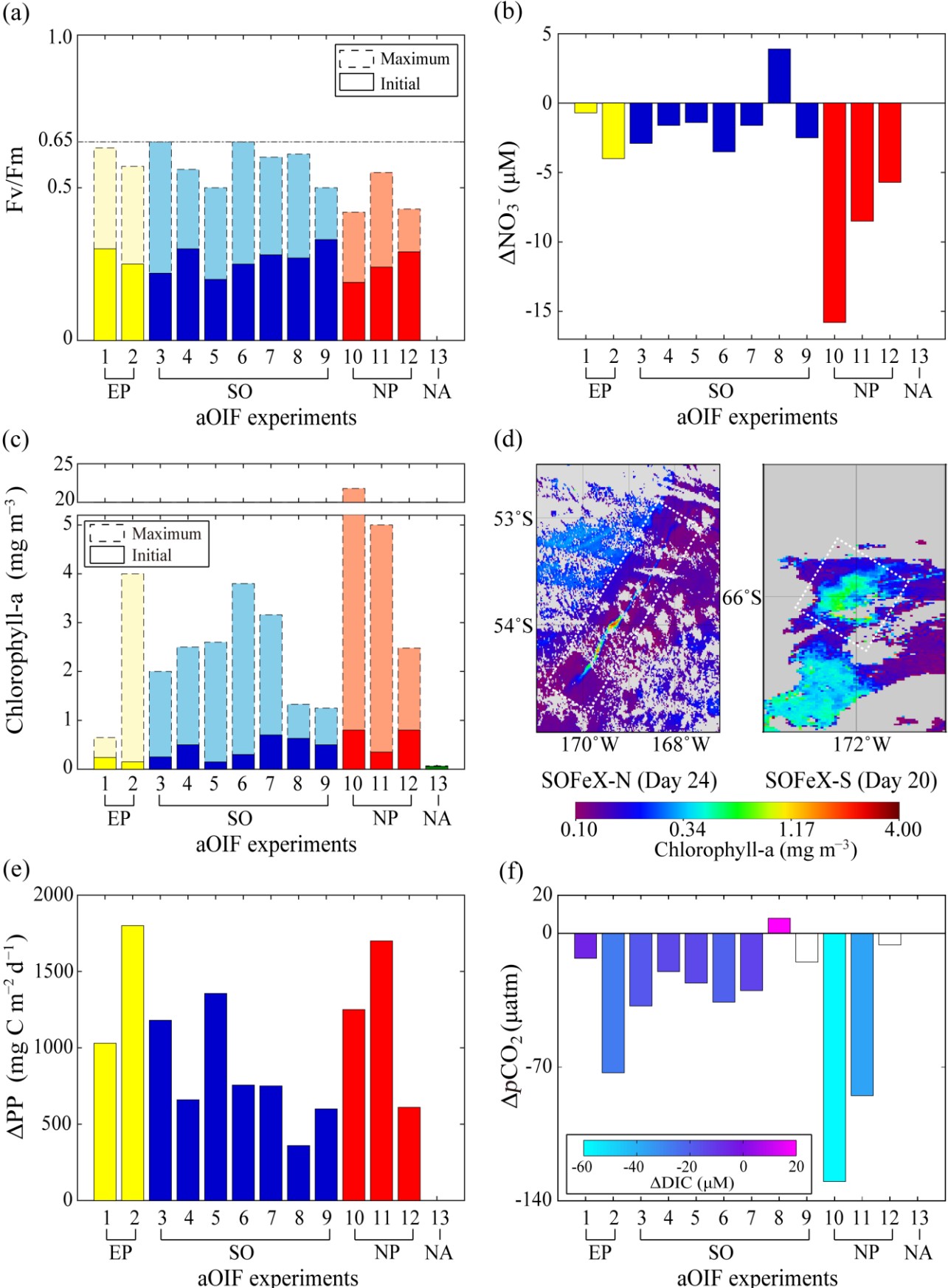

(a)

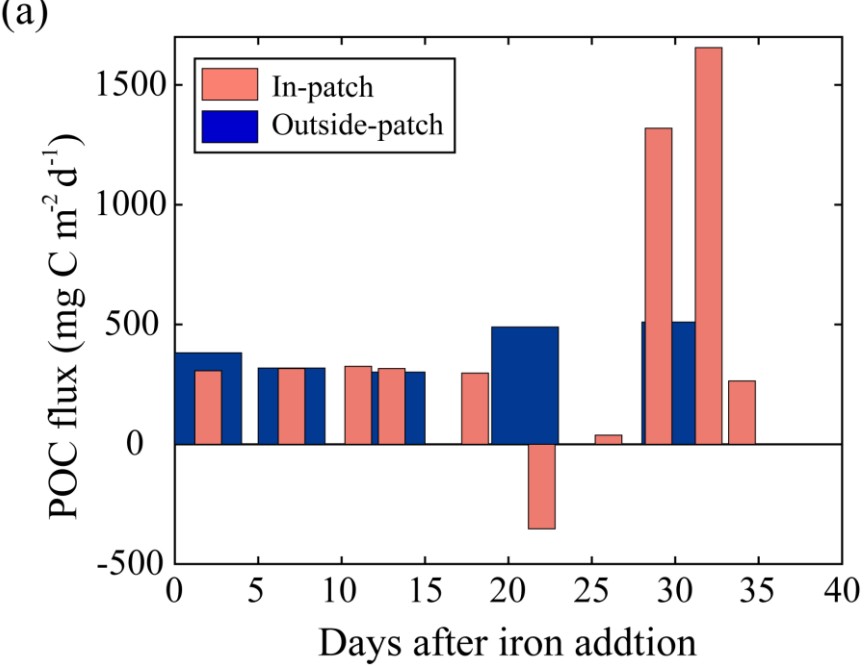

(b)

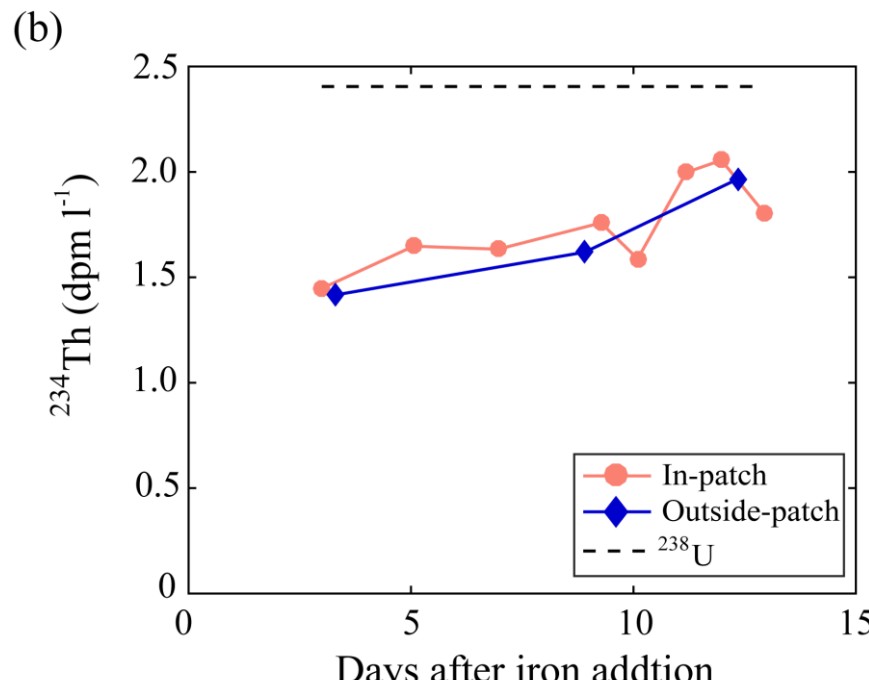

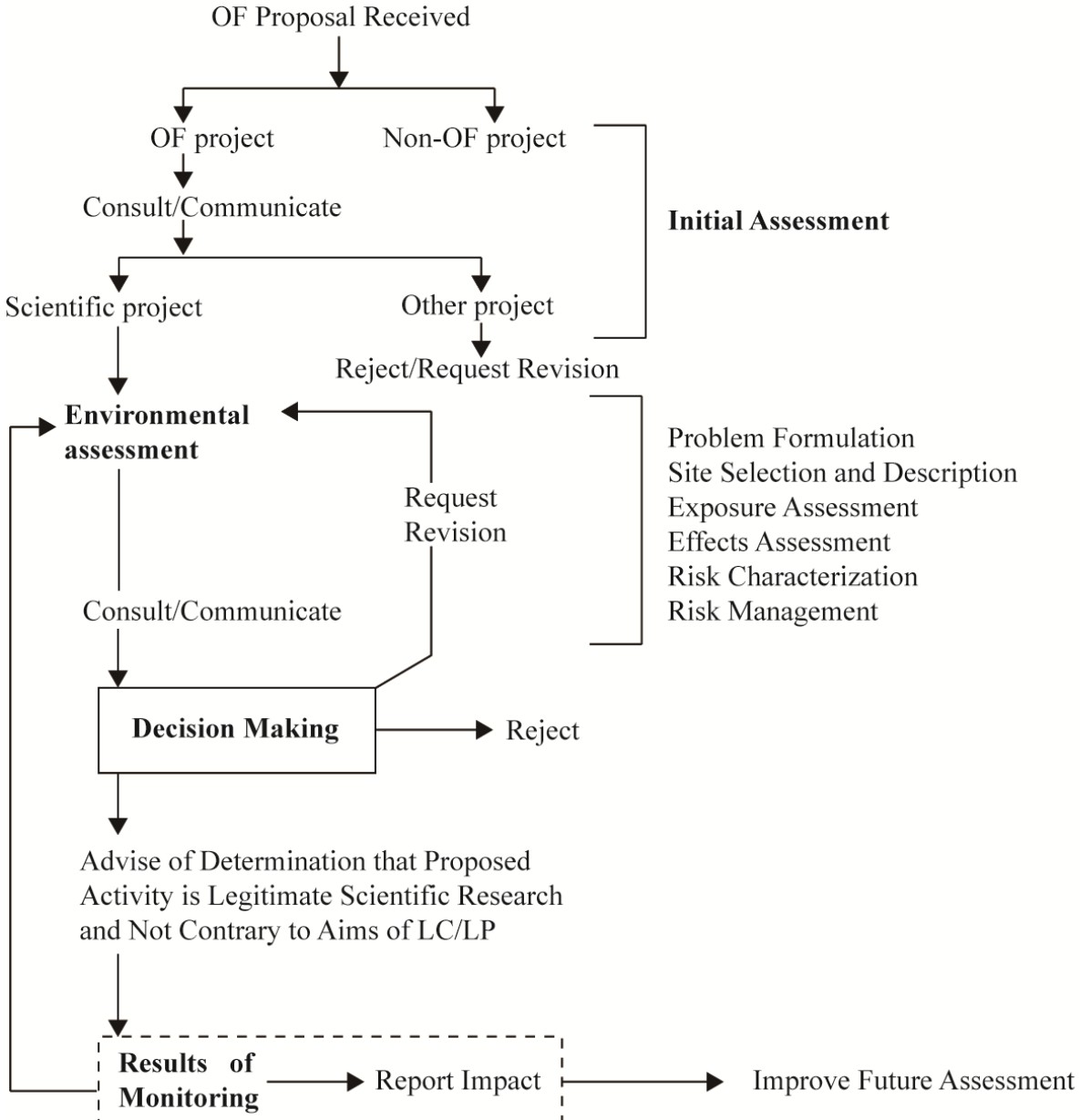

OF Proposal Received

OF project      Non-OF project

Consult/Communicate

**Initial Assessment**

Scientific project      Other project

Reject/Request Revision

**Environmental assessment**

Request Revision

Problem Formulation
Site Selection and Description
Exposure Assessment
Effects Assessment
Risk Characterization
Risk Management

Consult/Communicate

**Decision Making**      Reject

Advise of Determination that Proposed
Activity is Legitimate Scientific Research
and Not Contrary to Aims of LC/LP

**Results of Monitoring**      Report Impact      Improve Future Assessment

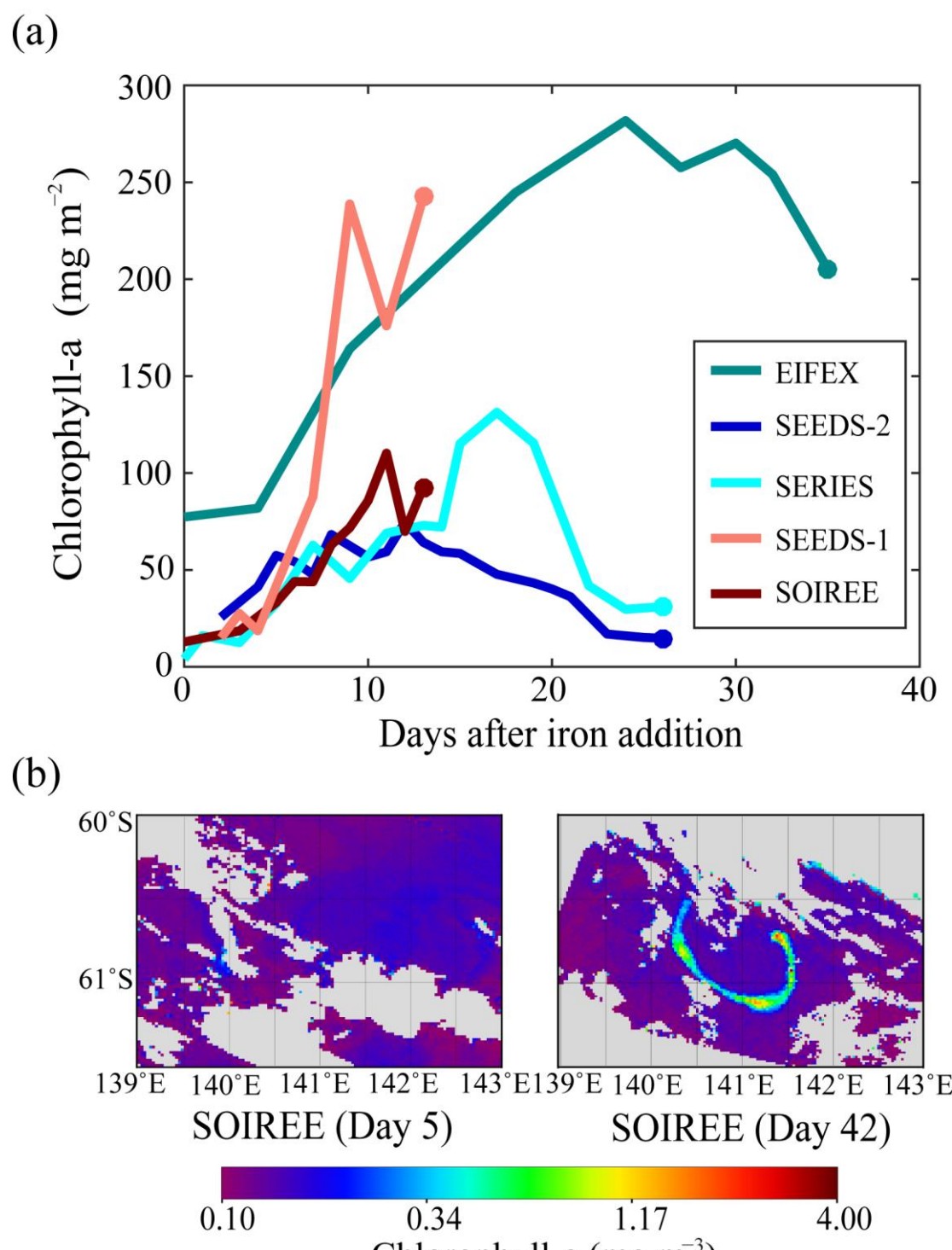

(a)

(b)

SOIREE (Day 5)          SOIREE (Day 42)

Chlorophyll-a (mg m⁻³)

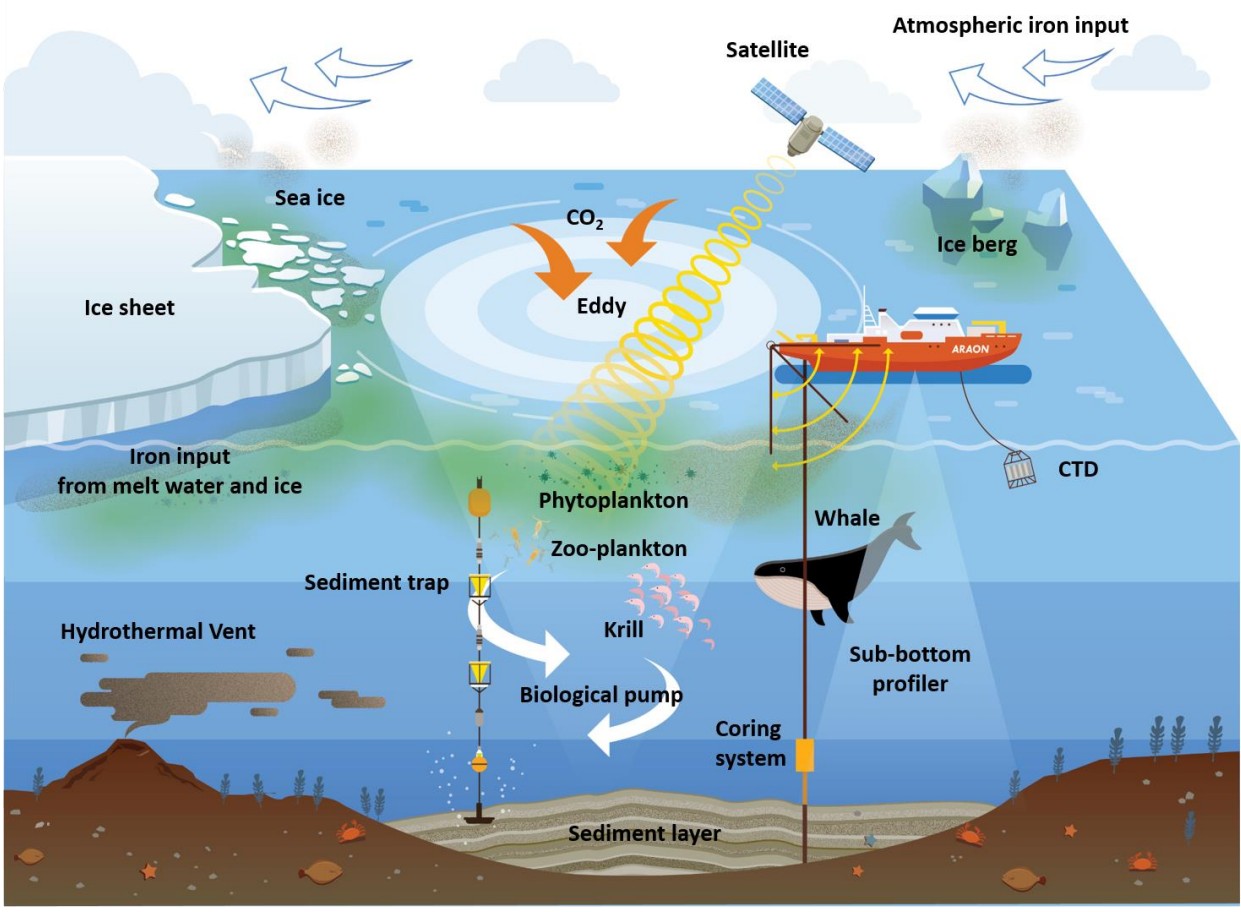