# Peer review of "Ocean Iron Fertilization Experiments: Past-Present-Future looking to a future Korean Iron Fertilization Experiment in the Southern Ocean (KIFES) Project"

_Biogeosciences, 2016_

## Referee Comment (RC1) · Anonymous Referee #1 · 5 Dec 2016

Review of Yoon et al. Ocean Iron Fertilization Experiments

The manuscript by Yoon et al. consists of two distinct parts. In the first, the authors summarise the findings of all ocean iron fertilisation experiments conducted to date. In the second, they briefly describe the plans for a proposed new iron fertilisation experiment that they are currently planning as a Korean-lead experiment with international participation.

Given the overall sensitive nature of this topic, and the fact that new iron fertilisation experiments have been placed under an approval and risk assessment process, I believe that the authors are taking a commendable approach in wishing to publish their plans well in advance in an open-access scientific journal. Biogeosciences would be an appropriate place to publish such a paper, and I believe that publication of this paper will help to foster a constructive and transparent discussion of the proposed study. The writing and presentation of figures and tables are of good quality, making the manuscript easy to follow, and the structure is logical and appropriate.

However, in my opinion the manuscript does not go far enough in discussing the results of previous studies and drawing possible conclusions. The review offered in Section 2 and in Section 3 feels like too much of a list of results, and the discussion that is offered is frequently inconclusive. This may reflect a desire on the part of the authors to avoid stoking controversy, but I do think that a more critical discussion is needed in the context of proposing a new study. Moreover, I think the authors could outline the objectives of the proposed new study more clearly with explicit definitions of what they mean by terms such as "effectiveness of OIF" or "efficiency" of OIF. Together with the rather brief outline of the proposed new study, the inconclusive nature of their discussion and the lack of clear definitions of what the new study aims to do I found myself wondering to what extent the new study will really go beyond the previous studies in allowing us to draw conclusions about the potential for OIF in removing atmospheric CO2 and the potential for negative side-effects. In other words, I think the paper would make a much stronger contribution if the authors could outline the most important gaps in our knowledge more clearly, identify specifically how future experiments should be designed to fill those gaps, and then explain how KIFES is designed to be able to address these questions.

I have decided to make a recommendation of "major revisions" because I think that providing a more insightful discussion goes beyond just minor changes. That said, I do not think that what I am recommending is overly onerous.

Specifically, I would recommend that the authors address the following points:

1. Downward carbon fluxes can be quantified using diverse metrics. The really important one from the point of view of geoengineering would be the amount of carbon sequestered below the depth of deepest winter mixing in the study region, which most previous OIF did not measure. The article frequently uses terms such as "efficiency of OIF at reducing atmospheric $CO_2$", but the authors never define clearly what they mean by this. The efficiency of the biological carbon pump can be quantified using several approaches, but from a geoengineering point of view the efficiency is less important that the absolute amount. The article would be more helpful if the authors defined clearly which metrics really matter. Moreover, it would be useful if the authors more explicitly assessed which of the experiments conducted to date were actually capable of detecting an enhancement of export if it had occurred (based on duration of the experiment relative to the phases of the bloom and the type of measurements that were taken), which of these did find a response in particle flux (e.g. EIFEX, SERIES), and how to what depth the carbon flux was followed.

2. The previous OIF differed significantly in experimental design, especially in terms of patch size, duration, location, and also in terms of which measurements were taken. I found the discussion of these aspects in Section 3.2 rather unsatisfying: especially since the authors are in advanced stages of planning a new experiment, what have they concluded from this literature about how best to design an OIF? What are their recommendations in terms of best patch size, minimum duration, and which measurements are required to quantify the effect on carbon sequestration? I think that discussion of these points is important, especially since the authors are clearly interested in persuading the scientific (and, presumably, wider) community that their proposed experiment will provide answers about the scope for geoengineering via OIF. The clear conclusion that they do appear to have drawn is that the experiment should be located inside an eddy. However, to accurately measure downward carbon flux out of the patch at the depth of maximum winter mixing will require a large patch to ensure that sediment traps potentially several hundred metres below the surface are not at too high a risk to actually miss possible particle fluxes. What is their conclusion about the

minimum duration that is needed? Given the results of SERIES, SEEDS-2, EIFEX, and LOHAFEX, it would seem to me that one should aim at between 35 and 40 days post-fertilisation. Further, what recommendations can be made about measurement approaches to quantify carbon fluxes? An important point to me is that having multiple redundant methods is very important, e.g. thorium profiles, frequent deployments of sediment traps at multiple depths (ideally neutrally buoyant traps), and high-frequency measurements of properties such as pCO2 and O2:Ar ratios. It also strikes me that autonomous platforms should play a much greater role in future OIF than they have in the past, e.g. a combination of gliders and Lagrangian floats equipped with biogeochemical sensors. Especially bio-optical sensors such as fluorescence and backscatter can be extremely useful to help constrain downward particle fluxes and their vertical and horizontal variations.

3. The discussion of possible unintended side-effects could be similarly improved by trying to draw clearer conclusions rather than just summarising results from the previous literature. For example, it seems to me that the main conclusion about domoic acid is that it is very variable regardless of fertilisation, with the cited Smith et al. paper actually reporting higher per-cell quotas from natural than from artificially fertilised waters (the cited Trick et al. paper relied on bottle incubations and extrapolations based on claims about likely bloom size made by geoengineering companies on an internet site). Moreover, while a degree of oxygen consumption would certainly result from OIF, the sentence that "Box model solutions have further suggested that anoxic conditions may develop after OIF" is quite misleading: the cited reference is actually a much more realistic 3-dimensional model that only found anoxia developing in part of the western Indian Ocean, and only after many years of sustained complete nutrient utilisation in the Southern Ocean. This is probably a significantly more extreme scenario than could be achieved in practice, suggesting that anoxic conditions are actually quite unlikely. Conversely, increased production of other relevant gases, such as N2O, is clearly an important concern (though the discussion of DMS could do with some reference to the fact that its role in climate seems to be rather more complex than originally thought).

As with the question of experimental design, a more critical discussion of these factors would make this a more insightful and more useful paper, though of course I would not dispute that all of these possible side-effects need to be monitored.

In addition to my general comments above, I also have the following specific comments:

1. Abstract Line 10, and page 10 final paragraph Line 1: make it clear that these side-effects are possible side effects, and that changes in community composition may have unintended consequences.

2. Page 4 Paragraph 3: > and < signs for latitude are the wrong way round

3. Page 12 final paragraph: given the large number and large scale of natural mesoscale blooms in HNLC regions (e.g. due to iceberg-derived iron), I think it is fair to say that the risks to the environment from small-scale OIF experiments is very small indeed, and I think that the authors should be prepared to make that case. The risks of large-scale OIF for geoengineering purposes are the risks that are not understood, and small-scale studies are what we therefore need to undertake at this point to assess these risks better.

4. Page 14 Paragraph 2: Sentence starting "To data . . ." should read "To date, the only OIF experiment . . .".

5. Page 14 Section 4.2.3: What do the authors mean by "rehearsal"? Will they add only a tracer, such as SF6, or will iron be added as well?

6. Page 14 Section 4.2.4: As I indicated in one of my general comments, I think that future OIF could benefit greatly from using autonomous platforms, such as gliders, equipped with biogeochemical sensors. If this is not planned at present, I would urge the project leaders to consider their use.

7. Page 15 Section 4.2.5: What is the second stage of KIFES?

8. In Figure 4, the authors could consider marking the study region proposed for KIFES.

9. Figure 8 provides a summary of carbon flux-related data for two experiments, EIFEX and SOIREE (though Fig 8b is referred to in the context of IronEx-2 in the text). Several other experiments did report comparable data, either with sediment traps, thorium deficits, or both. Comparison of these data is obviously complicated by the fact that different experiments measured flux at different depths, but trying to summarise the results of all of the studies that reported particle fluxes might be helpful. Moreover, when the authors state on Page 9 Paragraph 2 that "That being said, EIFEX was the exception. Significant changes in export production were not found in any of the other OIF experiments", it should be made clear that only a sub-set of all OIF experiments was actually designed in such a way that an enhancement of downward particle flux could be detected (especially given the short duration of several experiments).

In summary, I think that the authors are doing the right thing by laying out their plans for a new experiment in the open-access scientific literature, providing a justification for a new experiment by summarising the current state of our knowledge. However, I think that a more careful and detailed discussion of previous results, combined with a clearer explanation of how the new experiment will overcome the limitations faced by previous experiments, would make for a significantly more useful contribution. This should be prefaced with a more explicit explanation of the necessary aims of a new OIF and of the measurements that are needed to accomplish these aims.

End of review

---

## Referee Comment (RC2) · Anonymous Referee #2 · 6 Dec 2016

The review article by Yoon et al. presents a summary of results from all ocean iron fertilization experiments (OIF) carried to date and introduces the plan for a new OIF experiment in the Bransfield Basin under the leadership of the Korean Polar Research Institute.

Artificial Ocean Iron fertilization experiments carried out in the previous decade received strong opposition from environmental groups as well as parts of the scientific community. The main concerns at the time were 1) the lack of control and regulatory mechanisms ensuring that such experiments were carried out with a solid scientific

basis and with a thorough assessment of the impacts on the marine environment. 2) The fear that governments and business would resort to OIF as a quick fix to compensate for CO2 emissions rather than focus on emission reduction. The recent amendment to the London Convention/Protocol provides now the necessary framework to regulate marine geoengineering activities (including scientific iron fertilization experiments). With a carbon sequestration potential estimated at around 1 GtC / year (albeit with large uncertainties) and the increasing evidence that negative emissions will be necessary to keep warming at or below 1.5°C (Hansen et al., Earth Syst. Dynam. Discuss., doi:10.5194/esd-2016-42) a resumption of OIF research seems timely. Beyond the scientific and regulatory framework, it is of importance that such activities should be done in a transparent manner. I, therefore, fully support publication of this manuscript in biogeosciences. The manuscript is well written, I have, however, several comments that I believe the authors should address:

1) The presentation of results from previous experiments seems too much like a catalog of data, but there is no thorough discussion on why the outcomes of the experiments were so different, and what has been learned from these experiments. Further, given that KIFES is planned to take place in the Southern Ocean, it is not obvious to the reader how the detailed presentation of results from experiments carried out in other oceanic basin is relevant here.

2) In the same line of thought, the rationale for artificial vs. natural iron experiments could also be discussed.

3) Overall, model studies are poorly represented in this review. Given that C sequestration estimates, as well as large scale and long term impacts of OIF are mostly determined through model studies, it might be relevant to mention them and how additional experiments might help constrain such models (see also comments below).

Other comments: p. 7, line 28: the authors explain Fv/Fm but the term is used much earlier in the text. I suggest shifting the explanation to the first time Fv/Fm is used. I am

also not sure that the description as written is very useful for people outside the field.

p.8, lines 16-28: Given the large differences in mixed layer depth between experiments, I would suggest the authors also discuss mixed layer integrated chlorophyll stocks as these better reflect the real biomass built up (i.e. standing stocks accumulated during EIFEX were similar to those for SEEDS even though concentrations were an order of magnitude lower).

p.8, line 26: there is a mistake in the sentence ("were appeared at"?), and the message is not clear.

p. 9, line 1: add "the" before "surface"

p. 10 lines 17-21: More recent model studies do not show development of anoxic conditions for large-scale iron fertilization in the Southern Ocean (see for instance Oschlies et al. Biogeosciences, 7, 4017–4035, 2010; Keller et al. Nature Communications, doi: 10.1038/ncomms4304, 2013).

p.10, line 36: Change "also has" to "could also have"

p. 11, line 3: Change "even though generally..." to "even though diatom species of the genus Pseudo-nitzschia were dominant numerically ".

p. 11, lines 20-28: I feel that the question how is somehow too easily brushed aside. This review could be used to discuss protocols and relevant parameters that should be measured, applied or developed. Not all experiments followed similar protocols, or measured all parameters.

p.13 lines 23-31: Can the authors give a reference for the mentioned studies. Further, the rationale for doing the experiment in the Bransfield Basin is not clear.

Table 2: It would be more useful if the authors provided with initial nutrient and DIC and the delta values (rather than the final concentrations).

Figure 3. I do not understand how oxygen is part of the settling component.

Legend Figure 4, line 3: Change "nitrate and silicate were presented" to "nitrate and silicate were plotted"

Figure 5 legend: Change to "Picture for iron addition procedure" to "Illustration" or "Photographs of iron the addition procedure. Panels a-e taken during EIFEX and LO-HAFEX."

Figure 5a legend: Change legend: a) Iron (II) sulfate bags

Figure 5b legend: The photograph shows the funnel where iron and HCl was poured, not the HCl.

Figure 5f: I am not sure were this picture was taken (the corresponding web page gives no information) but I find it misleading as the iron mixture is released in much lower quantities than depicted here (compare with the size of the hose in panel d taken during EIFEX) and has a different appearance too. I would recommend removing this panel, unless reliable information of its provenance can be provided.

---

## Author Comment (AC1) · 6 Feb 2017

**Dear The Editor and Anonymous Reviewers,**

First of all, we greatly appreciate the time and effort spent in reviewing of our manuscript. Unfortunately, we recently heard that the Korean Iron Fertilization Experiment in the Southern Ocean (KIFES) project, which was the inspiration for this review, has lost its funding. However, we here at Korea Polar Research Institute, as well as our domestic and international collaborators, fully intend to seek other sources of funding to revitalize the KIFES project. We apologize in advance that our responses to reviewer questions on KIFES are necessarily vague due to the current state of uncertainty surrounding the future of the project. Nevertheless, we believe that this manuscript is still worthy of publication as a detailed review of the history of ocean iron fertilization experiment designs and results. We have changed the title as follows: **"Ocean Iron Fertilization Experiment: Past-Present-Future looking to a future Korean Iron Fertilization Experiment in the Southern Ocean (KIFES) Project"** and still provide design guidelines for KIFES even though it is not presently funded.

We provide our responses (plain text with blue and red colors) to all comments (*italic text*) below.

**- Response to Reviewer Comments -**

**Reviewer #1**

**-Response to General Comments:**

☐1 *Downward carbon fluxes can be quantified using diverse metrics. The really important one from the point of view of geoengineering would be the amount of carbon sequestered below the depth of deepest winter mixing in the study region, which most previous OIF did not measure. The article frequently uses terms such as "efficiency of OIF at reducing atmospheric CO2", but the authors never define clearly what they mean by this. The efficiency of the biological carbon pump can be quantified using several approaches, but from a geoengineering point of view the efficiency is less important that the absolute amount. The article would be more helpful if the authors defined clearly which metrics really matter. Moreover, it would be useful if the authors more explicitly assessed which of the experiments conducted to date were actually capable of detecting an enhancement of export if it had occurred (based on duration of the experiment relative to the phases of the bloom and the type of measurements that were taken), which of these did find a response in particle flux (e.g. EIFEX, SERIES), and how to what depth the carbon flux was followed.*

→ Thank you for pointing this out. To reflect the Reviewer's suggestion, we added the following to defining the "efficiency of OIF" in the Introduction:

"To evaluate whether OIF has potential as a geoengineering strategy for carbon sequestration, not only the amount of carbon fixed by phytoplankton at the ocean surface but also the amount of carbon sequestered to the deep ocean must be considered in determining OIF efficiency (Buesseler and Boyd, 2003)."

Based on the Reviewer's comment, we have created a new Table (Table 5, page 32 of this response) that includes where available: absolute magnitude of export carbon flux; measurement depth; and the methods applied to detect an enhancement of export. Using Table 5, we also have created new Section "2.5 Assessment of export carbon flux" to more explicitly assess the enhancement of carbon flux in the manuscript as follows:

- 2.5 Assessment of export carbon flux

[revised manuscript text omitted]

☐2 *The previous OIF differed significantly in experimental design, especially in terms of patch size, duration, location, and also in terms of which measurements were taken. I found the discussion of these aspects in Section 3.2 rather unsatisfying: especially since the authors are in advanced stages of planning a new experiment, what have they concluded from this literature about how best to design an OIF? What are their recommendations in terms of best patch size, minimum duration, and which measurements are required to quantify the effect on carbon sequestration? I think that discussion of these points is important, especially since the authors are clearly interested in persuading the scientific (and, presumably, wider) community that their proposed experiment will provide answers about the scope for geoengineering via OIF. The clear conclusion that they do appear to have drawn is that the experiment should be located inside an eddy. However, to accurately measure downward carbon flux out of the patch at the depth of maximum winter mixing will require a large patch to ensure that sediment traps potentially several hundred metres below the surface are not at too high a risk to actually miss possible particle fluxes. What is their conclusion about the minimum duration that is needed? Given the results of SERIES, SEEDS-2, EIFEX, and LOHAFEX, it would seem to me that one should aim at between 35 and 40 days post-fertilisation. Further, what recommendations can be made about measurement approaches to quantify carbon fluxes? An important point to me is that having multiple redundant methods is very important, e.g. thorium profiles, frequent deployments of sediment traps at multiple depths (ideally neutrally buoyant traps), and high-frequency measurements of properties such as pCO2 and O2:Ar ratios. It also strikes me that autonomous platforms should play a much greater role in future OIF than they have in the past, e.g. a combination of gliders and Lagrangian floats equipped with biogeochemical sensors. Especially bio-optical sensors such as fluorescence and backscatter can be extremely useful to help constrain downward particle fluxes and their vertical and horizontal variations.*

→ We agree. More discussion is necessary. As suggested by Reviewer, we have moved this Section to "4 Future:

Considerations for designing future OIF experiments" to supplement discussion for designing a future OIF experiment that would maximize the effectiveness of OIF. We have revised Section 4 to consider the methods for iron addition (
[revised manuscript text omitted]

**3** *The discussion of possible unintended side-effects could be similarly improved by trying to draw clearer conclusions rather than just summarising results from the previous literature. For example, it seems to me that the main conclusion about domoic acid is that it is very variable regardless of fertilisation, with the cited Smith et al. paper actually reporting higher per-cell quotas from natural than from artificially fertilised waters (the cited Trick et al. paper relied on bottle incubations and extrapolations based on claims about likely bloom size made by geoengineering companies on an internet site). Moreover, while a degree of oxygen consumption would certainly result from OIF, the sentence that "Box model solutions have further suggested that anoxic conditions may develop after OIF" is quite misleading: the cited reference is actually a much more realistic 3-dimensional model that only found anoxia developing in part of the western Indian Ocean, and only after many years of sustained complete nutrient utilisation in the Southern Ocean. This is probably a significantly more extreme scenario than could be achieved in practice, suggesting that anoxic conditions are actually quite unlikely. Conversely, increased production of other relevant gases, such as N2O, is clearly an important concern (though the discussion of DMS could do with some reference to the fact that its role in climate seems to be rather more complex than originally thought).*

→ We apologize for the confusion. Based on the Reviewer's comments, we have provided conclusions about side effects based on recent modeling results and have revised Section "3.1 Environmental side effects" as follows:

[revised manuscript text omitted]

The changes of phytoplankton community composition after iron addition discussed in Section 2.4 may also have unintended consequences, in particular, toxin production (Silver et al., 2010; Trick et al., 2010). Some OIF experiments (including IronEx-2, SOIREE, EisenEx, and SOFeX-N/S) generated large blooms of diatoms dominated by pennate diatoms belonging to the genus 'Pseudo-nitzschia' (de Baar et al., 2005; Trick et al., 2010). Some species of the genus 'Pseudo-nitzschia' have the capacity to produce the neurotoxin domoic acid (DA) that is known to detrimentally affect marine ecosystems. For example, during IronEx-2 and SOFeX-S, high cell abundances of 'Pseudo-nitzschia' ($10^6$ and $10^5$ cells $l^{-1}$, respectively) combined with moderate DA quotas (0.05

and 1 pg DA cell$^{-1}$, respectively) produced toxin levels as high as 45 ng DA l$^{-1}$ in IronEx-2 and 220 ng DA l$^{-1}$ in SOFeX-S; i.e., toxin levels that have been shown cause harm to marine communities in coastal waters (Silver et al., 2010). However, no DA was found during EisenEx, even though '*Pseudo-nitzschia*' were the dominant diatom species (Gervais et al., 2002; Assmy et al., 2007).

The direct and indirect environmental consequences of OIF remain unresolved due to inconsistent, highly uncertain outcomes (Williamson et al., 2012; Johnson and Karl, 2002; Chisholm et al., 2001), suggesting that we have yet to reached a conclusion as to whether OIF is a feasible carbon removal strategy (Boyd et al., 2007). Therefore, evaluation and prediction are paramount. It continues to be a valuable exercise to seek answers to scientific questions about the efficiency of OIF as a means of reducing atmospheric $CO_2$ as well as to quantify possible OIF side effects. In particular, potential trace gas emissions such as $N_2O$ and DMS, which are influenced by the remineralization of sinking particles that follows OIF-induced blooms, are important to understand. They can directly and indirectly modify the desired carbon sequestration effect and they can do so both positively and negatively. Therefore, monitoring of $N_2O$ and DMS to evaluate the effectiveness of OIF as a geoengineering approach is essential."

**-Specific Comments:**

**4** *Abstract Line 10, and page 10 final paragraph Line 1: make it clear that these side- effects are possible side effects, and that changes in community composition may have unintended consequences.*

→ We have made it clear by changing "including side effects" to "including possible side effects" in the Abstract Line 10 and "The changes of phytoplankton community after iron addition discussed in Section 2.4 also has unintended consequences" to "The changes of phytoplankton community composition after iron addition discussed in Section 2.4 may also have unintended consequences" in page 10 final paragraph Line 1.

**5** *Page 4 Paragraph 3: > and < signs for latitude are the wrong way round.*

→ We corrected > and < signs for latitude.

**6** *Page 12 final paragraph: given the large number and large scale of natural mesoscale blooms in HNLC regions (e.g. due to iceberg-derived iron), I think it is fair to say that the risks to the environment from small-scale OIF experiments is very small indeed, and I think that the authors should be prepared to make that case. The risks of large-scale OIF for geoengineering purposes are the risks that are not understood, and small-scale studies are what we therefore need to undertake at this point to assess these risks better.*

→ The reviewer is correct. We have revised Section "3.2 International law of the sea to OIF" by rephrasing sentences as follows:

- 3.2 International law of the sea to OIF

"However, this effort has not been able to move forward because we have little knowledge about the potential magnitude of possible side effects related to large-scale geoengineering OIF. It remains difficult to extrapolate findings from the small-scale OIF experiments because the environmental/ecosystem side effects from these miniature studies are themselves quite variable and not yet clearly understood. However, presently available studies do indicate that the known side effects from small-scale studies are themselves small-scale. It therefore seems reasonable that we should continue to undertake small-scale studies to better assess these risks and so lay the groundwork for evaluating the potential efficacy and impacts of large-scale OIF as a geoengineering solution to anthropogenic change."

**7** *Page 14 Paragraph 2: Sentence starting "To data . . ." should read "To date, the only OIF experiment . . ."*

→ Done.

**8** *Page 14 Section 4.2.3: What do the authors mean by "rehearsal"? Will they add only a tracer, such as SF6, or will iron be added as well?*

→ We apologize for the confusion. By "rehearsal', we intended to indicate that we would conduct hydrographic surveys outside/inside the eddy structure in the eastern Bransfield Basin and employ drifting-buoys prior to the actual (eddy structure) OIF experiment without iron addition. Basically, we would not plan to add $SF_6$ as a chemical tracer in the KIFES project. We deleted the word and revised the KIFES section. Please, find it below (page 25–28 of this response).

**9** *Page 14 Section 4.2.4: As I indicated in one of my general comments, I think that future OIF could benefit greatly from using autonomous platforms, such as gliders, equipped with biogeochemical sensors. If this is not planned at present, I would urge the project leaders to consider their use.*

→ Thank you. Yes. We would plan to use autonomous platforms (page 27 of this response).

**10** *Page 15 Section 4.2.5: What is the second stage of KIFES?*

→ We meant the second 5-years of the KIFES project (2021–2025). Given the current status of KIFES funding, we deleted mention of the second stage of KIFES.

**11** *In Figure 4, the authors could consider marking the study region proposed for KIFES.*

→ We have included the number "14" to indicate KIFES in the Antarctic Peninsula in Figure 4 (page 35 of this response).

**12** *Figure 8 provides a summary of carbon flux related data for two experiments, EIFEX and SOIREE (though Fig 8b is referred to in the context of IronEx-2 in the text). Several other experiments did report comparable*

*data, either with sediment traps, thorium deficits, or both. Comparison of these data is obviously complicated by the fact that different experiments measured flux at different depths, but trying to summarise the results of all of the studies that reported particle fluxes might be helpful. Moreover, when the authors state on Page 9 Paragraph 2 that "That being said, EIFEX was the exception. Significant changes in export production were not found in any of the other OIF experiments", it should be made clear that only a subset of all OIF experiments was actually designed in such a way that an enhancement of downward particle flux could be detected (especially given the short duration of several experiments).*

→ We now compare carbon flux from all the studies in Table 5. Please refer our Response (1).

**Reviewer #2**

**-General Comments:**

☐13 **(1)** *The presentation of results from previous experiments seems too much like a catalog of data, but there is no thorough discussion on why the outcomes of the experiments were so different, and what has been learned from these experiments. (2) Further, given that KIFES is planned to take place in the Southern Ocean, it is not obvious to the reader how the detailed presentation of results from experiments carried out in other oceanic basin is relevant here.*

→ (1) We have created a new "Table 2 Summary of OIF experiments; objective, significant results, and limitation" summarizing Section 2 including OIF design, biogeochemical response, and limitations (page 29–30 of this response). We have also modified manuscript with new Section "2.6 Significant results and limitations in previous OIF experiments" to discuss why the outcomes of the experiments were so different, and what has been learned from these experiments as follows:

- 2.6 Significant results and limitations in previous OIF experiments

"To understand how various physical and biogeochemical properties response in HNLC regions by artificially adding iron, previous OIF experiments have been conducted with various objectives (Table 2). These various objectives have contributed to having idea to find out optimal conditions that have potential capacity to efficiently sequester carbon by studying whether organic carbon produced by iron enrichment is retained into deep ocean export flux or recycled in the water column by grazing pressure and remineralization in each HNLC regions (Smetacek et al., 2012). To test iron hypothesis, initial artificial OIF experiments have focused on whether iron supply limits phytoplankton growth in HNLC regions based on "bottom up" approach and confirmed increased phytoplankton biomass by showing maximum drawdown of $p$CO$_2$ by 130 ppm in SEEDS-1 and increase in primary production by 1800 mg C m$^{-2}$ d$^{-1}$ in IronEx-2 (Figure 7e and f) (de Baar et al., 2005; Boyd et al., 2007). Massive phytoplankton bloom was due to abrupt rise in diatom (Coale et al., 1996; Boyd et al., 2000). Especially, in "bottom up" approach, there were multiple efforts to detect export fluxed to deep layer of iron-induced massive

phytoplankton bloom to confirm the second condition of iron hypothesis (Bidigare et al., 1999; Charette and Buesseler, 2000; Coale et al., 2004; Smetacek et al., 2012). EIFEX only showed significant export carbon to deep layer of 3000 m by aggregate formation with highly fast sinking rates (Table 5) (Smetacek et al., 2012). In the case of OIF experiments with low amounts of carbon fluxed below mixed layer in spite of highly increased phytoplankton production in the water column, OIF experiments were focused on recycled carbon which is determined by grazing pressure (SEEDS-2) and bacterial remineralization (SERIES) based "top down" approach (Boyd et al., 2004; Tsuda et al., 2007). Relatively slight increase in primary production to iron addition (~500 mg C $m^{-2}$ $d^{-1}$), occurred during OIF experiments such as SAGE, LOHAFEX, designed to confirm biogeochemical response by added iron in very low silicate concentrations (<~2 nM) (Table 2) (Coale et al., 2004; Harvey et al., 2010; Martin et al., 2013)."

→ (2) The reason we are producing a comprehensive review of previous OIF experiments is to lay an efficient groundwork for new projects by determining the advantages and disadvantages, successes, and failures of earlier efforts. We hope that understanding the history of OIF experimentation will lead to more efficient experimental strategies and designs, which will in turn produce successful OIF experiments through the selection/adoption of useful approaches and tools that have already had verifiable success in the field.

Contrast to the previous experiments, the idea that the KIFES project was developed was initiated from deep-sediment core information obtained in the eastern Bransfield Basin. The paleoclimate team at Korea Polar Research Institute (KOPRI) found geological evidence of intensive organic carbon burial in the sediments (Yoo et al., 2016), which removes atmospheric $CO_2$, in the eastern Bransfield Basin on the Antarctic Peninsula. The diatomaceous ooze layer was well preserved in the buried sediments of the Bransfield Basin (Bahk et al., 2003; Kang et al., 2003; Bak et al., 2015), and represents the fast sinking of diatoms within a short time. Scientists at KOPRI suspect that enhancement of the diatom flux may be related to input of bioavailable iron that controls phytoplankton population by allowing efficient use of surface nutrients. In addition, this unique increase in diatom production, the fast sinking rate of the organic matter, and the remarkably well preserved organic carbon sediments in this area, suggest the existence of a strong 'biological pump (i.e., significant export production)'. This type of 'bottom-up' approach (see potential for a surface source by looking at the sedimentary evidence) has not been considered in the location selection for previous experiments. Therefore, it is expected that OIF in diatom-dominated eastern Bransfield Basin will be effective for carbon export. Please refer Section "5.1 Background for future KIFES suggestion" (page 25–26 of this response).

14 *In the same line of thought, the rationale for artificial vs. natural iron experiments could also be discussed.*

→ Good point. We have added the rationale for artificial vs. natural iron experiment in 3.2 Section as follows:

- 3.2 International law of the sea to OIF

"Nevertheless, as these small-scale OIF experiments have demonstrated considerable potential for easily and

efficiently reducing atmospheric $CO_2$ levels, physical/biogeochemical/ecological models and natural (long-term) iron fertilization experiments have been studied in an effort to overcome some of the limitations of short-term iron-addition experiments and to predict the effect of long-term and large-scale fertilization (Aumont and Bopp, 2006; Blain et al., 2007; Denman, 2008; Pollard et al., 2009)."

"Natural OIF experiments also showed much higher carbon sequestration rates than the small-scale OIF experiments (Morris and Charette, 2013), suggesting that there may be scaling or timing issues in the smaller experiments that preclude simple scaling-up as a prediction tool (see discussion in Section 4)."

→ We also have added the rationale for artificial vs. natural iron experiment in 4 Section. Please refer our Response (2).

15 *Overall, model studies are poorly represented in this review. Given that C sequestration estimates, as well as large scale and long term impacts of OIF are mostly determined through model studies, it might be relevant to mention them and how additional experiments might help constrain such models (see also comments below).*

→ Thank you for pointing out this deficiency. We have added the model results for carbon sequestration estimates in Section 3.2 as follows:

- 3.2 International law of the sea to OIF

"Earlier simplistic global biogeochemical models suggested that massive fertilization could draw down atmospheric $CO_2$ by as much as 107 ppm in 100 years (Joos et al., 1991; Peng and Broecker, 1991; Sarmiento and Orr, 1991; Kurz and Maier-Reimer, 1993). Recent global models with a more realistic ecosystem and biogeochemical cycles predict values closer to 33 ppm drawdown in atmospheric $CO_2$. These results suggest that the amount of carbon sequestration resulting from OIF would represent only a modest offset, a contribution less than 10 % for the range of IPCC future emissions scenarios (Aumont and Bopp, 2006; Denman, 2008). Natural OIF experiments also showed much higher carbon sequestration rates than the small-scale OIF experiments (Morris and Charette, 2013), suggesting that there may be scaling or timing issues in the smaller experiments that preclude simple scaling-up as a prediction tool (see discussion in Section 4)."

We have also added modelling results in our discussion of side effects in Section 3.1. Please refer our Response (3).

**-Specific Comments:**

16 *p. 7, line 28: the authors explain Fv/Fm but the term is used much earlier in the text. I suggest shifting the explanation to the first time Fv/Fm is used. I am also not sure that the description as written is very useful for people outside the field.*

→ Done.

**17** *p.8, lines 16-28: Given the large differences in mixed layer depth between experiments, I would suggest the authors also discuss mixed layer integrated chlorophyll stocks as these better reflect the real biomass built up (i.e. standing stocks accumulated during EIFEX were similar to those for SEEDS even though concentrations were an order of magnitude lower).*

→ Thank you for suggestion. We have added some sentences about integrated chlorophyll stocks in Section 2.4 as follows:

- 2.4 Biogeochemical responses

"However, added iron influences the growth of phytoplankton from surface to euphotic depth because added iron is mixed within the mixed layer by physical processes (Coale et al., 1998). Although maximum chlorophyll-a concentration during SEEDS-1 (22 mg m$^{-3}$) was much higher than EIFEX (3.16 mg m$^{-3}$), mixed layer integrated chlorophyll-a concentrations was similar with ~250 mg m$^{-2}$. There were distinct differences between mixed layer integrated chlorophyll-a concentration and surface chlorophyll-a concentration. Therefore, during previous OIF experiment, to quantify the exact changes in phytoplankton biomass to iron addition, it would be important to detect the changes in integrated primary production within water column by iron added within mixed layer."

**18** *p.8, line 26: there is a mistake in the sentence ("were appeared at"?), and the message is not clear.*

→ We apologize for this confusion. We have revised the sentences explaining satellite chlorophyll-a concentration images to clearly deliver the message as follows:

- 2.4 Biogeochemical responses

"Spatial changes in chlorophyll-a concentration as a result of iron addition were detected in SOFeX-N/S using Sea-viewing Wide Field-of-view Sensor (SeaWiFS) and MODerate resolution Imaging Spectrometer (MODIS) Terra Level-2 chlorophyll-a images. The chlorophyll image at ~28 days after iron addition in the SOFeX-N showed a phytoplankton bloom distribution resembling a long thread shape (1.0 mg m$^{-3}$), while chlorophyll image at ~20 days in the SOFeX-S suggested a somewhat broader bloom pattern (0.4 mg m$^{-3}$) (Fig. 7d) (Westberry et al., 2013)."

**19** *p. 9, line 1: add "the" before "surface".*

→ Done.

**20** *p. 10 lines 17-21: More recent model studies do not show development of anoxic conditions for large-scale iron fertilization in the Southern Ocean (see for instance Oschlies et al. Biogeosciences, 7, 4017–4035, 2010; Keller et al. Nature Communications, doi: 10.1038/ncomms4304, 2013).*

→ We apologize for confusion. We have added more recent model studies in Section 3.1. Please refer our Response (3).

**21** *p.10, line 36: Change "also has" to "could also have"*

→ Done.

**22** *p. 11, line 3: Change "even though generally…" to "even though diatom species of the genus Pseudo-nitzschia were dominant numerically".*

→ Done.

**23** *p. 11, lines 20-28: I feel that the question how is somehow too easily brushed aside. This review could be used to discuss protocols and relevant parameters that should be measured, applied or developed. Not all experiments followed similar protocols, or measured all parameters.*

→ We have modified manuscript. Please refer our Response (13).

**24** *p.13 lines 23-31: (1) Can the authors give a reference for the mentioned studies. (2) Further, the rationale for doing the experiment in the Bransfield Basin is not clear.*

→ (1) Thanks. We included references.

→ (2) Please find the revised KIFES parts below (page 25–26 of this response).

**25** *Table 2: It would be more useful if the authors provided with initial nutrient and DIC and the delta values (rather than the final concentrations).*

→ Done (page 31 of this response).

**26** *Figure 3. I do not understand how oxygen is part of the settling component.*

→ We have removed this path from Figure 3 (page 34 of this response).

**27** *Legend Figure 4, line 3: Change "nitrate and silicate were presented" to "nitrate and silicate were plotted"*

→ Thank you. Done.

**28** *Figure 5 legend: Change to "Picture for iron addition procedure" to "Illustration" or "Photographs of iron the addition procedure. Panels a-e taken during EIFEX and LOHAFEX*

→ Done.

**29** *Figure 5a legend: Change legend: a) Iron (II) sulfate bags*

**→** Done.

**30** *Figure 5b legend: The photograph shows the funnel where iron and HCl was poured, not the HCl.*

**→** Done.

**31** *Figure 5f: I am not sure were this picture was taken (the corresponding web page gives no information) but I find it misleading as the iron mixture is released in much lower quantities than depicted here (compare with the size of the hose in panel d taken during EIFEX) and has a different appearance too. I would recommend removing this panel, unless reliable information of its provenance can be provided.*

**→** The figure has been removed (page 36 of this response).

[revised manuscript text omitted]

---

## Author Comment (AC2) · 6 Feb 2017

The comment was uploaded in the form of a supplement:
http://www.biogeosciences-discuss.net/bg-2016-472/bg-2016-472-AC2-supplement.pdf
* * *

---

## Author Response (AR1)

**Dear The Editor and Anonymous Reviewers,**

First of all, we greatly appreciate the time and effort spent in reviewing of our manuscript. Unfortunately, we recently heard that the Korean Iron Fertilization Experiment in the Southern Ocean (KIFES) project, which was the inspiration for this review, has lost its funding. However, we here at Korea Polar Research Institute, as well as our domestic and international collaborators, fully intend to seek other sources of funding to revitalize the KIFES project. We apologize in advance that our responses to reviewer questions on KIFES are necessarily vague due to the current state of uncertainty surrounding the future of the project **(Reviewer 1- Comments  8,  9, 10, 11 and Reviewer 2- Comments 13-2, 24)**. Nevertheless, we believe that this manuscript is still worthy of publication as a detailed review of the history of ocean iron fertilization experiment designs and results. We have changed the title as follows: **"Ocean Iron Fertilization Experiment: Past-Present-Future looking to a future Korean Iron Fertilization Experiment in the Southern Ocean (KIFES) Project"** and still provide design guidelines for KIFES even though it is not presently funded.

We provide our responses (plain text with blue and red colors) to all comments (*italic text*) below.

**- Response to Reviewer Comments -**

**Reviewer #1**

**-General Comments:**

**1** *Downward carbon fluxes can be quantified using diverse metrics. The really important one from the point of view of geoengineering would be the amount of carbon sequestered below the depth of deepest winter mixing in the study region, which most previous OIF did not measure. The article frequently uses terms such as "efficiency of OIF at reducing atmospheric CO2", but the authors never define clearly what they mean by this. The efficiency of the biological carbon pump can be quantified using several approaches, but from a geoengineering point of view the efficiency is less important that the absolute amount. The article would be more helpful if the authors defined clearly which metrics really matter. Moreover, it would be useful if the authors more explicitly assessed which of the experiments conducted to date were actually capable of detecting an enhancement of export if it had occurred (based on duration of the experiment relative to the phases of the bloom and the type of measurements that were taken), which of these did find a response in particle flux (e.g. EIFEX, SERIES), and how to what depth the carbon flux was followed.*

→ Thank you for pointing this out. To reflect the Reviewer's suggestion, we added the following to define the "effectiveness of OIF" in the Introduction:

- 1 Introduction (from line 40, page 2 to line 2, page 3):

"To evaluate whether OIF has potential as a geoengineering strategy for carbon sequestration, not only the amount of carbon fixed by phytoplankton at the ocean surface but also the amount of carbon sequestered to the deep ocean must be considered in determining the effectiveness of OIF (Buesseler and Boyd, 2003)."

→ Based on the Reviewer's comment, we have created a new "Table 5" (Please refer lines 1–5, page 36) that includes where available: absolute magnitude of export carbon flux; measurement depth; and the methods applied to detect an enhancement of export. Using "Table 5", we also have created new Section "2.5 Assessment of export carbon flux" to more explicitly assess the enhancement of carbon flux in the manuscript as follows:

- 2.5 Assessment of export carbon flux (from line 21, page 9 to line 30, page 10):

[revised manuscript text omitted]

**2** *The previous OIF differed significantly in experimental design, especially in terms of patch size, duration, location, and also in terms of which measurements were taken. I found the discussion of these aspects in Section 3.2 rather unsatisfying: especially since the authors are in advanced stages of planning a new experiment, what have they concluded from this literature about how best to design an OIF? What are their recommendations in terms of best patch size, minimum duration, and which measurements are required to quantify the effect on carbon sequestration? I think that discussion of these points is important, especially since the authors are clearly interested in persuading the scientific (and, presumably, wider) community that their proposed experiment will provide answers about the scope for geoengineering via OIF. The clear conclusion that they do appear to have drawn is that the experiment should be located inside an eddy. However, to accurately measure downward carbon flux out of the patch at the depth of maximum winter mixing will require a large patch to ensure that sediment traps potentially several hundred metres below the surface are not at too high a risk to actually miss possible particle fluxes. What is their conclusion about the minimum duration that is needed? Given the results of SERIES, SEEDS-2, EIFEX, and LOHAFEX, it would seem to me that one should aim at between 35 and 40 days post-fertilisation. Further, what recommendations can be made about measurement approaches to quantify carbon fluxes? An important point to me is that having multiple redundant methods is very important, e.g. thorium profiles, frequent deployments of sediment traps at multiple depths (ideally neutrally buoyant traps), and high-frequency measurements of properties such as pCO2 and O2:Ar ratios. It also strikes me that autonomous platforms should play a much greater role in future OIF than they have in the past, e.g. a combination of gliders and Lagrangian floats equipped with biogeochemical sensors. Especially bio-optical sensors such as fluorescence and backscatter can be extremely useful to help constrain downward particle fluxes and their vertical and horizontal variations.*

→ We agree. More discussion is necessary. As suggested by Reviewer, we have moved this Section to "4 Future: Considerations for designing future OIF experiments" to supplement discussion for designing a future OIF experiment that would maximize the effectiveness of OIF. We have revised Section 4 to consider the methods for iron addition (
[revised manuscript text omitted]

**3** *The discussion of possible unintended side-effects could be similarly improved by trying to draw clearer conclusions rather than just summarising results from the previous literature. For example, it seems to me that the main conclusion about domoic acid is that it is very variable regardless of fertilisation, with the cited Smith et al. paper actually reporting higher per-cell quotas from natural than from artificially fertilised waters (the cited Trick et al. paper relied on bottle incubations and extrapolations based on claims about likely bloom size made by geoengineering companies on an internet site). Moreover, while a degree of oxygen consumption would certainly result from OIF, the sentence that "Box model solutions have further suggested that anoxic conditions may develop after OIF" is quite misleading: the cited reference is actually a much more realistic 3-dimensional model that only found anoxia developing in part of the western Indian Ocean, and only after many years of sustained complete nutrient utilisation in the Southern Ocean. This is probably a significantly more extreme scenario than could be achieved in practice, suggesting that anoxic conditions are actually quite unlikely. Conversely, increased production of other relevant gases, such as N2O, is clearly an important concern (though the discussion of DMS could do with some reference to the fact that its role in climate seems to be rather more complex than originally thought).*

→ We apologize for the confusion. Based on the Reviewer's comments, we have provided conclusions about side effects based on recent modeling results and have revised Section "3.1 Environmental side effects" as follows:

- 3.1 Environmental side effects (from line 13, page 11 to line 21, page 13):

[revised manuscript text omitted]

**-Specific Comments:**

**4** *Abstract Line 10, and page 10 final paragraph Line 1: make it clear that these side- effects are possible side effects, and that changes in community composition may have unintended consequences.*

→ We have made it clear by changing "including side effects" to "including possible side effects" in the Abstract (Please refer line 24, page 1) and "The changes of phytoplankton community after iron addition discussed in Section 2.4 also has unintended consequences" to "The changes of phytoplankton community composition after iron addition discussed in Section 2.4 may also have unintended consequences" (Please refer lines 3–4, page 13).

**5** *Page 4 Paragraph 3: > and < signs for latitude are the wrong way round.*

→ We corrected ">" and "<" signs for latitude (Please refer lines 21–22, page 4).

**6** *Page 12 final paragraph: given the large number and large scale of natural mesoscale blooms in HNLC regions (e.g. due to iceberg-derived iron), I think it is fair to say that the risks to the environment from small-scale OIF experiments is very small indeed, and I think that the authors should be prepared to make that case. The risks of large-scale OIF for geoengineering purposes are the risks that are not understood, and small-scale studies are what we therefore need to undertake at this point to assess these risks better.*

→ The reviewer is correct. We have revised Section "3.2 International law of the sea to OIF" by rephrasing sentences as follows:

- 3.2 International law of the sea to OIF (lines 1–8, page 14):

"However, this effort has not been able to move forward because we have little knowledge about the potential magnitude of possible side effects related to large-scale geoengineering OIF. It remains difficult to extrapolate findings from the small-scale OIF experiments because the environmental/ecosystem side effects from these miniature studies are themselves quite variable and not yet clearly understood. However, presently available studies do indicate that the known side effects from small-scale studies are themselves small-scale. It therefore seems reasonable that we should continue to undertake small-scale studies to better assess these risks and so lay the groundwork for evaluating the potential efficacy and impacts of large-scale OIF as a geoengineering solution to anthropogenic change."

**7** *Page 14 Paragraph 2: Sentence starting "To data . . ." should read "To date, the only OIF experiment . . ."*

→ Done (Please refer line 17, page 17).

**8** *Page 14 Section 4.2.3: What do the authors mean by "rehearsal"? Will they add only a tracer, such as SF6,*

*or will iron be added as well?*

→ We apologize for the confusion. By "rehearsal", we intended to indicate that we would conduct hydrographic surveys outside/inside the eddy structure in the eastern Bransfield Basin and employ drifting-buoys without iron addition prior to the actual (eddy structure) OIF experiment. Basically, we would not plan to add $SF_6$ as a chemical tracer in the KIFES project. We deleted the word and revised the KIFES section (Please refer from line 11, page 17 to line 30, page 19).

**9** *Page 14 Section 4.2.4: As I indicated in one of my general comments, I think that future OIF could benefit greatly from using autonomous platforms, such as gliders, equipped with biogeochemical sensors. If this is not planned at present, I would urge the project leaders to consider their use.*

→ Thank you. Yes. We would plan to use autonomous platforms (Please refer line 10, page 19).

**10** *Page 15 Section 4.2.5: What is the second stage of KIFES?*

→ We meant the second 5-years of the KIFES project (2021–2025). Given the current status of KIFES funding, we deleted mention of the second stage of KIFES.

**11** *In Figure 4, the authors could consider marking the study region proposed for KIFES.*

→ Thank you for your comment. However, we do not mark the study region proposed for KIFES project due to its halt (Please refer the letter at first page of our responses).

**12** *Figure 8 provides a summary of carbon flux related data for two experiments, EIFEX and SOIREE (though Fig 8b is referred to in the context of IronEx-2 in the text). Several other experiments did report comparable data, either with sediment traps, thorium deficits, or both. Comparison of these data is obviously complicated by the fact that different experiments measured flux at different depths, but trying to summarise the results of all of the studies that reported particle fluxes might be helpful. Moreover, when the authors state on Page 9 Paragraph 2 that "That being said, EIFEX was the exception. Significant changes in export production were not found in any of the other OIF experiments", it should be made clear that only a subset of all OIF experiments was actually designed in such a way that an enhancement of downward particle flux could be detected (especially given the short duration of several experiments).*

→ We now compare carbon flux from all the studies in new "Table 5". Please refer our Response (1).

**Reviewer #2**

**-General Comments:**

13 *(1) The presentation of results from previous experiments seems too much like a catalog of data, but there is no thorough discussion on why the outcomes of the experiments were so different, and what has been learned from these experiments. (2) Further, given that KIFES is planned to take place in the Southern Ocean, it is not obvious to the reader how the detailed presentation of results from experiments carried out in other oceanic basin is relevant here.*

→ (1) We have created a new "Table 2 Summary of OIF experiments; objective, significant results, and limitation" summarizing Section 2 including OIF design, biogeochemical response, and limitations (from line 1, page 32 to line 5, page 33). We have also modified manuscript with new Section "2.6 Significant results and limitations in previous OIF experiments" to discuss why the outcomes of the experiments were so different and what has been learned from these experiments as follows:

- 2.6 Significant results and limitations in previous OIF experiments (from line 32, page 10 to line 10, page 11):

"To understand how various physical and biogeochemical properties response to artificial iron addition in HNLC regions, previous OIF experiments have been conducted with various objectives (Table 2). These various objectives have contributed to develop ideas/approaches to find optimal conditions that have potential capacity to efficiently sequester carbon (Smetacek et al., 2012). To test iron hypothesis, initial artificial OIF experiments (e.g., SEEDS-1 and IronEx-2) have focused on whether iron supply limits phytoplankton growth in HNLC regions and have confirmed increases in phytoplankton biomass by showing maximum drawdown of $p$CO$_2$ by 130 ppm in SEEDS-1 and in primary production by 1800 mg C m$^{-2}$ d$^{-1}$ in IronEx-2 (Fig. 7e and f) (de Baar et al., 2005; Boyd et al., 2007). Massive phytoplankton bloom was due to rapid increase in diatom production (Coale et al., 1996; Boyd et al., 2000). There were multiple efforts to detect deep export production from surface iron-induced massive phytoplankton bloom, as the second step of iron hypothesis (Bidigare et al., 1999; Charette and Buesseler, 2000; Coale et al., 2004; Smetacek et al., 2012). EIFEX only showed significant export carbon to deep layer of 3000 m by aggregate formation with highly fast sinking rates (Table 5) (Smetacek et al., 2012). Despite highly increased phytoplankton production in the mixed layer by OIF experiments (e.g., SEEDS-1, SOFeX-N/S, SERIES, and SEEDS-2), export production was relatively low. Thus, the study focus was on high bacterial remineralization (SERIES) and/or grazing pressure (SEEDS-2) in the upper water columns (Boyd et al., 2004; Tsuda et al., 2007). Relatively slight increase in primary production to iron addition (~500 mg C m$^{-2}$ d$^{-1}$) occurred in SAGE and LOHAFEX experiments, which were designed to investigate biogeochemical responses to iron addition in very low silicate concentrations (<~2 nM) (Table 2) (Coale et al., 2004; Harvey et al., 2010; Martin et al., 2013)."

→ (2) The reason we are producing a comprehensive review of previous OIF experiments is to lay an efficient groundwork for new projects by determining the advantages and disadvantages, successes, and failures of earlier efforts. We hope that understanding the history of OIF experimentation will lead to more efficient experimental

strategies and designs, which will in turn produce successful OIF experiments through the selection/adoption of useful approaches and tools that have already had verifiable success in the field.

Contrast to the previous experiments, the idea that the KIFES project was developed was initiated from deep-sediment core information obtained in the eastern Bransfield Basin. The paleoclimate team at Korea Polar Research Institute (KOPRI) found geological evidence of intensive organic carbon burial in the sediments (Yoo et al., 2016), which removes atmospheric $CO_2$, in the eastern Bransfield Basin on the Antarctic Peninsula. The diatomaceous ooze layer was well preserved in the buried sediments of the Bransfield Basin (Bahk et al., 2003; Kang et al., 2003; Bak et al., 2015), and represents the fast sinking of diatoms within a short time. Scientists at KOPRI suspect that enhancement of the diatom flux may be related to input of bioavailable iron that controls phytoplankton population by allowing efficient use of surface nutrients. In addition, this unique increase in diatom production, the fast sinking rate of the organic matter, and the remarkably well preserved organic carbon sediments in this area, suggest the existence of a strong 'biological pump (i.e., significant export production)'. This type of 'bottom-up' approach (see potential for a surface source by looking at the sedimentary evidence) has not been considered in the location selection for previous experiments. Therefore, it is expected that OIF in diatom-dominated eastern Bransfield Basin will be effective for carbon export. Please refer Section "5.1 Background - Bransfield Basin" (Please refer from line 12, page 17 to line 5, page 18).

**14** *In the same line of thought, the rationale for artificial vs. natural iron experiments could also be discussed.*

→ Good point. We have added the rationale for artificial vs. natural iron experiment in Section 3.2 as follows:

- 3.2 International law of the sea to OIF (lines 26–30, page 13):

"Nevertheless, as these small-scale OIF experiments have demonstrated considerable potential for easily and efficiently reducing atmospheric $CO_2$ levels, physical/biogeochemical/ecological models and natural (long-term) iron fertilization experiments have been studied in an effort to overcome some of the limitations of short-term iron-addition experiments and to predict the effectiveness of long-term and large-scale fertilization (Aumont and Bopp, 2006; Blain et al., 2007; Denman, 2008; Pollard et al., 2009)."

- 3.2 International law of the sea to OIF (lines 35–38, page 13):

"Natural OIF experiments also showed much higher carbon sequestration rates than the small-scale OIF experiments (Morris and Charette, 2013), suggesting that there may be scaling or timing issues in the smaller experiments that preclude simple scaling-up as a prediction tool (see discussion in Section 4)."

→ We also have added the rationale for artificial vs. natural iron experiment in Section 4 as follows:

- 4 Future: Considerations for designing future OIF experiments (lines 14–16, page 15):

"Blain et al. (2007) explained that the higher carbon sequestration effectiveness of natural OIF experiments compared to artificial OIF experiments partly resulted from the slow and continuous iron addition that occurs in

the natural environment."

- 4 Future: Considerations for designing future OIF experiments (from line 40, page 16 to line 1, page 17):

"Furthermore, long-term observation period covering the later stage of bloom development during natural OIF experiments has made it possible to obtain high carbon sequestration effectiveness (Blain et al., 2007; Pollard et al., 2009)."

**15** *Overall, model studies are poorly represented in this review. Given that C sequestration estimates, as well as large scale and long term impacts of OIF are mostly determined through model studies, it might be relevant to mention them and how additional experiments might help constrain such models (see also comments below).*

→ Thank you for pointing out this deficiency. We have added the model results for carbon sequestration estimates in Section 3.2 as follows:

- 3.2 International law of the sea to OIF (lines 30–38, page 13):

"Earlier simplistic global biogeochemical models suggested that massive fertilization could draw down atmospheric $CO_2$ by as much as 107 ppm in 100 years (Joos et al., 1991; Peng and Broecker, 1991; Sarmiento and Orr, 1991; Kurz and Maier-Reimer, 1993). Recent global models with a more realistic ecosystem and biogeochemical cycles predict values closer to 33 ppm drawdown in atmospheric $CO_2$. These results suggest that the amount of carbon sequestration resulting from OIF would represent only a modest offset, a contribution less than 10 % for the range of IPCC future emissions scenarios (Aumont and Bopp, 2006; Denman, 2008). Natural OIF experiments also showed much higher carbon sequestration rates than the small-scale OIF experiments (Morris and Charette, 2013), suggesting that there may be scaling or timing issues in the smaller experiments that preclude simple scaling-up as a prediction tool (see discussion in Section 4)."

We have also added modelling results in our discussion of side effects in Section 3.1 as follows:

- 3.1 Environmental side effects (lines 33–34, page 11):

"Simulated Southern Ocean large-scale iron fertilization has suggested that enhancement of $CH_4$ emission would offset only <1 % of the resulting carbon sequestration (Oschlies et al., 2010)."

- 3.1 Environmental side effects (lines 2–7, page 12):

"Model estimates suggested that potential $N_2O$ production on longer timescales (6 weeks) would subsequently offset by 6–12 % increased carbon reduction benefits resulting from remineralization of additional carbon fixed during SOIREE (Law and Ling, 2001). This estimate is in the same range as the $N_2O$ offset of 6–18 % suggested by an earlier modeling study (Jin and Gruber, 2003) and the 5–9 % suggested by a more recent modeling study investigating the effectiveness of long-term and large-scale Southern Ocean OIF (Oschlies et al., 2010)."

- 3.1 Environmental side effects (from line 39, page 12 to line 2, page 13):

"Early studies using box model solutions have further suggested that anoxic conditions may develop after OIF (Sarmiento and Orr, 1991). However, OIF-induced reductions in oxygen concentration based on more sophisticated and realistic models have been smaller resulting in well-oxygenated end-conditions rather than oceanic anoxia (Oschlies et al., 2010; Keller et al., 2014)."

**-Specific Comments:**

**16** *p. 7, line 28: the authors explain Fv/Fm but the term is used much earlier in the text. I suggest shifting the explanation to the first time Fv/Fm is used. I am also not sure that the description as written is very useful for people outside the field.*

→ Done (Please refer lines 29–30, page 6 and line 29, page 7).

**17** *p.8, lines 16-28: Given the large differences in mixed layer depth between experiments, I would suggest the authors also discuss mixed layer integrated chlorophyll stocks as these better reflect the real biomass built up (i.e. standing stocks accumulated during EIFEX were similar to those for SEEDS even though concentrations were an order of magnitude lower).*

→ Thank you for suggestion. We have added some sentences about integrated chlorophyll stocks in Section 2.4 as follows:

- 2.4 Biogeochemical responses (lines 31–37, page 8):

"However, influence of iron addition on the phytoplankton growth covers from surface to euphotic depth as added iron is mixed within the mixed layer by physical processes (Coale et al., 1998). Although maximum chlorophyll-a concentrations during SEEDS-1 (~22 mg m$^{-3}$) were much higher than EIFEX (~3.2 mg m$^{-3}$), mixed layer integrated chlorophyll-a concentrations were similar to ~250 mg m$^{-2}$. There were distinct differences between mixed layer integrated chlorophyll-a concentration and surface chlorophyll-a concentration. Therefore, during previous OIF experiments, to quantify the exact changes in phytoplankton biomass to iron addition, it would be important to detect the change in integrated primary productions within MLDs."

**18** *p.8, line 26: there is a mistake in the sentence ("were appeared at"?), and the message is not clear.*

→ We apologize for this confusion. We have revised the sentences explaining satellite chlorophyll-a concentration images to clearly deliver the message as follows:

- 2.4 Biogeochemical responses (lines 25–30, page 8):

"Spatial changes in chlorophyll-a concentration as a result of iron addition were detected in SOFeX-N/S using Sea-viewing Wide Field-of-view Sensor (SeaWiFS) and MODerate resolution Imaging Spectrometer (MODIS) Terra Level-2 chlorophyll-a images. The chlorophyll-a image at ~28 days after iron addition in the SOFeX-N showed a phytoplankton bloom distribution resembling a long thread shape (~1.0 mg m$^{-3}$), while chlorophyll-a image at ~20

days in the SOFeX-S suggested a somewhat broader bloom pattern (~0.4 mg m$^{-3}$) (Fig. 7d) (Westberry et al., 2013)."

**19** *p. 9, line 1: add "the" before "surface".*

→ Done (line 38, page 8).

**20** *p. 10 lines 17-21: More recent model studies do not show development of anoxic conditions for large-scale iron fertilization in the Southern Ocean (see for instance Oschlies et al. Biogeosciences, 7, 4017–4035, 2010; Keller et al. Nature Communications, doi: 10.1038/ncomms4304, 2013).*

→ Thanks for the information. We have added more recent model studies in Section 3.1. Please refer our Response (3).

**21** *p.10, line 36: Change "also has" to "could also have"*

→ We have changed "also has" to "may also have" (also refer Response (4) for this revision).

**22** *p. 11, line 3: Change "even though generally…" to "even though diatom species of the genus Pseudo-nitzschia were dominant numerically".*

→ Done (Please refer lines 11–12, page 13).

**23** *p. 11, lines 20-28: I feel that the question how is somehow too easily brushed aside. This review could be used to discuss protocols and relevant parameters that should be measured, applied or developed. Not all experiments followed similar protocols, or measured all parameters.*

→ We have modified manuscript. Please refer our Response (13).

**24** *p.13 lines 23-31: (1) Can the authors give a reference for the mentioned studies. (2) Further, the rationale for doing the experiment in the Bransfield Basin is not clear.*

→ (1) Thanks. We included references (lines 23–25, page 17).

→ (2) Please find the revised KIFES parts. Please refer our Response (13-2).

**25** *Table 2: It would be more useful if the authors provided with initial nutrient and DIC and the delta values (rather than the final concentrations).*

→ Done (lines 1–6, page 34).

**26** *Figure 3. I do not understand how oxygen is part of the settling component.*

→ We have removed this path from "Figure 3" (lines 1–2, page 41).

**27** *Legend Figure 4, line 3: Change "nitrate and silicate were presented" to "nitrate and silicate were plotted"*

→ Thank you. Done (line 13, page 37).

**28** *Figure 5 legend: Change to "Picture for iron addition procedure" to "Illustration" or "Photographs of iron the addition procedure. Panels a-e taken during EIFEX and LOHAFEX*

→ Done (line 18, page 37).

**29** *Figure 5a legend: Change legend: a) Iron (II) sulfate bags*

→ Done (line 18, page 37).

**30** *Figure 5b legend: The photograph shows the funnel where iron and HCl was poured, not the HCl.*

→ Done (line 19, page 37).

**31** *Figure 5f: I am not sure were this picture was taken (the corresponding web page gives no information) but I find it misleading as the iron mixture is released in much lower quantities than depicted here (compare with the size of the hose in panel d taken during EIFEX) and has a different appearance too. I would recommend removing this panel, unless reliable information of its provenance can be provided.*

→ The figure has been removed (lines 1–2, page 43).

[revised manuscript text omitted]

---

## Referee Report (RR1)

**Review of Yoon *et al. Ocean Iron Fertilization Experiments**

I am sorry to hear that the funding for the proposed Korean Iron Fertilisation Experiment has fallen through, and I wish the authors good luck in trying to secure future funds to conduct this study.

While the initial justification of the manuscript was the presentation of this proposed experiment, I believe that publication of the manuscript even in the absence of secured funding is appropriate. The paper seeks to provide a useful synthesis of previous experiments, as well as to provide design guidelines for future experiments. A key point is that the previous experiments still leave us with many open questions as to the response of marine ecosystems to iron addition.

The authors have made quite extensive changes to the manuscript, which go some way towards addressing my previous comments. For example, the two new tables (now Table 2 and Table 5) help a lot to provide a better overview of the different iron fertilisation studies.

However, I still feel that the actual manuscript itself could be significantly improved if the authors could draw clearer, more specific conclusions to highlight the gaps in our understanding, and identify specifically what is needed to address these gaps. In its present form, I find that the manuscript is often rather long-winded (*e.g.* explaining very basic concepts such as the $^{234}$Th method, or mentioning in the text which research ships were used for which experiments) without actually drawing clear and specific conclusions. Numerous reviews of iron fertilisation experiments have been published previously – albeit not of all 13 experiments that have now been conducted – and what I really miss in the present paper is a coherent argument and set of justified conclusions emerging from this review. In principle, I think this manuscript would be a good opportunity to examine specifically what key questions remain unanswered and what specifically would need to be done to address them. The manuscript at present still describes a lot of results, but I think without really drawing more than rather vague conclusions.

I will first go through the manuscript section-by-section to point out where and why I think the manuscript could be improved. I will end the review proposing a possible structure for the authors to re-work their manuscript into a more useful review that manages to move the field forward further. I hope that this is helpful for the authors – I do not intend to do this effort down, because I think that a review such as this, coupled with a proposal for a new experiment could be genuinely useful. However, I fear that the present manuscript misses a valuable opportunity, and I think that in its present form the manuscript will not achieve as much impact as it might if the authors attempt a more thorough discussion and evaluation, rather than just listing of results.

1. The abstract does not include any summary of the main conclusions that the authors have drawn from their synthesis.

2.  It is not clear to me what the purpose of the entire Section 2.1 is. Ostensibly, the authors attempt to describe the objectives of each experiment, but I'm not convinced that this information is really important enough to merit a 2-page section, nor am I convinced that the authors explain these objectives terribly clearly. The section mixes general introduction to each experiment (*e.g.* descriptions of the oceanographic conditions of each site) with usually only a vague, general statement of purpose for the experiments (*e.g.* "To investigate the unexpected responses revealed in IronEx-1, a second OIF was conducted", or "To measure biologically-driven gas fluxes"). For some experiments, the authors only list briefly the main conclusions, but don't explain much about the objectives and design.

3. Section 2.2 is basically a long list of measurements of initial conditions for the various experiments. Many of these have been reviewed previously, and the authors don't draw any conclusions from this 1.5-page-long section. The authors have done an excellent job of summarising this information in their tables (which I think is very useful). But since the main message seems to be that OIF have been conducted under a diverse range of initial conditions, wouldn't it be better if the authors just made that point in, maybe, 1–2 paragraphs, and then focused on advancing an argument and drawing conclusions?

4. The subject matter of Section 2.3 is an important one: how many additions of iron should be made, and how should the fertilised patch be traced? What I miss in this section is to see the authors actually draw conclusions from these results. For example, are multiple additions the right way to design an experiment? At what intervals should they be made? The experiments listed vary a lot in duration, so simply listing the total number of additions, rather than the intervals at which they were made, seems uninformative. Likewise, it might be useful to see some discussion of the amount of iron added per square kilometre of patch – I believe that the additions for the various experiments varied a lot also on a per-area basis, but this is not discussed. Can we determine an optimum rate of application from the various experiments? If not, is this something that warrants further experimental work, and if so, of what kind? Likewise, the discussion of tracing methods just boils down to listing the fact that $SF_6$ and buoys can be used, without drawing conclusions that could be formulated as practical recommendations. For example, drifting buoys in previous experiments were found to ultimately leave the patch, I believe due to wind forcing, and one of the recommendations of this paper is that experiments should be conducted that last upwards of 30 days, under which conditions multiple $SF_6$ injections might become more necessary.

5. Section 2.4 I think mostly re-hashes just the main findings of previous experiments, and again does so in an often rather long-winded way (*e.g.* the information about spatial pattern of the SOFEX blooms doesn't seem to contribute much). All in all, this section does not really build up an integrated understanding of the biogeochemical responses to iron addition, but reads rather more like a selective catalogue of results picked rather haphazardly from the different experiments: *e.g.* the difference in maximum chlorophyll between SEEDS-I and SEEDS-II is pointed out, but this isn't part of a broader discussion of chlorophyll levels across the different OIF and what their causes might be. In the next paragraph, SEEDS-I and EIFEX are compared with max chl-a concentration and integrated chl-a stock, but this is not compared with other experiments. My point here is not to ask the authors to add specifically a discussion of chlorophyll, I am just pointing this out as an instance of where the paper rather shies away from reaching interesting new insights. The entire section then

simply ends, without any attempt to conclude anything from all this information, aside from pointing out in the middle of the penultimate paragraph that integrated primary production ought to be monitored in future experiments.

6. I had previously requested that the authors make explicit definitions of terms like "efficiency" and "sequestration" in the context of carbon fluxes, but these terms are still used loosely without specific definitions. I think this is a significant problem when discussing the planning of future iron fertilisation work, as it is critical to achieving the stated objectives to ensure that the correct measurements are made, and this can only be done if we define clearly what we need to measure. I would suggest that the authors refer to the Lampitt et al. paper in the 2008 special issue of the Philosophical Transactions (same issue as the cited Smetacek & Naqvi paper), titled "Ocean fertilization: a potential means of geoengineering?". This paper discusses explicit definitions of terms such as carbon sequestration with reference to the depth of winter mixing, and discusses how they can be measured.

7. Section 2.5 again leaves me rather unsatisfied: the authors present a list of findings, but don't make any argument or properly discuss reasons for the divergent outcomes. As a result, by the end of the section, it is not clear to the reader why "the effectiveness of iron addition on this component of the biological pump remains a question". In other words, the authors should be using this section to discuss why the previous 13 experiments have not managed to yield clear answers to the effectiveness of carbon sequestration. The answer lies in a combination of experiment duration, measurement methods (measuring only shallow or also deeper fluxes), patch size (with a very small patch, I suspect that deeper traps might miss export that may be occurring because the plume of sinking particles is confined to such a small area), and patch movement (tracking deep export is easy in a stationary patch, but very hard in a large patch). Although the authors do draw these conclusions in a very general sense, saying at the end that future experiments need to last long enough, fertilise a largeish area, and use multiple methods, the justification of publishing a review paper like this one must lie in a more detailed analysis of how which of the previous OIF were unable to achieve these requirements. To do this, the authors would need to delve more into the details of each experiment, rather than spending most of the time reviewing the basic biogeochemical responses of each experiment. Again, terms like "effectiveness of iron addition" could be defined in very specific ways, and using language as loosely as this does not help with achieving insights.

8. Section 2.6 starts as a promising paragraph (in fact, I think this could be a good introductory paragraph to the entire Section 2), but then falls flat for the same reasons pointed out above. What are we to conclude from these diverse outcomes? Which experiments may have missed an export event, or deep fluxes of sinking particles, and for what reasons? The issue of ecosystem responses and grazing is clearly an important one, but receives hardly any discussion.

9. After all this extensive review of previous OIF in Section 2, the review suddenly moves on to possible side-effects. What conclusions can be drawn from all of the data that the authors have reviewed in Section 2?

10. Section 3.1 is slightly better, as it at least ends with a proper attempt at conclusions. However, the section is again mostly a cataloguing of results from previous experiments. What would be more useful is if the authors attempted to synthesise the insights and discussions of the publications about the individual studies so that we can at try and find patterns in the large variability between studies.

11. Section 3.2 implies a discussion of the international legal situation, but the first paragraph is not about legal matters at all. The second paragraph does a decent job of summarising the legal situation. However, since the London Convention explicitly grants exemption only to scientific experiments in "coastal waters" I wonder where that would leave open-ocean experiments in practice. Of course, iron fertilising coastal waters is pointless, as everyone in the oceanographic community knows, but are the authors confident that legitimate research projects in the open ocean will get approval? Some clarification of this seems called for here, at least to the best of the authors' ability.

12. Section 4 finally attempts to put forward some practical suggestions. However, I think that this needs to be expanded on and made more specific. Simply concluding that "multiple additions of iron are more efficient" is not the same as formulating specific guidelines about fertilisation levels (e.g. minimum addition per $km^2$) and application intervals. Likewise, the third paragraph in this section ("What") is much too brief a discussion to be genuinely useful and to move the field forward. A long list of measurements is put forward without discussion of the relative merits of each, or a real discussion of the difficulty of tracking a fertilised patch, and whether we've learnt how to do this better over the 13 experiments that have been conducted. Likewise, the discussion of how to measure carbon fluxes is too indiscriminate a list to be of much use. What would really help would be an evaluation of what the various measurements contribute to our understanding, what their pitfalls are (especially, for example, surface-tethered sediment traps, which are probably better avoided in favour of their neutrally buoyant counterparts), and, again, where in the water column we really need to be measuring. The fact that the sections concludes in saying that carbon fluxes need to be monitored using "both trap fluxes and/or $^{234}$Th deficiency" is somewhat worrying: surely, one of the lessons from previous experiments is that both measurements really are needed, and that traps must be at multiple depths so that we can track both the export out of the surface as well as the sequestration below the depth of winter mixing? In discussing possible candidate regions, it would be useful if the authors proposed specific regions, rather than just making general observations such as recommending "regions with high silicate concentrations and low copepod abundances".

13. The manuscript ends with the design proposal for KIFES. Even though KIFES currently has no funding, it is an experiment that the authors clearly hope to conduct in more or less this form in the future. Therefore, I think it is valuable to have this section in the manuscript. What might be useful is if the authors used this opportunity to explain in slightly more detail how the design of KIFES will avoid the problems encountered in earlier experiments, i.e. explain the specifics of duration, patch size, measurement methods, etc.

There were additionally numerous minor points about specific phrasing that I felt could be improved throughout the manuscript. However, I think there is little point in listing all of these, since I suspect that the authors will need to re-write this manuscript in a relatively

major way anyway. I actually believe that this might not necessarily involve too much work: what I would recommend, however, is that the authors think carefully about what their main message is from reviewing all of the previous studies, and that they seek to systematically craft these arguments instead of rather indiscriminately listing previous results.

I would suggest that a revised structure might look something like this:

1. Overall introduction to OIF and statement of the purpose of the present paper.

2. A brief overview of each of the experiments, maybe written as a historical narrative (1–2 pages) that describes how the various experiments built on each other and what the main hypotheses were that each was designed to address. This section can be used to highlight the different physical and biogeochemical conditions of each experiment, and also highlight the main biogeochemical responses and findings. This experiment-by-experiment approach might help to give the reader are more integrated understanding of each experiment than the current parameter-by-parameter approach used throughout Section 2. The section could then end with a paragraph highlighting the key outstanding questions that future experiments (including KIFES) need to address. These include questions relating to the amount of carbon sequestration, trace gas production, and plankton community shifts.

2. A detailed discussion of each of these questions in turn, highlighting why the previous 13 experiments have failed to reach consensus, and what needs to be done to move our understanding forward. This needs to be an issue-based discussion, for example going into real detail about how carbon fluxes need to be measured, which experiments managed to take these measurements (but failed to find effects), which experiments were maybe hampered by size and/or duration, and also a discussion of the uncertainties relating to each method (*e.g.* methodological problems with sediment traps, issues with the thorium technique, problems with the estimation of net community production from $O_2$:Ar ratios, etc.).

3. A section that reaches specific conclusions and makes recommendations about the design of future experiments.

4. Specific proposal for KIFES and description of the logic for conducting this in the Bransfield Strait. Do the authors have some preliminary altimetry images to show stable eddies in this region?

Finally, I think the authors might not want to neglect the ecological arguments for conducting OIF experiments: terrestrial ecologists have learnt a large amount about the functioning of terrestrial ecosystems from conducting nutrient manipulation experiments. OIF are effectively our only way of manipulatively testing similar hypotheses about marine pelagic ecosystems, and I believe that OIF could be a useful tool purely for scientific purposes, regardless of geoengineering feasibility.

End of review.

---

## Referee Report (RR2)

[revised manuscript text omitted]

Halogenated volatile organic compounds (HVOCs, such as CH$_3$Cl, CH$_3$Br, and CH$_3$I), well known for their ability to destroy ozone in the lower stratospheric ozone and marine boundary layer (Solomon et al., 1994), were also measured during  OIF experiments (Wingenter et al., 2004; Liss et al., 2005). During SOFeX-N , iron addition  HVOC were complicated: CH$_3$Cl concentrations remained unchanged; CH$_3$Br concentrations increased by ~14 %; and while  CH$_3$I concentrations decreased by ~23 % (Wingenter et al., 2004), CH$_3$I concentrations increased 2-fold  EisenEx (Liss et al., 2005). Therefore, as  DMS , further study is needed to understand  HVOC response.

Decomposition of iron addition-enhanced biomass may cause decreased oxygen concentrations in the subsurface waters (Williamson et al., 2012). Although mid-water oxygen depletion has not been reported during the OIF experiments to date, . Early studies using  model  suggested that anoxic conditions may develop after  (Sarmiento and Orr, 1991). However, more sophisticated and realistic models  OIF-

[revised manuscript text omitted]

a climate mitigation strategy and a means to gain carbon credits (Chisholm et al., 2001; Buesseler and Boyd, 2003). However,  effort  move forward because  related to large-scale  OIF.

5  However, presently available studies  side effects  small-scale  It therefore  continue  small-scale studies to better assess these risks and so lay the groundwork for evaluating the potential efficacy and impacts of large-scale OIF as a geoengineering solution to anthropogenic change.

With potential risks and benefits of OIF,  legal  raised to support the further study

10 and increase understanding of OIF (Williamson et al., 2012). At present, large-scale and/or commercial OIF experiments are banned by international regulation. The international Convention on the Prevention of Marine Pollution by Dumping of Wastes and Other Matter (London Convention, 1972) and Protocol to the London Convention (London Protocol, 1996) placed legal restrictions on dumping of wastes and other matter that cause hazard, harm, and damage in the ocean and/or interfere with the marine environment. In 2007, the London Convention & Protocol (LC/LP) scientific groups released a statement of concern

15 about ocean fertilization and recommended that ocean fertilization activities be evaluated carefully to ensure that such operations were not contrary to the aims of the LC/LP. Under the LC/LP, commercial activities are prohibited, and only 'small-scale' legitimate scientific research in 'coastal waters' is allowed (Resolution LC-LP.1 (2008), 2008). LC/LP also developed an assessment framework for scientific ocean fertilization research to be applied on a case-by-case basis founded on the agreed definition and compliance with the aims and objectives of Resolution LC-LP.1 (2008) (Fig. 9) (Assessment Framework for

20 Scientific Research Involving Ocean Fertilization, 2010). This framework demands preliminary scientific research to  for OIF experimentation as transparent/reasonable scientific rationale/purpose and risk analysis undertaken using parameters such as problem formulation, site selection, exposure assessment, effects assessment, risk characterization, and risk management must be provided (Assessment Framework for Scientific Research Involving Ocean Fertilization, 2010). Monitoring is also required an integral component of all approved (i.e., legitimate) scientific research activity to assess

25 ecological impacts and to review actual versus intended geoengineering benefits (ACE CRC, 2015). In October 2013, LC/LP categorized artificial ocean fertilization as marine geoengineering, thereby prohibiting operational OIF activities, but enabling OIF scientific research that meets the permit conditions through the environmental assessment framework (Resolution LP.4 (8), 2013).

30 **4 Future: Considerations for designing future OIF experiments**

Scientific research on OIF has focused on improving our understanding of the effectiveness, capacity, and risks of OIF as an atmospheric $CO_2$ removal strategy. Although the first OIF experiments took place more than twenty years ago, the legal and economic aspects of such a strategy in terms of international laws of the sea and carbon offset markets are not yet clear (ACE CRC, 2015). It is therefore of paramount importance that future OIF experiments continue to focus on the effectiveness

35 and capacity of OIF as a means of reducing of atmospheric $CO_2$, but in doing so should carefully consider iron addition  (i.e., 'How'), tracking methods and  parameters (i.e., 'What'), location (i.e., 'Where'), timing (i.e., 'When'), and duration (i.e., 'How long') to build on the results of OIF experiments, develop our understanding of the magnitude and sources of uncertainties, and in so doing build confidence in our ability to reproduce results.

How: The first consideration for a successful OIF experiment lies in the strategy/approach to maintain added iron within

40 the upper mixed layer. During the first OIF experiment, IronEx-1, the patch was fertilized with acidified iron(II) sulfate

 to  target concentrations of 3.6 nM because iron-enrichment bottle incubation experiments performed in deck- incubators using ocean water suggested maximum phytoplankton growth rates in response to iron additions of 1–2 nM (Fitzwater et al., 1996). However,  concentrations of iron  rapidly decreased  to 0.25 nM  four days. Further, the magnitude of the  biogeochemical response was

5  than bottle enrichment experiments suggested (Coale et al., 1998).  to sustain enhanced iron concentrations in patches,  repeated (2 to 4) iron infusions has been  in all OIF experiments except SEEDS-1 and FeeP (de Baar et al., 2005; Boyd et al., 2007). Like IronEx-1, SOIREE showed  losses in dissolved iron after the first  infusion  due to horizontal dispersion, and also noted loss due to oxidation of the additional iron(II) to iron(III) (Bowie et al., 2001).  SOIREE demonstrated that four additions of iron with intervals of about 3

10 days led to a  of both dissolved and particulate iron within the mixed layer   fast reduction combined with an increase in the concentration of iron-binding ligands . Both EIFEX and SOFeX-S also found that multiple iron(II) infusions allowed iron to persist in the mixed layer longer than its expected oxidation  relatively low oxidation rates were related to a combination of photochemical production, slow oxidation, and possibly organic complexation (Croot et al., 2008). Blain et al. (2007) explained

15 that the higher carbon sequestration effectiveness of natural OIF experiments compared to artificial OIF experiments partly resulted from the slow and continuous iron addition that occurs in the natural environment. Short-term infusions of large amounts of iron tend to lead the substantial loss of artificially added iron. Therefore, to increase  the  of carbon  exported to  iron supplied, multiple additions of iron  more efficient.

What: The second consideration for a successful OIF experiment is effective tracing of fertilized patch including

20 detection of carbon sequestration (Buesseler and Boyd, 2003) and monitoring of possible side effects. OIF side effects include emission of climate-relevant gases such as $N_2O$ and DMS that directly contribute to warming and cooling of the environment, respectively (Law, 2008). During IronEx-1, the fertilized patch was subsequently traced with large variety of physical-biogeochemical techniques  such as GPS and ARGO equipped drifting buoys, $SF_6$, Fv/Fm ratio, $pCO_2$, and chlorophyll fluorescence using underway sampling systems, and satellite images (Martin et al., 1994; Coale et al., 1998). As

25 IronEx-1 provided potential evidence to support Martin's iron hypothesis by showing an increase in phytoplankton bloom with iron enrichment, many subsequent OIF experiments adopted the tracing methods introduced by IronEx-1, and were similarly able to detect environmental changes through the observation of both physical and biogeochemical parameters before and after iron addition (Martin et al., 1994; Coale et al., 1996; Boyd et al., 2000; Tsuda et al., 2005; Coale et al., 2004; Boyd et al., 2004; Smetacek et al., 2012). Carbon export fluxes can be detected using $^{234}$Th, $^{13}$C$_{org}$, free-drifting sediment traps, beam-

30 transmissometers,  UVPs (Table 5) (Bidigare et al., 1999; Nodder et al., 2001; Boyd et al., 2004; Buesseler et al., 2004; Coale et al., 2004; Aono et al., 2005; Tsuda et al., 2007; Smetacek et al., 2012; Martin et al., 2013). In particular, it is possible to evaluate the temporal evolution of iron-induced export carbon fluxes into deeper waters  the thorium deficiency method  sediment trap  used during previous OIF experiments (Table 5).  transmissometer,  camera that photographs particles,

[revised manuscript text omitted]

(a)

(b)

[Figure]

[Figure]

[Figure]

[Figure]

[Figure]

(a)

[Figure]

(b)

[Figure]

[Figure]

[Figure]

[Figure]

---

## Referee Report (RR3)

**Review of Yoon et al. Ocean Iron Fertilization Experiments: Past–Present–Future looking to a future Korean Iron Fertilization Experiment in the Southern Ocean (KIFES) Project**

The revised version of the manuscript by Yoon *et al.* is very much improved. The authors have made extensive changes that address all of the points that I raised in my previous review. The main part of the manuscript is a review of previous experiments, and this section is now far more clearly and logically structured, and overall the authors are doing a far better job of actually evaluating the results of the studies they are reviewing.

I now only have a small number of minor requests that I would like the authors to undertake before publishing the manuscript. These points are as follows:

1. Most importantly, the authors are still only giving vague directions as to the depths at which particle export needs to be measured. These measurements, however, are absolutely crucial if we are to judge the effectiveness of iron fertilisation as a means of carbon sequestration, and there has been substantial debate in the particle flux literature precisely about this point. Unfortunately, the authors only talk vagues of "intermediate and deep waters" (e.g. Page 4 Line 1, P 26 L 28, and P 27 L 6; clear directions on depths are also lacking in Section 3.1). Measurement depths for particle flux absolutely need to be spelled out explicitly and as much as possible standardised in future experiments so that meaningful comparisons can be made. In the abstract, and on P 29 L 18, the authors indicate that in KIFES, they would want to place one sediment trap "within the mixed layer and anotehr below it". This is absolutely not appropriate: sediment traps are known to perform poorly within the mixed layer, so the shallowest depth for a trap should be just *below* the mixed layer, so that mixed layer export can be measured. The second critical depth for a sediment trap is the depth of deepest winter-time convective mixing, which can be estimated from Argo float data for a given study area. In LOHAFEX, that was how the depth of the deep sediment traps at around 400 m was chosen. My recommendation would be that the authors make clear recommendations that particle flux must be measured just below the mixed layer (*i.e.* 10–20 m below) and at the depth of the winter mixed layer. I am open to alternative suggestions from the authors, but whatever they intend to do absolutely must be spelled out clearly, and the importance of these depths should be explained properly in the manuscript. As it is described at present, I would have real concerns as to whether KIFES would actually collect the correct measurements. It is critical that this manuscript makes appropriate, and explicit, recommendations in this regard.

2. The authors in several places talk about "effectiveness" of aOIF, but I don't remember that being properly defined in the manuscript. As above, I think this does need to be defined, and in my view an appropriate definition would be the amount of additional carbon exported below the winter mixed layer depth as a result of iron addition. The importance of other metrics, such as the amount of carbon sequestered relative to the amount of iron added, might also be mentioned in this context.

3. In the abstract, Line 27, the authors might want to briefly spell out what these questions are. The abstract can probably be shortened by editing the language carefully throughout. I would also recommend that the depths at which carbon sequestration should be monitored

are explicitly spelled out in the abstract (and, I reiterate, these depths are *not* inside the mixed layer and [somewhere] below the mixed layer).

4. In the first paragraph of the introduction, I am quite strongly of the opinion that the authors should also state that there is an urgent need to reduce global greenhouse gas emissions.

5. P 3 L 12: why is "ocean fertilisation" italicised?

6. P 6 L 10: Presumably you mean "By the end of the 20th century"?

7. P 10 L 26: Correct spelling of the name is "Behrenfeld"

8. P 16 L35: What is the difference between "tracking" and "quantifying" export flux? I think one will do. Again, please do be explicit throughout the manuscript as to what depths you are thinking about.

9. P 18 L 12: Saying that thorium and sediment traps are "of limited use" in determining the fate of POC is not appropriate; the implication of saying "of limited use" is that these measurements are not very useful. The authors should re-phrase this to say "and, therefore, these methods should ideally be complemented with additional techniques that can measure particle stocks at high depth resolution throughout the water column".

10. P 18 L 17: The UVP, and also transmissometer and other optical measurements, actually do not give a particle flux as such, they show the particle concentration. There are ways to estimate fluxes from these measurements using assumptions about particle sinking rates, but in the first instance they provide information about stocks. The authors might also want to refer to the study by Briggs et al. (2011) *High-resolution observations of aggregate flux during a sub-polar North Atlantic spring bloom* in Deep-Sea Research 1. This paper provides a very nice example of using backscattering and chlorophyll sensors to gain high-resolution data about a particle flux event in a Lagrangian study, so a similar scenario to an iron fertilisation.

11. P 20 L 10: This sentence isn't quite clear. What does the 20% refer to: a 20% increase of the total SO DMS flux? Or a 20% increase in the 2% of the SO?

12. P 21 L 12: The statement about domoic acid levels needs a reference. In several places throughout this paragraph, the authors rely on the conclusions reached by Trick et al. 2010 – however, I don't think that this study gives a reliable guide to possible impacts of *Pseudonitzschia* blooms. Trick et al. chiefly conducted bottle experiments that yielded a range of several orders of magnitude in cell quotas of DA. Their conclusions were then chiefly based on extrapolating their highest measured cell quota in a bottle incubation with quite unrealistic estimates of likely surface biomass levels to estimate a possible in-water DA concentration. If the authors do wish to cite this paper, I would strongly recommend that these caveats to their conclusions should be pointed out explicitly.

13. P 24 L1: better to say "along with sufficient levels of solar radiation" instead of using "receipt"

14. P 26 Line 39: MIT and WHOI are entirely separate institutions, despite their collaborative PhD programme.

---

## Referee Report (RR4)

**Ocean Iron Fertilization Experiments: Past–Present–Future looking to a future Korean Iron Fertilization Experiment in the Southern Ocean (KIFES) Project**

Joo-Eun Yoon[1], Kyu-Cheul Yoo[2], Alison M. Macdonald[3], Ho-Il Yoon[2], Ki-Tae Park[2], Eun Jin Yang[2], Hyun-Cheol Kim[2], Jae Il Lee[2], Min Kyung Lee[2], Jinyoung Jung[2], Jisoo Park[2], Jiyoung Lee[1], Soyeon Kim[1], Kitae Kim[2*], and Il-Nam Kim[1*]

[1]Department of Marine Science, Incheon National University, Incheon 22012, Republic of Korea
[2]Korea Polar Research Institute, Incheon 21990, Republic of Korea
[3]WHOI, MS 21, 266 Woods Hold Rd., Woods Hole, MA 02543, USA

*Correspondence to*: Il-Nam Kim (ilnamkim@inu.ac.kr) and Kitae Kim (ktkim@kopri.re.kr)

**Abstract.** Since the start of the industrial revolution, human activities have caused a rapid increase in atmospheric $CO_2$ concentrations, which have, in turn, had an impact on climate leading to global warming and ocean acidification. Various approaches have been proposed to reduce atmospheric $CO_2$ concentrations. The 'Martin (or Iron) Hypothesis' suggests that ocean iron fertilization (OIF) could be an effective method for stimulating oceanic carbon sequestration through the biological pump in iron-limited, high-nutrient, low-chlorophyll regions. To test the Martin hypothesis, 13 artificial OIF (aOIF) experiments have been performed since 1990 in the Southern Ocean (seven experiments), in the subarctic Pacific (three experiments), in the equatorial Pacific (two experiments), and in the subtropical Atlantic (one experiment). These aOIF field experiments have demonstrated that primary production can be significantly enhanced by the artificial addition of iron. However, the effectiveness of export production (i.e., the export of carbon from surface waters into intermediate/deep waters) revealed by the aOIF experiments was unexpectedly low compared to that achieved by natural phytoplankton blooms, except in the Southern Ocean European Iron Fertilization Experiment. These results, including possible side effects (e.g., changes in climate-relevant gas emissions and an increase in toxic phytoplankton species) have been debated amongst those who support and oppose aOIF experimentation, but many questions remain. In the context of increasing global and political concerns associated with climate change, it is valuable to examine the validity and usefulness of the aOIF experiments. To maximize the effectiveness of aOIF experiments under international OIF regulations in the future, we suggest a design that incorporates several conditions. (1) Experiments are conducted in the center of an eddy structure when grazing pressure is low and silicate levels are high (e.g., in the case of the Southern Ocean, at the south of polar front during the early summer). (2) Shipboard observations are made during a minimum of ~40 days, with multiple iron injections (iron infusions of ~2,000 kg at least three times, with an interval of ~10–15 days, to fertilize a patch of 300 km$^2$ to obtain a ~2 nM concentration). (3) The iron fertilized patch is traced using both physical (e.g., a drifting buoy) and biogeochemical (e.g., sulfur hexafluoride and the Fv/Fm ratio, where Fm is the maximum chlorophyll fluorescence yield and Fv is the difference between Fm and the minimum chlorophyll fluorescence yield) tracers. (4) A neutrally buoyant sediment trap system and water-column derived $^{234}$Thorium method are employed at two depths (one within the mixed layer and another below it), with autonomous profilers equipped with underwater video profiler and transmissometer to estimate accurately the carbon export flux. (5) The side effects on marine/ocean ecosystems are monitored, including the production of climate-relevant gases (e.g., $N_2O$, dimethyl sulfide, and halogenated volatile organic compounds) and an increase in the abundance of toxic phytoplankton species (e.g.,

[revised manuscript text omitted]
., 1996) and, therefore, this basin became the next region selected for an aOIF experiment (Frost, 1996). With the hypothesis that iron and light availability may act as key factors that control phytoplankton dynamics, community structure, and grazing in the SO, the Southern Ocean Iron Release Experiment (SOIREE) (Table 1 and Fig. 4a), which was the first *in situ* aOIF experiment performed in the SO, took place in February 1999 (13 days) in the Australasian-Pacific sector (Boyd et al., 2000).

The following year, a second aOIF experiment in the SO, EisenEx ('Eisen' means iron in German), was performed in November within an Antarctic Circumpolar Current eddy in the Atlantic sector (Smetacek, 2001). This region is considered to have a relatively high iron supply, which is supported by dust inputs and sea-ice melt (de Baar et al., 1995; Quéguiner et al., 1997; Smetacek et al., 2002). EisenEx was designed to test the hypothesis that atmospheric dust, an important source of iron in ocean environments, might have led to a dramatic increase in ocean productivity during the LGM due to the relief of iron-limiting conditions for phytoplankton growth.

In addition to iron availability, the supply of silicate is also considered to be an important factor controlling PP in the SO. Silicate-requiring diatoms, which are large-sized phytoplankton, have an important role in the biological pump and are responsible for ~75% of the annual PP in the SO (Tréguer et al., 1995). The silicate concentrations in the SO have a decreasing northward gradient, in particular, on either side of the Antarctic Polar Front (PF), with low silicate concentrations (<5 μM) in the sub-Antarctic waters north of the PF (<61°S) and high silicate concentrations (>60 μM) to the south of the PF (Fig. 4c). Therefore, to address the potential for iron and silicate interactions to regulate the diatom bloom, two aOIF experiments were conducted during January–February 2002 in two distinct regions: the Southern Ocean iron experiment-north (SOFeX-N) and -south (SOFeX-S) of the PF (Table 1) (Coale et al., 2004; Hiscock and Millero, 2005). In these two experiments, it was hypothesized that conditions that provided sufficient silicate and iron would lead to high diatom production, while sufficient iron alone would not lead to a diatom bloom (Coale et al., 2004).

Two years later, the Surface Ocean Lower Atmosphere Study (SOLAS) Air–Sea Gas Exchange (SAGE) experiment was conducted during March–April 2004 (15 days) in sub-Antarctic waters, which are HNLC and low silicate concentration waters (HNLCLSi). The aim was to determine the response of phytoplankton dynamics to iron addition in an HNLCSi region (Fig. 4c) (Law et al., 2011). SAGE was designed with the assumption that the response of phytoplankton blooms to artificial iron addition could be detected by enhanced air-sea exchanges of climate-relevant gases (e.g., $CO_2$ and dimethyl sulfide (DMS)) (Harvey et al., 2010; Law et al., 2011).

These early aOIF experiments demonstrated clear increases in phytoplankton biomass, but the association with export production (i.e., exported carbon from the surface waters into intermediate/deep waters) was obscure (de Baar et al., 2005; Boyd et al., 2007). Therefore, to determine if aOIF could increase export production, EIFEX was conducted during February–March 2004 in a cyclonic eddy core near the PF (Fig. 5). Because it was designed to investigate export production, EIFEX was a much longer experiment (39 days) compared to earlier studies (~28 days or less) (Smetacek et al., 2012).

To trace the fate of an iron-stimulated phytoplankton bloom and deep carbon export, the Indo-German iron fertilization experiment (LOHAFEX; 'Loha' is iron in Hindi) was conducted during January–March 2009 (40 days), also in a PF cyclonic eddy (Smetacek and Naqvi, 2010; Martin et al., 2013).

**2.1.3 Subarctic North Pacific**

By the 20th century, the subarctic NP was the only HNLC region in which an aOIF experiment had not been performed (Table 1) (de Baar et al., 2005; Boyd et al., 2007). 
[revised manuscript text omitted]
, also increased simultaneously, substantially increasing the grazing effect (~50%) (Coale et al., 1996). However, the grazing force of the increased biomass was insufficient to suppress the diatom bloom over eight days early in the IronEx-2 experiment (Table 4) (Coale et al., 1996). The iron-induced diatom bloom began to decline after day ~8 of the experiment. The decline was probably associated with the combined effects of both the elevated grazing pressure and the onset of nutrient depletion (i.e., limitation in silicate and/or iron) (Cavender-Bares et al., 1999; Boyd, 2002).

To determine whether the biological pump (i.e., export production) is enhanced after iron addition, the export flux of particulate organic carbon (POC) can be estimated using, individually or in combination, chemical tracers such as the natural radiotracer thorium-234 ($^{234}$Th; half-life = 24.1 days) and the stable carbon isotope of particulate organic matter ($\delta^{13}C_{org}$), sediment traps, transmissometers, and underwater video profilers (UVPs) (Table 5) (Bidigare et al., 1999; Nodder et al., 2001; Boyd et al., 2004; Buesseler et al., 2004; Coale et al., 2004; Aono et al., 2005; Tsuda et al., 2007; Smetacek et al., 2012; Martin et al., 2013). The $^{234}$Th radionuclide has a strong affinity for particles, and the extent of $^{234}$Th removal in the water column is indicative of the export of POC associated with surface PP out of the mixed layer (Buesseler, 1998). IronEx-2 was the first aOIF experiment in which the POC flux was estimated (Bidigare et al., 1999). The $^{234}$Th deficiency from the surface to 25 m was measured in the iron-fertilized patch to estimate iron-stimulated export production in the surface layer (Table 5). However, no $^{234}$Th measurements were made in the unfertilized patch for comparison, and no measurements in the deep ocean were undertaken to demonstrate deep carbon export (Bidigare et al., 1999).

**2.4.2 Southern Ocean**

As in the EP IronEx-1/-2 experiments, there were initial rapid increases in the Fv/Fm ratio within 24 hours of iron addition, indicating that phytoplankton growth was mainly limited by iron availability. Maximum values of the Fv/Fm ratio ranged from 0.5 (SOFeX-N and LOHAFEX) to 0.65 (SOIREE and SOFeX-S) (Table 4 and Fig. 7a). However, the time taken to reach the maximum Fv/Fm ratio was usually longer than ~10 days, i.e., much slower than in IronEx-1/-2 (~2 days) (Boyd and Abraham, 2001; Gervais et al., 2002; Coale et al., 2004; Smetacek et al., 2005; Peloquin et al., 2011; Martin et al., 2013). The slower response time in the SO compared to the EP might be attributed to the colder temperatures (~5°C vs. ~24°C) and/or the deeper MLDs (~60 m vs. ~30 m) (Figs. 6c and d), which were indicative of active physical mixing (Boyd and Abraham, 2001; Boyd, 2002).

The aOIF experiments in the SO recorded >2-fold increases in chlorophyll-a concentrations compared to initial levels (<0.7 mg m$^{-3}$), and maximum values between 1.25 mg m$^{-3}$ (LOHAFEX) and ~3.8 mg m$^{-3}$ (SOFeX-S) were obtained after artificial iron additions (Table 4 and Fig.7c). Satellite observations were used to investigate the changing spatial and temporal distribution of chlorophyll-a concentrations in response to iron fertilization in the fertilized patches compared to the surrounding waters (Boyd et al., 2000; Coale et al., 2004; Boyd et al., 2005; Westberry et al., 2013). For example, spatial changes in chlorophyll-a resulting from SOFeX-N/S iron addition were detected using Sea-viewing Wide Field-of-view Sensor (SeaWiFS) and MODerate-resolution Imaging Spectrometer (MODIS) Terra Level-2 chlorophyll-a images. The chlorophyll-a image on day 24 after iron addition in the SOFeX-N showed a phytoplankton bloom distribution resembling a

~~long thread with 10-fold higher concentrations (1.0 mg m⁻³) than the surrounding waters (0.1 mg m⁻³), while a chlorophyll-a image on day 20 of SOFeX-S suggested a somewhat broader bloom pattern (0.5 mg m⁻³), with concentrations elevated ~5-fold over the surrounding levels (~0.1 mg m⁻³) (Fig. 7d) (Westberry et al., 2013).~~

Following artificial iron enrichment in the SO, $\Delta$PP ranged from 360 (SAGE) to ~1356 mg C m⁻² d⁻¹ (SOFeX-N) (Table 4 and Fig. 7e).  SOIREE, EisenEx, and SOFeX-N/S, the PP increased continuously throughout the duration of the experiments (Boyd et al., 2000; Gall et al., 2001a; Gervais et al., 2002; Coale et al., 2004). However, in EIFEX, SAGE, and LOHAFEX there was a significant increase in PP for ~10 (SAGE) – 20 (EIFEX) days in response to the iron addition, and decreasing trends after day ~12 (SAGE) – 25 (EIFEX) due to various , such as  export  (e.g., EIFEX), lateral dilution with surrounding waters (e.g., SAGE), and high grazing pressure and  bacterial respiration (e.g., LOHAFEX) (Boyd, 2002; Gervais et al., 2002; Buesseler et al., 2004; Coale et al., 2004; Peloquin et al., 2011; Smetacek et al., 2012; Martin et al., 2013).

Using both microscopes and high-performance liquid chromatography pigment analysis, changes in phytoplankton community affected by iron addition have also been investigated. SO iron additions have resulted in blooms of  (Boyd et al., 2007). During SOIREE and EisenEx, the dominant phytoplankton community shifted from pico- and nano-phytoplankton (e.g., pico-eukaryotes and prymnesiophytes) to micro-phytoplankton (i.e., diatoms) (Gall et al., 2001a; Gervais et al., 2002). In SOFeX-S and EIFEX, diatoms were already the most abundant group prior to iron addition (Coale et al., 2004; Hoffmann et al., 2006). The contribution of large diatoms became especially clear in EIFEX where ~97% of the phytoplankton bloom was attributed to  (Smetacek et al., 2012). However, no taxonomic shift toward diatom-dominated  communities (<5% of total phytoplankton community)  observed  SAGE and LOHAFEX, which were conducted under silicate-limited conditions (Harvey et al., 2010; Peloquin et al., 2011; Martin et al., 2013; Ebersbach et al., 2014). Although SOFeX-N was conducted under low silicate conditions, the diatom biomass increased remarkably making up ~44% of the total phytoplankton community (Coale et al., 2004). This result was partly influenced by the temporary relief of silicate limitation through lateral mixing of the iron-fertilized waters with surrounding waters, with relatively higher silicate concentrations (Coale et al., 2004).

Iron-mediated increases in PP resulted in a significant uptake in macronutrients and $p$CO$_2$ throughout the aOIF experiments in the SO (except for SAGE) (Table 3, Figs. 7b and f). The $\Delta$NO$_3^-$ ranged from -3.5 μM (e.g., SOFeX-S) to -1 μM (e.g., EisenEx) and $\Delta p$CO$_2$ ranged from -38 μatm (e.g., SOIREE) to -7 μatm (e.g., LOHAFEX). Although SOFeX-S had a somewhat greater $\Delta$NO$_3^-$ (-3.5 μM) and $\Delta p$CO$_2$ (-36 μatm) than EIFEX ($\Delta$NO$_3^-$: -1.6 μM and $\Delta p$CO$_2$: -30 μatm) both ==results suggested that diatoms were abundant in the two experiments.== However, the smaller $\Delta$Si observed during SOFeX-S (-4 μM, compared to EIFEX -11 μM) was associated with a decrease in silicification (i.e.,  frustule thickness ) of the dominant diatom species ( *Fragilariopsis* sp. (Twining et al., 2004).  EIFEX, the ratio of heavily silicified diatoms (e.g., *Thalassiothrix antarctica*) to total diatom biomass increased from 0.24 (day 0) to 0.46 (day 37) leading to the  (Hoffmann et al., 2006). Interestingly, the biogeochemical responses in SAGE were totally different from those seen in other experiments, in particular  in $\Delta$NO$_3^-$ (+3.9 μM), $\Delta p$CO$_2$ (+8 μatm), and $\Delta$DIC (+25 μM) (Table 3, Figs. 7b and f). These contrasting results were thought to be the result of entrainment through vertical and horizontal physical mixing into the iron-fertilized patch of  waters with higher  concentrations (Currie et al., 2011; Law et al., 2011).

SOIREE was the first aOIF experiment in the SO to estimate the downward carbon flux into deep waters (Fig. 3c).  comprehensive suite of methods  drifting trap, ²³⁴Th and δ¹³C$_{org}$ estimates derived from high-volume pump sampling, and a beam transmissometer (Nodder and Waite, 2001). However, no measurable change

in carbon export was observed in response to iron-stimulated PP (Table 5 and Fig. 8b) (Charette and Buesseler, 2000; Nodder and Waite, 2001; Trull and Armand, 2001; Waite and Nodder, 2001). During EisenEx, an increased downward carbon flux estimated from $^{234}$Th deficiency was observed in the iron-fertilized patch as the experiment progressed. However, there were no clear differences between in- and outside-patch carbon fluxes (Buesseler et al., 2005). During SOFeX-S, significantly enhanced POC fluxes below the mixed layer after iron enrichment were  from $^{234}$Th  (Buesseler et al., 2005).  similar to that observed in natural blooms.  SOFeX-N  equipped with transmissometers  downward carbon flux  between day ~27 and ~45 after the first iron addition (Bishop et al., 2004; Coale et al., 2004). However,  SAGE and LOHAFEX , which were conducted under silicate limited conditions (Table 3, Figs. 4c and 6f), Table 5) (Peloquin et al., 2011; Martin et al., 2013). This result was likely to be  pico-plankton , which led to rapid recycling in the mixed layer . In contrast to the other aOIF experiments, EIFEX, which was conducted within the core of an eddy,  clear evidence of carbon export stimulated by artificial iron addition (Jacquet et al., 2008; Smetacek et al., 2012). During EIFEX, the initial export flux, estimated from $^{234}$Th in the upper 100 m of the fertilized patch, was ~340 mg C m$^{-2}$ d$^{-1}$ (Table 5 and Fig. 8a) (Smetacek et al., 2012). This value remained constant for about 24 days after iron addition. Between day 28 and 32 a massive increase in carbon export flux (maximum of ~1692 mg C m$^{-2}$ d$^{-1}$) was observed in the fertilized patch, while the initial value remained constant in the unfertilized patch (Table 5 and Fig. 8a). The profiling transmissometer with high-resolution coverage confirmed this result, showing an increase in exported POC below 200 m after day 24. At least half the iron-induced biomass sank (via the formation of aggregates of diatom species, in particular 'Chaetoceros dichaeta') to a depth of 1,000 m, with a tenfold higher sinking rate (500 m d$^{-1}$), compared to the initial conditions (Smetacek et al., 2012). Significant changes in export production were not found in any of the other aOIF experiments and, therefore, the impact of artificial iron addition on this component of the biological pump needs to be resolved in future OIF experiments (Boyd et al., 2004; Smetacek et al., 2012; Martin et al., 2013).

**2.4.3 Subarctic North Pacific**

The observed increase in the Fv/Fm ratio in response to artificial iron addition in the subarctic NP suggests that the relief in iron limitation may have assisted phytoplankton growth (Table 4 and Fig. 7a). SEEDS-1/-2, which were conducted in the western basin, showed continuous increases in the Fv/Fm ratio, with a maximum value of ~0.4 approximately 10 days after the first iron addition (Tsuda et al., 2003; Tsuda et al., 2007). During SERIES, which was conducted in the eastern basin, the Fv/Fm ratio rapidly increased and reached a maximum value of 0.55 within 24 hours of the first iron addition (Boyd et al., 2005). However, the Fv/Fm ratio returned toward the initial value of <0.3 as the dissolved iron concentrations decreased to background levels (<0.2 nM) after about day 10 (Tsuda et al., 2003; Boyd et al., 2005; Tsuda et al., 2007).

Increases in chlorophyll-a concentrations were detected in the subarctic NP aOIF experiments in both basins after about the fifth day (Tsuda et al., 2003; Boyd et al., 2004; Suzuki et al., 2009). These increases were especially apparent in SEEDS-1, where they reached a maximum value of 21.8 mg m$^{-3}$ (27 times the initial value of 0.8 mg m$^{-3}$) (Table 4 and Fig. 7c). ==This augmentation was the largest among all the aOIF experiments (Tsuda et al., 2003). The dramatic chlorophyll-a increase observed during SEEDS-1 was partly attributed to the particular range of seawater temperature in the region, which was conducive to diatom growth (i.e., 8–13°C) as well as to the shallower MLD (~10 m), which provided a relatively longer surface water residence time for the additional iron (Figs. 6c and d) (Noiri et al., 2005; Takeda and Tsuda, 2005; Tsumune et al., 2005).== During SERIES, chlorophyll-a concentrations increased substantially from the initial value of 0.35 to 5 mg m$^{-3}$

over 17 days, and the second highest concentration of all aOIF experiments was recorded (Table 4 and Fig. 7c) (Boyd et al., 2004). However, on the 18th day there was a downturn in chlorophyll-a as silicate concentrations decreased to <2 μM (Boyd et al., 2005). Although SEEDS-2 was conducted under similar initial conditions to SEEDS-1 (refer to Section 2.2.3), there was a minimal increase in chlorophyll-a (i.e., maximum value of less than 3 mg m$^{-3}$) (Fig. 7c). This smaller increase was thought to be the result of extensive copepod grazing (SEEDS-2 had almost five times more copepod biomass than SEEDS-1) (Table 4) (Tsuda et al., 2007). A similar spread was seen in depth-integrated PP, which increased by 7-fold or more after various iron enrichments in the subarctic NP aOIF experiments (e.g., from 300–420 to 1,000–2,000 mg C m$^{-2}$ d$^{-1}$) (Table 4 and Fig. 7e).

Changes in the composition of phytoplankton groups were investigated in the subarctic NP aOIF experiments. In SEEDS-1 there was a shift from oceanic diatoms (e.g., *Pseudonitzschia turgidula*), with growth rates of 0.5–0.9 d$^{-1}$, to faster-growing neritic diatoms (e.g., *Chaetoceros debilis*, 1.8 d$^{-1}$) (Tsuda et al., 2005). The shift in the dominant phytoplankton species during the SEEDS-1 experiment was an important contributor to what became the greatest aOIF-induced increase in phytoplankton biomass yet recorded. During SERIES, the phytoplankton community changed from *Synechococcus* and haptophytes to diatoms, and the highest SERIES chlorophyll-a concentration (day 17) was associated with a peak in diatom abundance (Boyd et al., 2005). However, during SEEDS-2, no significant iron-induced diatom bloom was observed. Instead, pico-phytoplankton (e.g., phytoflagellates) (67% of the total community) dominated throughout the duration of the experiment due to the heavy grazing pressure on diatoms (Table 4) (Tsuda et al., 2007).

In the subarctic NP experiments, significant decreases in macro-nutrient uptake (i.e., $\Delta NO_3^-$ and $\Delta Si$), $\Delta DIC$, and $\Delta pCO_2$ in response to artificial iron addition were observed (Table 3 and Figs. 7b and f). SEEDS-1, which saw the largest increases in chlorophyll-a concentrations, also had the largest $\Delta pCO_2$ (-130 μatm) and $\Delta DIC$ (-58 μM) (Table 3 and Fig. 7f). These changes led, in turn, to the largest $\Delta NO_3^-$ (-15.8 μM) (Fig. 7b) and $\Delta Si$ (-26.8 μM) (Table 3) (Tsuda et al., 2003). The second largest increase in the chlorophyll-a concentration was observed in SERIES, where drawdowns of $pCO_2$ (-85 μatm), DIC (-37 μM), nitrate (-8.5 μM), and silicate (-13.6 μM) were recorded. During SEEDS-2, the nitrate concentration decreased remarkably from 18.4 μM to 12.7 μM after day 5; however, there was no significant change in silicate concentrations, which would have been expected as a signal of an iron-induced diatom bloom (Tsuda et al., 2007; Suzuki et al., 2009).

Despite the formation of a massive iron-induced phytoplankton bloom during SEEDS-1, there was no large POC export flux during the observation period (Table 5) (Tsuda et al., 2003; Aono et al., 2005; Aramaki et al., 2009). During SERIES and SEEDS-2, which allowed comprehensive time-series measurements of the development and decline of the iron-stimulated bloom, POC fluxes estimated by the drifting traps in the fertilized patch displayed temporal variations (Boyd et al., 2004; Aramaki et al., 2009). However, the results suggested that only a small part of the decrease in the mixed layer POC was subsequently captured by the drifting trap, and POC flux losses were mainly governed by bacterial remineralization and mesozooplankton grazing (Boyd et al., 2004; Tsuda et al., 2007).

**2.4.4 Subtropical North Atlantic**

Not much is known about the biogeochemical responses to OIF in the subtropical NA. The FeeP experiment reported that pico-plankton abundances increased after iron and phosphate additions (Rees et al., 2007); however, no other details of the biogeochemical response to iron addition in FeeP have been reported.

**2.5 Summary of the significant results from aOIF experiments**

To test the hypothesis that the addition of iron to the surface layer will effectively reduce atmospheric $CO_2$ by increasing PP and enhancing, in turn, the carbon export flux to the deep ocean, aOIF experiments have usually been conducted in HNLC regions: the EP, SO, and subarctic NP. The one exception was the FeeP experiment, which was performed in the subtropical NA. The initial environmental conditions associated with the physical and biogeochemical properties were determined at these OIF sites over 1–7 days before iron addition to allow the responses to the aOIF to be evaluated and quantified. Preliminary surveys confirmed that all sites, except FeeP in the subtropical NA, were subject to iron-limited HNLC conditions, with typical levels of iron <0.4 nM, nitrate >~10 μM, and chlorophyll-a <1 mg m$^{-3}$. The initial Fv/Fm ratios were <~0.3, suggesting that phytoplankton growth was severely iron-limited. In SEEDS-1, SOFeX-S, and EIFEX, prior to the addition of iron, the micro-phytoplankton (e.g., diatom) community accounted for half of the population and this was thought to be beneficial to the enhancement of export production. In the other experiments, pico- and nano-phytoplankton (e.g., *Synechococcus* and haptophytes) initially dominated; they are associated with rapid recycling in the mixed layer through the microbial loop rather than export production (Michaels and Silver, 1988; Coale et al., 1998; Landry et al., 2000; Boyd and Law, 2001; Gervais et al., 2002; Coale et al., 2004; Boyd et al., 2005; Tsuda et al., 2005; Hoffmann et al., 2006; Tsuda et al., 2007; Harvey et al., 2010; Martin et al., 2013).

Iron-sulfates dissolved in acidified seawater have been commonly used for artificial iron addition because they are both highly bioavailable and inexpensive. The mixture is generally released into the ship's wake over a period of 24 hours. The amount of added iron was determined so to reach a target dissolved-iron concentration (at least >1 nM) by volume (defined as the MLD × patch area). To achieve this, a wide range of 225–2,000 kg was applied. Except in IronEx-1 and SEEDS-1, the experiments used multiple (2–4) iron additions to reinforce the increased iron levels. To trace the iron-fertilized patches, physical tracers (i.e., ARGO or other GPS-tracked drifting buoys) and/or chemical tracers such as $SF_6$ were used. In addition, biogeochemical parameters, such as the Fv/Fm ratio, macro-nutrients, and $CO_2$ variables, were used to detect responses through a comparison of before and after conditions (i.e., Δ = [parameter]$_{postf}$ − [parameter]$_{pref}$). In particular, it should be noted that the Fv/Fm ratios promptly increased from <~0.3 to 0.56 (± 0.08) in the two days following iron addition, indicating a relief in the iron-limitation on phytoplankton growth. The subarctic NP SEEDS-1 experiment, which was conducted under temperature conditions ideal for diatom growth (~8°C) and with shallow MLDs (~10 m), produced the greatest changes in biogeochemical parameters.

The aOIF experiments have generally led to changes in the size of the phytoplankton community from pico and nano-phytoplankton to micro-phytoplankton. This effect was particularly noticeable as diatoms became the dominant species during IronEx-2, SOIREE, EisenEx, SEEDS-1, SOFeX-S, EIFEX, and SERIES, with micro-phytoplankton accounting for ~44% of total phytoplankton community in SOFeX-N. The shift to a diatom-dominated community appears be related to the initial availability of silicate (i.e., initial silicate was >~3 μM in all the experiments just listed). Diatom-dominated blooms induced >4.5-fold increases in chlorophyll-a concentrations and accounted for >65% of the chlorophyll-a increase (Boyd et al., 2000; Gervais et al., 2002; Coale et al., 2004; Smetacek et al., 2012). However, as silicate concentrations decreased to <2 μM due to removal by the elevated diatom abundance, the extent of diatom blooms rapidly declined. In SAGE and LOHAFEX, had low initial levels of silicate (< 2 μM), pico and nano-phytoplankton dominated communities, and diatom growth was limited by the lack of available silicate. These results suggest that to develop a massive phytoplankton bloom, a changeover to a diatom-dominated community after iron addition is needed. A necessary, but not sufficient condition, for such a change to occur is the availability of silicate. Silicate alone is not expected to be sufficient because diatom-dominated blooms with distinct increases in the chlorophyll-a concentration were not observed in all experiments with high initial silicate concentrations. IronEx-1 and SEEDS-2 had high initial silicate levels (>~4 μM), which were conducive to the

development of a diatom-dominated bloom, but the bloom was suppressed due to high grazing pressure. Taken together, the OIF results suggest that both mesozooplankton grazing rates as well as the initial silicate concentrations played a role in limiting the stimulation of diatom-dominated blooms after artificial iron enrichment.

In some experiments (IronEx-1, SEEDS-2, SAGE, and LOHAFEX) there was little change in the carbon export flux, while in others (IronEx-2, SOIREE, EisenEx, SEEDS-1, SOFeX-S, EIFEX, and SERIES) there was a >2-fold increase in PP within the mixed layer, with massive diatom-dominated blooms. However, even in the latter, changes in the carbon export flux differed from experiment to experiment. In SEEDS-1 and SOIREE there was little increase in export flux. However, it has been reported that changes in in the POC concentrations following an increase in PP can take three to four weeks (Buesseler et al., 2005), whereas these two experiments were conducted over only about two weeks, which suggests that the duration of each experiment was too short to detect downward carbon export. In SERIES, there was a distinct increase in the carbon export flux within the mixed layer (30 m), but there was no increase in the export flux below this because the abundance of heterotrophic bacteria was elevated after iron addition rapidly remineralized POC within the mixed layer (
[revised manuscript text omitted]
 this increase was caused by sampling biases (Nodder et al., 2001; Nodder and Waite, 2001). Likewise, SEEDS-1 $^{234}Th$-based POC fluxes at 50-m depth over day 9–13 were estimated to be 423 mg m$^{-2}$ d$^{-1}$, but the drifting trap only recorded 141 mg m$^{-2}$ d$^{-1}$ at 40-m depth over day 12–14, 3 times lower (Table 5) (Aono et al., 2005; Aramaki et al., 2009). This large discrepancy between the two methods might be caused by the under-sampling of POC into the drifting traps (Nodder and Waite, 2001; Aono et al., 2005).

To resolve the potential biases in traditional sediment traps, a neutrally buoyant (and freely drifting) sediment trap (NBST) was developed (Valdes and Price, 2000; Valdes and Buesseler, 2006; Lampitt et al., 2008). Through preliminary experiments conducted in June and October 1997 at the Bermuda Atlantic Time-series Study site, Buesseler et al. (2000) showed that an NBST system could reduce the invasion/inclusion of zooplankton into the trap samplers, and that NBST-based $^{234}$Th fluxes were comparable with water-column based estimates. LOHAFEX has been the only OIF experiment so far that has measured particle export using PELAGRA (Particle Export measurement using a LAGRAngian trap) sediment traps based on the NBST system (Martin et al., 2013). However, the PELAGRA sediment traps deployed below the mixed layer (at 200 m and 450 m) did not detect iron fertilization-induced carbon export even though PP did increase within the mixed layer. Water-column based $^{234}$Th measurements estimated the POC flux at a 100-m depth to be ~94 mg m$^{-2}$ d$^{-1}$, whereas the PELAGRA sediment traps estimated the flux at 200 m and 450 m to be only ~12 mg m$^{-2}$ d$^{-1}$. It should be noted that both sediment traps and water-column based $^{234}$Th measurements have a limited ability to fully scan the vertical profile of POC fluxes and, therefore, are of limited used in determining the fate of iron-induced POC in the water column.

To resolve the full column more effectively, LOHAFEX employed a UVP, which provided photographic evidence of sinking particles (particle size ≥100 μm) from the surface down to ~3,000-m depth, with ~0.2 m vertical resolution (Smetacek et al., 2010; Martin et al., 2013). Through an analysis of particle size distributions, the UVP also allowed particles to be classified into fecal pellets, aggregates, and live zooplankton. Vertical total particle volume profiles obtained from UVP indicated the maximum particle flux at a 75-m depth (~0.3 mm$^3$ L$^{-1}$), with a gradual decrease to 150 m (~0.15 mm$^3$ L$^{-1}$). Interestingly, large particles (i.e., zooplankton) were copious between 75-m and 100-m depth, suggesting that there might be high grazing pressure, which might explain the large discrepancy between the 100 m (water-column based $^{234}$Th method) and 200 m / 450 m (PELAGRA sediment trap) POC flux estimates (i.e., rather than a sampling bias in sediment trap data) (Martin et al., 2013). To continuously monitor vertical changes in POC flux following iron addition, EIFEX used a transmissometer, providing high vertical resolution (~24 data points per meter) and tracking of the iron-induced flux down to ~3,000 m, even though, unlike UVPs, transmissometers do not allow classification of particles (Smetacek et al., 2012). Improving on this method, SOFeX-N applied autonomous carbon flux explorers equipped with transmissometers, designed to float along with the currents. Three autonomous carbon flux explorers were deployed, two explored the 'iron fertilized in-patch' and one acted as a 'control' outside the patch. Carbon flux explorers could continuously monitor in the field for up to 18 months beyond the initial deployment, which allowed SOFeX-N to observe 'episodic raining' in the iron-fertilized waters (Bishop et al., 2004), indicating a high carbon export flux long after artificial iron addition.

The combination of multiple approaches is essential to the successful detection of the POC produced in response to iron addition and its fate. NBST systems (e.g., the PELAGRA sediment trap) are appropriate for quantifying the aOIF-induced POC flux in the upper waters (<~400-m depth), especially 
[revised manuscript text omitted]

Secondly, the effectiveness of aOIF may also be offset leading to changes in the ocean ecosystem following OIF, such as a decrease in dissolved oxygen and an increase in domoic acid (DA) levels. The decomposition of iron addition-enhanced biomass may cause decreased oxygen concentrations in subsurface waters (Williamson et al., 2012). Although mid-water oxygen depletion has not been reported from aOIF experiments to date, early modeling studies suggest that anoxic conditions may develop after long-term and large scale OIF (Sarmiento and Orr, 1991). However, more sophisticated and realistic models suggest that OIF produces well-oxygenated conditions, without the development of anoxic conditions, even under climate change scenarios (Oschlies et al., 2010; Keller et al., 2014). Thus, hypoxia/anoxia development in response to iron additions is unlikely to be a primary concern.

The changes in phytoplankton community composition after iron addition discussed in Section 2.4 may also have unintended consequences; for example, they could lead to potentially toxic species dominating plankton assemblages (Silver et al., 2010; Trick et al., 2010). Some aOIF experiments (e.g., IronEx-2, SOIREE, EisenEx, SOFeX-N/S, and SERIES) generated large blooms dominated by pennate diatoms belonging to the genus 'Pseudo-nitzschia' (de Baar et al., 2005; Trick

et al., 2010). Some 'Pseudo-nitzschia' species have the capacity to produce the neurotoxin DA that  detrimentally affect marine ecosystems. However, no DA was found during EisenEx and SERIES, even though 'Pseudo-nitzschia' were dominant (Gervais et al., 2002; Marchetti et al., 2008; Assmy et al., 2007). Phytoplankton samples used to estimate DA production had been stored for a long time before the analysis, for example, 12 years in IronEx-2 and four years in SOFeX-S

5 (Silver et al., 2010). Trick et al. (2010) argued that  an under-estimation in DA . This implies that accurate information about changes in DA production in response to iron addition might not be available. However,  IronEx-2, and SOFeX-S experiments  discernable changes in DA production,  (Silver et al., 2010). ==It is likely that several phytoplankton samples (e.g., *Pseudo-nitzschia* abundance: $1.3 \times 10^6$ cells $L^{-1}$ in IronEx-2 and $7.5 \times 10^4$ cells $L^{-1}$ in SOFeX-==

10 ==S) collected with a net tow were suitable to detect these changes.== During IronEx-2 and SOFeX-S, high cell abundances of '*Pseudo-nitzschia*' ($10^6$ and $10^5$ cells $l^{-1}$, respectively) combined with moderate DA  (0.05 and 1 pg DA cell$^{-1}$, respectively) produced toxin levels as high as 45 ng DA $l^{-1}$ in  and 220 ng DA $l^{-1}$ in , i.e., ==toxin levels high enough to damage marine communities in coastal waters. Therefore, it is necessary to quantify DA production in response to iron additions, with concentrated phytoplankton samples (i.e., large numbers of cells) using a net tow.== This, once again,

[revised manuscript text omitted]

Sources are Martin et al. (1994); Coale et al. (1996); Coale et al. (1998); Boyd et al. (2000); Boyd and Law (2001); Gervais et al. (2002); Tsuda et al. (2003); Boyd et al. (2004); Coale et al. (2004); Bakker et al. (2005); Boyd et al. (2005); de Baar et al. (2005); Nishioka et al. (2005); Hoffmann et al. (2006); Law et al. (2006); Blain et al. (2007); Boyd et al. (2007); Rees et al. (2007); Tsuda et al. (2007); Pollard et al. (2009); Strong et al. (2009); Harvey et al. (2010); Smetacek et al. (2012); and  Martin et al. (2013).

**Table 2.** Summary of artificial ocean iron fertilization (aOIF) experiments; objectives, significant results, and limitations.

| | Experiment | Objectives | Significant results | Limitations |
|---|---|---|---|---|
| 1 | IronEx-1 | • To test the hypothesis that artificial iron addition will increase phytoplankton productivity by relieving the iron limitation of phytoplankton in high-nutrient low chlorophyll regions | • Small increases in the Fv/Fm ratio, chlorophyll-a concentration, and primary production (PP)
 • Insignificant changes in nutrients and $pCO_2$ concentrations | • Single iron addition
 • Insufficient experimental periods to observe the full phases of biogeochemical responses from the onset to termination after iron additions
 • Micro/macro-nutrient limitations |
| 2 | IronEx-2 | • To test three hypotheses that were advanced to explain the weak biogeochemical response observed during IronEx-1 | • Dramatic changes in biogeochemical responses; close to support for Martin's hypothesis
 • Taxonomic shift toward diatom-dominated phytoplankton communities | • No export flux measurements in the deep ocean
 • Insufficient experimental duration |
| 3 | SOIREE | • To test the iron hypothesis in the Southern Ocean | • Diatom-dominated bloom
 • No measurable change in carbon export | • Insufficient experimental duration |
| 4 | EisenEx | • To test the hypothesis that atmospheric dust inputs might have led to a dramatic increase in ocean productivity during the Last Glacial Maximum due to the relief of iron-limited conditions for phytoplankton growth | • Diatom-dominated bloom
 • No clear differences in carbon flux between in-patch and outside-patch | • Light limitation by storms
 • Insufficient experimental duration |
| 5 | SOFeX-N | • To address the potential for iron and silicate interactions to regulate the diatom bloom | • Remarkable increase in diatom biomass
 • Observation of large export flux event with transmissometers | • Entrainment of dissolved silicate into the fertilized patch by physical mixing
 • No direct measurement of export fluxes with sediment traps |
| 6 | SOFeX-S | • To address the potential for iron and silicate interactions to regulate the diatom bloom | • Significantly enhanced export fluxes out of the mixed layer, but similar to those for natural blooms | • Insufficient experimental duration |
| 7 | EIFEX | • To confirm that aOIF experiments can increase export production | • Observation of all the phases of the phytoplankton bloom from onset to termination
 • Significant carbon export to deeper layers (down to 3,000 m) due to the formation of aggregates with rapid sinking rates
 • The occurrence of rapidly sinking large aggregates | |

To be continued

| | Experiment | Objective | Significant results | Limitations |
|---|---|---|---|---|
| 8 | SAGE | • To determine the response of phytoplankton dynamics to iron addition in high nutrient low chlorophyll and low silicate (HNLCLSi) regions
• To test the assumption that the response of phytoplankton blooms to artificial iron addition can be detected by the enhanced air-sea exchanges of climate-relevant gases | • No shift to a diatom-dominated community
• No detection of fertilization-induced export | • High dilution rate by small patch size |
| 9 | LOHAFEX | • To trace the fate of iron-stimulated phytoplankton blooms and deep carbon export in HNLCSi regions | • Observation of all the phases of the phytoplankton bloom from onset to termination
• No shift to a diatom-dominated community
• No detection of fertilization-induced export
• High grazing pressure and active bacterial respiration | |
| 10 | SEEDS-1 | • To investigate the relationship between phytoplankton biomass/community and dust deposition in the subarctic North Pacific (NP)
• To investigate changes in phytoplankton composition and vertical carbon flux | • A shift from oceanic diatoms to fast-growing neritic ones
• The largest changes in biogeochemical parameters of all aOIF experiments
• No detection of large POC export flux | • Single iron addition
• Insufficient experimental duration |
| 11 | SERIES | • To compare the response of phytoplankton in eastern subarctic with that in the western subarctic ecosystem
• To investigate the most significant factor that controls the beginning to the ending of the phytoplankton bloom induced by iron addition | • Observation of all phases of the phytoplankton bloom from onset to termination
• No significant increases in export fluxes below the mixed layer depth
• High bacterial remineralization and mesozooplankton grazing pressure | |
| 12 | SEEDS-2 | • To investigate the most significant factor that controls the beginning to the ending of the phytoplankton bloom induced by iron addition | • Observation of all phases of the phytoplankton bloom from onset to termination
• No shift to a diatom-dominated community
• No significant increases in export fluxes
• Extensive copepod grazing | |
| 13 | FeeP | • To investigate the impact of iron and phosphate co-limitation on PP | • Increases in pico-phytoplankton abundances | |

[revised manuscript text omitted]

(a)

[Figure]

(b)

(c)

[Figure]

[Figure]

[Figure]

(a)

[Figure]

(b)

[Figure]

[Figure]

[Figure]

(a)

(b)

SOIREE (Day 5)    SOIREE (Day 45)

Chlorophyll-a (mg m$^{-3}$)

---

## Author Response (AR2)

We provide our responses (Plain text with blue and red colors) to all comments (*italic text*) below.

**- Response to Reviewer Comments -**

**Reviewer #1**

**-General Comments:**

**1** *The abstract does not include any summary of the main conclusions that the authors have drawn from their synthesis.*

☞ We have modified Abstract (from line 21, page 1 to line 1, page 2).

**2** *It is not clear to me what the purpose of the entire Section 2.1 is. Ostensibly, the authors attempt to describe the objectives of each experiment, but I'm not convinced that this information is really important enough to merit a 2-page section, nor am I convinced that the authors explain these objectives terribly clearly. The section mixes general introduction to each experiment (e.g. descriptions of the oceanographic conditions of each site) with usually only a vague, general statement of purpose for the experiments (e.g. "To investigate the unexpected responses revealed in IronEx-1, a second OIF was conducted", or "To measure biologically-driven gas fluxes"). For some experiments, the authors only list briefly the main conclusions, but don't explain much about the objectives and design.*

☞ Thank you for pointing out this deficiency. We have revised Section 2.1 to clearly explain objectives and hypotheses of the previous artificial OIF experiments (from line 28, page 4 to line 28, page 6). We have also removed the main results from previous OIF experiments in Section 2.1.

**3** *Section 2.2 is basically a long list of measurements of initial conditions for the various experiments. Many of these have been reviewed previously, and the authors don't draw any conclusions from this 1.5-page-long section. The authors have done an excellent job of summarising this information in their tables (which I think is very useful). But since the main message seems to be that OIF have been conducted under a diverse range of initial conditions, wouldn't it be better if the authors just made that point in, maybe, 1–2 paragraphs, and then focused on advancing an argument and drawing conclusions?*

☞ Reviewer is correct. We have revised Section 2.2 (from line 30, page 6 to line 24, page 8) to explain why/how initial environmental conditions were different in each experiment. Initial environmental conditions (e.g., silicate concentration and zooplankton abundance) were determined to test hypotheses and to achieve objectives for each

artificial OIF experiment and were related to biogeochemical responses after iron additions. We overall reorganised Section 2.2, and added the conclusions derived from Section 2.2 to Section 2.5 (lines 1–15, page 15).

**4** *The subject matter of Section 2.3 is an important one: how many additions of iron should be made, and how should the fertilised patch be traced? What I miss in this section is to see the authors actually draw conclusions from these results. For example, are multiple additions the right way to design an experiment? At what intervals should they be made? The experiments listed vary a lot in duration, so simply listing the total number of additions, rather than the intervals at which they were made, seems uninformative. Likewise, it might be useful to see some discussion of the amount of iron added per square kilometre of patch– I believe that the additions for the various experiments varied a lot also on a per-area basis, but this is not discussed. Can we determine an optimum rate of application from the various experiments? If not, is this something that warrants further experimental work, and if so, of what kind? Likewise, the discussion of tracing methods just boils down to listing the fact that SF6 and buoys can be used, without drawing conclusions that could be formulated as practical recommendations. For example, drifting buoys in previous experiments were found to ultimately leave the patch, I believe due to wind forcing, and one of the recommendations of this paper is that experiments should be conducted that last upwards of 30 days, under which conditions multiple SF6 injections might become more necessary.*

☞ Good point. We have substantially revised Section 2.3 Iron addition and tracing methods by describing how previous OIF experiments injected iron and traced fertilized patches, why most previous OIF experiment conducted multiple iron addition, and pros and cons of tracing methods (from line 26, page 8 to line 6, page 10). We have also added the conclusions derived from Section 2.3 to Section 2.5 (lines 16–25, page 15). In Section 4 Future: Designing future aOIF experiments, we have added discussion/suggestion about intervals of iron addition and the amount of iron added per km$^2$ of patch, tracing method to maximize the effectiveness of OIF experiments based on Section 2.3 (line 33, page 24 to line 38, page 25).

**5** *Section 2.4 I think mostly re-hashes just the main findings of previous experiments, and again does so in an often rather long-winded way (e.g. the information about spatial pattern of the SOFEX blooms doesn't seem to contribute much). All in all, this section does not really build up an integrated understanding of the biogeochemical responses to iron addition, but reads rather more like a selective catalogue of results picked rather haphazardly from the different experiments: e.g. the difference in maximum chlorophyll between SEEDS-I and SEEDS-II is pointed out, but this isn't part of a broader discussion of chlorophyll levels across the different OIF and what their causes might be. In the next paragraph, SEEDS-I and EIFEX are compared with max chl-a concentration and integrated chl-a stock, but this is not compared with other experiments. My point here is not to ask the authors to add specifically a discussion of chlorophyll, I am just pointing this out as an instance of where the paper rather shies away from reaching interesting new insights. The entire section then simply ends, without any attempt to conclude anything from all this information, aside from pointing out in the middle of the penultimate paragraph that integrated primary production ought to be monitored in future experiments.*

☞ Thank you for your comment. We have substantially revised Section 2.4 Biogeochemical responses (from line 8, page 10 to line 37, page 14). To reflect the Reviewer's suggest, we have added the integrated conclusion derived from Section 2.4 to Section 2.5 (from line 22, page 15 to line 25, page 16). For your information, three conditions, at least, are inevitable to detect significant carbon export to deep waters following increase in PP as follows: (1) a shift to a diatom-dominated community, (2) low bacterial respiration and grazing pressure rates within the mixed layer, and (3) a sufficient experimental duration, enabling both immediate and delayed responses to iron addition to be observed.

6 *I had previously requested that the authors make explicit definitions of terms like "efficiency" and "sequestration" in the context of carbon fluxes, but these terms are still used loosely without specific definitions. I think this is a significant problem when discussing the planning of future iron fertilisation work, as it is critical to achieving the stated objectives to ensure that the correct measurements are made, and this can only be done if we define clearly what we need to measure. I would suggest that the authors refer to the Lampitt et al. paper in the 2008 special issue of the Philosophical Transactions (same issue as the cited Smetacek & Naqvi paper), titled "Ocean fertilization: a potential means of geoengineering?". This paper discusses explicit definitions of terms such as carbon sequestration with reference to the depth of winter mixing, and discusses how they can be measured.*

☞ We apologize this confusing. To exactly explain definition about "sequestration" (i.e., organic carbon flux exported from the mixed layer into intermediate/deep waters) in Section 1 introduction, we have cited Lampitt et al. (2008) and have added new Figure 3c (page 57). We have also added a sentence to define the effectiveness of artificial OIF as follows:

- 1. Introduction (from line 40, page 3 to line 2, page 4):

"However, for aOIF to be considered as a useful geoengineering approach (IPCC, 2007), in the long run, the most critical issue is whether the substantial amounts of organic carbon produced by aOIF in the surface waters lead to a significant export to intermediate/deep layers and long-term (~1,000 years) storage (Fig. 3c) (Lampitt et al., 2008)."

7 *Section 2.5 again leaves me rather unsatisfied: the authors present a list of findings, but don't make any argument or properly discuss reasons for the divergent outcomes. As a result, by the end of the section, it is not clear to the reader why "the effectiveness of iron addition on this component of the biological pump remains a question". In other words, the authors should be using this section to discuss why the previous 13 experiments have not managed to yield clear answers to the effectiveness of carbon sequestration. The answer lies in a combination of experiment duration, measurement methods (measuring only shallow or also deeper fluxes), patch size (with a very small patch, I suspect that deeper traps might miss export that may be occurring because the plume of sinking particles is confined to such a small area), and patch movement (tracking deep export is easy in a stationary patch, but very hard in a large patch). Although the authors do draw these conclusions in a very general sense, saying at*

*the end that future experiments need to last long enough, fertilise a largeish area, and use multiple methods, the justification of publishing a review paper like this one must lie in a more detailed analysis of how which of the previous OIF were unable to achieve these requirements. To do this, the authors would need to delve more into the details of each experiment, rather than spending most of the time reviewing the basic biogeochemical responses of each experiment. Again, terms like "effectiveness of iron addition" could be defined in very specific ways, and using language as loosely as this does not help with achieving insights.*

☞ We agree to the Reviewer's comment. More discussion is necessary. We have moved this Section to 2.4 Biogeochemical responses to describe the results of export fluxes after iron additions as follows:

- 2.4.1. Equatorial Pacific (lines 12–22, page 11):

"To determine whether the biological pump (i.e., export production) is enhanced after iron addition, the export flux of particulate organic carbon (POC) can be estimated using, individually or in combination, chemical tracers such as the natural radiotracer thorium-234 ($^{234}$Th; half-life = 24.1 days) and the stable carbon isotope of particulate organic matter ($\delta^{13}C_{org}$), sediment traps, transmissometers, and underwater video profilers (UVPs) (Table 5) (Bidigare et al., 1999; Nodder et al., 2001; Boyd et al., 2004; Buesseler et al., 2004; Coale et al., 2004; Aono et al., 2005; Tsuda et al., 2007; Smetacek et al., 2012; Martin et al., 2013). The $^{234}$Th radionuclide has a strong affinity for particles, and the extent of $^{234}$Th removal in the water column is indicative of the export of POC associated with surface PP out of the mixed layer (Buesseler, 1998). IronEx-2 was the first aOIF experiment in which the POC flux was estimated (Bidigare et al., 1999). The $^{234}$Th deficiency from the surface to 25 m was measured in the iron-fertilized patch to estimate iron-stimulated export production in the surface layer (Table 5). However, no $^{234}$Th measurements were made in the unfertilized patch for comparison, and no measurements in the deep ocean were undertaken to demonstrate deep carbon export (Bidigare et al., 1999)."

- 2.4.2. Southern Ocean (from line 38, page 12 to line 25, page 13):

"SOIREE was the first aOIF experiment in the SO to estimate the downward carbon flux into deep waters (Fig. 3c). They used a comprehensive suite of methods such as the deployment of a drifting trap, $^{234}$Th and $\delta^{13}C_{org}$ estimates derived from high-volume pump sampling, and a beam transmissometer (Nodder and Waite, 2001). However, no measurable change in carbon export was observed in response to iron-stimulated PP (Table 5 and Fig. 8b) (Charette and Buesseler, 2000; Nodder and Waite, 2001; Trull and Armand, 2001; Waite and Nodder, 2001). During EisenEx, an increased downward carbon flux estimated from $^{234}$Th deficiency was observed in the iron-fertilized patch as the experiment progressed. However, there were no clear differences between in- and outside-patch carbon fluxes (Buesseler et al., 2005). During SOFeX-S, significantly enhanced POC fluxes below the mixed layer after iron enrichment were obtained from $^{234}$Th observations (Buesseler et al., 2005). However, the absolute magnitude of these flux increases was similar to that observed in natural blooms. Uniquely, SOFeX-N used only free-profiling robotic Lagrangian carbon explorers equipped with transmissometers to estimate the downward carbon flux without employing chemical tracers, and observed large POC flux events between day ~27 and ~45 after the first iron addition (Bishop et al., 2004; Coale et al., 2004). However, it was unclear whether surface-fixed carbon was well and truly delivered into intermediate/deep depths. For SAGE and LOHAFEX experiments, which were conducted

under silicate limited conditions (Table 3, Figs. 4c and 6f), there was no detection of fertilization-induced export by any method (Table 5) (Peloquin et al., 2011; Martin et al., 2013). This result was likely to be associated with the pico-plankton dominated community, which led to rapid recycling in the mixed layer and less downward carbon flux. In contrast to the other aOIF experiments, EIFEX, which was conducted within the core of an eddy, provided clear evidence of carbon export stimulated by artificial iron addition (Jacquet et al., 2008; Smetacek et al., 2012). During EIFEX, the initial export flux, estimated from $^{234}$Th in the upper 100 m of the fertilized patch, was ~340 mg C m$^{-2}$ d$^{-1}$ (Table 5 and Fig. 8a) (Smetacek et al., 2012). This value remained constant for about 24 days after iron addition. Between day 28 and 32 a massive increase in carbon export flux (maximum of ~1692 mg C m$^{-2}$ d$^{-1}$) was observed in the fertilized patch, while the initial value remained constant in the unfertilized patch (Table 5 and Fig. 8a). The profiling transmissometer with high-resolution coverage confirmed this result, showing an increase in exported POC below 200 m after day 24. At least half the iron-induced biomass sank (via the formation of aggregates of diatom species, in particular '*Chaetoceros dichaeta*') to a depth of 1,000 m, with a tenfold higher sinking rate (500 m d$^{-1}$), compared to the initial conditions (Smetacek et al., 2012). Significant changes in export production were not found in any of the other aOIF experiments and, therefore, the impact of artificial iron addition on this component of the biological pump needs to be resolved in future OIF experiments (Boyd et al., 2004; Smetacek et al., 2012; Martin et al., 2013)."

- 2.4.3. Subarctic North Pacific (lines 27–33, page 14):

"Despite the formation of a massive iron-induced phytoplankton bloom during SEEDS-1, there was no large POC export flux during the observation period (Table 5) (Tsuda et al., 2003; Aono et al., 2005; Aramaki et al., 2009). During SERIES and SEEDS-2, which allowed comprehensive time-series measurements of the development and decline of the iron-stimulated bloom, POC fluxes estimated by the drifting traps in the fertilized patch displayed temporal variations (Boyd et al., 2004; Aramaki et al., 2009). However, the results suggested that only a small part of the decrease in the mixed layer POC was subsequently captured by the drifting trap, and POC flux losses were mainly governed by bacterial remineralization and mesozooplankton grazing (Boyd et al., 2004; Tsuda et al., 2007)."

☞ We have added the discussion about the effectiveness of carbon sequestration influenced by experiment design and experiment condition (i.e., initial silicate concentrations, diatom bloom, grazing pressure, bacterial respiration, and experiment duration described in Section 2 Past: Overview of previous aOIF experiments to Section 2.5 Significant results in previous OIF experiments (lines 4–25, page 16). Also, to discuss/suggest which of export flux measurement methods can be optimal to exactly track and quantify carbon export flux, we have created new Section 3.1 Export flux measurement methods (from line 9, page 17 to line 36, page 18).

**8** *Section 2.6 starts as a promising paragraph (in fact, I think this could be a good introductory paragraph to the entire Section 2), but then falls flat for the same reasons pointed out above. What are we to conclude from these diverse outcomes? Which experiments may have missed an export event, or deep fluxes of sinking particles, and*

*for what reasons? The issue of ecosystem responses and grazing is clearly an important one, but receives hardly any discussion.*

☞ We have moved the first paragraph in Section 2.6 to the introductory paragraph for Section 2 Past: Overview of previous aOIF experiments (lines 17–26, page 4). Revised Section 2.5 Summary of the significant results from aOIF experiments (from line 1, page 15 to line 25, page 16) may answer to the questions on "what are we to conclude from these diverse outcomes?, which experiments may have missed an export event?, and for what reasons?". In addition, please refer Responses of 3–5 and 7.

9 *After all this extensive review of previous OIF in Section 2, the review suddenly moves on to possible side-effects. What conclusions can be drawn from all of the data that the authors have reviewed in Section 2?*

☞ Thank you for comment. We have revised Section 2.5 Summary of the significant results from aOIF experiments (from line 1, page 15 to line 25, page 16) to draw the conclusions reviewed in Section 2 Past: Overview of previous aOIF experiments.

10 *Section 3.1 is slightly better, as it at least ends with a proper attempt at conclusions. However, the section is again mostly a cataloguing of results from previous experiments. What would be more useful is if the authors attempted to synthesise the insights and discussions of the publications about the individual studies so that we can at try and find patterns in the large variability between studies.*

☞ Thank you. We have moved/revised this Section to Section 3.2 Considering environmental side effects (from line 37, page 18 to line 27, page 21). We have classified possible side effects into climate-relevant gases emission (i.e., $CH_4$, $N_2O$, dimethylsulfide, and halogenated volatile organic compounds) that is produced by biological activities and/or photochemical reactions before and after iron additions, and decrease in dissolved oxygen and increase in domoic acid that lead to the changes in the ocean ecosystem following OIF. Finally, we suggested that monitoring of the productions of climate-relevant gases and domoic acid is essential to evaluate the effectiveness of OIF as a geoengineering approach, as they may directly modify the desired carbon sequestration effectiveness and can do so both positively and negatively.

11 *Section 3.2 implies a discussion of the international legal situation, but the first paragraph is not about legal matters at all. The second paragraph does a decent job of summarising the legal situation. However, since the London Convention explicitly grants exemption only to scientific experiments in "coastal waters" I wonder where that would leave open-ocean experiments in practice. Of course, iron fertilising coastal waters is pointless, as everyone in the oceanographic community knows, but are the authors confident that legitimate research projects in the open ocean will get approval? Some clarification of this seems called for here, at least to the best of the*

*authors' ability.*

☛ We apologize for the confusion. Based on the Reviewer's comments, we have removed the first paragraph. We have also moved this Section to Section 3.3 Regulation of aOIF: International law of the sea as it applies to aOIF and have revised Section 3.3 to clarify the legal situation for artificial OIF experiment in the open ocean (from line 29, page 21 to line 24, page 22).

**12** *Section 4 finally attempts to put forward some practical suggestions. However, I think that this needs to be expanded on and made more specific. Simply concluding that "multiple additions of iron are more efficient" is not the same as formulating specific guidelines about fertilisation levels (e.g. minimum addition per km2) and application intervals. Likewise, the third paragraph in this section ("What") is much too brief a discussion to be genuinely useful and to move the field forward. A long list of measurements is put forward without discussion of the relative merits of each, or a real discussion of the difficulty of tracking a fertilised patch, and whether we've learnt how to do this better over the 13 experiments that have been conducted. Likewise, the discussion of how to measure carbon fluxes is too indiscriminate a list to be of much use. What would really help would be an evaluation of what the various measurements contribute to our understanding, what their pitfalls are (especially, for example, surface-tethered sediment traps, which are probably better avoided in favour of their neutrally buoyant counterparts), and, again, where in the water column we really need to be measuring. The fact that the sections concludes in saying that carbon fluxes need to be monitored using "both trap fluxes and/or 234Th deficiency" is somewhat worrying: surely, one of the lessons from previous experiments is that both measurements really are needed, and that traps must be at multiple depths so that we can track both the export out of the surface as well as the sequestration below the depth of winter mixing? In discussing possible candidate regions, it would be useful if the authors proposed specific regions, rather than just making general observations such as recommending "regions with high silicate concentrations and low copepod abundances".*

☛ Thank you for comments. We have modified Section 4 Future: Designing future aOIF experiments (from line 26, page 22 to line 31, page 26) to put forward some practical suggestions for location (i.e., 'Where'), timing (i.e., 'When'), duration (i.e., 'How long'), modes of iron addition (i.e., 'How'), tracking methods/parameters measurements/protocols (i.e., 'What'), and side effects on marine/ocean ecosystems (i.e., 'What concern') including discussion of "fertilization levels (e.g. minimum addition per km$^2$), application intervals, the difficulty of tracking a fertilized patch, how to measure carbon fluxes, and possible candidate regions" to maximize the effectiveness of OIF experiments in the future based on the descriptions of Sections 2 and 3.

**13** *The manuscript ends with the design proposal for KIFES. Even though KIFES currently has no funding, it is an experiment that the authors clearly hope to conduct in more or less this form in the future. Therefore, I think it is valuable to have this section in the manuscript. What might be useful is if the authors used this opportunity to explain in slightly more detail how the design of KIFES will avoid the problems encountered in earlier experiments,*

*i.e. explain the specifics of duration, patch size, measurement methods, etc.*

☞ Thank you for pointing this out. To reflect the Reviewer's suggestion, we have revised Section 5 Design of the Korean Iron Fertilization Experiment in the Southern Ocean (KIFES). Based on previous artificial OIF experiments, in Section 5.1 Background - Bransfield Basin (from line 34, page 26 to line 36, page 27), we have set up the purposes of KIFES; (1) implementation of a scientific artificial OIF experiment complying with "Assessment Framework for Scientific Research Involving Ocean Fertilization" for the first time after assessment framework was accepted from LC/LP in 2010 (Resolution LC-LP.2, 2010), (2) evaluation the effectiveness of scientific artificial OIF in terms of atmospheric carbon sequestration (i.e., to identify/quantify significant increase in iron-induced carbon export fluxes into intermediate/deeper waters) in the Southern Ocean, (3) determination the environmental conditions that would maximize effectiveness of artificial OIF, and (4) monitoring quantitatively and qualitatively short- and long-term possible side effects derived from previous artificial OIF experiments. With the purpose, we have also added some sentences to discuss why in/near the eastern Bransfield Basin was considered for the site of KIFES based on following three criteria: (1) possibility of diatom bloom, (2) proximity to meso-scale eddies, and (3) availability of historical oceanographic data and to establish the hypothesis that input of bioavailable iron allowing an increase in productivity and export would have led to the massive enhancement of the diatom flux in this basin and an artificial OIF in the diatom-dominated region with high sinking rates near the eastern Bransfield Basin will be more effective for carbon export compared to the previous artificial OIF experiments conducted in the Southern Ocean. Specific plans for KIFES experiment (e.g., duration, patch size, and measurement methods) in Section 5.2 A plan for the future: KIFES (from line 38, page 27 to line 30, page 29) have been modified based on Section 4 Future: Designing future OIF experiment.

**14** *Overall introduction to OIF and statement of the purpose of the present paper.*

☞ We have modified some sentences to define effectiveness of OIF and to introduce purpose of this paper in Section 1 Introduction (from line 1, page 3 to line 15, page 4).

**15** *A brief overview of each of the experiments, maybe written as a historical narrative (1–2 pages) that describes how the various experiments built on each other and what the main hypotheses were that each was designed to address. This section can be used to highlight the different physical and biogeochemical conditions of each experiment, and also highlight the main biogeochemical responses and findings. This experiment-by-experiment approach might help to give the reader are more integrated understanding of each experiment than the current parameter-by-parameter approach used throughout Section 2. The section could then end with a paragraph highlighting the key outstanding questions that future experiments (including KIFES) need to address. These include questions relating to the amount of carbon sequestration, trace gas production, and plankton community shifts.*

☞ Good point. Based on the Reviewer's comment, overall we have reorganized Section 2. Also, please refer Responses 2–11.

**16** *A detailed discussion of each of these questions in turn, highlighting why the previous 13 experiments have failed to reach consensus, and what needs to be done to move our understanding forward. This needs to be an issue-based discussion, for example going into real detail about how carbon fluxes need to be measured, which experiments managed to take these measurements (but failed to find effects), which experiments were maybe hampered by size and/or duration, and also a discussion of the uncertainties relating to each method (e.g. methodological problems with sediment traps, issues with the thorium technique, problems with the estimation of net community production from O2:Ar ratios, etc.).*

☞ Thank you for pointing out this deficiency. We have added the discussion about "why the previous 13 experiments have failed to reach consensus and which experiments were maybe hampered by size and/or duration" to Section 2.5 Summary of the significant results from aOIF experiments (from line 1, page 15 to line 25, page 16) and Section 4 Future: Designing future aOIF experiments (from line 39, page 24 to line 7, page 25). What needs to be done to move our understanding forward was discussed in Section 3 Present: Unanswered aOIF questions - export flux, possible side effects, and international law (from line 27, page 16 to line 24, page 22). Also, please refer Response 9. We have also created new Section 3.1 Export flux measurement methods (from line 9, page 17 to line 36, page 18) to discuss "how carbon fluxes need to be measured, which experiments managed to take these measurements, and the uncertainties relating to each method". Also, please refer Response 7.

**17** *A section that reaches specific conclusions and makes recommendations about the design of future experiments.*

☞ Thank you for suggestion. Please refer Response 12.

**18** *Specific proposal for KIFES and description of the logic for conducting this in the Bransfield Strait. Do the authors have some preliminary altimetry images to show stable eddies in this region?*

☞ Please refer Response 13 for "specific proposal for KIFES and description of the logic for conducting this in the Bransfield Strait".

- For stable eddies in the Bransfield Strait, Thompson et al. (2009) showed that a large standing eddy (~40 km in diameter) was centered at ~62°S and 54°W and remained for ~30 days using historical drifters released during the period 1989–2005, and 40 drifters released in February 2007 as part of the Antarctic Drifter Experiment: Links to Isobaths and Ecosystems project (Fig. R1).

[Figure]

Fig. R1 The mean velocity field for historical drifters released during a period between 1989 and 2007 and forty drifters released in February 2007 as the Antarctic Drifter Experiment: Links to Isobaths and Ecosystems project. The bin size is 0.1° lat x 0.2° lon. The shading indicates bathymetry with color change every 500 m and contour lines at every 1000 m between 1000 m and 4000 m depth (from Thompson et al., 2009).

- Altimetry images obtained from Archiving, Validation and Interpretation of Satellite Oceanographic (AVISO) (https://www.aviso.altimetry.fr/) also show stable eddies (>~50 km in diameter) near/in the eastern Bransfield Strait (Fig. R2). This image is Level-4 sea level anomaly daily image with a spatial resolution of 0.25°. The reference period of the sea level anomaly is based on a 20-year (1993–2012) period.

[Figure]

Fig. R2 Sea level anomaly (cm) image in 20-Jan-2015 (https://www.aviso.altimetry.fr/).

**Reviewer #2**

**-General Comments:**

19 *The question of why the outcomes of previous experiments were so different and how this new project could improve on past experience is only superficially addressed. What are the hypothesis? how the KIFES project improves on previous experiments? and why?*

☞ Good point. We have revised Section 5.1 Background - Bransfield Basin to describe more specifically "what are the hypothesis and how the KIFES project improves on previous experiments" (from line 34, page 26 to line 36, page 27). Also, please refer Response 13.

20 *In the same line of thought, the knowledge gained by artificial compared to natural iron experiments (within the context of geoengeneering and potentially negative side-effects) could also be more thoroughly analyzed. As an example, one would be hard pressed to find 'potential long term negative effects' by studying the naturally fertilized systems in the Southern Ocean as these systems are key to sustaining Southern Ocean food webs and biogeochemistry.*

☞ Thank you for suggestion. We have added some sentences about "the knowledge that could be gained by artificial compared to natural iron experiments" in Section 4 as follows:

- Section 4 Future: Designing future aOIF experiments (from line 40, page 22 to line 12, page 23):

"The nOIF experiments have also produced much higher carbon sequestration rates than the small-scale aOIF experiments (Morris and Charette, 2013). Furthermore, the results from nOIF experiments do not support the potential negative impacts proposed for OIF experiments, even at larger scales (Belviso et al., 2008). However, these nOIF results do not guarantee that aOIF as a geoengineering approach is able to achieve the high effectiveness associated with carbon sequestration and enable a simple scaling-up as a prediction tool, because the nOIF experiments differ from the aOIF experiments in the mode of iron supply. In particular, nOIF is a continuous and slow process and its iron source is based on the upwelling of iron-rich subsurface waters to the surface layer, whereas aOIF is intended to be episodic, with massive short-term iron additions (Blain et al., 2007). In addition, in nOIF it is difficult to accurately identify iron sources due to the complexity of the system, whereas in aOIF there is quantitative and qualitative information about iron additions and sources (Blain et al., 2008). Contrary to the results of aOIF experiments in the SO (e.g., SOIREE and SOFeX-N), no increase in DMS emissions was found in nOIF experiments in the SO (i.e., Kerguelen Ocean and Plateau compared Study: KEOPS) (Belviso et al., 2008), suggesting that it might be difficult to identify the potential long-term negative effects of aOIF by studying the naturally fertilized systems in the SO."

**-Specific Comments:**

**21** *In order to simplify comparison, I would recommend to first group the experiments by region with HNLC regions first (Eq. Pac, Subarctic Pac. SO, North Atlantic) and then by date at which they took place in all figures and tables.*

☞ Thank you for your recommendation. We presented the data "by region with HNLC regions (Equatorial Pacific-Southern Ocean-Subarctic North Pacific-North Atlantic) first and then by date at which they took place" in Tables 1–5 (pages 44–52), Figure 4 (page 58) and Figures 6–7 (pages 60–61).

**22** *Table 2: The authors definition of 'limitations' is somewhat unusual. Finding no export is not a 'limitation', just a result that might not support the iron hypothesis. Limitation would be for instance that the experiment did not last long enough to see the terminal phase of the bloom when most export would occur.*

☞ We have removed the results that might not support the iron hypothesis in "limitations" from Table 2 (page 47–48). We have added limitations for experiment design in "limitations".

**23** *Page 13, lines 17-22: Why do the authors focus only on N2O and DMS. From the previous paragraphs, there are many more 'issues' to focus on (those include export, plankton assemblage composition (toxic species), production of other greenhouse gazes, reaction of upper trophic levels. All those are poorly understood.*

☞ Thank you for pointing out this deficiency. Please refer Response **10**.

**24** *Page 16, lines 19-26: The authors would strengthen their case by present scientific evidence to the claim that March is the ideal period (i.e the late growth season can also be challenging due to increased zooplankton biomass and potential Si limitation). Also the 'best' period is most probably linked to the location.*

☞ We agree. PP in the SO, a representative HNLC region, is subject to co-limitation by micro/macro-nutrients (i.e., iron and/or silicate) and light availability (Mitchell et al., 1991). In the south PF of SO, phytoplankton blooms usually occur during the early summer (i.e., from late December to early January) due to the increasing nutrient flux from the subsurface waters to the surface waters by the shoaling of MLD, along with the receipt of sufficient solar radiation (Moore and Abbott, 2002). Prior to December, phytoplankton growth is mainly limited due to light availability (Mitchell et al., 1991; Veth et al., 1997; Abbott et al., 2000), while after January (i.e., during late summer and early autumn from February to March) it is mainly limited due to silicate availability. In previous aOIF experiments in the SO that have been conducted between spring and early autumn, PP was mainly limited by iron and/or silicate availability rather than light availability (de Baar et al., 2005; Smetacek and Naqvi, 2008; Peloquin et al., 2011). In addition, the grazing pressure of mesozooplankton (i.e., copepods) on large diatoms was also a major limiting factor in diatom production (Coale et al., 2004; Martin et al., 2013), and was generally higher during late summer and early autumn (February to March) (Le Quéré et al., 2016). Therefore, by considering the key factors (i.e., micro/macro nutrient availability, light availability, and grazing pressure) controlling PP in the SO, we have modified the most appropriate timing for an aOIF experiment to start in the SO from 'March' to 'late

December–early January' in "When" paragraph in 4 Future: Designing future aOIF experiments (from line 37, page 23 to line 11, page 24).

**25** *Page 17, lines 1-3: Similarly, the planning of the length of the experiment could also be dealt with in a more robust manner and take into account the longevity of the fertilization signal from previous studies (after the patch was left) and include the fact that new autonomous observation technology is available.*

☞ Thank you for comments. We have modified "How long" paragraph in 4 Future: Designing future aOIF experiments (lines 12–32, page 24). SOFeX-S experiments didn't show large increase in export fluxes compared to natural fluxes due to insufficient experiment period (28 days) to cover the termination of phytoplankton bloom. However, only EIFEX (experiment period: 39 days), which did fully monitor all the phases of phytoplankton bloom from onset to termination, showed high export fluxes compared to outside-patch fluxes and this increase in export fluxes occurred between day 28 and day 32 after iron addition. Further, SOIREE showed the longest iron-induced bloom (i.e., the longevity of >40 days). Therefore, experiment period should be at least >~40 days based on the previous OIF experiments.

**26** *Page 17, lines 13-21: I am not sure I understand the point of this paragraph.*

☞ We have removed this paragraph and rewrote Section 5 (from line 33, page 26 to line 7, page 30). Also, please refer Response **13**.

**27** *Table 1: - The description in the legend does not seem to mirror the table headings: Background Fe = Initial Fe in the heading; Fe concentrations after Fe addition = After Fe; ...etc...- 'Day of Fe addition from the beginning of OIF experiment' refers to what? Duration of Fe addition? Or time between start of the experiment and 1st Fe addition?- 'Period' should be 'Duration' instead.- In the title HNLCLSi could be explained as it is not a commonly used acronym.*

☞ We apologize for this confusion. We have revised the legend of Table 1 (lines 1–3, page 44). We have also added annotation to explain "HNLCLSi" (line 1, page 46).

**28** *Table 2: In the title 'limitation' should be 'limitations' (plural). Objectives and name of the experiments are not aligned. This can lead to confusion.*

☞ We revised them in Table 2 (pages 47–48).

**29** *Table 3 legend: 'Changes of chemical parameters…' should be 'Initial conditions and changes (Δ values) in*

*chemical parameters during the OIF experiments'.*

☞ We revised it (lines 1–2, page 49).

**30** *Table 4 legend: replace with 'Response of biological parameters to iron fertilization (maximum difference between initial conditions and conditions after fertilization). Note that PP (mg C m$^{-2}$ d$^{-1}$) was estimated by multiplying PP (mg C m$^{-3}$ d$^{-1}$) with mixed layer depth (m)'.*

☞ We apologize for the confusion. Maximum in Table 4 legend indicates maximum values after fertilization. To prevent any confusion, we have replaced 'Maximum' with 'After' in Table 4 heading and 'Changes of biological parameters from initial to after (maximum) concentrations by OIF experiments. Note that *PP (mg C m$^{-2}$ d$^{-1}$) was estimated by multiplying PP (mg C m$^{-3}$ d$^{-1}$) with mixed layer depth (m).' with 'Initial values of biological parameters and the values after fertilization. Note that maximum values were attained after fertilization.' in Table 4 legend (line 1, page 50).

**31** *Table 5: Data and methods are not aligned in the table. This can lead to confusion as to which method was used to estimate fluxes during the experiments.*

☞ Thank you for comments. We removed method description column from Table 5 (page 52). However, we left method column because export fluxes estimates depend on measurement methods. Please refer Response **16** for discussion about export flux measurement methods

**32** *Table 5 legend: replace with 'Initial and maximum export fluxes (mg C m-2 d-1) and corresponding depth in and outside the fertilized patch of OIF experiments. Values in brackets correspond the day of measurement after fertilization."*

☞ We revised it (lines 1–3, page 52).

**33** *Table 5: Export values for the SERIES out patch seem wrong, shouldn't it be 120/48 same values as the initial in-patch values?*

☞ Boyd et al. (2004) showed that 50 m 'outside-patch' export fluxes were 192 mg C m$^{-2}$ d$^{-1}$ on day 3 and 139 mg C m$^{-2}$ d$^{-1}$ on day 15 with deployment of drifting trap. We defined initial export fluxes in SERIES experiment as export fluxes on day 3 during SERIES experiment. In-patch export flux on day 3 was measured in patch center while outside-patch export flux on day 3 was measured in the surrounding waters. Therefore, in-patch export flux on day 3 (~120 mg C m$^{-2}$ d$^{-1}$) was different from outside-patch export flux on day 3 (192 mg C m$^{-2}$ d$^{-1}$).

**34** *Figure 5: the photograph in panel e should be in panel c and vice versa.*

☞ We changed the photograph in panel e to panel c in Figure 5 (lines 20–26, page 53).

**35** *Figures 6 & 7: For comparison purposes it would be helpful to present the data on OIFs grouped by regions rather than the time-sequence in which the experiments took place. As well as maybe demarcate (maybe using vertical lines) each regions (i.e. Eq Pac/Sub. Pac/SO) in the graphs.*

☞ Please refer Response **21**.

To determine whether the biological pump (i.e., export production) is enhanced after iron addition, the export flux of particulate organic carbon (POC) can be estimated using, individually or in combination, chemical tracers such as the natural radiotracer thorium-234 ($^{234}$Th; half-life = 24.1 days) and the stable carbon isotope of particulate organic matter ($\delta^{13}C_{org}$), sediment traps, transmissometers, and underwater video profilers (UVPs) (Table 5) (Bidigare et al., 1999; Nodder et al., 2001; Boyd et al., 2004; Buesseler et al., 2004; Coale et al., 2004; Aono et al., 2005; Tsuda et al., 2007; Smetacek et al., 2012; Martin et al., 2013). The $^{234}$Th radionuclide has a strong affinity for particles, and the extent of $^{234}$Th removal in the water column is indicative of the export of POC associated with surface PP out of the mixed layer (Buesseler, 1998). IronEx-2 was the first aOIF experiment in which the POC flux was estimated (Bidigare et al., 1999). The $^{234}$Th deficiency from the surface to 25 m was measured in the iron-fertilized patch to estimate iron-stimulated export production in the surface layer (Table 5). However, no $^{234}$Th measurements were made in the unfertilized patch for comparison, and no measurements in the deep ocean were undertaken to demonstrate deep carbon export (Bidigare et al., 1999).

**2.4.2 Southern Ocean**

As in the EP IronEx-1/-2 experiments, there were initial rapid increases in the Fv/Fm ratio within 24 hours of iron addition, indicating that phytoplankton growth was mainly limited by iron availability. Maximum values of the Fv/Fm ratio ranged from 0.5 (SOFeX-N and LOHAFEX) to 0.65 (SOIREE and SOFeX-S) (Table 4 and Fig. 7a). However, the time taken to reach the maximum Fv/Fm ratio was usually longer than ~10 days, i.e., much slower than in IronEx-1/-2 (~2 days) (Boyd and Abraham, 2001; Gervais et al., 2002; Coale et al., 2004; Smetacek et al., 2005; Peloquin et al., 2011; Martin et al., 2013). The slower response time in the SO compared to the EP might be attributed to the colder temperatures (~5°C vs. ~24°C) and/or the deeper MLDs (~60 m vs. ~30 m) (Figs. 6c and d), which were indicative of active physical mixing (Boyd and Abraham, 2001; Boyd, 2002).

The aOIF experiments in the SO recorded >2-fold increases in chlorophyll-a concentrations compared to initial levels (<0.7 mg m$^{-3}$), and maximum values between 1.25 mg m$^{-3}$ (LOHAFEX) and ~3.8 mg m$^{-3}$ (SOFeX-S) were obtained after artificial iron additions (Table 4 and Fig.7c). Satellite observations were used to investigate the changing spatial and temporal distribution of chlorophyll-a concentrations in response to iron fertilization in the fertilized patches compared to the surrounding waters (Boyd et al., 2000; Coale et al., 2004; Boyd et al., 2005; Westberry et al., 2013). For example, spatial changes in chlorophyll-a resulting from SOFeX-N/S iron addition were detected using Sea-viewing Wide Field-of-view Sensor (SeaWiFS) and MODerate resolution Imaging Spectrometer (MODIS) Terra Level-2 chlorophyll-a images. The chlorophyll-a image on day 24 after iron addition in the SOFeX-N showed a phytoplankton bloom distribution resembling a long thread with 10-fold higher concentrations (1.0 mg m$^{-3}$) than the surrounding waters (0.1 mg m$^{-3}$), while a chlorophyll-a image on day 20 of SOFeX-S suggested a somewhat broader bloom pattern (0.5 mg m$^{-3}$), with concentrations elevated ~5-fold over the surrounding levels (~0.1 mg m$^{-3}$) (Fig. 7d) (Westberry et al., 2013).

Following artificial iron enrichment in the SO, ΔPP ranged from 360 (SAGE) to ~1356 mg C m$^{-2}$ d$^{-1}$ (SOFeX-N) (Table 4 and Fig. 7e). In SOIREE, EisenEx, and SOFeX-N/S, the PP increased continuously throughout the duration of the experiments (Boyd et al., 2000; Gall et al., 2001a; Gervais et al., 2002; Coale et al., 2004). However, in EIFEX, SAGE, and LOHAFEX there was a significant increase in PP for ~10 (SAGE) – 20 (EIFEX) days in response to the iron addition, and decreasing trends after day ~12 (SAGE) – 25 (EIFEX) due to various influences, such as high export production (e.g., EIFEX), lateral dilution with surrounding waters (e.g., SAGE), and high grazing pressure and active bacterial respiration (e.g., LOHAFEX) (Boyd, 2002; Gervais et al., 2002; Buesseler et al., 2004; Coale et al., 2004; Peloquin et al., 2011; Smetacek et al., 2012; Martin et al., 2013).

Using both microscopes and high-performance liquid chromatography pigment analysis, changes in phytoplankton community affected by iron addition have also been investigated. SO iron additions have resulted in blooms of relatively large-sized phytoplankton (Boyd et al., 2007). During SOIREE and EisenEx, the dominant phytoplankton community shifted from pico- and nano-phytoplankton (e.g., pico-eukaryotes and prymnesiophytes) to micro-phytoplankton (i.e., diatoms) (Gall et al., 2001a; Gervais et al., 2002). In SOFeX-S and EIFEX, diatoms were already the most abundant group prior to iron addition (Coale et al., 2004; Hoffmann et al., 2006). The contribution of large diatoms became especially clear in EIFEX where ~97% of the phytoplankton bloom was attributed to these species (Smetacek et al., 2012). However, no taxonomic shift toward diatom-dominated phytoplankton communities (<5% of total phytoplankton community) was observed in SAGE and LOHAFEX, which were conducted under silicate-limited conditions (Harvey et al., 2010; Peloquin et al., 2011; Martin et al., 2013; Ebersbach et al., 2014). Although SOFeX-N was conducted under low silicate conditions, the diatom biomass increased remarkably making up ~44% of the total phytoplankton community (Coale et al., 2004). This result was partly influenced by the temporary relief of silicate limitation through lateral mixing of the iron-fertilized waters with surrounding waters, with relatively higher silicate concentrations (Coale et al., 2004).

Iron-mediated increases in PP resulted in a significant uptake in macronutrients and $p$CO$_2$ throughout the aOIF experiments in the SO (except for SAGE) (Table 3, Figs. 7b and f). The ΔNO$_3^-$ ranged from -3.5 μM (e.g., SOFeX-S) to -1 μM (e.g., EisenEx) and Δ$p$CO$_2$ ranged from -38 μatm (e.g., SOIREE) to -7 μatm (e.g., LOHAFEX). Although SOFeX-S had a somewhat greater ΔNO$_3^-$ (-3.5 μM) and Δ$p$CO$_2$ (-36 μatm) than EIFEX (ΔNO$_3^-$: -1.6 μM and Δ$p$CO$_2$: -30 μatm) both results suggested that diatoms were abundant in the two experiments. However, the smaller ΔSi observed during SOFeX-S (-4 μM, compared to EIFEX -11 μM) was associated with a decrease in silicification (i.e., the adjustment of frustule thickness toward thinner frustules) of the dominant diatom species (i.e., *Fragilariopsis* sp.) (Twining et al., 2004). In EIFEX, the ratio of heavily silicified diatoms (e.g., *Thalassiothrix antarctica*) to total diatom biomass increased from 0.24 (day 0) to 0.46 (day 37) leading to the larger ΔSi (i.e., more demand for silicate) (Hoffmann et al., 2006). Interestingly, the biogeochemical responses in SAGE were totally different from those seen in other experiments, in particular of increases in ΔNO$_3^-$ (+3.9 μM), Δ$p$CO$_2$ (+8 μatm), and ΔDIC (+25 μM) (Table 3, Figs. 7b and f). These contrasting results were thought to be the result of entrainment through vertical and horizontal physical mixing into the iron-fertilized patch of the waters, with higher biogeochemical concentrations (Currie et al., 2011; Law et al., 2011).

SOIREE was the first aOIF experiment in the SO to estimate the downward carbon flux into deep waters (Fig. 3c). They used a comprehensive suite of methods such as the deployment of a drifting trap, $^{234}$Th and δ$^{13}$C$_{org}$ estimates derived from high-volume pump sampling, and a beam transmissometer (Nodder and Waite, 2001). However, no measurable change in carbon export was observed in response to iron-stimulated PP (Table 5 and Fig. 8b) (Charette and Buesseler, 2000; Nodder and Waite, 2001; Trull and Armand, 2001; Waite and Nodder, 2001). During EisenEx, an increased downward carbon flux estimated from $^{234}$Th deficiency was observed in the iron-fertilized patch as the experiment progressed. However, there were

no clear differences between in- and outside-patch carbon fluxes (Buesseler et al., 2005). During SOFeX-S, significantly enhanced POC fluxes below the mixed layer after iron enrichment were obtained from [234]Th observations (Buesseler et al., 2005). However, the absolute magnitude of these flux increases was similar to that observed in natural blooms. Uniquely, SOFeX-N used only free-profiling robotic Lagrangian carbon explorers equipped with transmissometers to estimate the downward carbon flux without employing chemical tracers, and observed large POC flux events between day ~27 and ~45 after the first iron addition (Bishop et al., 2004; Coale et al., 2004). However, it was unclear whether surface-fixed carbon was well and truly delivered into intermediate/deep depths. For SAGE and LOHAFEX experiments, which were conducted under silicate limited conditions (Table 3, Figs. 4c and 6f), there was no detection of fertilization-induced export by any method (Table 5) (Peloquin et al., 2011; Martin et al., 2013). This result was likely to be associated with the pico-plankton dominated community, which led to rapid recycling in the mixed layer and less downward carbon flux. In contrast to the other aOIF experiments, EIFEX, which was conducted within the core of an eddy, provided clear evidence of carbon export stimulated by artificial iron addition (Jacquet et al., 2008; Smetacek et al., 2012). During EIFEX, the initial export flux, estimated from [234]Th in the upper 100 m of the fertilized patch, was ~340 mg C $m^{-2}$ $d^{-1}$ (Table 5 and Fig. 8a) (Smetacek et al., 2012). This value remained constant for about 24 days after iron addition. Between day 28 and 32 a massive increase in carbon export flux (maximum of ~1692 mg C $m^{-2}$ $d^{-1}$) was observed in the fertilized patch, while the initial value remained constant in the unfertilized patch (Table 5 and Fig. 8a). The profiling transmissometer with high-resolution coverage confirmed this result, showing an increase in exported POC below 200 m after day 24. At least half the iron-induced biomass sank (via the formation of aggregates of diatom species, in particular 'Chaetoceros dichaeta') to a depth of 1,000 m, with a tenfold higher sinking rate (500 m $d^{-1}$), compared to the initial conditions (Smetacek et al., 2012). Significant changes in export production were not found in any of the other aOIF experiments and, therefore, the impact of artificial iron addition on this component of the biological pump needs to be resolved in future OIF experiments (Boyd et al., 2004; Smetacek et al., 2012; Martin et al., 2013).

**2.4.3 Subarctic North Pacific**

The observed increase in the Fv/Fm ratio in response to artificial iron addition in the subarctic NP suggests that the relief in iron limitation may have assisted phytoplankton growth (Table 4 and Fig. 7a). SEEDS-1/-2, which were conducted in the western basin, showed continuous increases in the Fv/Fm ratio, with a maximum value of ~0.4 approximately 10 days after the first iron addition (Tsuda et al., 2003; Tsuda et al., 2007). During SERIES, which was conducted in the eastern basin, the Fv/Fm ratio rapidly increased and reached a maximum value of 0.55 within 24 hours of the first iron addition (Boyd et al., 2005). However, the Fv/Fm ratio returned toward the initial value of <0.3 as the dissolved iron concentrations decreased to background levels (<0.2 nM) after about day 10 (Tsuda et al., 2003; Boyd et al., 2005; Tsuda et al., 2007).

Increases in chlorophyll-a concentrations were detected in the subarctic NP aOIF experiments in both basins after about the fifth day (Tsuda et al., 2003; Boyd et al., 2004; Suzuki et al., 2009). These increases were especially apparent in SEEDS-1, where they reached a maximum value of 21.8 mg $m^{-3}$ (27 times the initial value of 0.8 mg $m^{-3}$) (Table 4 and Fig. 7c). This augmentation was the largest among all the aOIF experiments (Tsuda et al., 2003). The dramatic chlorophyll-a increase observed during SEEDS-1 was partly attributed to the particular range of seawater temperature in the region, which was conducive to diatom growth (i.e., 8–13°C) as well as to the shallower MLD (~10 m), which provided a relatively longer surface water residence time for the additional iron (Figs. 6c and d) (Noiri et al., 2005; Takeda and Tsuda, 2005; Tsumune et al., 2005). During SERIES, chlorophyll-a concentrations increased substantially from the initial value of 0.35 to 5 mg $m^{-3}$ over 17 days, and the second highest concentration of all aOIF experiments was recorded (Table 4 and Fig. 7c) (Boyd et al., 2004). However, on the 18th day there was a downturn in chlorophyll-a as silicate concentrations decreased to <2 μM (Boyd et al., 2005). Although SEEDS-2 was conducted under similar initial conditions to SEEDS-1 (refer to Section 2.2.3), there was a minimal increase in chlorophyll-a (i.e., maximum value of less than 3 mg $m^{-3}$) (Fig. 7c). This smaller increase was thought to be the

result of extensive copepod grazing (SEEDS-2 had almost five times more copepod biomass than SEEDS-1) (Table 4) (Tsuda et al., 2007). A similar spread was seen in depth-integrated PP, which increased by 7-fold or more after various iron enrichments in the subarctic NP aOIF experiments (e.g., from 300–420 to 1,000–2,000 mg C m$^{-2}$ d$^{-1}$) (Table 4 and Fig. 7e).

Changes in the composition of phytoplankton groups were investigated in the subarctic NP aOIF experiments. In SEEDS-1 there was a shift from oceanic diatoms (e.g., *Pseudonitzschia turgidula*), with growth rates of 0.5–0.9 d$^{-1}$, to faster-growing neritic diatoms (e.g., *Chaetoceros debilis*, 1.8 d$^{-1}$) (Tsuda et al., 2005). The shift in the dominant phytoplankton species during the SEEDS-1 experiment was an important contributor to what became the greatest aOIF-induced increase in phytoplankton biomass yet recorded. During SERIES, the phytoplankton community changed from *Synechococcus* and haptophytes to diatoms, and the highest SERIES chlorophyll-a concentration (day 17) was associated with a peak in diatom abundance (Boyd et al., 2005). However, during SEEDS-2, no significant iron-induced diatom bloom was observed. Instead, pico-phytoplankton (e.g., phytoflagellates) (67% of the total community) dominated throughout the duration of the experiment due to the heavy grazing pressure on diatoms (Table 4) (Tsuda et al., 2007).

In the subarctic NP experiments, significant decreases in macro-nutrient uptake (i.e., $\Delta NO_3^-$ and $\Delta Si$), $\Delta DIC$, and $\Delta pCO_2$ in response to artificial iron addition were observed (Table 3 and Figs. 7b and f). SEEDS-1, which saw the largest increases in chlorophyll-a concentrations, also had the largest $\Delta pCO_2$ (-130 μatm) and $\Delta DIC$ (-58 μM) (Table 3 and Fig. 7f). These changes led, in turn, to the largest $\Delta NO_3^-$ (-15.8 μM) (Fig. 7b) and $\Delta Si$ (-26.8 μM) (Table 3) (Tsuda et al., 2003). The second largest increase in the chlorophyll-a concentration was observed in SERIES, where drawdowns of $pCO_2$ (-85 μatm), DIC (-37 μM), nitrate (-8.5 μM), and silicate (-13.6 μM) were recorded. During SEEDS-2, the nitrate concentration decreased remarkably from 18.4 μM to 12.7 μM after day 5; however, there was no significant change in silicate concentrations, which would have been expected as a signal of an iron-induced diatom bloom (Tsuda et al., 2007; Suzuki et al., 2009).

Despite the formation of a massive iron-induced phytoplankton bloom during SEEDS-1, there was no large POC export flux during the observation period (Table 5) (Tsuda et al., 2003; Aono et al., 2005; Aramaki et al., 2009). During SERIES and SEEDS-2, which allowed comprehensive time-series measurements of the development and decline of the iron-stimulated bloom, POC fluxes estimated by the drifting traps in the fertilized patch displayed temporal variations (Boyd et al., 2004; Aramaki et al., 2009). However, the results suggested that only a small part of the decrease in the mixed layer POC was subsequently captured by the drifting trap, and POC flux losses were mainly governed by bacterial remineralization and mesozooplankton grazing (Boyd et al., 2004; Tsuda et al., 2007).

**2.4.4 Subtropical North Atlantic**

Not much is known about the biogeochemical responses to OIF in the subtropical NA. The FeeP experiment reported that pico-plankton abundances increased after iron and phosphate additions (Rees et al., 2007); however, no other details of the biogeochemical response to iron addition in FeeP have been reported.

**2.5 Summary of the significant results from aOIF experiments**

To test the hypothesis that the addition of iron to the surface layer will effectively reduce atmospheric $CO_2$ by increasing PP and enhancing, in turn, the carbon export flux to the deep ocean, aOIF experiments have usually been conducted in HNLC regions: the EP, SO, and subarctic NP. The one exception was the FeeP experiment, which was performed in the subtropical NA. The initial environmental conditions associated with the physical and biogeochemical properties were determined at these OIF sites over 1–7 days before iron addition to allow the responses to the aOIF to be evaluated and quantified. Preliminary

surveys confirmed that all sites, except FeeP in the subtropical NA, were subject to iron-limited HNLC conditions, with typical levels of iron <0.4 nM, nitrate >~10 μM, and chlorophyll-a <1 mg m$^{-3}$. The initial Fv/Fm ratios were <~0.3, suggesting that phytoplankton growth was severely iron-limited. In SEEDS-1, SOFeX-S, and EIFEX, prior to the addition of iron, the micro-phytoplankton (e.g., diatom) community accounted for half of the population and this was thought to be beneficial to the enhancement of export production. In the other experiments, pico- and nano-phytoplankton (e.g., *Synechococcus* and haptophytes) initially dominated; they are associated with rapid recycling in the mixed layer through the microbial loop rather than export production (Michaels and Silver, 1988; Coale et al., 1998; Landry et al., 2000; Boyd and Law, 2001; Gervais et al., 2002; Coale et al., 2004; Boyd et al., 2005; Tsuda et al., 2005; Hoffmann et al., 2006; Tsuda et al., 2007; Harvey et al., 2010; Martin et al., 2013).

Iron-sulfates dissolved in acidified seawater have been commonly used for artificial iron addition because they are both highly bioavailable and inexpensive. The mixture is generally released into the ship's wake over a period of 24 hours. The amount of added iron was determined so to reach a target dissolved-iron concentration (at least >1 nM) by volume (defined as the MLD × patch area). To achieve this, a wide range of 225–2,000 kg was applied. Except in IronEx-1 and SEEDS-1, the experiments used multiple (2–4) iron additions to reinforce the increased iron levels. To trace the iron-fertilized patches, physical tracers (i.e., ARGO or other GPS-tracked drifting buoys) and/or chemical tracers such as SF$_6$ were used. In addition, biogeochemical parameters, such as the Fv/Fm ratio, macro-nutrients, and CO$_2$ variables, were used to detect responses through a comparison of before and after conditions (i.e., Δ = [parameter]$_{postf}$ – [parameter]$_{pref}$). In particular, it should be noted that the Fv/Fm ratios promptly increased from <~0.3 to 0.56 (± 0.08) in the two days following iron addition, indicating a relief in the iron-limitation on phytoplankton growth. The subarctic NP SEEDS-1 experiment, which was conducted under temperature conditions ideal for diatom growth (~8°C) and with shallow MLDs (~10 m), produced the greatest changes in biogeochemical parameters.

The aOIF experiments have generally led to changes in the size of the phytoplankton community from pico and nano-phytoplankton to micro-phytoplankton. This effect was particularly noticeable as diatoms became the dominant species during IronEx-2, SOIREE, EisenEx, SEEDS-1, SOFeX-S, EIFEX, and SERIES, with micro-phytoplankton accounting for ~44% of total phytoplankton community in SOFeX-N. The shift to a diatom-dominated community appears be related to the initial availability of silicate (i.e., initial silicate was >~3 μM in all the experiments just listed). Diatom-dominated blooms induced >4.5-fold increases in chlorophyll-a concentrations and accounted for >65% of the chlorophyll-a increase (Boyd et al., 2000; Gervais et al., 2002; Coale et al., 2004; Smetacek et al., 2012). However, as silicate concentrations decreased to <2 μM due to removal by the elevated diatom abundance, the extent of diatom blooms rapidly declined. In SAGE and LOHAFEX, had low initial levels of silicate (< 2 μM), pico and nano-phytoplankton dominated communities, and diatom growth was limited by the lack of available silicate. These results suggest that to develop a massive phytoplankton bloom, a changeover to a diatom-dominated community after iron addition is needed. A necessary, but not sufficient condition, for such a change to occur is the availability of silicate. Silicate alone is not expected to be sufficient because diatom-dominated blooms with distinct increases in the chlorophyll-a concentration were not observed in all experiments with high initial silicate concentrations. IronEx-1 and SEEDS-2 had high initial silicate levels (>~4 μM), which were conducive to the development of a diatom-dominated bloom, but the bloom was suppressed due to high grazing pressure. Taken together, the OIF results suggest that both mesozooplankton grazing rates as well as the initial silicate concentrations played a role in limiting the stimulation of diatom-dominated blooms after artificial iron enrichment.

In some experiments (IronEx-1, SEEDS-2, SAGE, and LOHAFEX) there was little change in the carbon export flux, while in others (IronEx-2, SOIREE, EisenEx, SEEDS-1, SOFeX-S, EIFEX, and SERIES) there was a >2-fold increase in PP within the mixed layer, with massive diatom-dominated blooms. However, even in the latter, changes in the carbon export flux

differed from experiment to experiment. In SEEDS-1 and SOIREE there was little increase in export flux. However, it has been reported that changes in in the POC concentrations following an increase in PP can take three to four weeks (Buesseler et al., 2005), whereas these two experiments were conducted over only about two weeks, which suggests that the duration of each experiment was too short to detect downward carbon export. In SERIES, there was a distinct increase in the carbon export flux within the mixed layer (30 m), but there was no increase in the export flux below this because the abundance of heterotrophic bacteria was elevated after iron addition rapidly remineralized POC within the mixed layer (
[revised manuscript text omitted]
 this increase was caused by sampling biases (Nodder et al., 2001; Nodder and Waite, 2001). Likewise, SEEDS-1 $^{234}$Th-based POC fluxes at 50-m depth over day 9–13 were estimated to be 423 mg m$^{-2}$ d$^{-1}$, but the drifting trap only recorded 141 mg m$^{-2}$ d$^{-1}$ at 40-m depth over day 12–14, 3 times lower (Table 5) (Aono et al., 2005; Aramaki et al., 2009). This large discrepancy between the two methods might be caused by the under-sampling of POC into the drifting traps (Nodder and Waite, 2001; Aono et al., 2005).

To resolve the potential biases in traditional sediment traps, a neutrally buoyant (and freely drifting) sediment trap (NBST) was developed (Valdes and Price, 2000; Valdes and Buesseler, 2006; Lampitt et al., 2008). Through preliminary experiments conducted in June and October 1997 at the Bermuda Atlantic Time-series Study site, Buesseler et al. (2000) showed that an NBST system could reduce the invasion/inclusion of zooplankton into the trap samplers, and that NBST-based $^{234}$Th fluxes were comparable with water-column based estimates. LOHAFEX has been the only OIF experiment so far that has measured particle export using PELAGRA (Particle Export measurement using a LAGRAngian trap) sediment traps based on the NBST system (Martin et al., 2013). However, the PELAGRA sediment traps deployed below the mixed layer (at 200 m and 450 m) did not detect iron fertilization-induced carbon export even though PP did increase within the mixed layer. Water-column based $^{234}$Th measurements estimated the POC flux at a 100-m depth to be ~94 mg m$^{-2}$ d$^{-1}$, whereas the PELAGRA

sediment traps estimated the flux at 200 m and 450 m to be only ~12 mg m$^{-2}$ d$^{-1}$. It should be noted that both sediment traps and water-column based $^{234}$Th measurements have a limited ability to fully scan the vertical profile of POC fluxes and, therefore, are of limited used in determining the fate of iron-induced POC in the water column.

To resolve the full column more effectively, LOHAFEX employed a UVP, which provided photographic evidence of sinking particles (particle size ≥100 μm) from the surface down to ~3,000-m depth, with ~0.2 m vertical resolution (Smetacek et al., 2010; Martin et al., 2013). Through an analysis of particle size distributions, the UVP also allowed particles to be classified into fecal pellets, aggregates, and live zooplankton. Vertical total particle volume profiles obtained from UVP indicated the maximum particle flux at a 75-m depth (~0.3 mm$^3$ L$^{-1}$), with a gradual decrease to 150 m (~0.15 mm$^3$ L$^{-1}$). Interestingly, large particles (i.e., zooplankton) were copious between 75-m and 100-m depth, suggesting that there might be high grazing pressure, which might explain the large discrepancy between the 100 m (water-column based $^{234}$Th method) and 200 m / 450 m (PELAGRA sediment trap) POC flux estimates (i.e., rather than a sampling bias in sediment trap data) (Martin et al., 2013). To continuously monitor vertical changes in POC flux following iron addition, EIFEX used a transmissometer, providing high vertical resolution (~24 data points per meter) and tracking of the iron-induced flux down to ~3,000 m, even though, unlike UVPs, transmissometers do not allow classification of particles (Smetacek et al., 2012). Improving on this method, SOFeX-N applied autonomous carbon flux explorers equipped with transmissometers, designed to float along with the currents. Three autonomous carbon flux explorers were deployed, two explored the 'iron fertilized in-patch' and one acted as a 'control' outside the patch. Carbon flux explorers could continuously monitor in the field for up to 18 months beyond the initial deployment, which allowed SOFeX-N to observe 'episodic raining' in the iron-fertilized waters (Bishop et al., 2004), indicating a high carbon export flux long after artificial iron addition.

The combination of multiple approaches is essential to the successful detection of the POC produced in response to iron addition and its fate. NBST systems (e.g., the PELAGRA sediment trap) are appropriate for quantifying the aOIF-induced POC flux in the upper waters (<~400-m depth), especially 
[revised manuscript text omitted]

Secondly, the effectiveness of aOIF may also be offset leading to changes in the ocean ecosystem following OIF, such as a decrease in dissolved oxygen and an increase in domoic acid (DA) levels. The decomposition of iron addition-enhanced biomass may cause decreased oxygen concentrations in subsurface waters (Williamson et al., 2012). Although mid-water oxygen depletion has not been reported from aOIF experiments to date, early modeling studies suggest that anoxic conditions may develop after long-term and large scale OIF (Sarmiento and Orr, 1991). However, more sophisticated and realistic models suggest that OIF produces well-oxygenated conditions, without the development of anoxic conditions, even under climate change scenarios (Oschlies et al., 2010; Keller et al., 2014). Thus, hypoxia/anoxia development in response to iron additions is unlikely to be a primary concern.

The changes in phytoplankton community composition after iron addition discussed in Section 2.4 may also have unintended consequences; for example, they could lead to potentially toxic species dominating plankton assemblages (Silver et al., 2010; Trick et al., 2010). Some aOIF experiments (e.g., IronEx-2, SOIREE, EisenEx, SOFeX-N/S, and SERIES) generated large blooms dominated by pennate diatoms belonging to the genus '*Pseudo-nitzschia*' (de Baar et al., 2005; Trick et al., 2010). Some '*Pseudo-nitzschia*' species have the capacity to produce the neurotoxin DA that is known to detrimentally affect marine ecosystems. However, no DA was found during EisenEx and SERIES, even though '*Pseudo-nitzschia*' were dominant (Gervais et al., 2002; Marchetti et al., 2008; Assmy et al., 2007). Phytoplankton samples used to estimate DA production had been stored for a long time before the analysis, for example, 12 years in IronEx-2 and four years in SOFeX-S (Silver et al., 2010). Trick et al. (2010) argued that phytoplankton samples stored for a long time would have degraded, leading to an under-estimation in DA production. This implies that accurate information about changes in DA production in response to iron addition might not be available. However, the IronEx-2, and SOFeX-S experiments found discernable changes in DA production, even if the original DA might have degraded (Silver et al., 2010). It is likely that several phytoplankton samples (e.g., *Pseudo-nitzschia* abundance: $1.3 \times 10^6$ cells $L^{-1}$ in IronEx-2 and $7.5 \times 10^4$ cells $L^{-1}$ in SOFeX-S) collected with a net tow were suitable to detect these changes. During IronEx-2 and SOFeX-S, high cell abundances of '*Pseudo-nitzschia*' ($10^6$ and $10^5$ cells $l^{-1}$, respectively) combined with moderate DA quotas (0.05 and 1 pg DA $cell^{-1}$, respectively) produced toxin levels as high

as 45 ng DA $l^{-1}$ in IronEx-2 and 220 ng DA $l^{-1}$ in SOFeX-S, i.e., toxin levels high enough to damage marine communities in coastal waters. Therefore, it is necessary to quantify DA production in response to iron additions, with concentrated phytoplankton samples (i.e., large numbers of cells) using a net tow. This, once again, indicates that such processes 
[revised manuscript text omitted]
 the increasing nutrient flux from the subsurface waters to the surface waters by the shoaling of MLD, along with the receipt of sufficient solar radiation (Moore and Abbott, 2002). Prior to December, phytoplankton growth is mainly limited due to light availability (Mitchell et al., 1991; Veth et al., 1997; Abbott et al., 2000), while after January (i.e., during late summer and early autumn from February to March) it is mainly limited due to silicate availability. In previous aOIF experiments in the SO that have been conducted between spring and early autumn, PP was mainly limited by iron and/or silicate availability rather than light availability (de Baar et al., 2005; Smetacek and Naqvi, 2008; Peloquin et al., 2011). In addition, the grazing pressure of mesozooplankton (i.e., copepods) on large diatoms was also a major limiting factor in diatom production (Coale et al., 2004; Martin et al., 2013), and was generally higher during late summer and early autumn (February to March) (Le Quéré et al., 2016). Considering the key factors (i.e., micro/macro nutrient availability, light availability, and grazing pressure) controlling PP in the SO, the most appropriate timing for an aOIF experiment to start in the SO is likely to be the early summertime (i.e., late December to early January).

How long: The third consideration for a successful aOIF experiment is the duration. Although the periods that phytoplankton blooms have been maintained by OIF have lasted from ~10 to ~40 days (Kolber et al., 1994; Martin et al., 1994; Coale et al., 1996; Boyd et al., 2000; Tsuda et al., 2005; Coale et al., 2004; Boyd et al., 2004; Smetacek et al., 2012), it has

also been suggested that most aOIF experiments did not cover the full response times from onset to termination (Boyd et al., 2005). For example, SOIREE and SEEDS-1 had relatively short observation periods (13 days) and saw increasing trends in PP throughout the experiments (Fig. 10a), suggesting that the observation period should have been extended. Furthermore, after the end of SOIREE, ocean color satellite images showed continued high chlorophyll-a concentrations (>1 mg m$^{-3}$) in the iron fertilized patch, which was visible as a long ribbon shape that extended some 150 km for >40 days (~6 weeks) after the initial iron addition (Fig. 10b) (Abraham et al., 2000; Westberry et al., 2013). This indicates that short experimental durations may not be sufficient for detecting the full influence of artificial iron additions on PP and ecosystem (Figs. 8b and 10) (Boyd et al., 2000; Tsuda et al., 2003; de Baar et al., 2005). SOFeX-S also resulted in relatively low export production despite the high PP due to the experimental duration being insufficient to cover the termination of the phytoplankton bloom. However, SERIES, SEEDS-2, EIFEX, and LOHAFEX did fully monitor all phases of the phytoplankton bloom from onset to termination. EIFEX, the third-longest aOIF experiment, at 39 days, was the only one that observed iron-induced deep export production between day 28 and 32 (Table 5 and Fig. 8a) (Smetacek et al., 2012; Assmy et al., 2013). Furthermore, long-term observations covering the later stage of bloom development during nOIF experiments resulted in much higher Fe:C export efficiencies compared to the short-term aOIF (Blain et al., 2007; Pollard et al., 2009). Based on previous aOIF experiments, it would, therefore, be important to detect the full phase of a phytoplankton bloom to determine accurately the amount of iron-induced POC exported out of the mixed layer. The observation period is, therefore, an important factor to consider in budget and effectiveness estimates. It is suggested that the experimental duration should be a minimum of ~40 days based on the SOIREE experiment, which produced the longest iron-induced bloom (i.e., the longevity of >40 days).

How: The fourth consideration for a successful aOIF experiment lies in the strategy/approach of adding and maintaining dissolved iron within the upper mixed layer to produce a phytoplankton bloom. First, the chemical form for iron addition should be acidified iron-sulfate, which is less expensive and more bioavailable than other iron compounds. The amount of iron-sulfate required is calculated according to the target concentration of the dissolved iron and volume (MLD × patch size). Based on bottle incubation experiments, target iron concentrations of ~2–4 nM are recommended to stimulate maximum phytoplankton growth due to the rapid losses of added iron by horizontal advection/diffusion and oxidation to poorly bioavailable iron(III) (Coale et al., 1996; Coale et al., 1998; Bowie et al., 2001). For patch size, a biogeochemical model study showed that a fertilized patch size of 156 km$^2$ maintained an iron concentration above 0.3 nM for 56 days, while a longer period of 194 days required a fertilized patch size of 160,000 km$^2$ (Xiu and Chai, 2010). This is because, compared to larger iron-fertilized patches, a smaller patch size tended to lose iron more rapidly due to dilution effects with unfertilized water. Previous aOIF experiments also produced similar results to this model study. The lateral dilution rate (<0.25 d$^{-1}$) during SAGE, which had the smallest fertilized patch size (36 km$^2$) of the SO experiments, was two times higher than the rates (<0.11 d$^{-1}$) in the SO experiments with a larger fertilized patch size (e.g., EIFEX fertilized with a patch size of 167 km$^2$ and SOFeX-N/S fertilized with a patch size of 225 km$^2$) (Coale et al., 2004; Harvey et al., 2010; Law et al., 2011; Smetacek et al., 2012). Therefore, it would be more appropriate to fertilize a large area (e.g., LOHAFEX had the largest aOIF experiment at 300 km$^2$), which would reduce the dilution effect with unfertilized waters (Xiu and Chai, 2010). Based on a ~2 nM iron concentration for a patch size of 300 km$^2$ and MLD of ~60 m, it would need ~2,000 kg of iron(II) to be applied in a fertilization experiment. Iron should be released into the wake of a ship, with the release track describing an expanding spiral (or square) in the eddy center, with a regular interval of ~1 km throughout the patch, because it is easier to locate a fertilized patch than a point release (Watson et al., 1991). In addition, it should be completed within ~24 hours because of the time-dependent phytoplankton response within the iron-fertilized patch. Previous aOIF experiments have shown that multiple iron additions (≥2 infusions) are needed to maintain the dissolved iron concentration required to derive maximum phytoplankton growth within the fertilized patch. For example, in SOIREE it was found that four additions of iron at intervals of about three days led to persistently high levels of both dissolved and particulate iron within the mixed layer, with a rapid reduction at the end of the experiment,

combined with an increase in the concentration of iron-binding ligands (Bowie et al., 2001). In both EIFEX and SOFeX-S, it was also found that multiple iron(II) infusions (in particular, two infusions with intervals of 13 days in EIFEX and four infusions with intervals of four days in SOFeX-S) allowed iron to persist in the mixed layer longer than its expected oxidation kinetics. The relatively low oxidation rates were related to a combination of photochemical production, slow oxidation and, possibly, organic complexation (Croot et al., 2008). Blain et al. (2007) explained that the higher carbon sequestration effectiveness of nOIF experiments compared to aOIF experiments partly resulted from the slow and continuous iron addition that occurs in the natural environment. Large amounts of iron addition at one time can lead to a substantial loss of artificially added iron. Therefore, for an experimental duration of >~40 days, a minimum of three iron infusions at intervals of ~10–15 days would be required to prevent the iron limitation on phytoplankton growth, based on the EisenEx and EIFEX experiments (Nishioka et al., 2005; Smetacek et al., 2012).

What: The fifth consideration for a successful aOIF experiment is the effective tracing of the fertilized patch, including the detection of carbon sequestration (Buesseler and Boyd, 2003). All previous aOIF experiments used physical tracers, in particular GPS and ARGO equipped drifting buoys, to follow the iron fertilized patch. A drifting buoy is a natural and passive system moving along with the currents, but it can be escaped from the fertilized patch due to the action of strong winds (Tsumune et al., 2005). Therefore, the release of GPS and ARGO equipped drifting buoys at the center of the patch after the iron infusions would provide a visual map showing the tracked positions of the fertilized patch. An inert chemical tracer, such as $SF_6$, would also be an excellent option for following the fertilized patch after iron addition. Previous aOIF experiments have shown that the $SF_6$ measurements based on underway sampling systems can be used to determine accurately time-dependent vertical and lateral transport of iron-fertilized patches. Many subsequent aOIF experiments have also used tracing methods based on the observation of biogeochemical parameters (such as the Fv/Fm ratio, chlorophyll fluorescence, and underway $pCO_2$) before and after iron addition (Martin et al., 1994; Coale et al., 1996; Boyd et al., 2000; Coale et al., 2004; Boyd et al., 2004; Tsuda et al., 2005; Smetacek et al., 2012). The Fv/Fm ratio can be easily and promptly used as an indicator to track the fertilized patch due to its rapid response to iron addition. Direct measurements of carbon export fluxes to determine the effectiveness of aOIF should be conducted by deploying an NBST at two depths: (1) within the mixed layer to detect increases in iron-induced POC in the surface layer along with the calibration of a water-column based $^{234}$Th method, and (2) below the depth of the winter MLD to detect iron-induced export carbon fluxes into intermediate/deeper waters (Bidigare et al., 1999; Nodder et al., 2001; Boyd et al., 2004; Buesseler et al., 2004; Coale et al., 2004; Aono et al., 2005; Buesseler et al., 2005; Tsuda et al., 2007; Smetacek et al., 2012; Martin et al., 2013). Sinking-particle profiling systems mounted on autonomous floats, such as a transmissometer and UVP that measure and photograph sinking particles, could provide a record of the temporal and vertical evolution of iron-induced POC stocks through successive depth layers down to ~3,000-m depth for ~20 months after deployment, once calibrated using POC fluxes measured from sediment traps and/or a water-column based $^{234}$Th method (Bishop et al., 2004; Smetacek et al., 2012; Martin et al., 2013). Future OIF experiments would benefit from these technological advances, enabling a more efficient tracing of the carbon export flux and particle size and composition at higher vertical and temporal resolution than has been possible in the past. Hence, the application of an NBST system and water-column based $^{234}$Th method to direct flux estimates, combined with autonomous sinking-particle profilers of a transmissometer and UVP, will enable the quantitative and qualitative evaluation of the effectiveness of aOIF and direct observation of iron-induced carbon export fluxes after artificial iron additions.

[revised manuscript text omitted]

*High Nutrient Low Chlorophyll and Low Silicate (HNLCLSi) region; **natural OIF experiments (CROZet natural iron bloom and EXport experiment: CROZEX; KErguelen Ocean and Plateau compared Study: KEOPS).

Sources are Martin et al. (1994); Coale et al. (1996); Coale et al. (1998); Boyd et al. (2000); Boyd and Law (2001); Gervais et al. (2002); Tsuda et al. (2003); Boyd et al. (2004); Coale et al. (2004); Bakker et al. (2005); Boyd et al. (2005); de Baar et al. (2005); Nishioka et al. (2005); Hoffmann et al. (2006); Law et al. (2006); Blain et al. (2007); Boyd et al. (2007); Rees et al. (2007); Tsuda et al. (2007); Pollard et al. (2009); Strong et al. (2009); Harvey et al. (2010); Smetacek et al. (2012); and  Martin et al. (2013).

**Table 2.** Summary of artificial ocean iron fertilization (aOIF) experiments; objectives, significant results, and limitations.

| | Experiment | Objectives | Significant results | Limitations |
|---|---|---|---|---|
| 1 | IronEx-1 | • To test the hypothesis that artificial iron addition will increase phytoplankton productivity by relieving the iron limitation of phytoplankton in high-nutrient low chlorophyll regions | • Small increases in the Fv/Fm ratio, chlorophyll-a concentration, and primary production (PP)
• Insignificant changes in nutrients and $pCO_2$ concentrations | • Single iron addition
• Insufficient experimental periods to observe the full phases of biogeochemical responses from the onset to termination after iron additions
• Micro/macro-nutrient limitations |
| 2 | IronEx-2 | • To test three hypotheses that were advanced to explain the weak biogeochemical response observed during IronEx-1 | • Dramatic changes in biogeochemical responses; close to support for Martin's hypothesis
• Taxonomic shift toward diatom-dominated phytoplankton communities | • No export flux measurements in the deep ocean
• Insufficient experimental duration |
| 3 | SOIREE | • To test the iron hypothesis in the Southern Ocean | • Diatom-dominated bloom
• No measurable change in carbon export | • Insufficient experimental duration |
| 4 | EisenEx | • To test the hypothesis that atmospheric dust inputs might have led to a dramatic increase in ocean productivity during the Last Glacial Maximum due to the relief of iron-limited conditions for phytoplankton growth | • Diatom-dominated bloom
• No clear differences in carbon flux between in-patch and outside-patch | • Light limitation by storms
• Insufficient experimental duration |
| 5 | SOFeX-N | • To address the potential for iron and silicate interactions to regulate the diatom bloom | • Remarkable increase in diatom biomass
• Observation of large export flux event with transmissometers | • Entrainment of dissolved silicate into the fertilized patch by physical mixing
• No direct measurement of export fluxes with sediment traps |
| 6 | SOFeX-S | • To address the potential for iron and silicate interactions to regulate the diatom bloom | • Significantly enhanced export fluxes out of the mixed layer, but similar to those for natural blooms | • Insufficient experimental duration |
| 7 | EIFEX | • To confirm that aOIF experiments can increase export production | • Observation of all the phases of the phytoplankton bloom from onset to termination
• Significant carbon export to deeper layers (down to 3,000 m) due to the formation of aggregates with rapid sinking rates
• The occurrence of rapidly sinking large aggregates | |

| | Experiment | Objective | Significant results | Limitations |
|---|---|---|---|---|
| 8 | SAGE | • To determine the response of phytoplankton dynamics to iron addition in high nutrient low chlorophyll and low silicate (HNLCLSi) regions
• To test the assumption that the response of phytoplankton blooms to artificial iron addition can be detected by the enhanced air-sea exchanges of climate-relevant gases | • No shift to a diatom-dominated community
• No detection of fertilization-induced export | • High dilution rate by small patch size |
| 9 | LOHAFEX | • To trace the fate of iron-stimulated phytoplankton blooms and deep carbon export in HNLCSi regions | • Observation of all the phases of the phytoplankton bloom from onset to termination
• No shift to a diatom-dominated community
• No detection of fertilization-induced export
• High grazing pressure and active bacterial respiration | |
| 10 | SEEDS-1 | • To investigate the relationship between phytoplankton biomass/community and dust deposition in the subarctic North Pacific (NP)
• To investigate changes in phytoplankton composition and vertical carbon flux | • A shift from oceanic diatoms to fast-growing neritic ones
• The largest changes in biogeochemical parameters of all aOIF experiments
• No detection of large POC export flux | • Single iron addition
• Insufficient experimental duration |
| 11 | SERIES | • To compare the response of phytoplankton in eastern subarctic with that in the western subarctic ecosystem
• To investigate the most significant factor that controls the beginning to the ending of the phytoplankton bloom induced by iron addition | • Observation of all phases of the phytoplankton bloom from onset to termination
• No significant increases in export fluxes below the mixed layer depth
• High bacterial remineralization and mesozooplankton grazing pressure | |
| 12 | SEEDS-2 | • To investigate the most significant factor that controls the beginning to the ending of the phytoplankton bloom induced by iron addition | • Observation of all phases of the phytoplankton bloom from onset to termination
• No shift to a diatom-dominated community
• No significant increases in export fluxes
• Extensive copepod grazing | |
| 13 | FeeP | • To investigate the impact of iron and phosphate co-limitation on PP | • Increases in pico-phytoplankton abundances | |

[revised manuscript text omitted]

To be continued

| - | Experiment | Time | Location | Research Vessel | Fe (kg) (day) | Initial Fe (nM) | After Fe (nM) | Tracer | Patch size (km²) | Period (days) | Region |
| --- | --- | --- | --- | --- | --- | --- | --- | --- | --- | --- | --- |
| 9 | EIFEX | Feb–Mar 2004 | 50° S, 2° E  Southern Ocean  Atlantic sector | RV *Polarstern* | ①1406 (0)  ②1406 (13) | 0.20 | 1.50  0.34 | | 167 | 39 | HNLC |

| No. | Name | Date | Location | Ship | Depth | | | | | |
|---|---|---|---|---|---|---|---|---|---|---|
| | | | 27.5° N 22.5° W | | | | | | | |
| 10 | FeeP | Apr–May 2004 | North Atlantic | RV *Charles Darwin* | ①1840 (0) | 0.20 | 3.00 | SF₆ | 25 | 21 | LNLC |
| | | | Subtropical north-east Atlantic | RV *Poseidon* | | | | | | | |
| | | | 46.7° S 172.5° E | | ①265 (0) | | 3.03 | | | | |
| 11 | SAGE | Mar–Apr 2004 | Southern Ocean | RV *Tangaroa* | ②265 (6) | 0.09 | 1.59 | SF₆ | 36 | 15 | HNLCLSi |
| | | | | | ③265 (9) | | 0.55 | | | | |
| | | | Southeast of New Zealand | | ④265 (12) | | 1.01 | | | | |
| | | | 48° N, 166° E | RV *Hakuho-Maru* | ①332 (0) | | 1.38 | | | | |
| 12 | SEEDS-2 | Jul–Aug 2004 | North Pacific | | | 0.17 | | SF₆ | 64 | 26 | HNLC |
| | | | Western subarctic gyre | RV *Kilo-Moana* | ②159 (6) | | | | | | |
| | | | 48° S, 15° W | RV *Polarstern* | ①2000 (0) | | 2.00 | | | | |
| 13 | LOHAFEX | Jan–Mar 2009 | Southern Ocean | | | - | | SF₆ | 300 | 40 | HNLCLSi |
| | | | | | ②2000 (21) | | | | | | |

Atlantic sector

[revised manuscript text omitted]

(a)

[Figure]

[Figure]

[Figure]

[Figure]

(a)

[Figure]

(b)

[Figure]

[Figure]

(a)

[Figure]

SOIREE (Day 05)    SOIREE (Day 46)

---

## Author Response (AR3)

Dear Reviewers,

Thank you for your very thoughtful review of our paper. We provide our responses (Plain text with black and blue colors) to all comments (*italic text*) below. We hope that the revision addresses your concerns.

**- Response to Reviewer Comments -**

**Reviewer #1**

**-Specific Comments:**

**1.** *Most importantly, the authors are still only giving vague directions as to the depths at which particle export needs to be measured. These measurements, however, are absolutely crucial if we are to judge the effectiveness of iron fertilisation as a means of carbon sequestration, and there has been substantial debate in the particle flux literature precisely about this point. Unfortunately, the authors only talk vagues of "intermediate and deep waters" (e.g. Page 4 Line 1, P 26 L 28, and P 27 L 6; clear directions on depths are also lacking in Section 3.1). Measurement depths for particle flux absolutely need to be spelled out explicitly and as much as possible standardised in future experiments so that meaningful comparisons can be made. In the abstract, and on P 29 L 18, the authors indicate that in KIFES, they would want to place one sediment trap "within the mixed layer and another below it". This is absolutely not appropriate: sediment traps are known to perform poorly within the mixed layer, so the shallowest depth for a trap should be just below the mixed layer, so that mixed layer export can be measured. The second critical depth for a sediment trap is the depth of deepest winter-time convective mixing, which can be estimated from Argo float data for a given study area. In LOHAFEX, that was how the depth of the deep sediment traps at around 400 m was chosen. My recommendation would be that the authors make clear recommendations that particle flux must be measured just below the mixed layer (i.e. 10–20 m below) and at the depth of the winter mixed layer. I am open to alternative suggestions from the authors, but whatever they intend to do absolutely must be spelled out clearly, and the importance of these depths should be explained properly in the manuscript. As it is described at present, I would have real concerns as to whether KIFES would actually collect the correct measurements. It is critical that this manuscript makes appropriate, and explicit, recommendations in this regard.*

● Thank you for pointing out this deficiency. We have modified **'intermediate and deep waters'** to **'below the winter mixed layer'**. We have added sentences explaining '*specific depth for particle flux measurement', 'the importance of these depths'*, and defining '*effectiveness of aOIF*'.

➔ Abstract (lines 21–23, page 1):

[revised manuscript text omitted]

➔ 5.2.4 Year four plan (lines 18–21, page 29):

"(5) Measure iron-induced carbon export fluxes for the regions both inside and outside the center of an eddy structure using NBST systems at two depths **(i.e., just below the *in situ* MLD and at the winter MLD)** along with the calibration of water-column based [234]Th measurements and autonomous profilers equipped with a transmissometer and an UVP."
* * *
**2.** *The authors in several places talk about "effectiveness" of aOIF, but I don't remember that being properly defined in the manuscript. As above, I think this does need to be defined, and in my view an appropriate definition would be the amount of additional carbon exported below the winter mixed layer depth as a result of iron addition. The importance of other metrics, such as the amount of carbon sequestered relative to the amount of iron added, might also be mentioned in this context.*

● We added the definition about '*effectiveness of aOIF*'. Please find the Abstract (lines 22–23, page 1) and 1 Introduction (lines 1–4, page 4).

**3.** *In the abstract, Line 27, the authors might want to briefly spell out what these questions are. The abstract can probably be shortened by editing the language carefully throughout. I would also recommend that the depths at which carbon sequestration should be monitored are explicitly spelled out in the abstract (and, I reiterate, these depths are not inside the mixed layer and [somewhere] below the mixed layer).*

● We have briefly spelled out *'what these questions are'*, shortened the *'abstract'*.

➔ Abstract (lines 15–40, page 1):

"Since the start of the industrial revolution, human activities have caused a rapid increase in atmospheric $CO_2$ concentrations, which have, in turn, had an impact on climate leading to global warming and ocean acidification. Various approaches have been proposed to reduce atmospheric $CO_2$. The 'Martin (or Iron) Hypothesis' suggests that ocean iron fertilization (OIF) could be an effective method for stimulating oceanic carbon sequestration through the biological pump in iron-limited, high-nutrient, low-chlorophyll **(HNLC)** regions. To test the Martin hypothesis, 13 artificial OIF (aOIF) experiments have been performed since 1990 in **HNLC regions**. These aOIF field experiments have demonstrated that primary production can be significantly enhanced by the artificial addition of iron. However, **except in the Southern Ocean European Iron Fertilization Experiment, the effectiveness of aOIF (i.e., the amount of iron-induced carbon export flux below the winter mixed layer depth) has been** unexpectedly low compared to that achieved by natural phytoplankton blooms. These results, including possible side effects have been debated amongst those who support and oppose aOIF experimentation, and many questions **such as effectiveness of scientific aOIF, environmental side effects, and international aOIF law frameworks** remain. In the context of increasing global and political concerns associated with climate change, it is valuable to examine the validity and usefulness of the aOIF experiments. To maximize the effectiveness of aOIF experiments under international aOIF regulations in the future, we suggest a design that incorporates several **components**. (1) Experiments conducted in the center of an eddy structure when grazing pressure is low and silicate levels are high (e.g., in the Southern Ocean south of polar front during early summer). (2) Shipboard observations extending over a minimum of ~40 days, with multiple iron injections (at least 2 (or 3) iron infusions of ~2,000 kg with an interval of ~10–15 days to fertilize a patch of 300 $km^2$ and obtain a ~2 nM concentration). (3) Tracing of the iron fertilized patch using both physical (e.g., a drifting buoy) and biogeochemical (e.g., sulfur hexafluoride and photosynthetic quantum efficiency) tracers. **(4) Employment of** neutrally buoyant sediment trap**s and application of the** water-column derived $^{234}$Thorium method at two depths **(i.e., just below the *in situ* mixed layer depth and at the winter mixed layer depth),** with autonomous profilers equipped with an underwater video profiler and a transmissometer. (5) Monitoring of side effects on marine/ocean ecosystems, including production of climate-relevant gases (e.g., $N_2O$, dimethyl sulfide, and halogenated volatile organic compounds), **decline in oxygen inventory**, and development of toxic algae blooms, **with optical sensor equipped**

**autonomous moored profilers and/or autonomous benthic vehicles.** Lastly, we introduce the scientific aOIF experimental design guidelines for a future Korean Iron Fertilization Experiment in the Southern Ocean."
* * *
**4.** *In the first paragraph of the introduction, I am quite strongly of the opinion that the authors should also state that there is an urgent need to reduce global greenhouse gas emissions.*

● We have added statement.

➔ 1 Introduction (lines 2–5, page 3):

"Since the start of the industrial revolution, human activities have caused a rapid increase in atmospheric carbon dioxide ($CO_2$, **a major greenhouse gas**) from ~280 ppm (pre-industrial revolution) to ~400 ppm (present day) (http://www.esrl.noaa.gov/), which has, in turn, led to global warming and ocean acidification, **indicating that there is an urgent need to reduce global greenhouse gas emissions** (IPCC, 2013) (Fig. 1)."
* * *
**5.** *P 3 L 12: why is "ocean fertilisation" italicised?*

● We changed to plain text.
* * *
**6.** *P 6 L 10: Presumably you mean "By the end of the 20th century"?*

● We apologize this confusing. Based on **Reviewer #2's editing**, we have removed this sentence.
* * *
**7.** *P 10 L 26: Correct spelling of the name is "Behrenfeld"*

● We corrected it.
* * *
**8.** *P 16 L35: What is the difference between "tracking" and "quantifying" export flux? I think one will do. Again, please do be explicit throughout the manuscript as to what depths you are thinking about.*

● We modified "through both **tracking and quantifying** export flux" to "through **quantification of** export fluxes". For '*specific depths*', please refer **Reviewer #1's Response 1.**

**9.** *P 18 L 12: Saying that thorium and sediment traps are "of limited use" in determining the fate of POC is not appropriate; the implication of saying "of limited use" is that these measurements are not very useful. The authors should re-phrase this to say "and, therefore, these methods should ideally be complemented with additional techniques that can measure particle stocks at high depth resolution throughout the water column".*

● Reviewer is correct. We have revised this sentence.

**10.** *P 18 L 17: The UVP, and also transmissometer and other optical measurements, actually do not give a particle flux as such, they show the particle concentration. There are ways to estimate fluxes from these measurements using assumptions about particle sinking rates, but in the first instance they provide information about stocks. The authors might also want to refer to the study by Briggs et al. (2011) High-resolution observations of aggregate flux during a sub-polar North Atlantic spring bloom in Deep-Sea Research 1. This paper provides a very nice example of using backscattering and chlorophyll sensors to gain high-resolution data about a particle flux event in a Lagrangian study, so a similar scenario to an iron fertilisation.*

● Thank you for pointing out this deficiency and recommending nice study. We have modified *'particle flux'* to *'stock'* or *'concentration'* and have cited Briggs et al. (2011).

➔ 3.1 Export flux measurement methods (lines 34–36, page 17):

"Total vertical particle volume profiles obtained from the UVP indicated **a maximum concentration** at 75 m (~0.3 mm$^3$ L$^{-1}$), with a gradual decrease to 150 m (~0.15 mm$^3$ L$^{-1}$)."

➔ 3.1 Export flux measurement methods (line 39, page 17 – line 2, page 18):

"To continuously monitor vertical changes in POC **stocks** following iron addition, EIFEX used a transmissometer, providing high vertical resolution (~24 data points per meter) and tracking of the iron-induced **stocks** down to ~3,000 m, even though, unlike UVPs, transmissometers do not allow classification of particles (Smetacek et al., 2012)."

➔ 3.1 Export flux measurement methods (lines 7–9, page 18):

**"Furthermore, recent studies also reported that use of optical spike signals in particulate backscattering and fluorescence, measured from autonomous platforms such as gliders and floats, can provide high-resolution observations of POC flux (Briggs et al., 2011; Dall'Olmo and Mork, 2014)."**

**11.** *P 20 L 10: This sentence isn't quite clear. What does the 20% refer to: a 20% increase of the total SO DMS flux? Or a 20% increase in the 2% of the SO?*

● We apologize for this confusion. We have modified this sentence.

➔ 3.2 Considering environmental side effects (lines 34–36, page 19):

"Estimates derived by the extrapolation of SOFeX-N DMS production results suggested that fertilizing ~2% of the SO **area** over the course of a week would **derive a 20% increase of the total SO DMS flux,** which would lead to a 2°C decrease in air temperature over the SO (Wingenter et al., 2007)."

**12.** *P 21 L 12: The statement about domoic acid levels needs a reference. In several places throughout this paragraph, the authors rely on the conclusions reached by Trick et al. 2010 – however, I don't think that this study gives a reliable guide to possible impacts of Pseudonitzschia blooms. Trick et al. chiefly conducted bottle experiments that yielded a range of several orders of magnitude in cell quotas of DA. Their conclusions were then chiefly based on extrapolating their highest measured cell quota in a bottle incubation with quite unrealistic estimates of likely surface biomass levels to estimate a possible in-water DA concentration. If the authors do wish to cite this paper, I would strongly recommend that these caveats to their conclusions should be pointed out explicitly.*

● Thank you for your suggestion. As Reivewer #1's comments, we have added *'these caveats to their conclusion'* in Section 3.2. We have also added some references.

➔ 3.2 Considering environmental side effects (lines 1–4, page 21):

**"Trick et al. (2010) suggested that large-scale OIF may induce DA accumulation with developing toxic *Pseudo-nitzschia* blooms. However, large uncertainties remain as Trick et al. (2010) simply extrapolated DA concentration based on bottle incubation experiments with HNLC surface waters to the DA production expected from large-scale OIF."**

**13.** *P 24 L1: better to say "along with sufficient levels of solar radiation" instead of using "receipt"*

● Thank you. Based on Reviewer #2's editing, we have modified this sentence as follows:

➔ 4 Future: Designing future aOIF experiments (lines 36–39, page 23):

"To the south of the SO PF, phytoplankton blooms usually occur during early summer (i.e., from late December to early January) due to an increase in the nutrient flux from subsurface waters **induced by winter mixing**, along with **the favorable light conditions provided by a shoaling of ML** (Moore and Abbott, 2002)."

**14.** *P 26 Line 39: MIT and WHOI are entirely separate institutions, despite their collaborative PhD programme.*

- We modified MIT-WHOI to WHOI.

**Reviewer #2**

**-General Comments:**

*This third version of review article by Yoon et al. presenting a summary of results from all ocean iron fertilization experiments (OIF) carried to date and introducing plans for a new OIF experiment in the Bransfield Basin under the leadership of the Korean Polar Research Institute. Despite some improvement in the presentation of results from previous experiments, this new version falls short of addressing the issues and comments of both reviewers: the discussion on why previous experiments led to different outcomes is still superficial and lacks a robust analysis of previous results/data and methods. As a consequence, the answers to the questions of why? where/when? and how? still rest mostly on broad statements (see comments below and in the annotated manuscript) and it is not clear how the plans for KIFES are anchored and improve on the lessons learned from previous experiments. finally I believe the issue of C sequestration (which seems to be the main aim in KIFES) is also largely informed by model simulations, therefore, how additional experiments might inform ocean biogeochemical models and affect their results might be an important aspect when presenting the scientific rationale.*

● Thank you very much for all your valuable comments. We have addressed all the comments as shown in the revised manuscript. Please see below for a description and/discussion of how we addressed your comments. For annotated editing, we have reflected almost your comments and edits.

**-Specific Comments:**

**1.** *p. 5, § 1: The hypothesis mentioned in Coale et al. (1996) are: 1) rapid loss of iron from the patch. 2) Insufficient light after subduction of the patch. 3) Zooplankton Grazing. 4) Limitation by other micro-nutrients or silicate.*

● We have modified *'hypothesis'* as follows:

➔ 2.1.1 Equatorial Pacific (lines 3–7, page 5):

"Four hypotheses were advanced to explain the weak responses observed: (1) the possibility of unforeseen micro-nutrient (e.g., zinc, cadmium, and manganese) or macro-nutrient (e.g., silicate) limitations, (2) the short residence time of bioavailable iron in the surface patch **due to colloidal aggregation and/or sinking of larger particles containing iron, (3) insufficient light brought about by subduction of the patch, and** (4) high grazing pressure **by zooplankton (Martin et al., 1994;** Cullen, 1995; **Coale et al., 1996; Gordon et al., 1998**)."

**2.** *p. 10: Given that the patch was subducted during IronEx 1, the response of the patch in terms of impact of aOIF on biogeochemistry, is not particularly relevant. I recommend the authors focus on the results of IronEx II. That would help shorten section 2.4.1 and keep the discussion to the point.*

● IronEx-1 was the first aOIF experiment to test Martin's hypothesis. As we mentioned specific objectives of this review in Introduction – "**to provide a thorough overview of the aOIF experiments conducted over the last 25 years**", the results of IronEx-1 should be included in this review, even though the magnitude of the biogeochemical response in IronEx-1 was not as large as expected due to the subduction of iron-fertilized patch. Therefore, we would like to keep the description on the results of IronEx-1 in Section 2.4.1.

● We have added the information on *"the patch was subducted during IronEx-1"* in Section 2.4.1 as follows:

➔ 2.4.1 Equatorial Pacific (lines 22–24, page 10):

"While the elevated IronEx-1 Fv/Fm ratios promptly disappeared, **suggesting** rapid iron loss **due to the subduction of the fertilized patch and/or adsorption onto colloidal particles** (perhaps indicative of insufficient iron supply),"

**3.** *p. 11-12, lines 36-38 and 1-3, respectively: I am not sure what point the authors want to make with this paragraph.*

● We apologize for this confusion. We have revised this paragraph.

➔ 2.4.2 Southern Ocean (lines 31–34, page 11):

"Satellite observations were used to investigate the changing spatial and temporal distribution of chlorophyll-a concentration in response to iron fertilization in the fertilized patches compared to the surrounding waters**; for example, SOFeX-N/-S found elevated chlorophyll-a concentrations in fertilized patches after iron addition through satellite images (Fig. 7d)** (Boyd et al., 2000; Coale et al., 2004; Westberry et al., 2013)."

**4.** *p. 15, Section 2.5: I would remove discussion of the FEP in the NA, since it is not relevant in this context (see also annotated manuscript).*

● We have removed *'discussion of the FeeP experiment in the NA'* in **Section 2.2.4, Section 2.4.4, and Section 2.5**.

**5.** *p. 15, Section 2.5, § 1 & 2: These paragraphs only provide background information that is already given in previous sections. I would remove and focus on the significant results as mentioned in the heading.*

● We have removed these paragraphs in Section 2.5.

**6.** *p.15, lines 35-38: The authors argue that Si is necessary for bloom formation. Looking at results from SOFEX-N in Fig. 6 and 7, it does not seem to be necessarily the case. As I have mentioned in a previous review, this might be due to the fact that the authors compared concentrations, when they should actually compare standing stocks (i.e. integrated values for the mixed layer).*

● We apologize for this confusion. SOFeX-N/SAGE/LOHAFEX were commonly conducted in the initial low silicate concentrations. However, unlike SAGE and LOHAFEX, during SOFeX-N, silicate limitation in the iron-fertilized waters was temporary relieved through lateral mixing with the surrounding waters that had relatively higher silicate concentrations (Coale et al., 2004), and it contributed to taxonomic shift toward diatom-dominated communities (from 16% to 44% of total phytoplankton community). Please refer the SOFeX-N results in Section 2.4.2 (lines 13–17, page 12). Finally, we modified this paragraph as follows:

➔ 2.5 Summary of the significant results from aOIF experiments (lines 28–38, page 14):

"This effect was particularly noticeable as diatoms became the dominant species during IronEx-2, SOIREE, EisenEx, SEEDS-1, SOFeX-S, EIFEX, and SERIES. Diatom-dominated blooms induced >4.5-fold increases in chlorophyll-a concentrations and accounted for >65% of the chlorophyll-a increase (Boyd et al., 2000; Gervais et al., 2002; Coale et al., 2004; Smetacek et al., 2012). **The shift to a diatom-dominated community appears to be related to initial availability of silicate (i.e., initial silicate was >~5 μM in all the experiments listed above).** However, as silicate concentrations decreased to <2 μM due to removal by **phytoplankton**, diatom blooms rapidly declined. SAGE and LOHAFEX had low initial levels of silicate (<2 μM). **As a consequence**, pico and nano-phytoplankton dominated **their** communities and diatom growth was limited by the lack of available silicate. **However during SOFeX-N, initial silicate limitation (< ~3 μM) in the iron-fertilized waters was temporarily relieved through lateral mixing with the surrounding waters that had relatively higher silicate concentrations (Coale et al., 2004), which contributed to a taxonomic shift toward diatom-dominated communities (from 16% to 44% of total phytoplankton community).**"

**7.** *p. 16, lines 34-56: the authors make no mention of model simulations. There are several studies on efficiency and potential negative impact from model studies. This should be mentioned when dealing with this topic.*

● Thank you for suggestion. We have added as follows:

➔ 3 Present: Unanswered aOIF questions (line 39, page 15 – line 16, page 16):

"However, the impact on the net transfer of $CO_2$ from the atmosphere to **below the winter MLD** through the 'biological pump' **(Fig. 3c)** is not yet fully understood or quantified and appears to vary with environmental conditions, export flux measurement techniques, and other unknown factors (Smetacek et al., 2012). **There have also been a wide range of the estimates of atmospheric $CO_2$ drawdown resulting from large-scale and long-term aOIF based on model simulations (Joos et al., 1991; Peng and Broecker, 1991; Sarmiento and Orr, 1991; Kurz and Maier-Reimer, 1993; Gnanadesikan et al., 2003; Aumont and Bopp, 2006; Denman, 2008; Jin et al., 2008; Zahariev et al., 2008; Strong et al., 2009; Sarmiento et al., 2010).** While it is generally agreed that OIF effectiveness needs to be determined through **quantification of export fluxes,** there has been no discussion about which export flux measurement techniques are the most effective. **Meanwhile,** concern has been expressed regarding possible environmental side effects in response to iron addition (Fuhrman and Capone, 1991). These side effects include the production of greenhouse gases (e.g., $N_2O$ and $CH_4$) (Lawrence, 2002; **Jin and Gruber, 2003;** Liss et al., 2005; Law, 2008; **Oschlies et al., 2010**), the development of hypoxia/anoxia in the water column (Sarmiento and Orr, 1991; **Oschlies et al., 2010; Keller et al., 2014**), and toxic algal blooms (e.g., *Pseudo-nitzschia*) (Silver et al., 2010; Trick et al., 2010). **These unwanted side effects** could lead to negative climate and ecosystem changes (Fuhrman and Capone, 1991; **Sarmiento and Orr, 1991; Jin and Gruber, 2003; Schiermeier, 2003; Oschlies et al., 2010). Model studies suggested that the unintended ecological and biogeochemical consequences in response to large-scale aOIF might cancel out the effectiveness of aOIF. For example, aOIF enhanced $N_2O$ production may have offset (up to ~40%) the benefits of $CO_2$ sequestration in the EP (Sarmiento and Orr, 1991; Jin and Gruber, 2003; Oschlies et al., 2010; Hauck et al., 2016).**"

● For more details on model studies on efficiency and potential negative impact, please refer $CH_4$ (lines 35–37, page 18), $N_2O$ (lines 8–12, page 19), DMS (lines 38–39, page 19), and Oxygen (lines 18–23, page 20) part in 3.2 Section and Section 4 (lines 25–34, page 22).

**8.** *Figure 6, panel b: Given that the size of the initial fertilized patches varied between experiments, it would be more informative to have the target iron concentrations in seawater rather than the total amount used.*

● Thank you. We have changed Figure 6b from *'total amount used'* to *'target iron concentrations'*.

➔ Figure captions (line 28, page 55):

"(b) **First target iron concentratins (nM).**"

**9.** *Section 3.1: I am entirely sure what the aim of this paragraph, broad presentations of methods used, or a discussion on the result (export fluxes) from previous experiments? While both issues might be relevant the authors do not really tackle the issues: why export fluxes varied so much in all experiments? Where the methods used appropriate?*

● The results of export flux varied with the environmental condition (e.g., silicate concentrations, grazing pressure rates within the mixed layer, and bacterial respiration), experimental duration, and export flux measurement techniques. Based on Reviewer #1's comments in the second revision (**Please refer Response 7 & 8 in author's response version 2**), we had added the discussion about the effectiveness of OIF *'why export fluxes varied so much in all experiments'* in Section 2.5 (line 26, page 14 – line 32, page 15) and Section 3.1 (line 8, page 17 – line 9, page 18).

● For '*Where the methods used appropriate*', we had also showed the limitations in each method (e.g., discrepancy between the $^{234}$Th and sediment traps which was caused by the under- or over-sampling of POC into the drifting traps during SOIREE and SEEDS-1 and limited ability to fully scan the vertical profile of POC fluxes in both sediment traps and $^{234}$Th methods during LOHAFEX) and methods used additionally to resolve these limitations (e.g., the deployment of neutrally buoyant sediment trap which could reduce the bias by sediment traps during LOHAFEX, the employment of optical sensors such as underwater video profile or transmissometers during LOHAFEX and EIFEX to monitor vertical changes in sinking particles with high resolution, and deployment of autonomous carbon explorers up to 18 months after initial deployment during SOFeX-N to continuously monitoring iron-induced POC fluxes after aOIF) in Section 3.1 (line 8, page 17 – line 9, page 18). Based on these methods, we concluded that the combination of multiple approaches is essential to accurately observe the POC produced in response to iron addition and its fate (Please refer the last paragraph of Section 3.1: lines 10–17, page 18 and 'What' of Section 4: line 38, page 25 – line 11, page 26). Also, we have added the mention about *'where the export flux should be detected'* based on Reviewer #1's comments. Please refer the **Reviewer #1's Response 1**.

**10.** *p. 17, lines 23-29: The authors seem to imply that 234Th is more reliable then sediment traps. This is not necessarily true: the 234Th method is sensitive to the type of model used (steady-state vs. non- steady state, C/234Th ratios, mixing, vertical resolution and uncertainties in measurements especially when sampling deeper layers). Further, the 234Th is not appropriate to measure deep fluxes. While deep fluxes (below the winter mixed layer down to the sediments) are mentioned in p.18, there is no real discussion of this highly relevant issue (which is also related to the monitoring of long-term effects).*

● We have added the limitations of the $^{234}$Th approach.

➔ 3.1 Export flux measurement methods (lines 4–7, page 17):

**"However, the water column-based $^{234}$Th method is sensitive to the characterization of the POC to $^{234}$Th ratio on sinking particles and/or the choice of $^{234}$Th flux models (Buesseler et al., 2006). Therefore, sampling to estimate the POC to $^{234}$Th ratio should be conducted below MLD to accurately detect downward carbon export flux into intermediate/deep waters."**

● For discussion on *'monitoring of long-term effects'*, we had mentioned the autonomous carbon export, allowing the continuous observation of POC fluxes during and after aOIF experiment in Section 3.1 (lines 2–17, page 18) and Section 4 (lines 1–11, page 26). We have also added the long-term monitoring methods for possible side effects after aOIF experiments as follows.

➔ 4 Future: Designing future aOIF experiments (lines 21–24, page 26):

**"The possible side effects after an aOIF experiment can be continuously monitored from optical sensors equipped autonomous moored profiler and/or autonomous benthic vehicle (e.g., crawler, which is capable to perform a long-term benthic oxygen measurements for ~12 months) (Dunne et al., 2002; Purser et al., 2013; Wenzhöfer et al., 2016)."**

➔ 4 Future: Designing future aOIF experiments (lines 33–35, page 26):

"(5) The side effects on marine/ocean ecosystems, including **decline in oxygen contents** and the production of climate-relevant gases (e.g., $N_2O$, DMS, and HVOCs) and toxic DA, **are monitored using optical sensors equipped autonomous moored profiler and/or autonomous benthic vehicle.**"

➔ 5.2.4 Year four plan (lines 21–23, page 29):

"(6) Monitor possible side effects, such as **the decline in oxygen contents** and the production of climate-relevant gases and toxic DA, **using optical sensors equipped autonomous moored profiler and/or autonomous benthic vehicle.**"

**11.** *Section 3.2: The authors make no mention of observations from natural fertilization studies.*

● Thank you for pointing this out. We have added the information about *'the observations from natural fertilization'* as follows:

➔ 3.2 Considering environmental side effects (lines 31–33, page 18):

**"The SO nOIF experiment conducted in 2011 year (i.e., Kerguelen Ocean and Plateau compared Study-2: KEOPS-2) (Table 1) showed that $CH_4$ concentrations were 4-fold higher in the naturally iron-fertilized patch than in the control area (Farías et al., 2015)."**

➔ 3.2 Considering environmental side effects (lines 12–13, page 19):

**"However, the SO nOIF experiment (i.e., KEOPS-2) suggested that nOIF acts as both a sink and a source for $N_2O$ (Farías et al., 2015)."**

➔ 3.2 Considering environmental side effects (lines 36–38, page 19):

**"On the other hand, the SO nOIF experiment (KEOPS-1) conducted in 2005 year (Table 1) showed that DMS production was not markedly higher in the naturally fertilized area compared to the surrounding waters (Belviso et al., 2008)."**

**12.** p. 20, lines 34-36: This statement is not correct. Keller et al, 2014 does find significant changes in ocean O2 under climate change scenarios.

● We apologize for the confusion. We have revised this paragraph and added the information about decline in oxygen content as follows:

➔ 3.2 Considering environmental side effects (lines 18–24, page 20):

"Early modeling studies suggest that anoxic conditions may develop after long-term, large-scale aOIF (**Fuhrman and Capone, 1991;** Sarmiento and Orr, 1991), **whereas a recent study based on more sophisticated models showed sustained** well-oxygenated conditions **($O_2 \approx 120$ μM) even under simulated aOIF south of 30°S on a 100 year timescale from 2010 to 2110 (Oschlies et al., 2010). Keller et al. (2014) found that simulated SO large-scale aOIF south of 40°S from the year 2020 to 2100 under a high $CO_2$ emissions scenario (Meinshausen et al., 2011) may develop suboxia ($O_2 <10$ μM) in the year 2125. Clearly, the circumstances under which a substantial decline in oxygen inventory can be caused by large-scale aOIF need further study."**

➔ Abstract (lines 36–39, page 1):

"(5) Monitoring of side effects on marine/ocean ecosystems, including production of climate-relevant gases (e.g., $N_2O$, dimethyl sulfide, and halogenated volatile organic compounds), **decline in oxygen inventory,** and development of toxic algae blooms, **with optical sensor equipped autonomous moored profilers and/or autonomous benthic vehicles."**

➔ 3.2 Considering environmental side effects (lines 9–13, page 21):

"The production of climate-relevant gases such as $N_2O$, DMS, and HVOCs, which is influenced by the remineralization of sinking particles that follows OIF-induced blooms, **the decline in oxygen inventory**, and the production of DA are particularly important to understand. These processes can directly and indirectly modify the effectiveness of carbon sequestration, with either positive or negative effects. Therefore, monitoring **declines in oxygen content** and production of climate-relevant gases and DA to evaluate the effectiveness of aOIF as a geoengineering approach is essential."

➔ 4 Future: Designing future aOIF experiments (lines 16–19, page 26):

"The emissions of climate-relevant gases, such as $N_2O$, DMS, and HVOCs, may directly contribute to warming or cooling effects, and **oxygen decrease and** toxic DA production may have a negative impact on marine/ocean ecosystems (Law, 2008; Silver et al., 2010; Trick et al., 2010; **Williamson et al., 2012**), resulting in significant offsets against the benefits of aOIF experiments."

➔ 4 Future: Designing future aOIF experiments (lines 33–35, page 26):

"(5) The side effects on marine/ocean ecosystems, including **decline in oxygen contents** and the production of climate-relevant gases (e.g., $N_2O$, DMS, and HVOCs) and toxic DA, are monitored using optical sensors equipped autonomous moored profiler and/or autonomous benthic vehicle."

➔ 5.2.2 Year two plan (lines 25–27, page 28):

"(1) Using the ice breaker RV *ARAON*, undertake a field investigation in/near the eastern Bransfield Basin to determine physical and biogeochemical parameters associated with both carbon sequestration and aOIF side effects (e.g., **decline in oxygen inventory** and production of $N_2O$, DMS, HVOCs, and DA), based on the first-year results."

➔ 6 Summary (lines 18–19, page 30):

"Likewise, the possible environmental negative side effects in response to iron addition, such as **decline in oxygen contents** and the production of climate-relevant gases and toxic DA, could not be fully evaluated due to the widely differing outcomes,"

**13.** p. 21, lines 17-27: I believe the topic of C sequestration is also largely influenced by model simulations, therefore, how additional experiments might inform ocean biogeochemical models might be an important aspect when presenting the scientific rationale (see for instance Losch et al., 2014, http://dx.doi.org/10.1016/j.jmarsys.2013.09.003).

● Thank you for your suggestion. However, this paragraph summarizes which environmental side effects should be essentially quantified to evaluate the effectiveness of aOIF as a means for reducing atmospheric $CO_2$ based on previous aOIF experiments, nOIF experiments, and modeling results. Improvement of the ocean biogeochemical models can be realized from future scientific additional aOIF experiment, like KIFES. Therefore, we think that adding the sentence about *'how additional experiments might inform ocean biogeochemical models might be an important aspect when presenting the scientific rationale'* to Section 5.2.5 KIFES section would be more reasonable. We added this part as follows:

➔ **5.2.5 Year five plan** (lines 33–35, page 29):

**"(5) Evaluate effectiveness and environmental side effects of large-scale SO aOIF via more realistic simulations under various scenarios with ocean biogeochemical models using the integrated results of KIFES."**

**14.** p. 23 § 2: The issue of zooplankton grazing is mentioned only in passing in the next page (§ 1 on timing of experiment), however, zooplankton composition and stocks should also guide the choice of location (for instance areas were krill or salps dominate?) and scientific rationale for KIFES.

● Good point. We have added the mention about *'the issue of zooplankton grazing'* in Section 4 as follows:

➔ 4 Future: Designing future aOIF experiments (lines 18–24, page 23):

"Previous aOIF experiments have shown that silicate concentration and mesozooplankton **stocks (i.e., copepods) are** the crucial **factors controlling** diatom **blooms (Boyd et al., 2000; Gervais et al., 2002; Coale et al., 2004; Tsuda et al., 2007; Smetacek et al., 2012).** Therefore, to obtain the greatest possible carbon export flux in response to iron addition, aOIF experiments should be designed in regions with high silicate concentrations **and low grazing pressure. It will be important to conduct initial surveys to measure the degree of grazing pressure in HNLC region with** high silicate concentrations such as in the subarctic NP (e.g., SEEDS-1 experiment) and the south of SO PF (e.g., SOFeX-S experiment) >~15 µM (Fig. 4c)."

**15.** *p. 24, § 2: Duration: The authors should consider the life span of eddies as well as the possibility of using autonomous observation platforms. In addition, risk assessment and monitoring as stipulated in the LC-LP.2 (2010) resolution should also be considered as a basis to determine the duration of the experiments. Further, carrying repeated fertilization might delay sinking events due to iron limitation of phytoplankton.*

● We provided the information about the life span of mesoscale eddies in 'Where' of Section 4 (lines 27-28, page 23).

● We have added the sentence about consideration about *'the possibility of using autonomous observation platforms'* in Section 4 as follows:

➔ 4 Future: Designing future aOIF experiments (lines 29–31, page 24):

**"In addition, autonomous observation platforms are essential to monitor post-assessment of effectiveness, capacity, and risks of aOIF for at least 12 months after experiment termination."**

● For the risk assessment and monitoring, please refer **Reviewer #2's Response 10**.

**16.** *p. 25, lines 14-25: The arguments for several iron infusions are not very robust given that despite a decrease in dissolved iron concentrations Fv/Fm remained high in all experiments before iron additions were repeated. Further, based on the EIFEX results, one could argue that a single iron addition to a final concentration of 1.5-2 nM might be enough to trigger both a bloom as well as lead to significant export (see also previous comment).*

● Thank you for your comments. As we had mentioned in Section 2.3.1 (lines 32–34, page 8), bottle incubation experiments showed that maximum phytoplankton growth rates in response to iron additions occurred when dissolved iron concentrations increased to the target concentrations, 1–2 nM (Fitzwater et al., 1996). However, IronEx-1, single iron addition experiment, showed that dissolved iron concentration rapidly decreased from 3.6 to 0.25 nM ~4 days after iron addition in the center of the fertilized patch due to subduction of the patch and sinking of colloidal aggregation and/or large particles containing iron, suggesting a limit to the level required for phytoplankton growth (please refer Section 2.3.1: lines 11–13, page 9). As we had mentioned in How of Section 4 (lines 10–15, page 25), SOIREE also showed that first infusion elevated the dissolved iron concentration up to 2.7 nM, and then dissolved iron concentrations was rapidly decreased to 0.2–0.3 nM in ~60 hours after first addition, requiring re-infusion (Bowie et al., 2001). During IronEx-1, Fv/Fm ratio, which indicates iron limitation on phytoplankton growth, showed that increased Fv/Fm after first iron addition continuously decreased ~4 days after first iron addition, but during IronEx-2 increased Fv/Fm after first iron addition remained throughout experiment duration following additional iron infusions (Barber and Hiscock, 2006) (please refer Section 2.4.1: lines 21–26, page 10). During SOIREE multiple additions of iron led to persistently high levels of both dissolved and particulate iron within the mixed layer, with a

rapid reduction at the end of the experiment, combined with an increase in the concentration of iron-binding ligands (Bowie et al., 2001) (please refer Section 4: lines 12–15, page 25). In both EIFEX and SOFeX-S, it was also found that multiple iron infusions allowed iron to persist in the mixed layer longer than its expected oxidation kinetics (Croot et al., 2008) (please refer Section 4: lines 15–18, page 25). Therefore, we think that the suggestion of multiple iron infusions would be more reasonable for aOIF experimentation.
* * *
**17.** *p. 27: I still find the argument for the location (Bransfield straight) somewhat tenuous as they could apply to other location as well. Further, the location seems to be situated in a sensitive area (CCAMLR subarea 48.1) with high krill stocks adding a whole new level of complexity in following both the evolution of the patch and impacts on higher trophic levels. I suggest the location should be re-evaluated based on considerations including food webs and hydrography (including what would be the fate of the patch and surface properties as well as exported organic matter once the eddy collapses).*

● Thank you for your suggestion. We believe that we had faithfully answered to Reviewers's comment on the location of Bransfield region with detailed explanations in **Response 13 in Author's response version 2**. Detailed background/information are found in *'specific proposal for KIFES', 'description of the logic for conducting this in the Bransfield Strait', 'what are the hypothesis', and 'how the KIFES project improve on previous experiments'*.

● For aOIF experiment in the krill habitat, Smetacek and Naqvi (2008) suggested that the krill habitat may provide a suitable site for a future SO aOIF experiment, as they have a potential (Tovar-Sanchez et al. 2007) for sustaining biological pump after the experiment and in turn recovering the whale populations, leading to positive effect on marine ecosystem. However, there has been little information about the relationship between mesoscale eddies and krill habitat in previous aOIF experiments. Therefore, we think that at present it is inappropriate dealing with the subject in this review.

**-Annotated Comments in the Manuscript:**

> ❖ **AC1: In 'caution is required because artificially high levels of SF6 injection may negatively impact the interpretation of low-level SF6 signals dissolved in seawater via air-sea exchange.',**
> *\* SF6 has other advantages: can help provide estimates of vertical and horizontal mixing/diffusivity.*
> *\* meaning of this sentence unclear. The main issue with SF6 is that concentrations decrease relatively rapidly (mixing and air-sea exchanged). Hence, it allows marking of the patch as a whole only once and for a limited period of time.*

● We have modified this sentence and added sentences concerning *'main issue with SF₆'* in Section 2.3.2 and Section 4 as follows:

➔ 2.3.2 Tracing iron-fertilized patch (lines 29–34, page 9):

"Although these earlier experiments demonstrated that the injection of artificial SF₆ is a useful technique for following iron-fertilized patches, **SF₆ can only be used for limited period (~2 weeks) due to the loss at the surface through air-sea gas exchange (Law et al., 2006; Tsumune et al., 2009; Martin et al., 2013). Furthermore,** caution is required because artificially high levels of SF₆ injection may negatively impact the interpretation of low-level SF₆ signals dissolved in seawater via air-sea exchange **to estimate tracer-based water mass ages for understanding physical circulation (Fine, 2011).**"

➔ 4 Future: Designing future aOIF experiments (lines 32–33, page 25):

**"However, tracing via SF₆ allows for only a limited period (~2 weeks) due to air-sea gas exchange (Law et al., 2006; Tsumune et al., 2009; Martin et al., 2013)."**

> ❖ **AC2: In '(~50%)',**
> *\* this value must be explained.*

● We have added explanation as follows:

➔ 2.4.1 Equatorial Pacific (lines 7–9, page 11):

"The biomass of meso-zooplankton (200–2,000 μm), such as copepods, grew simultaneously, substantially increasing the **community grazing effect of larger animals on phytoplankton standing stocks from 7.8% $d^{-1}$ outside patch to 11.4% $d^{-1}$ in the patch** (Coale et al., 1996)."

❖ **AC3: In '2.4.2 Southern Ocean',**
**\*** *As the Southern Ocean is the most relevant here, I would shift this section to the end of section 2.4.*

● Thank you. However, based on Reviewers's comments in second revision (please refer **Response 21 in Author's response version 2**), we had organized the Section 2 and Figures 6–7 *'by region with HNLC regions'*. As we had presented the data by the regions (EP-SO-Subarctic NP) in Section 2.1 and Section 2.2, it is likely appropriate to sustain original version (EP-SO-Subarctic NP) in Section 2.4.2.

❖ **AC4: In 'results suggested that diatoms were abundant in the two experiments.',**
**\*** *How so?*

● We have revised this sentence in Section 2.4.2 as follows:

➔ 2.4.2 Southern Ocean (lines 20–22, page 12):

"**Although both initially dominated by diatoms,** SOFeX-S had a somewhat greater $\Delta NO_3^-$ (-3.5 µM) and $\Delta pCO_2$ (-36 µatm) than EIFEX ($\Delta NO_3^-$: -1.6 µM and $\Delta pCO_2$: -30 µatm) (Coale et al., 2004; Hoffmann et al., 2006; Smetacek et al., 2012; Assmy et al., 2013)."

❖ **AC5: In 'This augmentation was the largest among all the aOIF experiments (Tsuda et al., 2003). The dramatic chlorophyll-a increase observed during SEEDS-1 was partly attributed to the particular range of seawater temperature in the region, which was conducive to diatom growth (i.e., 8–13°C) as well as to the shallower MLD (~10 m), which provided a relatively longer surface water residence time for the additional iron (Figs. 6c and d) (Noiri et al., 2005; Takeda and Tsuda, 2005; Tsumune et al., 2005).',**
**\*** *When comparing integrated values (over the mixed layer), other experiments reached similar high biomass. Given the broad range in mixed layer depths, comparing concentrations can give a wrong impression of the actual biomass increase.*

● We apologize for missing your comment in first revision (**Response 17 in Author's response version 1**) when we re-organized our manuscript during second revision. We have re-added mention about '*the difference between surface chlorophyll-a concentrations' and 'integrated chorophyll-a concentrations or/and primary production'* in Section 2.5, which we had revised in first revision and have added 'surface' word in the sentence.

➔ 2.4.3 Subarctic North Pacific (lines 29–31, page 13):

"The dramatic **surface** chlorophyll-a increase observed during SEEDS-1 was partly attributed to the particular range of seawater temperature in the region, which was conducive to diatom growth (i.e., 8–13°C) as well as to the shallower MLD (~10 m),"

➔ 2.5 Summary of the significant results from aOIF experiments (lines 7–14, page 15):

"**Among previous aOIF experiments,** the subarctic NP SEEDS-1 experiment, which was conducted under temperature conditions ideal for diatom growth (~8°C) and with shallow MLDs (~10 m), produced the greatest changes in **surface phytoplankton biomass. However, influence of iron addition on the phytoplankton growth covers from surface to euphotic depth as added iron is mixed within the ML by physical processes (Coale et al., 1998). Although maximum surface chlorophyll-a concentration during SEEDS-1 (~22 mg m$^{-3}$) was much higher than EIFEX (~3.2 mg m$^{-3}$), the MLD-integrated chlorophyll-a concentrations were similar to ~250 mg m$^{-2}$ between two experiments. Therefore, to quantify the exact changes in phytoplankton biomass in response to iron addition, it would be eligible to consider the MLD-integrated PP for comparison.**"

❖ **AC6: In '2.5 Summary of the significant results from a OIF experiments',**
\* *These are not results!*

● Please refer **Reviewer #2's Response 5.**

❖ **AC7: In 'changeover to a diatom-dominated community after iron addition is needed.',**
\* *How about SOFEX-N then?*

● Please refer **Reviewer #2's Response 6**.

❖ **AC8: In 'addition were not dramatic compared to natural values.',**
**\*** *I am not sure I understand the argument here: natural values are not necessarily low.*

● We had referred 'Buesseler et al. (2005)' and added it.

In 'Buesseler et al. (2005)',

*'Thus, the lack of a stronger inverse correlation between surface $SF_6$ and $^{234}Th$, is evidence that during the 4-week occupation of the SOFeX bloom, surface ocean export was not particularly elevated, or at least was small relative to other observations of particle export associated with natural blooms from other regions.'.*

*'When converted to POC and $bSiO_2$ export rates, these fluxes were not large relative to natural blooms and were, in fact, smaller than those observed under natural conditions at this site during a different field season.'.*

➔ 2.4.2 Southern Ocean (lines 39–40, page 12):

"During SOFeX-S, significantly enhanced POC fluxes below the **MLD** similar to those observed in natural blooms, were estimated from $^{234}Th$ measurements after iron enrichment **(Buesseler et al., 2005)**."

➔ 2.5 Summary of the significant results from aOIF experiments (lines 21–22, page 15):

"However, the changes in export flux, after iron addition, were not dramatic compared to natural values **(Buesseler et al., 2005)**."

❖ **AC9: In 'offset leading',**
**\*** *I would change the wording, as offset is used in the literature as counteracting the effects OIFs, This is not the case here.*

● We have modified this wording as follows:

➔ 3.2 Considering environmental side effects (lines 15–16, page 20):

"**Another important consideration is the extent to which** the effectiveness of **aOIF is cancelled out by its tendency to** lead to ocean ecosystem changes such as a decrease in dissolved oxygen and an increase in domoic acid (DA) levels."

❖ **AC10: In 'It is likely that several phytoplankton samples (e.g., Pseudo-nitzschia abundance: $1.3 \times 10^6$ cells $L^{-1}$ in IronEx-2 and $7.5 \times 10^4$ cells $L^{-1}$ in SOFeX-S) collected with a net tow were suitable to detect these changes.',**
* *unclear. please explain.*

● We apologize for the confusion. Unlike the results of Trick et al. (2010) that showed no DA change during EisenEx and SERIES, Silver et al. (2010) suggested that discernable change in DA during IronEx-2 and SOFeX-S could be found from phytoplankton samples taken with net tows (20- to 30-μm mesh phytoplankton nets) by allowing larger phytoplankton samples including *Pseudo-nitzschia* to concentrate. During IronEx-2 and SOFeX-S, DA concentrations were as high as 45 ng DA $l^{-1}$ and 220 ng DA $l^{-1}$ in the water, respectively, i.e., toxin levels high enough to damage marine communities in coastal waters (Scholin et al., 2000; Schnetzer et al., 2007). Based on these results, we concluded that it is necessary to quantify DA production in response to iron additions, with concentrated larger phytoplankton samples taken from a net tow. To explain clearly these contents, we have modified these sentences in Section 3.2.

➔ 3.2 Considering environmental side effects (line 35, page 20 – line 1, page 21):

"It is likely that **detection was possible because these samples were collected with net tows (20- to 30-μm mesh phytoplankton nets), which provided concentrated samples of larger phytoplankton including *Pseudo-nitzschia*** (e.g., *Pseudo-nitzschia* abundance: $1.3 \times 10^6$ cells $L^{-1}$ in IronEx-2 and $7.5 \times 10^4$ cells $L^{-1}$ in SOFeX-S). During IronEx-2 and SOFeX-S, high cell abundances of *Pseudo-nitzschia* ($10^6$ and $10^5$ cells $l^{-1}$, respectively) combined with moderate DA **cell** quotas (0.05 and 1 pg DA cell$^{-1}$, respectively) produced toxin levels as high as 45 ng DA $l^{-1}$ and 220 ng DA $l^{-1}$ **in the water**, respectively, i.e., toxin levels high enough to damage marine communities in coastal waters **(Scholin et al., 2000; Schnetzer et al., 2007)**."

➔ 3.2 Considering environmental side effects (lines 4–5, page 21):

"As a result, it is necessary to **clarify**/quantify DA production in response to aOIF, with concentrated **larger** phytoplankton samples **collected using** net tows **(20- to 30-μm mesh phytoplankton net)**."

❖ **AC11: In 'toxin levels high enough to damage marine communities in coastal waters.',**
* *add references.*

● Done. Please refer **Reviewer #2's AC10**.

❖ **AC12: In 'Therefore, it is necessary to quantify DA production in response to iron additions, with concentrated phytoplankton samples (i.e., large numbers of cells) using a net tow.',**
\* *Were does this come from? It is not the conclusion I would make from the previous. Rather handle the samples in such a way that DA remains unchanged. Also what about DA in the water?*

● Please refer **Reviewer #2's AC10**.

❖ **AC13: In 'Where: The first consideration for a successful aOIF experiment is the location. The dominance of diatoms in phytoplankton communities plays a major role in increasing the biological pump because diatom species can sink rapidly as aggregates or by forming resting spores (Tréguer et al., 1995).',**
\* *see also more recent literature:*
*Rembauville, Mathieu, et al. "Export fluxes in a naturally iron-fertilized area of the Southern Ocean–Part 2: Importance of diatom resting spores and faecal pellets for export." Biogeosciences 12.11 (2015): 3171-3195. Rembauville, M., et al. "Strong contribution of diatom resting spores to deep-sea carbon transfer in naturally iron-fertilized waters downstream of South Georgia." Deep Sea Research Part I: Oceanographic Research Papers 115 (2016): 22-35. Salter, Ian, et al. "Estimating carbon, silica and diatom export from a naturally fertilised phytoplankton bloom in the Southern Ocean using PELAGRA: A novel drifting sediment trap." Deep Sea Research Part II: Topical Studies in Oceanography 54.18 (2007): 2233-2259.*

● Thank you. We have added these references and added the sentence that mentions diatom resting spores.

➔ 4 Future: Designing future aOIF experiments (lines 14–18, page 23):

"The dominance of diatoms in phytoplankton communities plays a major role in increasing the biological pump because diatom species can sink rapidly as aggregates or **by forming resting spores to efficiently bypass the intense grazing pressure of mesozooplankton (e.g., copepods, salps, and krill) and export carbon out of the winter ML** (Tréguer et al., 1995; **Salter et al. 2007; Assmy et al., 2013; Rembauville et al., 2015; Rembauville et al., 2016**)."

❖ **AC14: In 'while after January (i.e., during late summer and early autumn from February to March) it is mainly limited due to silicate availability.',**
\* *please provide references*

● We have added references.

**❖ AC15: In '(de Baar et al., 2005;',**
***** *De Baar 2005 states that light is limiting*

● Thank you for pointing this out. We have modified this sentence.

➔ 4 Future: Designing future aOIF experiments (line 41, page 23 – line 3, page 24):

"In previous SO aOIF experiments conducted between spring and early autumn, PP was mainly limited by iron and/or silicate availability rather than light availability **(except when heavy clouds led to severe light limitation, only occurred for a few days during EisenEx) (Gervais et al., 2002; Bakker et al., 2005;** Smetacek and Naqvi, 2008; Peloquin et al., 2011b)."

**❖ AC16: In '(Coale et al., 2004; Martin et al., 2013),',**
***** *I have not found convincing evidence for grazing control of diatom growth this in the articles cited.*

● We have removed these references and added more appropriate references **(Schultes et al., 2006; Smetacek and Naqvi, 2010)**, reporting grazing control on diatom growth.

In 'Schultes et al. (2006)':

*'Overall grazing impact of copepods in these studies reached 25% of phytoplankton standing stock removed per day with the major fraction being attributable to C. simillimus.'.*

In 'Smetacek and Naqvi (2010)':

*'The second fertilization had no noticeable effect on growth rates or biomass of phyto- or bacterioplankton. Apparently, the main reason why biomass did not build up to higher levels was due to heavy grazing of the large copepod population. Incubation experiments indicated that the copepods increased their feeding and faecal production rates inside the patch.'.*

❖ **AC17: In '(Le Quéré et al., 2016).',**

*\* This is a model simulation paper. Should not be referred here.*

● We have removed this paper and added more appropriate references as follows:

**Hunt, B. P. V. and Hosie, G. W.: The seasonal succession of zooplankton in the Southern Ocean south of Australia, part II: The Sub-Antarctic to Polar Frontal Zones, Deep-Sea Res. Pt. I, 53, 1203-1223, 2006.**

**Rembauville, M., Blain, S., Armand, L., Quéguiner, B., and Salter, I.: Export fluxes in a naturally iron-fertilized area of the Southern Ocean – Part 2: Importance of diatom resting spores and faecal pellets for export, Biogeosciences, 12, 3171-3195, https://doi.org/10.5194/bg-12-3171-2015, 2015.**

❖ **AC18: In '(2) below the depth of the winter MLD to detect iron-induced export carbon fluxes into intermediate/deeper waters (Bidigare et al., 1999; Nodder et al., 2001; Boyd et al., 2004; Buesseler et al., 2004; Coale et al., 2004; Aono et al., 2005; Buesseler et al., 2005; Tsuda et al., 2007; Smetacek et al., 2012; Martin et al., 2013). Sinking-particle profiling systems mounted on autonomous floats, such as a transmissometer and UVP that measure and photograph sinking particles, could provide a record of the temporal and vertical evolution of iron-induced POC stocks through successive depth layers down to ~3,000-m depth for ~20 months after deployment, once calibrated using POC fluxes measured from sediment traps and/or a water-column based 234Th method (Bishop et al., 2004; Smetacek et al., 2012; Martin et al., 2013).',**

*\* What about deep-sea (moored) traps and benthic oxygen measurements?*

● Thank you for comments. Please refer **Reviewer #2's response 10**.

~~One exception to the focus on HNLC study sites was the FeeP experiment, which was conducted in the subtropical NA, a typically LNLC region (Figs. 4a and b, Tables 3 and 4). To test the effects of the co-limitation of iron and phosphate on PP, FeeP was conducted under much lower initial nutrient (
[revised manuscript text omitted]

~~For example, spatial changes in chlorophyll-a resulting from SOFeX-N/S iron addition were detected using Sea-viewing Wide Field of view Sensor (SeaWiFS) and MODerate resolution Imaging Spectrometer (MODIS) Terra Level 2 chlorophyll a images. The chlorophyll a image on day 24 after iron addition in the SOFeX N showed a phytoplankton bloom distribution resembling a long thread with 10-fold higher concentrations (1.0 mg m$^{-3}$) than the surrounding waters (0.1 mg m$^{-3}$), while a chlorophyll a image on day 20 of SOFeX S suggested a somewhat broader bloom pattern (0.5 mg m$^{-3}$), with concentrations elevated ~5-fold over the surrounding levels (~0.1 mg m$^{-3}$) (Fig. 7d) (Westberry et al., 2013).~~

Following artificial iron enrichment in the SO, ΔPP ranged from 360 (SAGE) to ~1356 mg C m$^{-2}$ d$^{-1}$ (SOFeX-N) (Table 4 and Fig. 7e). During SOIREE, EisenEx, and SOFeX-N/-S,  PP increased continuously throughout the duration of the experiments (Boyd et al., 2000; Gall et al., 2001a; Gervais et al., 2002; Coale et al., 2004; Assmy et al., 2007). However, in EIFEX, SAGE, and LOHAFEX there was a significant increase in PP for ~10 (SAGE) to 20 (EIFEX) days in response to the iron addition, and decreasing trends after day ~12 (SAGE) – 25 (EIFEX). The decrease was due to various processes such as export  (e.g., EIFEX), lateral dilution with surrounding waters (e.g., SAGE),

and high grazing pressure and  bacterial respiration (e.g., LOHAFEX) (Boyd, 2002; Gervais et al., 2002; Buesseler et al., 2004; Coale et al., 2004; Peloquin et al., 2011a; Smetacek et al., 2012; Thiele et al., 2012; Assmy et al., 2013; Martin et al., 2013; Latasa et al., 2014).

Using both microscopes and high-performance liquid chromatography pigment analysis, changes in phytoplankton community affected by iron addition have also been investigated. Most SO aOIF  experiments have resulted in blooms of diatoms  (Boyd et al., 2007). During SOIREE and EisenEx, the dominant phytoplankton community shifted from pico- and nano-phytoplankton (e.g., pico-eukaryotes and prymnesiophytes) to micro-phytoplankton (i.e., diatoms) (Gall et al., 2001a; Gervais et al., 2002; Assmy et al., 2007). In SOFeX-S and EIFEX, diatoms were already the most abundant group prior to iron addition (Coale et al., 2004; Hoffmann et al., 2006; Assmy et al., 2013). The contribution of large diatoms became especially clear in EIFEX where ~97% of the phytoplankton bloom was attributed to this group  (Smetacek et al., 2012; Assmy et al., 2013). However, no taxonomic shift toward diatom-dominated  communities (<5% of total phytoplankton community) was observed during  SAGE and LOHAFEX, which were conducted under silicate-limited conditions (Harvey et al., 2010; Peloquin et al., 2011a; Martin et al., 2013; Ebersbach et al., 2014). Although SOFeX-N was conducted under low silicate conditions (Fig. 6f), the diatom biomass increased remarkably making up ~44% of the total phytoplankton community (Coale et al., 2004). This result was partly influenced by the temporary relief of silicate limitation through lateral mixing of the iron-fertilized waters with surrounding waters, with relatively higher silicate concentrations (Coale et al., 2004).

Iron-mediated increases in PP resulted in a significant uptake in macronutrients and $p$CO$_2$ throughout the aOIF experiments in the SO (except for SAGE) (Table 3, Figs. 7b and f).  $\Delta$NO$_3^-$ ranged from -3.5 µM (e.g., SOFeX-S) to -1.4 µM (e.g., SOFeX-N)  and $\Delta p$CO$_2$ ranged from -38 µatm (e.g., SOIREE) to -7 µatm (e.g., LOHAFEX). Although both initially dominated by diatoms, SOFeX-S had a somewhat greater $\Delta$NO$_3^-$ (-3.5 µM) and $\Delta p$CO$_2$ (-36 µatm) than EIFEX ($\Delta$NO$_3^-$: -1.6 µM and $\Delta p$CO$_2$: -30 µatm) (Coale et al., 2004; Hoffmann et al., 2006; Smetacek et al., 2012; Assmy et al., 2013) . However, the smaller silicate uptake  ($\Delta$Si = [Si]$_{postf}$ − [Si]$_{pref}$ ) observed during SOFeX-S (-4 µM)  compared to EIFEX (-11 µM) was associated with a decrease in silicification (i.e., changes  the  in frustule thickness  of the dominant diatom species,  *Fragilariopsis* sp.)  Twining et al., 2004). During  EIFEX, the ratio of heavily silicified diatoms (e.g., *Thalassiothrix antarctica*) to total diatom biomass increased from 0.24 (day 0) to 0.46 (day 37) leading to the higher Si uptake  (Hoffmann et al., 2006; Assmy et al., 2013). Interestingly, the biogeochemical responses in SAGE were totally different from those seen in other experiments,  as increases  in $\Delta$NO$_3^-$ (+3.9 µM), $\Delta p$CO$_2$ (+8 µatm), and $\Delta$DIC (+25 µM)  were observed (Table 3, Figs. 7b and f). These contrasting results were thought to be the result of entrainment through vertical and horizontal physical mixing into the iron-fertilized patch of surrounding  waters, with higher nutrient and $p$CO$_2$  concentrations (Currie et al., 2011; Law et al., 2011).

SOIREE was the first aOIF experiment in the SO to estimate the downward carbon flux into deep waters (Fig. 3c). A  comprehensive suite of methods was used:  drifting traps, $^{234}$Th and the stable carbon isotope of particulate organic matter ($\delta^{13}$C$_{org}$)  estimates derived from high-volume pump sampling, and a beam transmissometer (Nodder and Waite, 2001). However, no measurable change in carbon export was observed in response to iron-stimulated PP (Table 5 and Fig. 8b) (Charette and Buesseler, 2000; Nodder and Waite, 2001; Trull and Armand, 2001; Waite and Nodder, 2001). During EisenEx, an increased downward carbon flux estimated from $^{234}$Th deficiency was observed in the iron-fertilized patch as the experiment progressed. However, there were no clear differences between in- and

outside-patch carbon fluxes (Buesseler et al., 2005). During SOFeX-S, significantly enhanced POC fluxes below the the mixed layerMLD similar to those observed in natural blooms, were after iron enrichment estimatedobtained from $^{234}$Th measurements after iron enrichmentobservations (Buesseler et al., 2005). However, the absolute magnitude of these flux increases  was similar to that observed in natural blooms (Buesseler et al., 2005). DuringUniquely, SOFeX-N autonomous profilersused only free profiling robotic Lagrangian carbon explorers equipped with transmissometers recorded ato estimate the downward carbon flux without employing chemical tracers, and observed large POC flux events between day ~27 and ~45 after the first iron addition (Bishop et al., 2004; Coale et al., 2004). However, it was unclear whether surface-fixed carbon was well and truly delivered into intermediate/deep depthsbelow the winter MLD. DuringFor SAGE and LOHAFEX experiments, which were conducted under silicate limited conditions (Table 3, Figs. 4c and 6f), no significant enhancement of carbon export following was detectedthere was no detection of fertilization-induced export by any method (Table 5) (Peloquin et al., 2011a; Martin et al., 2013). This result was likely to be due to the dominance ofassociated with the pico-plankton andplankton and grazingdominated community, which 
[revised manuscript text omitted]

5    Artificial   The one exception was the FeeP experiment, which was performed in the subtropical NA. The initial environmental conditions associated with the physical and biogeochemical properties were determined at these OIF sites over 1–7 days before iron addition to allow the responses to the aOIF to be evaluated and quantified. Preliminary surveys confirmed that all sites, except FeeP in the subtropical NA, were subject to iron limited HNLC conditions, with typical levels of iron <0.4 nM, nitrate >~10 μM, and chlorophyll a <1 mg m$^{-3}$. The initial Fv/Fm ratios were <~0.3, suggesting that

10   phytoplankton growth was severely iron limited. In SEEDS 1, SOFeX S, and EIFEX, prior to the addition of iron, the micro phytoplankton (e.g., diatom) community accounted for half of the population and this was thought to be beneficial to the enhancement of export production. In the other experiments, pico  and nano phytoplankton (e.g., *Synechococcus* and haptophytes) initially dominated; they are associated with rapid recycling in the mixed layer through the microbial loop rather than export production (Michaels and Silver, 1988; Coale et al., 1998; Landry et al., 2000; Boyd and Law, 2001;

15   Gervais et al., 2002; Coale et al., 2004; Boyd et al., 2005; Tsuda et al., 2005; Hoffmann et al., 2006; Tsuda et al., 2007; Harvey et al., 2010; Martin et al., 2013).

     Iron sulfates dissolved in acidified seawater have been commonly used for artificial iron addition because they are both highly bioavailable and inexpensive. The mixture is generally released into the ship's wake over a period of 24 hours. The amount of added iron was determined so to reach a target dissolved-iron concentration (at least >1 nM) by volume

20   (defined as the MLD × patch area). To achieve this, a wide range of 225–2,000 kg was applied. Except in IronEx 1 and SEEDS 1, the experiments used multiple (2–4) iron additions to reinforce the increased iron levels. To trace the iron fertilized patches, physical tracers (i.e., ARGO or other GPS-tracked drifting buoys) and/or chemical tracers such as SF$_6$ were used. In addition, biogeochemical parameters, such as the Fv/Fm ratio, macro nutrients, and CO$_2$ variables, were used to detect responses through a comparison of before and after conditions (i.e., Δ = [parameter]$_{post}$   [parameter]$_{pref}$). In

25   particular, it should be noted that the Fv/Fm ratios promptly increased from <~0.3 to 0.56 (± 0.08) in the two days following iron addition, indicating a relief in the iron limitation on phytoplankton growth. The subarctic NP SEEDS 1 experiment, which was conducted under temperature conditions ideal for diatom growth (~8°C) and with shallow MLDs (~10 m), produced the greatest changes in biogeochemical parameters.

     The aOIF experiments have generally led to changes in the size of the phytoplankton community from pico and nano-

30   phytoplankton to micro-phytoplankton. This effect was particularly noticeable as diatoms became the dominant species during IronEx-2, SOIREE, EisenEx, SEEDS-1, SOFeX-S, EIFEX, and SERIES. , with micro phytoplankton accounting for ~44% of total phytoplankton community in SOFeX N. The shift to a diatom dominated community appears be related to the initial availability of silicate (i.e., initial silicate was >~3 μM in all the experiments just listed). Diatom-dominated blooms induced >4.5-fold increases in chlorophyll-a concentrations and accounted for >65% of the chlorophyll-a increase (Boyd et

35   al., 2000; Gervais et al., 2002; Coale et al., 2004; Smetacek et al., 2012). The shift to a diatom-dominated community appears to be related to initial availability of silicate (i.e., initial silicate was >~5 μM in all the experiments just listed above). However, as silicate concentrations decreased to <2 μM due to removal by phytoplankton the elevated diatom abundance, the extent of diatom blooms rapidly declined. In SAGE and LOHAFEX, started with 
[revised manuscript text omitted]

Le Quéré, C., Buitenhuis, E. T., Moriarty, R., Alvain, S., Aumont, O., Bopp, L., Chollet, S., Enright, C., Franklin, D. J., Geider, R. J., Harrison, S. P., Hirst, A. G., Larsen, S., Legendre, L., Platt, T., Prentice, I. C., Rivkin, R. B., Sailley, S., Sathyendranath, S., Stephens, N., Vogt, M., and Vallina, S. M.: Role of zooplankton dynamics for Southern Ocean phytoplankton biomass and global biogeochemical cycles, Biogeosciences, 13, 4111-4133, 2016.

Levasseur, M., Scarratt, M. G., Michaud, S., Merzouk, A., Wong, C. S., Arychuk, M., Richardson, W., Rivkin, R. B., Hale, M., Wong, E., Marchetti, A., and Kiyosawa, H.: DMSP and DMS dynamics during a mesoscale iron fertilization experiment in the Northeast Pacific—Part I: Temporal and vertical distributions, Deep-Sea Res. Pt. II, 53, 2353-2369, 2006.

Liss, P., Chuck, A., Bakker, D., and Turner, S.: Ocean fertilization with iron: effects on climate and air quality, Tellus B, 57, 269-271, 2005.

London Convention: Convention on the Prevention of Marine Pollution by Dumping of Wastes and Other Matter 1972, 1972.

London Protocol: 1996 Protocol to the Convention on the Prevention of Marine Pollution by Dumping of Wastes and Other Matter, 1972, 1996.

Marchetti, A., Sherry, N. D., Kiyosawa, H., Tsuda, A., and Harrison, P. J.: Phytoplankton processes during a mesoscale iron enrichment in the NE subarctic Pacific: Part I—Biomass and assemblage, Deep-Sea Res. Pt. II, 53, 2095-2113, 2006.

Marchetti, A., Lundholm, N., Kotaki, Y., Hubbard, K., Harrison, P. J., and Virginia Armbrust, E.: Identification and assessment of domoic acid production in oceanic Pseudo-nitzschia (Bacillariophyceae) from iron-limited waters in the Northeast Subarctic Pacific, Journal of Phycology, J. Phycol., 650-661, 2008.

Marshall, J. and Speer, K.: Closure of the meridional overturning circulation through Southern Ocean upwelling, Nat. Geosci., 5, 171-180, 2012.

Martin, J. H. and Fitzwater, S. E.: Iron deficiency limits phytoplankton growth in the north-east Pacific subarctic, Nature, 331, 341-343, 1988.

Martin, J. H.: Glacial-interglacial $CO_2$ change: The Iron Hypothesis, Paleoceanography, 5, 1-13, 1990.

Martin, J. H. and Chisholm, P.: Design for a mesoscale iron enrichment experiment. Woods Hole Oceanographic Institution, U.S JGOFS Planning Report, 15, 1992.

Martin, J. H., Coale, K. H., Johnson, K. S., Fitzwater, S. E., Gordon, R. M., Tanner, S. J., Hunter, C. N., Elrod, V. A., Nowicki, J. L., Coley, T. L., Barber, R. T., Lindley, S., Watson, A. J., Van Scoy, K., Law, C. S., Liddicoat, M. I., Ling, R., Stanton, T., Stockel, J., Collins, C., Anderson, A., Bidigare, R., Ondrusek, M., Latasa, M., Millero, F. J., Lee, K., Yao, W., Zhang, J. Z., Friederich, G., Sakamoto, C., Chavez, F., Buck, K., Kolber, Z., Greene, R., Falkowski, P., Chisholm, S. W., Hoge, F., Swift, R., Yungel, J., Turner, S., Nightingale, P., Hatton, A., Liss, P., and Tindale, N. W.: Testing the iron hypothesis in ecosystems of the equatorial Pacific Ocean, Nature, 371, 123-129, 1994.

Martin, P., van der Loeff, M. R., Cassar, N., Vandromme, P., d'Ovidio, F., Stemmann, L., Rengarajan, R., Soares, M.,

González, H. E., Ebersbach, F., Lampitt, R. S., Sanders, R., Barnett, B. A., Smetacek, V., and Naqvi, S. W. A.: Iron fertilization enhanced net community production but not downward particle flux during the Southern Ocean iron fertilization experiment LOHAFEX, Global Biogeochem. Cycles, 27, 871-881, 2013.

Matthews, B.: Climate engineering: a critical review of proposals, their scientific and political context, and possible impacts, Compiled for scientists for global responsibility, 1996. (http://records.viu.ca/earles/geol312o/assignments/mitigation.htm, 1996).

McElroy, M. B.: Marine biological controls on atmospheric $CO_2$ and climate, Nature, 302, 328-329, 1983.

Meinshausen, M., Smith, S. J., Calvin, K., Daniel, J. S., Kainuma, M. L. T., Lamarque, J.-F., Matsumoto, K., Montzka, S. A., Raper, S. C. B., Riahi, K., Thomson, A., Velders, G. J. M., and van Vuuren, D. P. P.: The RCP greenhouse gas concentrations and their extensions from 1765 to 2300, Clim. Change, 109, 213, 2011.

Mengelt, C., Abbott, M. R., Barth, J. A., Letelier, R. M., Measures, C. I., and Vink, S.: Phytoplankton pigment distribution in relation to silicic acid, iron and the physical structure across the Antarctic Polar Front, 170°W, during austral summer, Deep-Sea Res. Pt. II, 48, 4081-4100, 2001.

Michaels, A. F. and Silver, M. W.: Primary production, sinking fluxes and the microbial food web, Deep-Sea Res. Pt. I, 35, 473-490, doi:10.1016/0198-0149(88)90126-4, 1988.

[revised manuscript text omitted]

(a)

[Figure]

(b)

(a)

[Figure]

(b)

[Figure]

[Figure]

[Figure]

(a)

(b)

SOIREE (Day 5)   SOIREE (Day 42)

Chlorophyll-a (mg m$^{-3}$)

---

## Author Response (AR4)

Dear Dr. Victor Smetacek,

Thank you very much for the constructive comments. We highly appreciate your time and efforts you put in reviewing our manuscript "Ocean Iron Fertilization Experiments: Past–Present–Future looking to a future Korean Iron Fertilization Experiment in the Southern Ocean (KIFES) Project". A major revision of our has been carried out to take all of them into account. And in the process, we believe the paper has been significantly improved.

We provide our responses (Plain text with black and blue colors) to all comments (*italic text*) below. We hope that the revision addresses your concerns.

**- Response to Reviewer Comments -**

**Reviewer #3**

**-General Comments:**

**1.** *I would like to see such a paper published in order to re-stimulate the discussion on using OIF as an experimental tool for hypothesis-testing in the fields of plankton ecology and ocean biogeochemistry*

● Thank you for your suggestion. As you gave comments, it will be important to judge the success of previous experiments on the criterion of not only "effectiveness" in sequestering $CO_2$ but also "plankton ecology and ocean biogeochemistry". We know that our manuscript is focusing on "effectiveness" that is geoengineering aspects of OIF. During the previous revision processes, Reviewer #1 had suggested a revised structure on our manuscript into a more useful review that manages to move the field forward further by focusing on "effectiveness of OIF" to clarify "why the previous 13 experiments have not managed to yield clear answers to the effectiveness of carbon sequestration", "The really important one from the point of view of geoengineering would be the amount of carbon sequestered below the depth of deepest winter mixing in the study region, which most previous OIF did not measure", "Moreover, it would be useful if the authors more explicitly assessed which of the experiments conducted to data were actually capable of detecting an enhancement of export if it had occurred (based on duration of the experiment relative to the phase of the bloom and the type of measurements that were taken), which of these did find a response in particle flux (e.g., EIFEX, SEREIS), and how to what depth the carbon flux was followed", and "What are their recommendations in terms of best patch size, minimum duration, and which measurements are required to quantify the effect on carbon sequestration?" (**Please refer Response 1 & 2 in author's**

**response version 1 and Response 4, 6, 7, 12 in author's response version 2**). Based on Reviewer #1, we have revised our manuscript by focusing on "effectiveness". Reviewer #1 finally suggested that this manuscript is acceptable for publication. So, in the stage of fourth revision, it is really hard to change the focus from "effectiveness" to "hypothesis-testing in the fields of plankton ecology and ocean biogeochemistry". However, based on Reviewer #3, in order to re-stimulate the discussion on using OIF as an experimental tool for hypothesis-testing in the fields of plankton ecology and ocean biogeochemistry, we have added the mentions on the "importance of OIF as an experimental tool for hypothesis-testing in the fields of plankton ecology and ocean biogeochemistry", "OIF experiments allowed one to study how plankton-based ecosystems work by providing insight into mechanisms operating in real time and under in situ conditions", "OIF experiments provide insights into the structure and functioning of pelagic ecosystems that cannot be acquired from observational cruises alone", and "Thus, each experiment has provided new results on basic processes pertaining to the relationship between pelagic ecology and biogeochemistry, such as selection of the dominant phytoplankton group or species, the effects of grazing by different zooplankters, interactions within the plankton community and other fundamental issues that came to light during each experiment.".

➔ Abstract (lines 27–28, page 1):

**"Furthermore, it is logical to carry out such experiments because they allow one to study how plankton-based ecosystems work by providing insight into mechanisms operating in real time and under *in situ* conditions."**

➔ 1 Introduction (lines 19–23, page 4):

**"Furthermore, aOIF experiments have provided insights into the structure and function of pelagic ecosystems that cannot be acquired from observational cruises alone. While observational cruises provide a jumble of snapshots from which the plot of the movie has to be guessed, carrying out an OIF experiments is like watching the movie by having one directly follow the processes triggered by addition of the crucial limiting element, iron. For these reasons, it is necessary to plan and carry out the next aOIF experiments within the framework of the international law."**

➔ 2.5 Summary of the significant results from aOIF experiments (lines 2–4, page 15):

**"Each aOIF experiment has provided new results on basic processes pertaining to the relationship between pelagic ecology and biogeochemistry, such as selection of the dominant phytoplankton group or species, the effects of grazing, interactions within the plankton community, and effects of nutrient concentrations on the growth of phytoplankton."**

➔ 4 Future: Designing future aOIF experiments (lines 26–29, page 23):

**"There is also a broad swathe of hypotheses in the fields of pelagic ecology/biogeochemistry that can be tested with OIF experiments. It could be derived from the correlations between temperature, $CO_2$ concentrations, and dust over the past 4 glacial/interglacial cycles on the one hand and bottle experiments showing iron limitation of phytoplankton growth in HNLC regions on the other."**

➔ 5.1.5 Year five plan (lines 12–14 and 17–18, page 29):

"Objective: (1) **To determine** the effectiveness of artificially iron-induced export production **and** any negative impacts on climate change. **(2) To assess the results on basic processes pertaining to the relationship between pelagic ecology and ocean biogeochemistry.**"

**"(5) Assess the results for hypothesis-testing in the fields of plankton ecology and biogeochemistry using the integrated results of KIFES."**

➔ 5.2 Final Remark (lines 27–29, page 29):

"A continuation of **next aOIF experiment** would provide fundamental information and guidelines for future scientific aOIF experiments in HNLC regions, as well as improving our understanding of SO **pelagic ecology**/biogeochemistry."

➔ 6 Summary (lines 12–15, page 30):

"Finally, we envisage a future where the KIFES project, or a similar alternative, becomes a reality so that we may determine whether aOIF is a promising geo-engineering solution **for climate change mitigation and/or an adequate experimental tool for hypothesis-testing in the fields of plankton ecology and ocean biogeochemistry."**

**2.** *Summing up, it is highly unlikely that the rationale presented in this manuscript will convince international bodies or institutions wary of their reputation to support KIFES, and for the LC reviewers and the LC itself to explain the reasons for giving permission for it to be carried out. Besides, the design suggested by the authors requires about 40 days in a stable eddy which requires a great deal of luck. I cannot support publication of the present version of this manuscript as the chances of KIFES acquiring permission based on the rationale presented here are vanishingly small.*

● Thank you for pointing this out. At present, unfortunately, the KIFES project has totally lost a research funding. Actually, we don't know when this project can be revitalized. As Reviewer #3's comments, the Bransfield Strait as selected region for KIFES lies in the heart of the highly sensitive CCAMLR region and it will be hard to find a suitable, young eddy for the experiment in the Bransfield Strait because the topography and current speed, hydrographical features are likely to be more complex and dynamic than along fronts in the open ACC. And, our emphasis on "effectiveness in sequestering $CO_2$" may make a conflict of opinions with many biologists and various NGOs in the Southern Ocean for carrying out KIFES in the heart of the animal-rich Bransfield Strait region. So, we modified the target area of KIFES or next aOIF experiment from a specific area *"Bransfield Strait"* to general area *"sub-polar front"* in **"Section 5 Design of the Korean Iron Fertilization Experiment in the Southern Ocean (KIFES)"**. Previous aOIF experiments (e.g., EisenEx, EIFEX, and LOHAFEX) were conducted in mesoscale eddies formed along the sub-polar front. These eddy showed a stable form. In selecting sites for aOIF, it is important to distinguish the iron-fertilized patch from the surrounding unfertilized waters to easily and efficiently observe iron-induced changes (Coale et al., 1996). Ocean eddies provide an excellent setting for aOIF experimentation because they tend to naturally isolate interior waters from the surrounding waters. Eddy centers also tend to be subject to relatively slow current speeds, with low shear and high vertical coherence, providing ideal conditions for tracing the same water from the surface to below the winter mixed layer depth, while simultaneously minimizing lateral stirring and advection (Smetacek et al, 2012). Therefore, we thought that it is better to suggest the general sites like sub-polar front where eddy can be stably maintained for a long time based on **"Section 4 Future: Designing future aOIF experiments (lines 6–14, page 24)"** rather than suggesting a specific region like Bransfield Basin where we planned KIFES. We also removed the **"Section 5.1 Background- Bransfield Basin"**. Please find the **Section 5 (line 21, page 27 – line 31, page 29)**.

**3.** Another problem created by excessive reliance on geoengineering as a justification for aOIF is not exploiting the valuable information delivered by aOIF experiments in general. The criterion used for success of the various experiments as exemplified by lines 22 – 24 of the abstract is misleading. Here the authors write that the "effectiveness of aOIF … has been unexpectedly low compared to …. natural blooms" with the exception of EIFEX. This uncertainty is used as a rationale to carry out more experiments. This is a flawed argument because OIF experiments should not be compared with each other as individual entities on equal footing but need to be judged in the context of their duration.

● Thank you for your comments. As Reviewer #3 mentioned, previous aOIF experiments have different experiment periods and each aOIF experiment has provided new results on basic processes pertaining to the relationship between pelagic ecology and biogeochemistry, such as selection of the dominant phytoplankton group or species, the effects of grazing, interactions within the plankton community, and effects of nutrient concentrations on the growth of phytoplankton. During IronEx-2, SOIREE, EisenEx, SEEDS-1, SOFeX-N/-S, EIFEX, and SERIES, a >2-fold increase in primary productivity within the mixed layer depth, with massive diatom-dominated blooms, was commonly observed. However, changes in the carbon export varied substantially and differed from experiment to experiment. For the results of export flux, we showed iron-induced export fluxes observed during each experiment in Section 2.4 before comparing different results of each experiment **(Section 2.4.1: lines 25–31, page 11, Section 2.4.2: lines 3–28, page 13, and Section 2.4.3: lines 33–39, page 14)**. Among previous aOIF experiments, EIFEX was the only aOIF experiment that produced significant carbon export to deeper layers (down to 3,000 m). Except in EIFEX, other aOIF experiments did not detect the significant increase in carbon export to deep layers. From these previous aOIF experiments, we have suggested that to detect significant carbon exported below the winter mixed layer depth following an increase in primary productivity, at least three conditions are necessary: (1) a shift to a diatom-dominated community, (2) low bacterial respiration and grazing pressure rates within the mixed layer, and (3) a sufficient experimental duration, enabling both immediate and delayed responses to iron addition to be observed **(Please refer line 33, page 15 – line 8, page 16)**. Finally, based on Reviewer #3's comments, we have revised a sentence to avoid confusion in **Abstract and Section 1 Introduction**.

➔ Abstract (lines 21–23, page 1):

"However, except in the Southern Ocean European Iron Fertilization Experiment, **no significant change in** the effectiveness of aOIF (i.e., the amount of iron-induced carbon export flux below the winter mixed layer depth) has been **detected**."

➔ 1 Introduction (lines 5–6, page 4):

"However, **no significant increase in** carbon exports **has been detected during all aOIF experiments** (de Baar et al., 2005; Boyd et al., 2007),"

**4.** *The process of acquiring permission for an aOIF from the LC is a multi-facetted enterprise involving not only ecology, biogeochemistry and climate science (Martin's iron hypothesis), but also social sciences (ethics and efficacy of climate engineering measures) and ocean governance (international law of the sea and its enforcement).*

*The developments in governance and political views on aOIF have been mentioned in this manuscript in a rather low-key way on pages 21-22. I would suggest putting these developments up front, also mentioning them in the Introduction, and pointing out that since a legal framework has been put in place to prevent venture capitalists from deploying large-scale OIF that particular threat no longer exists. Indeed, one might argue here that inaction on the part of scientists might be an incentive for others to go ahead with experiments as happened off Canada in 2012. Again, this OIF event is mentioned in the text but would do better in the introduction.*

● As suggested by Reviewer #3, we have added the mention about international law in the **"Section 1 Introduction" and "Section 3.3 Regulation of aOIF: International law of the sea as it applies to aOIF"**.

➔ 1 Introduction (lines 10–16, page 4):

**"A legal framework has been put in place to prevent venture capitalist from deploying large-scale OIF in any international waters because of the potential threat of commercialization and large-scale damage inflicted on the environment by venture capitalists acting primarily on profit motivation. No other marine scientific institutions are willing to take up the challenge of carrying out new experiments due to the fear of negative publicity. Consequently, inaction on the part of scientists might be an incentive for others to go ahead with illegal experiments as happened off Canada in 2012 (e.g., 'the 2012 Haida Gwaii Iron Dump' off the west coast of Canada)."**

➔ Section 3.3 Regulation of aOIF: International law of the sea as it applies to aOIF (lines 28–31, page 22):

**"The process of acquiring permission for an aOIF experiment from the LC/LP is a multi-facetted enterprise involving not only ecology, biogeochemistry and climate science (i.e., Martin's iron hypothesis), but also social sciences (i.e., ethics and efficacy of climate engineering measures) and ocean governance (i.e., international law of the sea and its enforcement)."**

**5.** *The chapter on the review of previous experiments is dissatisfying because, as mentioned above, it compares results of experiments directly with each other instead of accounting for their length. For instance it is mentioned that SOIREE had very different results compared to the other SO experiments. But SOIREE lasted only about 2 weeks, so it is not surprising that sinking losses were low. Absence of evidence is not evidence of absence. When experiments are compared with each other it is important to ensure that the comparisons are restricted to matching time periods, under consideration of temperature. If the first 2 weeks of SOFEX South, EisenEx and EIFEX are compared, the results would not be very different. The authors do mention later on that one has to consider the entire history of an iron fertilized bloom, but this should be self-understood. What they do not discuss is that the first 2 weeks have a decisive effect on the development and demise of the bloom. So, even if an experiment lasts only for a short while, useful information can be gained anyway from it. This is a critical point for KIFES because it is highly unlikely that the fertilized patch can be followed for much longer in that dynamic region.*

● The Reviewer #3 is correct. We have added the mention about the first 2 weeks and useful information for a short experiment period.

➔ 4 Future: Designing future aOIF experiments (lines 34–36, page 24):

**"Although the first 2 weeks have a decisive effect on the development and demise of the bloom,** it has been suggested that most aOIF experiments did not cover the full response times from onset to termination (Boyd et al., 2005)."

➔ 4 Future: Designing future aOIF experiments (line 41, page 24 – line 3, page 25):

"This indicates that short experimental durations may not be sufficient for detecting the full influence of aOIF on PP and ecosystem (Figs. 8b and 10), **although useful information can be gained only for a short period, including dominance of spore-forming *Chaetoceros* species during SEEDS-1 experiment** (Boyd et al., 2000; Tsuda et al., 2003; de Baar et al., 2005; Tsuda et al., 2005)."

**6.** *This paper gives little consideration to the species-specific level of the organisms involved, because of its geoengineering focus. However, it is well established that, depending on the species of diatom stimulated by the aOIF, the effect on the BCP can be quite different.*

● We have added the mention about the species-specific level of the organisms.

➔ 2.4.3 Subarctic North Pacific (lines 15–19, page 14):

**"The effect on the biological pump can be quite different depending on the species of diatom stimulated by the aOIF. *Chaetoceros debilis* known to be widespread in coastal environments intensifies the biological pump by forming resting spores in contrast to grazer-protected, thickly silicified oceanic species (e.g., *Fragilariopsis* sp. and *Thalassiothrix* sp.) that contribute silica but little carbon to the sediments."**

**7.** *Another point that needs to be cleared up is the belief prevalent in the public that even experimental aOIF can harm the environment. This was apparent during LOHAFEX and nothing much has changed since then regarding public education. However, even biologists are apprehensive that small-scale experiments can cause harm, for instance, by inducing harmful algal blooms (HABs) that could impact the food chains leading to birds and mammals. This fear also needs to be addressed up front by pointing out that (to the best of my knowledge) no impacts of harmful algal blooms on animal life have ever been reported from the SO, not even under conditions of natural OIF around islands and land masses. Besides, the species that contributed to aOIF blooms have always been local species whenever the species composition was recorded. If OIF were to stimulate domoic acid production by SO Pseudo-nitzschia species, then such blooms should be common place in the productive hot spots of the SO, but this has not been reported to date from the Peninsula region despite investigations going back many decades.*

● Thank you. However, discernable changes in domoic acid (DA) production were found in IronEx-2 and SOFeX-S experiments (Silver et al., 2010). During IronEx-2 and SOFeX-S, high cell abundances of *Pseudo-nitzschia* ($10^6$ and $10^5$ cells l$^{-1}$, respectively) combined with moderate DA cell quotas (0.05 and 1 pg DA cell$^{-1}$, respectively) produced toxin levels as high as 45 ng DA l$^{-1}$ and 220 ng DA l$^{-1}$ in the water, respectively, i.e., toxin levels high enough to damage marine communities in coastal waters (Scholin et al., 2000; Schnetzer et al., 2007). Therefore, we had suggested that it is necessary to clarify/quantify DA production in response to aOIF **(Please refer Section 3.2 Considering environmental side effects: lines 2–24, page 21)**.

**8.** *Trace gases such as N2O, methane and halogenated hydrocarbons are likely to be a problem only at the very large and long-term scales, implying that the answer to the Objective posed in lines 29-30 of page 29 will be "no".*

● Thank you for your comment. We have modified the objective in **"Section 5.1.5 Year five plan (lines 12–14 and 17–18, page 29)"**. **Please refer Reviewer #3's Response 1.**

**9.** *Other hypotheses pertain to the role of grazers. The Bransfield Strait is home to the iconic zooplankter krill, of which I could find no mention in the text.*

● Thank you for pointing this out. However, we have modified the target area for next aOIF experiment from "*Bransfield Strait*" to "*sub-polar front*" in **"Section 5 Design of the Korean Iron Fertilization Experiment in the Southern Ocean (KIFES)"**. **Please refer Reviewer #3's Response 2.**

> **10.** *I would suggest that in the pre-experiment investigations the authors select conspicuous natural blooms appearing as chlorophyll patches on satellite images for process studies and use surrounding water as controls whether located within an eddy or not, in addition to monitoring eddies present in that period and place. By studying mature, natural blooms of high chlorophyll concentrations and snapshots of their fate it will be possible to connect diatom species composition with sinking behaviour.*

● We have added the mention about that.

➔ 4 Future: Designing future aOIF experiments (lines 8–10, page 26):

[revised manuscript text omitted]

---

## Author Response (AR5)

Dear Dr. Victor Smetacek,

We greatly appreciate to your efforts and time to improve our manuscript. We provide our responses (Plain text with black and blue colors) to all comments (*italic text*) below.

**- Response to Reviewer Comments -**

**Reviewer #3**

> **1.** *The eddies can be reliably identified and tracked with satellite imagery of sea level height anomalies that would not have worked as well in the Bransfield Strait region. The authors need to include this highly important tool in their arsenal of methods listed in section 5, Page 27. PF eddies, generally influenced by bottom topography, appear prominently and accurately in satellite imagery; hence carrying out a thorough study of the past history of eddies in the potential experimental region should be mentioned here.*

- Thank you for your comment. We have added statements on the detection of eddies through satellite altimetry in the **"Section 4 Future: Designing future aOIF experiments" and "Section 5.1.1 Year one plan".**

➔ 4 Future: Designing future aOIF experiments (lines 17–18, page 24):

**"The mesoscale eddies can be reliably identified and tracked with satellite sea surface height anomalies (Smetacek et al., 2012)."**

➔ 5.1.1 Year one plan (lines 9–11, page 28):

**"**(6) Analyze eddy development and distribution **in the potential experiment region by carrying out a thorough study of eddy history using satellite sea surface height anomalies."**

> **2.** *In this connection it would be wise to promote public awareness of the need for such studies and their rationale. I would strongly advise adding public relations to the list of preparations for KIFES.*

● Thank you for your suggestion. We have included items specific to public relations in our list of preparations for KIFES in **"Section 5.1.1 Year one plan", "Section 5.1.3 Year three plan", and "Section 5.1.4 Year four plan"**.

➔ 5.1.1 Year one plan (line 36, page 27– line 16, page 28):

**"**Goals: (1) Data collection with regard to oceanographic conditions in the SO PF, including both eddy development and distribution. (2) Establishment of the study aims, hypothes**e**s, and site for the KIFES experiment. **(3) Announcement of KIFES plans and intentions to the public.**

Objectives: (1) To understand the physical and biogeochemical oceanography of relevance to the SO PF as an aOIF site through an analysis of earlier datasets and a review of published papers. **(2) To promote public awareness of the aims of KIFES and its rationale.**

Main tasks: (1) Review databases of physical and biogeochemical parameters from previous surveys conducted in the SO PF. (2) Review the SO PF oceanographic conditions using data analysis and references. (3) Establish the study aims, hypothes**e**s, and **potential locations** in the SO PF for an aOIF experiment, based on the results obtained from tasks (1) and (2). (4) Design an oceanographic cruise map for the first preliminary survey in the SO PF. (5) Study natural blooms appearing as high chlorophyll-a concentrations patches and their fate using satellite data in the SO PF. (6) Analyze eddy development and distribution **in the potential experiment region by carrying out a thorough study of eddy history using satellite sea surface height anomalies.** (7) Prepare scientific instruments for ocean physical and biogeochemical monitoring. (8) Establish an international collaborative aOIF network. **(9) Distribute a press release announcing KIFES plans and intentions and explain the scientific background to the public via short films and animated cartoons on the role of the ocean in maintaining atmospheric $CO_2$ levels, Martin's hypothesis, and results of previous aOIF experiments.** (10) Submit KIFES field program proposal for the 'Initial Assessment' to determine that KIFES falls within the remit of ocean fertilization and should be evaluated in the LC/LP assessment framework based on the results from tasks (1)–(6).**"**

➔ 5.1.3 Year three plan (line 26, page 28– line 3, page 29):

**"**Goals: (1) Preliminary hydrographic survey outside/inside the center of an eddy structure prior to the KIFES experiment. (2) Approval of KIFES from LC/LP. **(3) Preparation for making a film on KIFES.**

Objectives: (1) To compare oceanographic conditions inside and outside the center of an eddy structure formed in the SO PF prior to the KIFES experiment. (2) To obtain permission on the basis that the proposed KIFES is legitimate scientific research from the LC/LP. **(3) To prepare the groundwork for making a documentary film on the KIFES expedition.**

Main tasks: (1) Using the ice breaker RV *ARAON*, detect an eddy formed in the SO PF using observations from acoustic Doppler current profilers (ADCPs) and satellites. (2) Conduct intensive physical and biogeochemical field investigations both inside and outside the center of an eddy structure. (3) Assess the physical and biogeochemical properties outside vs. inside the center of an eddy structure prior to KIFES. (4) Establish a final design for KIFES. (5) Submit the research results for 'Environmental Assessment' stage of the LC/LP assessment framework and obtain approval for the KIFES experiment via the 'Decision Making' process from the LC/LP. **(6) Contact with the director of Korean science TV channel for making a film on the KIFES expedition.**

➔ 5.1.4 Year four plan (lines 22–23, page 29):

**(9) Prepare a documentary film during the KIFES expedition by a Korean government TV channel for a regular series on science for the public."**
* * *
**3.** *Remove the apostrophes around species names throughout the text.*
- Done.
* * *
**4.** *P4 line 11 and 13 replace "venture capitalists" with "private entities".*
- Done.
* * *
**5.** *P10 line 30. The main problem with IronEx-I was subduction of the fertilized surface layer by adjacent water. This needs to be mentioned here.*
- Thank you for pointing this out. We have added the following sentence.

➔ 2.4.1 Equatorial Pacific (lines 30–33, page 10):

**"Unexpected small responses during IronEx-1 were due to subduction of the fertilized surface layer by adjacent water (Coale et al., 1998). The contrasting** results from the two experiments are **also** likely to be associated with **whether or not there were** additional iron injections (IronEx-1: no extra addition; IronEx-2: 2 additional injections) and different experiment durations (IronEx-1: 10 days; IronEx-2: 17 days).**"**

**6.** *P13 line 27/28. Replace "this component of the biological pump" with "diatom aggregate formation" and "to be resolved" with "focused study"*

- Done.

**7.** *P15 Line 17 "blooms"*

- Done.

**8.** *P15 Line 20 delete "the" before "high"*

- Done.

**9.** *P15 Line 28 replace "covers" with "extends"*

- Done.

**10.** *P15 Line 30 add the "between the two"*

- Done.

**11.** *P15 Line 31 replace "eligible" with "appropriate"*

- Done.

**12.** *P15 Line 38 For the sake of accuracy replace "was" with "was reported to be" rapidly remineralized*

- Done.

**13.** *P16 Lines 2 and 3 replace "detecting – aggregates" with "for diatoms to form aggregates and sink" (i.e., carbon export).*

- Done.

**14.** *P16 Line 36 delete "the" before "these".*

- Done.

**15.** *P20 Line 7 Clarify "an early SOIREE experiment"*

● We clarified as follows:

➔ 4 Future: Designing future aOIF experiments (lines 10–12, page 20):

**"During the SOIREE experiment, the initial** dominant phytoplankton species were haptophytes **and they remained dominant until day 7. Since then,** DMS production was increased by micro-zooplankton grazing on DMSP-rich haptophyte **groups (e.g.,** *Phaeocystis***)** (Gall et al., 2001b)."

**16.** *P20 Line 9 Prymnesiophyceae are haptophytes. Do you mean "Phaeocystis" here?*

● Thanks. We have revised it (Please refer **Response 15** above).

**17.** *P20 Line 11 replace "derive" with "result in"*

● Done.

**18.** *P20 Line 15/16 The sentence citing Bopp et al. does not make sense.*

● We apologize for the confusion. We have modified the sentence as follows:

➔ 4 Future: Designing future aOIF experiments (lines 17–19, page 20):

**"In addition, a 20-year aOIF simulation through three-dimensional ocean biochemical model** did not **show significant increase in DMS emissions from the SO** (Bopp et al., 2008)."

**19.** *P24 paragraph lines 15 – 31: When: I would consult satellite chlorophyll climatology maps to determine the timing of blooms in the SO. North of the APF diatom blooms have been reported as early as October/mid-November by e.g. Bathmann et al. 1997 DSR II.*

● We have included a statement on the use of satellite chlorophyll climatology maps to determine the timing of blooms in the SO.

➔ 4 Future: Designing future aOIF experiments (lines 25–26, page 24):

**"Weekly and monthly climatological maps of chlorophyll-a concentrations derived from satellite data could provide the necessary information for determining the timing of blooms in the SO PF (Westberry et al., 2013)."**

● For **Reviewer #3's comment** about north of the APF diatom blooms, we recommended south of the APF as the location (i.e., 'where') for fertilizing iron (**Please refer lines 3–9, page 24**). To avoid confusion, we added the word 'south of the SO PF' in the last sentence of **Section 4 'When'**.

➔ 4 Future: Designing future aOIF experiments (lines 35–37, page 24):

**"**Considering the key factors (i.e., micro-/macro-nutrient availability, light availability, and grazing pressure) controlling PP in the SO, the most appropriate timing for the start of **an aOIF experiment to the south of the SO PF** is likely to be the early summertime (i.e., late December to early January).**"**

**20.** *Page 26 Line 10 Delete "During aOIF experiments," and start with "All aOIF experiments…"*

● Done.

**21.** *Page 26 Line 12 Ideally the surface-tethered GPS buoy with a low above-water profile should be deployed after deciding where to place the patch (e.g. at the centre of the eddy) prior to the first station before fertilization (the baseline) and fertilization carried out around this buoy.*

● Thank you for pointing this out. We have modified the sentences describing the use of drifting buoys in **Section 4 Future: Designing future aOIF experiments**.

➔ 4 Future: Designing future aOIF experiments (lines 16–21, page 26):

**"A**ll aOIF experiments used physical tracers **to follow the iron-fertilized patches,** in particular GPS and Argos equipped drifting buoys **that provide the tracked positions of a fertilized patch as a passive system moving with local currents. GPS and Argos equipped drifting buoys should be released before fertilization (to provide a baseline), and ensuing aOIF experiments should be carried out in the region described by the drifting buoys deployed. Drifting buoys are,** however, **not perfect representations of water motion and due to the effects of winds are likely to** escape **a** fertilized patch **within a few days to a week regardless of how deep their drogues (Watson et al., 1991; Law et al., 1998; Stanton et al., 1998)."**

**22.** *Page 26 Line 13 What is meant by "a natural and passive system"? Our experience has been that all the buoys used: a plastic balloon (EisenEx), a spar buoy (EIFEX) and a small, low-lying spherical buoy all drifted out of the patch within a few days to a week despite the drogue being placed at ~20 m depth, presumably due to slippage of the surface water from the underlying mixed layer by wind.*

● We apologize for this confusion. We have modified this sentence. Please refer **Response 21** above.

**23.** *Page 27, line 14 Include also online pCO₂ measurements as a tracer of the patch.*

● We have included online $p$CO$_2$ measurements as follows:

➔ Abstract (lines 33–35, page 1):

[revised manuscript text omitted]